



# State of Wildfires 2023-24

**Authors:**

Matthew W. Jones[1,★], Douglas I. Kelley[2,★], Chantelle A. Burton[3,★], Francesca Di Giuseppe[4,★], Maria Lucia F. Barbosa[5,6], Esther Brambleby[1], Andrew J. Hartley[3], Anna Lombardi[7], Guilherme Mataveli[8,1], Joe R. McNorton[4], Fiona R. Spuler[9], Jakob B. Wessel[10,11], John T. Abatzoglou[12], Liana O. Anderson[13], Niels Andela[14], Sally Archibald[15], Dolors Armenteras[16], Eleanor Burke[3], Rachel Carmenta[17], Emilio Chuvieco[18], Hamish Clarke[19], Stefan H. Doerr[20], Paulo M. Fernandes[21], Louis Giglio[22], Douglas S. Hamilton[23], Stijn Hantson[24], Sarah Harris[25], Piyush Jain[26], Crystal A. Kolden[27], Tiina Kurvits[28], Seppe Lampe[29], Sarah Meier[30], Stacey New[3], Mark Parrington[31], Morgane M. G. Perron[32], Yuquan Qu[33,34], Natasha S. Ribeiro [35], Bambang H. Saharjo[36], Jesus San-Miguel-Ayanz[37], Jacquelyn K. Shuman[38], Veerachai Tanpipat[39], Guido R. van der Werf[40], Sander Veraverbeke[33,1], Gavriil Xanthopoulos[41]

**Institutions:**

1. Tyndall Centre for Climate Change Research, School of Environmental Sciences, University of East Anglia, Norwich Research Park, Norwich, UK, NR4 7TJ
2. Hydro-climate risks, UK Centre for Ecology & Hydrology, Wallingford OX10 8BB, U.K.
3. Hadley Centre, Met Office, Fitzroy Road, Exeter, UK, EX1 3PB
4. Earth System Modelling Section, Forecast Department, European Centre for Medium-range Weather Forecast, Shinfield Park, Reading RG29AX, United Kingdom
5. Department of Remote Sensing, National Institute for Space Research, Avenida dos Astronautas, 1758. Jd. Granja - São José dos Campos - São Paulo, Brazil , 12227-010
6. Natural Sciences Center, Federal University of São Carlos, Rodovia Lauri Simões de Barros, km 12 - SP-189 - Aracaçu, Buri - São Paulo, Brazil, 18290-000
7. Climate Intelligence, Research Department, European Centre for Medium-range Weather Forecast, Shinfield Road, Reading, UK, RG29AX
8. Earth Observation and Geoinformatics Division, National Institute for Space Research, Avenida dos Astronautas, 1758. Jd. Granja - São José dos Campos - São Paulo, Brazil , 12227-010
9. Department of Meteorology, University of Reading, University of Reading, Earley Gate, Whiteknights Rd, Reading RG6 6ET
10. Department of Mathematics and Statistics, University of Exeter, Harrison Building, University of Exeter, North Park Road, Exeter, UK
11. The Alan Turing Institute, British Library, 96 Euston Road, London, UK
12. School of Engineering, University of California, Merced, 5200 N Lake Rd, Merced, CA, 95343, USA
13. Cemaden/MCTI, Estrada Doutor Altino Bondensan, 500 - Distrito de Eugênio de Melo, São José dos Campos - São Paulo, Brazil
14. BeZero Carbon, 25 Christopher Street, London, UK, EC2A 2BS
15. School of Animal Plant and Environmental Sciences, University of Witwatersrand Johannesburg, University Corner, Braamfontein, Johannesburg
16. Landscape Ecology and Ecosystem Modelling Group, Faculty of Sciences, Department of Biology, Universidad Nacional de Colombia, Cra. 30 # 45-03, Bogotá D.C., CP 111321, Colombia
17. Tyndall Centre for Climate Change Research, School of Global Development, University of East Anglia, Norwich Research Park, Norwich, UK, NR4 7TJ
18. Department of Geology, Geography and the Environment, Universidad de Alcalá, Colegios, 2 - 28801 Alcalá de Henares
19. FLARE Wildfire Research, School of Agriculture, Food and Ecosystem Sciences, University of Melbourne, Grattan St, Parkville, Australia, 3010
20. Centre for Wildfire Research, Swansea University , Singleton Park Swansea SA2 8PP Wales, UK
21. ForestWISE—Collaborative Laboratory for Integrated Forest & Fire Management, Centre for the Research and Technology of Agro-Environmental and Biological Sciences, Universidade de Trás-os-Montes e Alto Douro, Quinta de Prados, Vila Real, Portugal, 5000-801
22. Department of Geographical Sciences, University of Maryland, College Park, MD 20742
23. Marine, Earth and Atmospheric Science , North Carolina State University, Raleigh, North Carolina, USA, 27695
24. Program in Earth System Sciences, Faculty of Natural Sciences, Universidad del Rosario, Bogotá, Colombia



25. Fire Risk, Research and Community Preparedness, Country Fire Authority, Burwood East, Victoria, Australia
26. Northern Forestry Centre, Canadian Forest Service, Natural Resources Canada, 5320 122 St NW,
Edmonton, AB T6H 3S5, Canada
27. Wildfire Resilience Center, School of Engineering, University of California, Merced, 5200 N Lake Rd, Merced,
CA, 95343, USA
28. GRID-Arendal, P.O Box 183, N-4802, Arendal, NORWAY
29. Department of Water and Climate, Vrije Universiteit Brussel, Pleinlaan 2, 1050 Brussel, Belgium
30. Land, Environment, Economics and Policy Institute, Department of Economics, University of Exeter, Rennes
Drive, Exeter, United Kingdom, EX4 4ST
31. Atmospheric Composition Section, Research Department, European Centre for Medium-range Weather
Forecast, Robert-Schuman-Platz 3, 53175 Bonn, Germany
32. UMR 6539 CNRS/IRD/Ifremer/LEMAR, Institut Universitaire Européen de la Mer, University of Brest, F-29280
Plouzané, France
33. Department of Earth Sciences, Faculty of Science, Vrije Universiteit Amsterdam, De Boelelaan 1105, 1081
HV Amsterdam, Netherlands
34. Institute of Bio- and Geosciences: Agrosphere (IBG-3), Forschungszentrum Jülich, Wilhelm-Johnen-Straße,
52428 Jülich, Germany
35. Faculty of Agronomy and Forest Engineering, Eduardo Mondlane University, 3453 Avenida Julius Nyerere,
Maputo, Mozambique
36. Faculty of Forestry, Bogor Agricultural University, Kampus Ipb Darmaga, Bogor
37. European Commission Joint Research Center, European Commission, Rue du Champ de Mars 21, 1050
Brussels, Belgium
38. NASA Ames Research Center, PO Box 1 Moffett Field, CA 94035-1000
39. Upper ASEAN Wildland Fire Special Research Unit (WFSRU), Forestry Research Center, Faculty of Forestry,
Kasetsart University, 5th Floor, 72nd Anniversary of Faculty Forestry Building
40. Wageningen University, Droevendaalsesteeg 3, 6708PB Wageningen
41. Forest Fire Laboratory, Institute of Mediterranean Forest Ecosystems, Hellenic Agricultural Organization
(DIMITRA), Terma Alkmanos, Ilisia, 11528, Athens, Greece
★ These authors contributed equally to this work.
Correspondence to:
matthew.w.jones@uea.ac.uk
doukel@ceh.ac.uk
chantelle.burton@metoffice.gov.uk
Francesca.DiGiuseppe@ecmwf.int

**Key words:** Wildfire, Extreme Fire, Attribution, Climate Change

## Abstract

Climate change is increasing the frequency and intensity of wildfires globally, with significant
impacts on society and the environment. However, our understanding of the global distribution
of extreme fires remains skewed, primarily influenced by media coverage and regional
research concentration. This inaugural State of Wildfires report systematically analyses fire
activity worldwide, identifying extreme events from the March 2023-February 2024 fire season.
We assess the causes, predictability, and attribution of these events to climate change and
land use, and forecast future risks under different climate scenarios. During the 2023-24 fire
season, 3.9 million km² burned globally, slightly below the average of previous seasons, but
fire carbon (C) emissions were 16% above average, totaling 2.4 Pg C. This was driven by
record emissions in Canadian boreal forests (over 9 times the average) and dampened by
reduced activity in African savannahs. Notable events included record-breaking wildfire extent
and emissions in Canada, the largest recorded wildfire in the European Union (Greece),
drought-driven fires in western Amazonia and northern parts of South America, and deadly
fires in Hawai'i (100 deaths) and Chile (131 deaths). Over 232,000 people were evacuated in
Canada alone, highlighting the severity of human impact. Our analyses revealed that multiple
drivers were needed to cause areas of extreme fire activity. In Canada and Greece a



combination of high fire weather and an abundance of dry fuels increased the probability of
fires by 4.5-fold and 1.9-4.1-fold, respectively, whereas fuel load and direct human
suppression often modulated areas with anomalous burned area. The fire season in Canada
was predictable three months in advance based on the fire weather index, whereas events in
Greece and Amazonia had shorter predictability horizons. Formal attribution analyses
indicated that the probability of extreme events has increased significantly due to
anthropogenic climate change, with a 2.9-3.6-fold increase in likelihood of high fire weather in
Canada and a 20.0-28.5-fold increase in Amazonia. By the end of the century, events of similar
magnitude are projected to occur 2.22-9.58 times more frequently in Canada under high
emission scenarios. Without mitigation, regions like Western Amazonia could see up to a 2.9-
fold increase in extreme fire events. For the 2024-25 fire season, seasonal forecasts highlight
moderate positive anomalies in fire weather for parts of western Canada and South America,
but no clear signal for extreme anomalies is present in the forecast. This report represents our
first annual effort to catalogue extreme wildfire events, explain their occurrence, and predict
future risks. By consolidating state-of-the-art wildfire science and delivering key insights
relevant to policymakers, disaster management services, firefighting agencies, and land
managers, we aim to enhance society's resilience to wildfires and promote advances in
preparedness, mitigation, and adaptation.

## Short Summary

This inaugural State of Wildfires report catalogues extreme fires of the 2023-24 fire season.
For key events, we analyse their predictability and drivers and attribute them to climate change
and land use. We provide a seasonal outlook and decadal projections. Key anomalies
occurred in Canada, Greece, and western Amazonia, with other high-impact events
catalogued worldwide. Climate change significantly increased the likelihood of extreme fires,
and mitigation is required to lessen future risk.

## 1    Introduction


### 1.1    Background

The potential for wildfires is growing under climate change, with increases in the frequency
and intensity of drought and periods of fire-favourable weather driving reductions in vegetation
(fuel) moisture and priming landscapes to burn more regularly, severely, and intensely (Jones
et al., 2022; UNEP, 2022a; Abatzoglou et al., 2019). Additionally, human activities and land
use change can contribute to or exacerbate the risk of extreme fires, especially in tropical
forests where people are the primary cause of ignition and forest degradation (Lapola et al.,
2023). Recent years have been marked by a series of extreme wildfire events spanning the
globe, with record levels of burned area (BA) occurring in the 2019-2020 Australian "Black
Summer" bushfires (Abram et al., 2021) and a series of high-ranking wildfire seasons
occurring in quick succession in the western US (2020 and 2021; Higuera & Abatzoglou,
2020), Siberia (2020 and 2021; Zheng et al., 2023), Europe (2022; European Commission
Joint Research Centre, 2023) and South America (2019, 2020; Kelley et al., 2021; Ferreira
Barbosa et al., 2022; Silveira et al., 2020). The 2023-24 fire season was marked by
unprecedented wildfire extent and emissions in Canada, deadly fast-moving fires in Hawaii
and Chile, the largest individual wildfires on record in the European Union and Canada,
widespread fires in northwestern South America including parts of Amazonia such as Brazil,
Bolivia, Colombia, and Venezuela (Mataveli et al., 2024; Kolden et al., 2024; European
Commission EU Science Hub, 2023).
The prominence of recent extreme wildfires and wildfire seasons notably contrasts with overall
trends in the area burned by fires globally. Due mostly to a reduction in the global savannahs



tied to landscape fragmentation and changing rainfall patterns, global BA has fallen since the
beginning of this century by around one-quarter (Andela et al., 2017; Jones et al., 2022; Chen
et al., 2024). Critically, this decline in fire extent masks major shifts in the distribution of fires
globally, with regions such as eastern Siberia and the western US and Canada experiencing
a more than 40% increase in BA since 2000 (Jones et al., 2022) and regions such as southeast
Australia also showing significant increases over longer periods (Canadell et al., 2021).
Likewise, there have been shifts in the global distribution of BA and fire carbon (C) emissions
from non-forests to forests globally and from the tropics to the extratropics (Kelley et al., 2019).
Hence, focussing on global aggregated BA extent underplays the scale and magnitude of
changes in wildfire activity and impact on regional levels. An increase in forest and peatland
burning is particularly concerning due to the rich ecosystem services that these regions
provide, including C storage and biodiversity (UNEP, 2022b). The intensification of fire
regimes in environments that are less fire-adapted is significant, as these ecosystems are
expected to be least resilient to such changes (Grau-Andrés et al., 2024).
The extreme wildfire events of recent years have significantly impacted society and
ecosystems across the globe. Since 1990, wildfire disasters have directly killed or injured at
least ~18,000 people, a conservative measure based on incomplete records and reporting
biased to the global Northern countries (updated from Jones et al., 2022; Centre for Research
on the Epidemiology of Disasters, 2024). In 2023, 232,000 people were evacuated due to
wildfires in Canada alone (Jain et al., 2024; Kolden et al., 2024). Also since 1990, fires are
estimated to have caused on the order of 10 million premature deaths globally through
degraded air quality (Johnston et al., 2012). Degraded air quality related to fires is experienced
most strongly in the tropics (Pai et al., 2022) and often disproportionately affects traditional
communities with poor public services or means of protection (Carmenta et al., 2021). Yet,
images of North America, including the infamous Wall Street, blanketed in smoke during the
2023 fire season highlight the global nature of this problem.
As anthropogenic emissions of $CO_2$ remain persistently high, the world's natural C sinks in
forests, peatlands and other ecosystems are increasingly pivotal to moderating increases in
atmospheric $CO_2$ concentration and are often relied upon for delivering national plans for
reaching Net Zero (Smith et al., 2023) and offering sites for nature based solutions (NBS). Yet,
massive wildfire emissions from boreal forests and soils in Siberia and Canada across the
years 2020, 2021, and 2023 amount to over 1 billion tonnes of C, a gross flux comparable in
magnitude to annual $CO_2$ emissions from fossil fuel combustion in India, the EU27 or the USA
(Friedlingstein et al., 2023; Zheng et al., 2023). Ecosystem function is also impacted by
extreme wildfires through widespread mortality of forest stands that undermine the
regenerative capacity of forests (Nolan et al., 2021a) and the habitats of many endemic
species being degraded in biodiversity hotspots (Ward et al., 2020). Not only are these
emissions high, but they may represent a change in the fire-C cycle from sink to source,
undermining assumptions about post-fire regrowth offsetting fire emissions. The lands and
territories of traditional communities and Indigenous Peoples are degraded and transformed
through wildfires, raising climate justice issues (Garnett et al., 2018; Barlow et al., 2018;
Lapola et al., 2023). Further, conflating distinct fire types has also stigmatised *intentional*
small-scale intergenerational managed fire use and led to prohibitive fire governance that
affects local communities (Carmenta et al., 2021; Barlow et al., 2020).
Mitigating and adapting to increases in wildfire potential are growing priorities of policymakers
and involve coordination with many other stakeholders. National and international disaster
management centres are seeking to enhance predictive capacity, while fire management
agencies are expanding or re-allocating their resources to rapidly suppress fires to avoid them
becoming too large, fast or intense. A number of international organisations such as the UN
Environment Programme (UNEP, 2022a), the World Bank (2020, 2024), and the Organisation
for Economic Co-operation and Development (OECD, 2023) and a range of other inter- or
non- governmental organisations are producing reports that consolidate evidence on the





changing risk of extreme fires and identify best practices for mitigating their impacts, including
through land management and urban/rural planning. Many land managers are developing and
implementing approaches such as fuel reduction, a process subject to permit systems issued
by regional fire management agencies in some countries (Fernandes and Botelho, 2003;
Stephens et al., 2012; Moreira et al., 2020; Chuvieco et al., 2023). Wildfire response agencies
are exploring innovative approaches to detecting and responding to fires, and there is rising
interest in the prospect of integrated fire management around the world (Food and Agriculture
Organization of the United Nations, 2024). Operators of C market projects and forest carbon-
conservation initiatives, such as REDD+ are particularly wary of the risks that wildfires present
to the permanence of C offsets, seen as a key tool for achieving Net Zero emissions (Barlow
et al., 2012; Smith et al., 2023).
Amidst extreme wildfires and wildfire seasons, stakeholders increasingly turn to scientists for
answers to pressing questions that naturally arise. How extreme was this fire event in a
historical context? Is climate change to blame? Will we see more wildfires like this in the
future? Did land management exacerbate or ameliorate the problem? Can we predict events
like this in future to improve early warning? What is the role of climate policy in reducing risk
of extreme wildfires in future?
While observational, statistical, and modelling tools for assessing extreme wildfire drivers and
predicting wildfire occurrence are advancing rapidly, their application to studying extreme
wildfire seasons or events on timescales relevant to public and political interest remains
limited. The State of Wildfires report embodies a structure for routinely cataloguing extreme
wildfire events at annual frequency and explaining their occurrence and relation to climate
change. We incorporate recent advances in disentangling the drivers of fire to fuel dryness,
fuel load, and weather, and ignition and suppression factors, and by applying these methods
in conjunction with models of global change we quantify the change in likelihood of the past
year's events under climate and land use changes. Observable fire metrics (e.g. burned area)
are the target variable of our causal inference and attribution work, which thereby advances
on more common climate attribution studies that attribute change in fire-favourable
meteorological conditions to climate change. Overall, this report capitalises on recent
advances in the study of extreme fire events and seasons to provide timely information about
shifting fire regimes and their causes. The findings of the report are relevant to policymakers,
the media, and the wider public.
Due to the diversity of environmental settings in which fires occur and the range of ecological,
economic, or societal impacts caused by fires, defining an extreme fire or an extreme fire
season remains inherently challenging. To date, extreme fires have commonly been defined
by their BA extent, by their feedback on the global climate compared to climatological data
and on large-scale ecosystem changes and by their socio-economic impact. While an extreme
fire event may result in significant anomalies against historical Earth Observations in the same
study region, the scientific community currently seeks to apply a more comprehensive
definition of extreme fire, including its impacts on societies, fire-prone regions and on
downwind society and ecosystems.

## 1.2 Objectives of this Report

This inaugural edition of the State of Wildfires report aims to stimulate development of tools
for understanding and predicting extreme fires and to deliver actionable information to policy
and practice stakeholders and society. In this edition we:
1. Regionally identify extreme individual wildfires or extreme wildfire seasons of the
period since March 2023, and place them in context of recent trends.



2. Shortlist a selective number of extremes (extreme individual wildfires or extreme wildfire seasons) with notable impacts on society or the environment, which we term the 'focal events' in this report.
3. Diagnose the contributions of fuel dryness, fuel load, ignitions and suppression to the occurrence of each focal event.
4. Assess the capacity of operational predictive systems to predict each focal event.
5. Attribute each focal event to anthropogenic factors including climate change and land use.
6. Provide an outlook on the probability of extreme events in the coming fire season.
7. Project future changes in the probability of each focal event under future climate scenarios.

The State of Wildfires report will be an annually recurring report that can harness and adopt new methodologies brought forward by the global community of fire scientists between publications.

To achieve our objectives, we integrate a variety of advanced methods developed in recent years. For instance, we identify anomalies in past fire seasons using both regional BA and emissions records and a global dataset of individual fires (Giglio et al., 2018; van der Werf et al., 2017; Andela et al., 2019). This dual approach enables us to detect not only regional anomalies but also individual fires that are exceptionally large or fast-moving. Additionally, we leverage the latest advances in fire risk forecasting on seasonal to sub-seasonal timescales provided by the ECMWF. Furthermore, we employ two state-of-the-art machine learning techniques (Kelley et al., 2019; McNorton et al., 2024) to pinpoint the causes of the extreme fire events of 2023-24. We also apply multiple attribution methods to quantify the influence of climate change and land use change on these extreme events (e.g. Kelley et al., 2019, 2021; Burton, Lampe, et al., 2023). Fire is an inherently stochastic process, and any attempt to assess its causes needs to account for uncertainty. The strength of our approach in this report is in employing different techniques to address the complex questions of how unusual the fires of the last season were, and what the drivers might have been. By using multiple tools to measure different fire metrics and identify the factors driving fires, and by using different techniques to attribute their causes to climate and human influences, we can either build up a solid body of evidence or highlight aspects of extreme fires in recent years that we do not yet understand. While the tools are diverse, they address the questions from different angles, giving us a robust way to assess predictability or quantify uncertainty in their results - also vital components in building confidence in the statements made in this report. We demonstrate the range of tools we have available to tackle these important questions now, and highlight gaps where we still need to develop the research further. The combination of these diverse techniques marks a significant step toward the regular and comprehensive assessment of past and future extreme wildfire events.

Over the coming years and decades, we aim to develop and apply the tools presented in this report in near-real time, thus enhancing our capacity to transfer key insights to decision-makers at the time they most need it. Ultimately, this report serves as a vehicle that consolidates the novel advances and efforts of the fire science community to provide relevant information to stakeholders, enabling society to adapt to emerging fire extremes.

## 2 Methods

### 2.1 Extreme Wildfire Events of 2023-24

We catalogued the extreme regional wildfire events or annual fire seasons in the period March 2023-February 2024 based on a combination of anomalies evident in Earth Observations (**Section 2.1.1**), as well as expert assessment of wildfires that severely impacted people or



the environment (**Section 2.1.4**). This approach deliberately represents a broad definition of extreme wildfire events by allowing our expert panel to flexibly integrate diverse forms of wildfire impacts into their regional assessments of extreme wildfire events. In so doing, we allow for the fact that it is challenging to define a consistent threshold for 'extreme' fires given the diversity in the characteristics and impacts of fire across the many fire regimes (e.g. Archibald et al., 2009; Cunningham et al. 2024). Metrics such as BA and emissions are a good proxy for impact in some cases, yet they cannot fully account for the diverse relationships between fire, ecosystems, people and societal values. Hence our broad definition of "extreme fire" allows nuanced measures of fire impacts to be captured (e.g. fire severity or challenges to suppression; Keeley 2009) and careful consideration of the complex interplay between wildfire and the various forms of fire intentionally applied by humans and serving multiple purposes. It also allows for elaboration of the links between wildfire and the various values that benefit or are burdened by it, including ecosystem services, biodiversity, Indigenous cultural values and biocultural landscapes, heritage, recreation, social capital, the built environment, tourism, agriculture and other industries, and human health and wellbeing.

### 2.1.1 Earth Observations of Fire

#### 2.1.1.1 Input Datasets

We assembled observations of global daily burned area (BA) for the period March 2001-February 2024 from NASA's MODIS BA product (MCD64A1, collection 6.1) at 500m resolution (Giglio et al., 2018, 2021). We also produced a global record of individual fires for the period March 2001-February 2024 by updating the Global Fire Atlas (Andela et al., 2019; Andela and Jones, 2023) through February 2024, driven by the 500m MODIS BA data. The Global Fire Atlas algorithm clusters burned cells into individual fires, tracks their daily progression, and logs attributes such as fire size and mean daily rate of growth. Our updates are provided at Andela and Jones (2023).

In addition, we gathered estimates of fire carbon (C) emissions for the period March 2023-February 2024 from two models driven by Earth Observations of active fires or burned area: firstly, the Global Fire Assimilation System (GFAS) product, provided operationally by the Copernicus Atmospheric Services (CAMS) at 0.1 degree spatial resolution and daily temporal resolution (Kaiser et al., 2012; European Centre for Medium-Range Weather Forecasts, 2024), and; secondly, the Global Fire Emissions Database (GFED; version 4.1s) product at 0.25 degree spatial resolution and daily temporal resolution (van der Werf et al., 2017). GFAS is driven by the fire radiative power (FRP) retrievals in the MODIS active fire product MCD14A1 and biome-level relationships between FRP and biomass consumed based on GFED3 (Kaiser et al., 2012). For the 1997-2016 period, GFED4s is driven by MODIS BA data (MCD64A1 collection 5) supplemented with small fire BA based on MODIS active fire data, and a model for biomass productivity and fuel consumption (van der Werf et al., 2017). For the post-2016 period, emissions are based on active fire detections scaled to emissions using pixel-based scaling factors derived from the 2003-2016 overlapping period.

We note that the MODIS MCD64A1 BA data used in various facets of our analyses are known to be conservative due to the limitations to detecting small fires based on surface spectral changes at 500m resolution. Recent work has shown that including detections of small active fires increases global BA estimates by 93% (Chen et al., 2023). However, variability and trends in regional BA totals using datasets that include small fires do not differ significantly from the variability and trends present in the MODIS MCD64A1 BA data (Chen et al., 2023). In this report, regional BA and fire C emissions totals are conservative and the context of missing small fires must be considered when interpreting the threshold values of the upper quantiles of the fire size distribution from the Global Fire Atlas. Nonetheless, there is consistency in our



approach across years such that anomalies are robust with respect to prior observations, and
our focus on more extreme fires renders smaller fires obsolete in most of our analyses.
**2.1.1.2  Regional Burned Area, Carbon Emissions and Fire Count Totals**
We calculated regional totals of BA and C emissions based on a variety of regional layers
defined in **Table 1**. The regional layers represent a range of biogeographical boundaries (e.g.
biomes), geopolitical boundaries (e.g. countries), and values used in scientific reports (e.g. by
the Intergovernmental Panel on Climate Change; IPCC). We calculated monthly totals of BA
and fire C emissions for each region by aggregating monthly BA and daily C emissions data,
summing the data from the input datasets both spatially and temporally as required. In the
case of fire C emissions, we also calculated the mean estimate of fire C emissions from
GFED4.1s and GFAS, regionally.
We adopt a March-February definition of the global fire season (e.g. the latest global fire
season spans March 2023-February 2024). Due to an annual lull in the global fire calendar in
the boreal spring months, fire season BA totals are least sensitive to the shifts in fire season
cutoffs of 1-2 months if the fire season centres on spring (Boschetti and Roy, 2008). This
makes the global fire season centred on spring a pragmatic option for the study of interannual
variability or trends in fire extent (Boschetti and Roy, 2008). The period March-February is
specifically oriented at the end of the austral fire season and before widespread fires have
begun in the boreal extratropics. The regions where this global definition of the fire season is
most problematic are: northern hemisphere South America, Southeast Asia, and Central
America (Giglio et al., 2013).
In addition, we calculated totals of regional fire counts for each global fire season based on
the number of individual fire ignition points present within each region, using ignition point
vectors from the Global Fire Atlas. The resolution of the MODIS data supplied to the Global
Fire Atlas algorithm is 500 m and hence fires that are smaller in scale are omitted. Regional
or national systems may record greater fire counts due to the inclusion of smaller fires.
**Table 1:** Regional layers to which global Earth Observations were disaggregated and used to
define regions with extreme wildfire seasons or extreme individual wildfire attributes. Regional
layers are available from Jones et al. (2024).

| Layer | Short Form | Source | Notes |
|---|---|---|---|
| Biomes | NA | Olson et al. (2001) | |
| Continents | NA | ArcGIS Hub (2024) | |
| Continental Biomes | NA | See above | Spatial intersect of biomes and continents. |
| Countries | NA | EU Eurostat (2020) | |
| UC Davis Global Administrative Areas (GADM) Level 1 | GADM-L1 | UC Davis (2022) | First sub-national administrative level, such as states of the US or provinces of China. Version 4.1. |
| Intergovernmental Panel on Climate Change *Sixth Assessment Report (AR6) Working Group I (WGI)* Reference Regions | IPCC AR6 WGI Regions | IPCC (2021); SantanderMetGroup (2021) | |
| Global C Project *Regional C* | RECCAP2 | Ciais et al. (2022) | |



| | | | |
|---|---|---|---|
| *Cycle Assessment and Processes (RECCAP2) Reference Regions* | Regions | | |
| Global Fire Emissions Database (GFED) Basis Regions | GFED4.1s Regions | van der Werf et al. (2006) | |

### 2.1.2 Identifying Extreme Fire Seasons and Events from Earth Observations

#### 2.1.2.1 Regions with Extreme Wildfire Seasons

Anomalies in BA, fire C emissions, and fire counts in the latest global fire season (March 2023-February 2024) were calculated in several ways:

  I. as relative anomalies (expressed in %) from the annual mean during all previous March-February periods since 2001;
  II. as standardised anomalies (standard deviations) from the annual mean during all previous March-February periods since 2001;
  III. as a rank amongst all March-February periods since 2001 (2003 for C emissions), March 2023-February 2024 inclusive, and;

In this report, anomalies in fire C emissions are reported based on the two-model mean estimate from GFED4.1s and GFAS, however anomalies based on the GFED4.1s or GFAS estimates individually are also available via Jones et al. (2024).

We identified regions in which the latest fire season was potentially classifiable as 'extreme' based on the rank of BA, C emissions and fire count amongst all fire seasons. For visualisation purposes, we identified regions in which the latest fire season ranked in the top 5 of all annual fire seasons on record (see **Section 3.1**). The BA data for the period March 2001-February 2024 includes 23 fire seasons, while the C emissions data for the period March 2003-February 2024 includes 21 fire seasons. Hence, a top-5 ranking translates approximately to a fire season in the upper quartile of those on record. While a high-ranking BA or C emissions is indicative of an extreme fire season, we reiterate that our regional experts play a crucial role in defining a broader range of events that could be considered extreme.

We further characterised the onset, peak, and cessation of anomalous monthly BA in March 2023-February 2024. First, we identified the month of the event's peak as the maximum difference between monthly BA values in March 2023-February 2024 and the climatological mean monthly values from the prior March-February periods. Thereafter, the event's onset and cessation were defined as the bounds of consecutive months with above-average BA prior to and following the peak (relative to the monthly climatology).

#### 2.1.2.2 Regions with Extreme Individual Wildfire Attributes

We identified regions in which large or fast-moving fires occurred in the latest fire season based on records of individual fires from the Global Fire Atlas. For each region (**Table 1)** and year, we estimated the size of the largest fire, the daily rate of growth of the fire that spread most rapidly, the size of the 95th percentile fire, and the daily rate of growth of the 95th percentile fire. In the Global Fire Atlas, the daily rate of growth for any given fire is determined by calculating the average daily rate of growth at which the fire advanced across all its constituent cells. This includes cells burned by the head, flank, and backfire. This method





produces lower spread rates than if the calculation were based solely on the cells burned by
the head fire.
Anomalies in each fire attribute were calculated using the same metrics as for BA (see ***i-iii***
above), and we identified regions in which the latest fire season featured fires with potentially
extreme attributes based on the rank of BA and fire C emissions amongst all fire seasons.

### 2.1.3  Updating Regional Trends in Burned Area and Carbon Emissions

To place recent extremes in the context of fire trends of the past two decades, we update our
regional analyses of trends in annual BA from those published by Jones et al. (2022). In
contrast to Jones et al. (2022), we present trends that align more closely with global fire
seasons, spanning the period March 2001-February 2024 rather than trends over calendar
years. We quantified trends using the Theil-Sen slope estimator. The Theil-Sen approach is
useful when data may contain outliers or be non-normally distributed, as it calculates the
median of all possible pairwise slopes between data points. This makes it less sensitive to
outliers than a standard least squares regression slope. Changes were then assessed by
multiplying by the trends (unit year$^{-1}$) by the number of fire seasons in the period March 2001-
February 2024 (23 fire seasons). Relative changes were calculated as the absolute changes
divided by the mean annual BA during the period, which is a more conservative approach than
adopting the BA total in the first fire season as the denominator. The significance of the slopes
estimated by the Theil-Sen method was evaluated using the Mann-Kendall test, with a
confidence level set at 95%. The Mann-Kendall test assesses whether there is a statistically
significant trend in the series of data. This test does not require the data to follow any specific
distribution and is robust against outliers.
In addition to reporting trends in *total* BA, we also present trends in *forest* BA as these regularly
diverge from total BA trends (see **Section 3.1**). Forest BA is calculated as described in Section
**2.1.1.2** but after isolating burned cells in areas with tree cover exceeding 30% in NASA's
annual MODIS MOD44B collection 6.0 Continuous Vegetation Field product (250 m) (DiMiceli
et al., 2015). The 30% threshold is widely used amongst studies of forest cover change (e.g.
Li et al., 2017; Cunningham et al., 2020; Sexton et al., 2016).

### 2.1.4  Expert Consultation

We assembled a panel of regional experts (two from each continent, **Table 2**), to contribute to
the identification, description, and characterisation of extreme wildfire seasons or impactful
events in the latest fire season. A key role of the expert panel was to catalogue regional events
that significantly impacted society or the environment but which may not have been detected
by Earth-observing satellites due to issues such as scale, short duration, timing of overpass,
and cloud or canopy cover. This includes (but is not limited to) wildfires that impacted society
by causing fatalities, evacuations, displacement (e.g. homelessness), direct structure or
infrastructure loss or damage, degradation of air or water quality, loss of livelihood, cultural
practice or other ways of life, and loss of economic productivity. This definition also includes
(but is not limited to) wildfires that impact the environment via disturbance to vulnerable
ecosystems, biodiverse areas, or ecosystem services such as C storage. This approach
recognises that Earth Observations do not provide a complete record of all impactful fires. We
do not define ubiquitous quantitative thresholds of impact by any of the measures outlined
above, but rather invite in-region experts to identify events that triggered impacts that were
sufficient in magnitude to infiltrate public and political discourse. The sources of information
available for cataloguing regional events include national/regional fire records, fire service
reports, disaster management reports, news reports, and social media.





A second key role of our expert panel was to describe and contextualise the impacts of the
fire seasons highlighted as extreme by Earth Observations or regional assessment (see
**Section 3.1**). This includes the discussion of anomalies that rank highly in Earth Observations
but nonetheless did not have major impacts in regions. Metrics such as percentage difference
from the mean and standard deviations from the mean were also used by the expert panel to
further characterise the magnitude of anomalies in regional BA, fire C emissions, fire counts
and individual fire properties.
**Table 2:** Experts contributing to the identification of extreme events and characterisation of
the global fire season during March 2023-February 2024.

| Region | Expert | Country of Organisation / Nationality | Professional Background(s) | Others Consulted |
|---|---|---|---|---|
| Africa | Natasha Ribeiro | Mozambique | Research | Robert Ang'ila, Karatina University; Kebonyethata Dintwe, Botswana University; John Mendelsohn, Okavango Research Institute; Ronald Heath, Forestry South Africa; Helen De Klerk, Stellenbosch University |
| | Sally Archibald | South Africa | Research | |
| Asia | Bambang Saharjo | Indonesia | Research, Litigation | |
| | Veerachai Tanpipat | Thailand | Research, Fire Control and Management Instructor and Consultant | |
| Europe | Paulo Fernandes | Portugal | Research | Davide Ascoli, University of Turin, IT; Hellenic Agricultural Organization "DIMITRA"; Institute of Mediterranean Forest Ecosystems; Niall MacLennan, Scottish Fire and Rescue Service |
| | Stefan Doerr | UK / Germany | Research | |
| | Gavriil Xanthopoulos | Greece | Research | |
| North America | Crystal Kolden | USA | Research, Firefighting | |
| | Jacquelyn Shuman | USA | Research | |
| | Piyush Jain | Canada | Research | |
| Oceania | Hamish Clarke | Australia | Research, Environmental Management | Grant Pearce, Fire and Emergency NZ; Simeon Telfer, SA Country Fire Service; Agnes Kristina, Department of Fire and Emergency Services; Russell Stephens Peacock, QLD Fire and Emergency Services; Chris Collins, Tasmania Fire Service; David Field, NSW Rural |
| | Sarah Harris | Australia | Research, Emergency Management | |



| Region | Expert | Country of Organisation / Nationality | Professional Background(s) | Others Consulted |
|---|---|---|---|---|
| Africa | Natasha Ribeiro | Mozambique | Research | Robert Ang'ila, Karatina University; Kebonyethata Dintwe, Botswana University; John Mendelsohn, Okavango Research Institute; Ronald Heath, Forestry South Africa; Helen De Klerk, Stellenbosch University |
| | Sally Archibald | South Africa | Research | |
| | | | | Fire Service. |
| South America | Dolors Armenteras | Colombia | Research | The Chico Mendes Institute for Biodiversity Conservation (ICMBio), Brazil, Santarém Office |
| | Liana Anderson | Brazil | Research, Disaster risk reduction strategies | |

### 2.1.5   Bespoke Analysis: Air Quality Impacts in Canada

Media and grey literature reports highlighted that wildfires in Canada during 2023 led to extreme degradation of air quality across North America (see **Section 3.1.3.4**). We evaluated air quality impacts of Canadian wildfire emissions in 2023 in the context of the previous 20 years using PM2.5 surface concentrations from the CAMS reanalysis of global atmospheric composition (Inness et al., 2019). The CAMS reanalysis provides a 3-dimensional time-consistent dataset of atmospheric composition including aerosols, chemical species and greenhouse gases for the period 2003-2023). Estimated wildfire emissions for the reanalysis are provided by the GFAS v1.2 (Kaiser et al., 2012), based on fire radiative power observations from the Terra and Aqua MODIS instruments, in addition to inventories of anthropogenic and biogenic emissions, and natural emissions (such as desert dust). While the reanalysis PM2.5 includes contributions from all emissions sources, the most extreme values are strongly correlated with the estimated fire emissions. The extreme annual PM2.5 surface concentrations are calculated as the 95th, 97th and 99th percentiles of the daily maximum concentrations for each year in the reanalysis, and presented as the area-weighted average for Canada.

### 2.2   Shortlisting of Focal Events

In later sections of this report, we apply various models to understand the causes and predictability of a selection of extreme wildfire seasons or events during March 2023-February 2024, to attribute the roles of climate change and land use change, and project the future likelihood of events of a similar magnitude under various climate change scenarios (see Methods below). We limited the number of analyses to three globally prominent focal events of the 2023-24 global fire season in this report because our models are not currently operational products and must be trained and optimised regionally, which is time-consuming.

Our models are structured and parameterised to predict anomalies in regional BA or fire count, meaning that they can only be applied to events that feature in Earth Observations. In addition, the models have not yet been applied to study fire C emissions or wildfires with extreme attributes, and so we restrict their application to regions with anomalous BA in the latest fire season. In discussion with our expert panel, we prioritised the three events studied in this



report by weighing up the anomalies in Earth Observations during the latest fire season as
well as the impacts that these extremes had on people and the environment. The focal events
are notable for their international significance, attracting attention from the media and
policymakers both within and beyond their region.
**2.3 Diagnosing Drivers and Predictability**
**2.3.1 Assessing Predictability of Extremes across Time Scales**
We evaluated the time frame over which extreme events could have been forecasted using
existing forecasting products that produce the Canadian Fire Weather Index (FWI) as outputs
(van Wagner, 1987). Developed by the Canadian Forest Service as part of the Canadian
Forest Fire Danger Rating System (van Wagner, 1987), the FWI comprises various
components that consider the influence of dead fuel moisture and wind on fire ignition and
spread behaviour, with 2m temperature, 10m wind speed, precipitation, and 2m relative
humidity as prerequisite variables. The FWI specifically combines three fuel moisture codes
representing vegetation moisture state at different layers in the forest floor, as well as a spread
index influenced by fuel moisture state and wind speed (van Wagner, 1987). A higher FWI
indicates fire weather conditions more conducive to wildfires in environments with sufficient
fuel load. Owing to its original design for use in forest ecosystems, the FWI is especially useful
for predicting the likelihood and severity of extreme events in ecosystems where weather is
the primary limitation to fire (i.e. those mainly limited by moisture or temperature); its accuracy
in forecasting BA in fuel-limited ecosystems is more limited (Carvalho et al., 2008; Bedia et
al., 2015; Abatzoglou et al., 2018; Jones et al., 2022). Its applications encompass early
warning systems, pre-suppression and suppression planning, prescribed burn planning and
effectively alerting authorities and the public to abnormal fire danger conditions. FWI is
extensively used in operational global information platforms such as the European Forest Fire
Information System (EFFIS; https://forest-fire.emergency.copernicus.eu/, last access: 2 June
2024) and the Global Wildfire Information System (GWIS; https://gwis.jrc.ec.europa.eu/, last
access: 2 June 2024), and the Canadian Wildland Fire Information System (CWFIS;
http://cwfis.cfs.nrcan.gc.ca/, last access: 2 June 2024).
In addition to well established fire danger forecasts with lead times of a few days, skilful
predictions of fire danger can be made on sub-seasonal to seasonal time scale (S2S) for
Mediterranean Europe (Bedia et al., 2015), United States (Roads et al., 2010) and Asia
(Spessa et al., 2015). Drought and fire weather conditions throughout the world have been
found to correlate with large scale climate patterns such as the El Niño Southern Oscillation
(ENSO) (Field et al., 2016; Chen et al., 2017), the Indian Ocean Dipole (Cai et al., 2009) and
other climate modes such as the Atlantic Multidecadal Oscillation and the Pacific Decadal
Oscillation implying predictability of fire-favourable weather conditions for various seasons and
regions (Aragão et al., 2018; Turco et al., 2018). Following the concept of seamless (Wetterhall
et al., 2018) prediction of fire weather on S2S timescales after Di Giuseppe et al. (2020), Di
Giuseppe et al. (2024) and Dowdy (2020), we collated FWI data from reanalyses and forecasts
designed to operate on S2S lead times of 10 days to 7 months. Specifically, the FWI is derived
from the operational monitoring provided by ERA5-land (Muñoz-Sabater et al., 2021), the
forecasts from the operational high-resolution ECMWF weather system, and the seasonal
predictions from the ECMWF's long-range forecasting system, SEAS5 (Johnson et al., 2019;
Di Giuseppe et al., 2020; Di Giuseppe et al., 2024). The predictions are compared to recorded
peaks in fire activity, both in terms of burned areas and active fires as observed by the MODIS
satellites. These prediction systems vary in spatial and temporal resolutions. Short to medium-
range FWI forecasts (up to 10 days) are available daily at a resolution of 9 km, while the FWI
seasonal forecast is available monthly at a resolution of approximately 25 km, however
seasonal skill is limited to 2-3 months in normal conditions (Di Giuseppe et al., 2024).





Predictability is assessed qualitatively by visually examining how extreme FWI was as a
precursor of the event, replicating the use of this indicator by fire agencies during the fire
season. Most fire agencies would have local information on fuel conditions and would be able
to interpret FWI values in an educated manner, thus reducing intervention on high FWI values
in fuel conditions that are not hazardous. This fuel consideration would not be included in a
validation of the FWI using traditional skill scores, as these would be dominated by false
alarms. We retain that FWI is an index representing flammability and cannot fairly be validated
against fire activities.
**2.3.2 Identifying Key Drivers of Focal Events**
*2.3.2.1 Overview*
We used two modelling systems with similar fire predictors to diagnose the direct drivers of
each focal event that was shortlisted: the Probability of Fire (PoF) model (McNorton et al.,
2024) and the Controls on Fire (ConFire) attribution framework (Kelley et al., 2019; Kelley et
al., 2021). The PoF model diagnoses the drivers of active fire (AF) observations from the
MODIS MCD14ML active fire product (collection 6.1; 1 km resolution; Giglio et al., 2016; NASA
FIRMS, 2020) using Shapley values (Lundberg and Lee, 2017), while ConFire diagnoses
drivers of BA from the MODIS MCD64A1 BA product (collection 6.1; 500 m resolution; Giglio
et al., 2018) regridded to 0.5°. Fires flagged as low confidence in the AF product were not
used. Although AF and BA have been used widely in global and regional scientific studies,
there are substantial differences between the two products (Roy et al., 2008; Di Giuseppe et
al., 2021; Chuvieco et al., 2019) and the strength of the relationship between them can vary
regionally (van der Werf et al., 2017; Hantson et al., 2013). Our use of two observational fire
products and two distinct model approaches provides a way to account for inherent
uncertainties in the definition of the fire events and the uncertainties in the methodologies.
Both methods use a number of predictors (we refer to the single variables as the drivers) of
AF or BA. The drivers are grouped into four main categories (we refer to grouped drivers as
the controls): weather; fuel abundance; fuel moisture, and; other drivers for PoF and human
drivers for ConFire (see **Table 3**). PoF drivers in "Other" include ignition and suppression
efforts as well as the residual error between forecast and observed fire activity. Grouping the
set of drivers between the four identified controls—weather, fuel moisture, fuel load, and
ignitions—is not always straightforward, as fuel moisture and weather variables are strongly
correlated, and fuel load is also related to weather conditions. Hence, some drivers can be
associated with more than one control. The categorisation stems mostly from the way the
driving datasets have been obtained and their underlying resolutions. We have also
considered the traditional approach of assessing fire weather in isolation within most fire
danger assessment metrics. We believe that grouping these metrics under the umbrella term
'control weather' offers a concise way to reference the drivers of the Fire Weather Index
(Matthews, 2009). Despite this, it is important to note that the techniques employed ensure
contribution from specific variables cannot be double counted between categories. Both
ConFire and PoF are also capable of quantifying the contribution of individual drivers (Kelley
et al., 2019; McNorton et al., 2024). However, we will focus our analysis on the impact of the
controls.
The selected weather parameters are consistent with each other and are available at a base
resolution of 9 km. Land cover maps are available at a higher resolution of 1 km. Additionally,
fuel load is constrained by external (to the weather model) observations from the ESA CCI
above ground biomass product and satellite inversion estimates of net ecosystem exchange
of C, while the fuel moisture is parameterized using land surface parameters constrained by
the weather variables.



### 2.3.2.2 Drivers and Controls Used in Fire Event Analysis

For our assessment of the contribution of weather and fuel moisture to the anomalous events we take several predictors from ERA5-Land (9 km resolution; Muñoz-Sabater et al., 2021), specifically variables that are known to correlate with AF or BA (Bistinas et al., 2014; Haas et al., 2022; Haas et al., 2022). The drivers considered for each control are listed in **Table 3**. For the weather component in isolation, we use 2m temperature, 2m dewpoint temperature, 10m wind speed and daily total precipitation (note that these are the prerequisite variables used in the formulation of the FWI; van Wagner, 1987). We use a fuel characteristic model to estimate the fuel load and fuel moisture components following McNorton and Di Giuseppe (2024), with model estimates of fuel moisture constrained by estimates of leaf area index (LAI) from the ECMWF's Integrated Forecast System (IFS) and model estimates of fuel loads constrained by aboveground biomass estimates from the ESA Climate Change Initiative (CCI; Santoro and Cartus, 2021) and net ecosystem exchange estimates from the Copernicus Atmosphere Monitoring Service (CAMS; Agustí-Panareda et al., 2019). Additional predictors regarding fuel load and state include vegetation cover and type (**Table 3**). Proxies for ignition and suppression controls, placed within the "Other" set of controls, are more challenging to establish. Currently, we use population density, urban fraction, cropland fraction, pasture fraction, lightning, orography (**Table 3**). For consistency all variables are interpolated to 9 km resolution. See **"Modelling frameworks"** in the **Extended Methods Supplement** for a detailed description. PoF accurately captures the spatiotemporal patterns of fire activity in Canada and Western Amazonia and effectively identifies areas of high fire danger in Alexandroupolis, Greece. However, it underestimates the severity of fire events in that area. See **"Evaluation" in the Extended Methods Supplement** for detailed evaluation.



**Table 3:** Drivers of fire and their parent control group included in the event fire analyses using
ConFire and PoF. Drivers are individual explanatory variables, which are grouped into
weather, fuel load, fuel moisture, and other controls. ** The 'Other' category includes proxies
for ignition and suppression controls plus the missed prediction. +ive or -ive under "ConFire
controls" describes if a driver increases or decreases BA in ConFire.

| Driver | POF control | ConFire controls | Frequency | Time Period | Source | Reference |
|---|---|---|---|---|---|---|
| 2m Temperature | Weather | Weather +ive | Daily | Jan 2014-NRT | ERA5-Land | Muñoz-Sabater et al. 2021 |
| 2m Dewpoint Temperature | Weather | Weather -ive | Daily | Jan 2014-NRT | ERA5-Land | Muñoz-Sabater et al. 2021 |
| 10m Wind Speed+ | Weather | Not used | Daily | Jan 2014-NRT | ERA5-Land | Muñoz-Sabater et al. 2021 |
| Precipitation | Weather | Weather -ive | Daily | Jan 2014-NRT | ERA5-Land | Muñoz-Sabater et al. 2021 |
| Live Leaf Fuel Load | Fuel Load | Not used | Daily | 2014- 2020 | Fuel Model | McNorton et al. 2024a |
| Live Wood Fuel Load | Fuel Load | Not used | Daily | 2014- 2020 | Fuel Model | McNorton et al. 2024a |
| Dead Foliage Fuel Load | Fuel Load | Not used | Daily | 2014- 2020 | Fuel Model | McNorton et al. 2024a |
| Dead Wood Fuel Load | Fuel Load | Not used | Daily | 2014- 2020 | Fuel Model | McNorton et al. 2024a |
| Mean & Max Vegetation Optical Depth (VOD) of the last 12 months | Not used | Fuel Load +ive | Monthly | 2014-NRT | Satellite (SMOS) | Wigneron et al 2021 |
| Vegetation Optical Depth (VOD) | Moisture | Moisture -ive | Monthly | 2014-NRT | Satellite (SMOS) | Wigneron et al 2021 |
| Low Vegetation (LAI) | Fuel Load/Moisture | Not used | Monthly | 2002-2020 climatology | Satellite (multi-sensor) | Boussetta et al., 2021 |
| High Vegetation (LAI) | Fuel Load/Moisture | Not used | Monthly | 2002-2020 climatology | Satellite (multi-sensor) | Boussetta et al., 2021 |
| Live Fuel Moisture Content | Fuel Moisture | Fuel Moisture -ive | Daily | 2014-NRT | Fuel Model | McNorton et al. 2024a |
| Dead Fuel Moisture Content | Fuel Moisture | Fuel Moisture -ive | Daily | 2014-NRT | Fuel Model | McNorton et al. 2024a |
| Snow cover | Fuel Moisture | Snow -ive | Daily | 2014-NRT | ERA5-Land | Muñoz-Sabater et al. 2021 |
| Pasture | Not used | Ignitions +ive Suppression -ive | Annual | 2014-2023 | HYDE | Klein Goldewijk et al., 2011 |
| Cropland | Not used | Ignitions +ive Suppression -ive | Annual | 2014-2023 | HYDE | Klein Goldewijk et al., 2011 |
| Urban population | Not used | Ignitions +ive Suppression -ive | Annual | 2014-2023 | HYDE | Klein Goldewijk et al., 2011 |
| Rural populations | Not used | Ignitions +ive Suppression -ive | Annual | 2014-2023 | HYDE | Klein Goldewijk et al., 2011 |
| Lightning | Not used | Ignitions +ive | Monthly | 2000- 2020 climatology | LIS/OTD | Cecil et al., 2014 |
| Type of Vegetation | Other** | Not used | Fixed | Jan 2014-NRT | ECLand | Boussetta et al., 2021 |
| Urban Fraction | Other** | Not used | Fixed | Jan 2014-NRT | ECLand | McNorton et al. 2023 |
| Orography | Other** | Not used | Fixed | Jan 2014-NRT | ECLand | Boussetta et al., 2021 |






***2.3.2.3  Analysis of Fire Drivers***

The PoF system uses gradient-boosted decision trees from the XGBoost library on detected AF (McNorton et al., 2024). The training iteratively adds decision trees to an ensemble of models to correct for errors made by previous iterations, resulting in a computationally efficient optimization (Chen and Guestrin, 2016). The system training uses a classifier approach which defines a positive hit as an AF detection within the grid cell on a given day. The driver attribution is performed using the SHapley Additive exPlanations (SHAP) method taken from the SHAP library (Lundberg and Lee, 2017). These are then combined to provide overall attribution to one of the four controls for AF predictions.

ConFire is an evidence-based uncertainty attribution framework that uses Bayesian Inference to evaluate the likelihood of different control strengths and their influence on observed BA. Rather than produce a single number output at a given location/time, ConFire produces a probability distribution based on relationships found between driving data input into a simple semi-empirical process-representation fire model, and BA observations. ConFire's hierarchical design permits each variable representing a fire driver to contribute to multiple controls (weather, fuel moisture, fuel abundance, or others) simultaneously. ConFire operates on a monthly time step and all drivers are aggregated to monthly daily means using the FLAME system (Barbosa, 2024). Each control operates as an optimised weighted linear combination of its respective drivers. The influence of each control is then represented using a logistic function, with BA calculated as the product of the four controls. Bayesian inference is used to optimise the weighted contribution of drivers to each control, as well as the manner in which each control affects BA. ConFire also includes a stochasticity term that represents the inherent unpredictability in fire occurrence and resultant BA which can occur under similar bioclimatic conditions (Kelley et al., 2021). We use the mean logit transformed BA distance between distributions with (equation 8 in supplement) and without (equation 7 in supplement) this stochasticity term provides a measure of uncertainty that has not been captured within the drivers considered. See **"Modelling frameworks" in the Extended Methods Supplement** for a detailed description.

The model is trained and ran between 2014-2023, the common period of all driving datasets. The model is trained on 50% of BA before being run in a predictive model for the full dataset. We also perform a separate run, training on all data from 2014-2018 and evaluating against 2019-2023 following the protocol outlined in Barbosa (2024). This evaluation protocol aims to test the ConFire model's ability to accurately represent the range of uncertainties and generate observed distributions through its probabilistic framework. This tells us that the framework can capture and reflect inherent uncertainties in fire processes during optimization. The model consistently captures observations within its uncertainty range across all regions, strongly aligning with real-world data and effectively representing BA anomalies and the stochastic nature of fire in our regions. However, it consistently exhibits a low bias, often underestimating BA, particularly in high-burn regions like deforestation areas in Western Amazonia and northern Canada. It can also capture the anomalies in BA, particularly in 2023, though it sometimes underestimates the anomaly's magnitude when not using stochasticity. Therefore the size of BA anomalies may be underestimated. See **"Evaluation"** in the **Extended Methods Supplement** for detailed evaluation**.**

## 2.4  Attribution to Global Change

### 2.4.1  Role of Global Change Factors

In addition to direct drivers of fire events as outlined in the previous section (e.g. weather, fuel, moisture, ignition and suppression), fires are also influenced by global change factors such as climate and land-use change which vary over longer timescales. Since the pre-industrial era,





global mean temperature has increased by nearly 1.3°C (Betts et al., 2023; Forster et al.,
2023), with regional temperatures even higher in some areas, providing additional potential
for fuel drying and facilitating fire ignition and propagation. Climate change has also resulted
in altered precipitation patterns, with total rainfall and dry season length increasing or
decreasing variably across regions (Polade et al., 2014; Swain et al., 2018; IPCC, 2023a).
Meanwhile, changes to fuel load and ignition rates are driven by climate change and
anthropogenic land-use, with varying effects regionally (Finney et al., 2018; Romps, 2019).
For example, in fuel-limited savannah biomes land-use change can drive more fragmented
fuel loads and a reduction in fire (or an increase in fire resulting from land abandonment),
whereas in forest ecosystems fragmentation provides more potential for ignition and leads to
increases in fire occurrence (Andela et al., 2017; Rosan et al., 2022).

### 2.4.2 Overview of Attribution Approaches

In this report, we apply different modelling techniques with consistent meteorological and fire
observations to attribute (i) regional changes in the probability of high fire weather to
anthropogenic forcing (**Section 2.4.3**) and (ii) changes in monthly BA to total climate forcing,
socio-economic change, and all forcing (**Section 2.4.4**), specifically targeting our focal regions
in both cases. Our analyses require separate approaches to attributing change to each of
these global change factors. Our attribution to *anthropogenic forcing* explicitly targets the
changes driven by anthropogenic emissions from greenhouse gases and land-use change,
following the IPCC WGI definition (Hegerl et al., 2009; Mengel et al., 2021). We prescribe
these emissions in a model to specifically isolate human forcing from natural variability
(**Section 2.4.3**). Our attribution to *total climate forcing* considers changes driven by climate
change since the pre-industrial period, including both anthropogenic forcing and natural
variability in line with the IPCC WGII definition of climate change impact attribution (IPCC,
2023b; IPCC 2023c). This involves comparing simulations driven with historical reanalysis to
a detrended counterfactual simulation with the historical warming signal removed (with both
simulations including historical transient land-use change) and therefore only the impacts of
climate change are attributed, not distinguishing between anthropogenic or natural causes
(Mengel et al., 2021; Burton, Lampe et al., 2023). Our attribution to *socio-economic factors* is
applied via the same set of simulations as our attribution to *total climate forcing*. The role of
socio-economic factors is isolated by comparing the early industrial period to the late industrial
period in the counterfactual scenario, in which only land-use and population density are
allowed to change (Burton, Lampe et al., 2023). Finally, *all forcing* compares the early
industrial period in the counterfactual scenario to the last industrial period in the factual
scenario, which gives the net effect of all forcings combined. These are summarised in the
**Table 4** below.

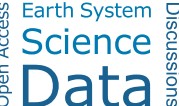

**Table 4**: Summary of the attribution approaches used in this report.

| Term | Definition | Experiments compared | Framework | Application |
|---|---|---|---|---|
| **Event attribution for fire weather** | | | | |
| **Anthropogenic Forcing** | Change in fire weather driven by anthropogenic emissions from greenhouse gases, land-use change and aerosols. As per (Ciavarella et al., 2018; Li et al., 2021) | **ALL**: natural forcing plus human emissions<br>**NAT**: Natural-only forcing from solar variability and volcanoes | HadGEM3-A attribution ensemble. 0.5 degree resolution | Fire weather (FWI) |
| **Impacts attribution for burned area** | | | | |
| **Total climate forcing** | Changes in BA due to climate change, irrespective of the cause of warming. As per (Mengel et al., 2021 and Frieler et al., 2024) | **Factual (2003-2019)**: present-day climate (driven by GSWP3-W5E5 reanalysis), CO2, land-use and population<br>**Counterfactual (2003-2019)**: De-trended historical climate (warming signal removed), CO2 fixed at 1901 value, present-day land-use and population | ISIMIP3a impact attribution. 0.5 degree resolution | FireMIP ensemble and ConFire |
| **Socio-economic factors** | Changes in BA due to land-use and population change. As per (Burton, Lampe et al., 2023) | **Counterfactual (1901-1917)**: Warming trend removed, fixed 1901 CO2, limited land use and population change<br>**Counterfactual (2003-2019)**: Warming trend removed, fixed 1901 CO2, present-day land use and population | ISIMIP3a impact attribution | FireMIP ensemble and ConFire |
| **All forcing** | Changes in BA due to climate, land-use and population change. As per (Burton, Lampe et al., 2023) | **Counterfactual (1901-1917)**: Warming trend removed, fixed 1901 CO2, limited land use and population change<br>**Factual (2003-2019)**: Historical climate driven by reanalysis | ISIMIP3a impact attribution | FireMIP ensemble |


The tools described here enable us to assess the influence of climate and socio-economic
forcing on fire from 3 different perspectives. We use FWI to assess how the probability of
experiencing meteorological conditions conducive to high (>90th percentile) fire danger has
changed as a result of anthropogenic forcing. As climate change has a direct impact on fire
weather, this approach enables us to isolate its effects without confounding factors of land-
use change and ignitions, and tells us how a fire might develop once ignited. We use ConFIRE
to model high (>90th percentile) BA and its uncertainty as a result of climate change and socio-
economic factors, which other fire models are unable to capture well. We use an ensemble of
7 fire models from FireMIP to put these specific extreme events into the context of changes in
global median BA resulting from climate change and socio-economics in the recent present-
day period. The fire model ensemble takes into account biogeochemical feedbacks between
fire, climate, land-use and vegetation as a result of altered forcing across the 7 Dynamic Global
Vegetation Models (DGVMs). Each of these methods is described in more detail below.
**2.4.3  Attributing Change in Likelihood of Fire Weather to Anthropogenic Forcing**
We use an established approach to attribute change in probability of high (>90th percentile)
fire weather conditions to anthropogenic forcing. The approach uses estimates of FWI, as
used in previous studies from the World Weather Attribution (Barnes et al., 2023), using
outputs from the HadGEM3-A large ensemble (Christidis et al., 2013; Ciavarella et al., 2018).
It follows the approach introduced by Stott et al. (2004) for attributing extreme weather events,



and it has been employed in other attribution studies targeting fire weather, such as Li et al.
(2021).
As outlined in **Section 2.3.1**, the FWI is used operationally and in research contexts to rate
fire danger based on meteorological conditions. Due to the availability of model output
variables we use maximum daily temperature at 1.5 m as a proxy for noon values, total daily
precipitation, mean daily relative humidity at 1.5 m, and mean daily wind speed at 10 m,
following Perry et al. (2022). We calculate the daily FWI for the month of 2023-24 peak BA
anomaly for each focus region, using the same month and region for validation over the
historical timeseries (1960-2013) (**Supplementary Figures S41-S43**).
We validate and bias-correct the model estimates of high FWI for the period 1960-2013 by
comparing a 15-member HadGEM3-A ensemble with ERA5 reanalysis data (C3S, 2024)
representing "observed" FWI. The 0.25 degree resolution observed FWI from ERA5 was
coarsened by linear interpolation (calculated by extending the gradient of the closest two
points) to match the 0.5 degree model grid. We compare the timeseries of individual
components of the FWI **(Figure S40)**, and the distribution of the modelled and observed FWI
(**Supplementary Figures S41-S43**), and apply a simple linear regression to find the bias
correction required for the 2023 model output. Before bias-correction, the modelled FWI is
generally higher than the observed FWI, and some regions (e.g. Greece) require a larger
correction than others. The correction adjusts the trend and absolute value while maintaining
variability, and the model successfully reproduces the observed distribution after applying the
correction in each region (see **Extended Methods and Evaluation Supplement**).
For the events occurring in the 2023-24 fire season, we calculate the FWI from the HadGEM3-
A model simulations comprising 2 experiments of 525 members each, one driven by all
forcings including historical greenhouse gas emissions, aerosols, zonal-mean ozone
concentrations, land-use change and natural forcing (ALL), and a second counterfactual
simulation with natural-only forcing from solar variability and volcanic emissions, and 1850
land-use (NAT). By applying the bias correction from the previous step, and comparing the
fire weather in the two simulations to the 2023 observed FWI from ERA5, we calculate the
change in probability of high (>90th percentile) fire weather due to anthropogenic forcing.
**2.4.4  Attributing Change in Regional BA to Total Climate Forcing, Socio-economic**
**Factors and All Forcing**
*2.4.4.1  Monthly High Burned Area*
We use the ConFire attribution framework to attribute high (>95th percentile for Canada and
Western Amazonia and >90th percentile for Greece) BA in the month of peak anomaly to total
climate forcing and socio-economic factors using the ISIMIP3a protocol (see **Table 5**). We
used 90th percentile for Greece as using the 95th would restrict our analysis to just three grid
cells, and our uncertainty ranges are too large to make a clear statement on such a small
subset of data.
We trained ConFire on observed monthly BA from MODIS MCD64A1 during 2003-2019 at
0.5° across the entire region. For model training, we drive ConFire with GSWP3-W5E5
forcings provided at a 0.5° spatial resolution by ISIMIP3a, as outlined in **Table 5**. The land
surface information (tree cover and non-tree vegetated cover) is derived from the JULES-ES
model's ISIMIP configuration described by Mathison et al. (2023) driven by ISIMIP3a GSWP3-
W5E5 forcings. This model includes dynamic vegetation, i.e changing vegetation cover in
response to climate variables, growth, plant competition and mortality. So as not to double
count the impact of fire, we turn the interactive vegetation-fire model off. The bias in this land
surface information is adjusted to the MODIS Vegetation Continuous Fields collection 6.1





remote sensed data for <60°N DiMiceli et al. (2022) and collection 6 for <60°N DiMiceli et al.
(2015) using a trend-preserving empirical quantile mapping bias adjustment method
implemented with the ibicus software package Spuler et al. (2024). The bias adjustment
method calibrates a mapping between the empirical cumulative distribution function of each
surface cover type at each grid cell derived from the JULES-ES model output and the
corresponding quantiles in the MODIS remote sensed data at this grid cell over the reference
period (2003-2019). This method significantly reduces the model bias in the JULES-ES output
over the historical period for most regions and variables, ensuring accurate mean and
distribution shapes while preserving trends between historical and future periods (**Figure
S22**). See **"Data and Data Processing" in Extended Methods Supplement** for details.
ConFire was run in predictive mode on a monthly timestep and with a similar structure to that
used in **Section 2.3.2.3**, with variables representing specific drivers grouped into controls as
listed in **Table 5**. However, the specific driving variables used differed to **Section 2.3.2.3** due
to the selection of variables available in the forcing data. Specifically, fuel load controls were
represented by total vegetation cover and tree cover as drivers; fuel moisture controls were
represented by mean consecutive dry days within each month, the fraction of dry days within
the month, daily mean precipitation, mean and maximum monthly temperature, mean and
maximum VPD as drivers; ignition controls were represented by climatological lightning,
pasture, crop and population density as drivers; and suppression controls were represented
by pasture, crop and population density drivers. The Bayesian Inference procedure used here
is useful when using correlated drivers, which are accounted for in the weights given to driver
contributions (Gelman et al., 2013; Kelley et al., 2023). ConFire simulations presented here
were performed on all data, with the three experiments outlined under "simulation framework".
For each of the experiments detailed above, we output the probability distribution BA for the
cells with the top 5% burned fractions for Canada and Western Amazonia, and 10% burned
fractions for Greece, to target the most extreme levels of burning.
To determine the impact of total climate forcing and socioeconomic factors on the increased
BA during our focal events, we conducted a paired sampling of monthly burned areas in our
target month(s) throughout the test period. This involved comparing the factual and
counterfactual scenarios for total climate forcing, as well as the counterfactual and early
industrial scenarios for socioeconomic factors (see **Table 5**). As there is no climate influence
in the Early Industrial simulation, we first adjusted the target event (a monthly regional BA
value) to that expected without climate change. For this adjustment, we find the percentile of
the observed BA in the factual and find the BA at the same percentile in the counterfactual.
We used paired samples to account for the uncertainty in the underlying mechanisms
relating our drivers to burned area, which would co-vary between experiments as per Kelley
et al. (2021). In total, we took 200 samples over the 17 years of each simulation, resulting in
3400 pairs.
The likelihood was then simply determined by the number of ensemble members in the
factual scenario that predicted greater BA than the counterfactual for total climate forcing, or
the counterfactual predicting greater BA than the early industrial scenario for socioeconomic
factors. The relative increase in BA extent is the BA in factual over counterfactual (total
climate forcing) or counterfactual over early industrial (socioeconomic).
As per, **Section 2.3.2.3** we evaluated the model using the procedure outline in Barbosa
(2024). We do a separate training on 50% of the data between 2003-2011 and evaluation on
2012-2019. The framework using ISIMIP3a reanalysis data outperforms its near-real-time
counterpart in **Section 2.3.2.3** at simulating burned areas, effectively representing high
burned areas and extremes across all regions. It shows a high probability of observations in





areas with extreme fires, indicating reliability for attribution analysis. Further details of the
method can be found in supplement **Extended Methods and Evaluation Supplement**.
**Table 5:** Explanatory variables used for attributing extreme BA (**Section 2.4.4**) and decadal
outlook (**Section 2.5.2**). The explanatory variables are forcing data from the Inter-Sectoral
Impact Model Intercomparison Project (ISIMIP) protocols ISIMIP3a and ISIMIP3b (Frieler et
al., 2024). +ive or -ive under "Controls" describes if a driver increases or decreases BA.

| Variable | Control s | Construction | Source | Reference |
|---|---|---|---|---|
| Max. consecutive dry days | Moisture +ive | Monthly max of running count of days since rainfall > 0.1mm/m | Based on precipitation from ISIMIP3a/3b | Frieler et al. (2024) |
| No. dry days | Moisture +ive | fractional no. days of rainfall < 0.1mm/m | Based on precipitation from ISIMIP3a/3b | Frieler et al. (2024) |
| Maximum monthly temperature | Moisture +ive | maximum of maximum daily temperature within the month | Daily temperature appriximated os ISIMIP3a/3b daily mean temperature + 0.5xdaily temperature range | Frieler et al. (2024) |
| Mean monthly temperature | Moisture +ive | | ISIMIP3a/3b | Frieler et al. (2024) |
| Mean monthly VPD | Moisture +ive | mean of daily values constructed from specific humidity, surface pressure and max. temperature | ISIMIP3a/3b | Frieler et al. (2024); Barbosa (2024) |
| Maximum monthly VPD | Moisture +ive | max of daily values | ISIMIP3a/3b | Frieler et al. (2024); Barbosa (2024 |
| Tree Cover | Moisture -ive & Fuel +ive | JULES-ISIMIP annual mean tree cover bias-corrected to VCF vs JULES-ISIMIP3a factual interpolated to monthly from annual values | JULES-ES-ISIMIP VCF | DiMiceli et al. (2017); Adzhar et al. (2022); Mathison et al. (2023) |
| Total vegetation cover | Fuel +ive | tree cover plus none-tree vegetated cover simulated by JULES and bias-corrected as above | JULES-ES-ISIMIP VCF | DiMiceli et al. (2017); Adzhar et al. (2022); Mathison et al. (2023) |
| Lightning | Ignitions +ive | Climatology | LIS taken from ISIMIP3a | Kelley et al. (2014); Frieler et al. (2024) |
| Cropland | Ignitions +ive/ Suppression -ive | Interpolated from annual to monthly | ISIMIP3a/3b | Frieler et al. (2024)) |
| Pasture | Ignitions +ive/ Suppression -ive | Interpolated from annual to monthly | ISIMIP3a/3b | Frieler et al. (2024) |
| Population Density | Ignitions +ive/ Suppression -ive | Interpolated from annual to monthly | ISIMIP3a/3b | Frieler et al. (2024) |

### 2.4.4.2 *Monthly Median Burned Area*
Finally, we report changes in present-day monthly median BA to total climate forcing, socio-
economic factors and all forcings using a novel attribution method developed using state-of-
the-art global fire models from the Fire Model Intercomparison Project (FireMIP) (Burton,
Lampe et al., 2023). This employs the same ISIMIP3a simulation framework outlined above,





with 7 fire-enabled DGVMs (see **Table S1**) for the period 1901-2019 for the factual and
counterfactual experiments. For clarity, ConFire was not used in this element of our attribution
approaches; rather, the native fire modelling scheme of each fire-enabled DGVM was
employed. Model fire schemes are described in Burton, Lampe et al. (2023).
We assess the recent period 2003-2019 in the historical simulation against the counterfactual
simulation to find how BA has changed in response to total climate forcing. To attribute
changes in BA to socioeconomic factors, the early industrial period (1901-1917) is compared
to the recent period (2003-2019) in the counterfactual only simulation. To assess the impact
of all forcings the early industrial period in the counterfactual is compared to the recent period
in the factual simulation.
A weighted ensemble of the monthly outputs of BA, based on regional performance against
observational data from GFED5 and FireCCI is used for the analysis. Due to large differences
in absolute values of BA between the GFED5 and FireCCI observational datasets and across
the models, the weightings in the ensemble are based on model capability to capture relative
anomalies present in the observational datasets on a regional basis, and all changes are
reported as relative anomalies. We focus on the change in monthly median BA, as the fire
models underpredict the high tails of the distribution. The weighted models are randomly
resampled to generate uncertainty estimates for each region. The method and results are
reported in full for all 43 IPCC AR6 regions in Burton, Lampe et al. (2023), and in the current
report we select the IPCC regions that align most closely with our focus regions defined in
**Section 3.2**.
**2.5   Seasonal and Decadal Outlook**
**2.5.1   Seasonal Forecasts**
Among the modes of variability in the climate system most relevant to wildfire activity *globally*
is the El Niño-Southern Oscillation (ENSO) (Chen et al., 2017; Mariani et al., 2016; Fuller and
Murphy, 2006; Cardil et al., 2023). ENSO is a complex, naturally occurring climate
phenomenon characterised by fluctuations in sea surface temperatures and atmospheric
pressure across the Pacific Ocean. ENSO has far-reaching consequences generating dry
conditions in vast parts of South and Central America, the mediterranean and African savanna.
It also exacerbates dry conditions on the west coasts of continents such as California in the
United States, southern Europe, and parts of Australia, which can become more susceptible
to fires. The geographic areas with such fire favourable teleconnections differ by ENSO phase
(Chen et al., 2017).
Another phenomenon demonstrably linked to global fire activity is the Indian Ocean Dipole
(IOD), which occurs in the Indian Ocean and is characterised by differences in sea surface
temperatures (SST) between its western and eastern parts. The IOD has a notable influence
on weather patterns in various regions of the Southern Hemisphere, particularly affecting the
southern and eastern parts of Australia (Harris and Lucas, 2019). Positive IOD events typically
lead to reduced rainfall in these areas, resulting in drier-than-normal conditions. However,
there is still ongoing debate regarding the direct influence of the IOD on Australian fires, as
the signal is often obscured by changes in land management practices (Harris and Lucas,
2019).

ENSO and the IOD have a planetary scale influence on weather anomalies through
teleconnections. Other atmospheric modes of variability in the Southern, Northern hemisphere
and in the arctic regions can also have strong influence on the seasonal trend of regional
burned areas. For example, **Figure S1** shows the climate modes with strongest influence on
burned areas globally.





The large uncertainties in the analysis highlights that there are few regions in the word where
it is possible to establish statistically significant teleconnection between burned areas and
atmospheric modes. Hence, we utilised the Copernicus Climate Change Service (C3S) multi-
model seasonal prediction system to evaluate large-scale climate modes with most proven
links to variation in fire activity: ENSO and IOD. The C3S multimodel incorporates forecasts
from several prominent meteorological agencies, including the ECMWF, UK Met Office,
Météo-France, German Weather Service (DWD), Euro-Mediterranean Center on Climate
Change (CMCC), US National Weather Service's National Centers for Environmental
Prediction (NCEP), Japan Meteorological Agency (JMA), and Environment and Climate
Change Canada (ECCC). This multi-model system aids in defining the expected evolution of
these two modes of variability. These indices, calculated as Sea Surface Temperature (SST)
anomalies over the ocean, offer a succinct means of understanding the expected strengths of
atmospheric modes of variability in the coming months.
To look more closely at the impact that these two modes might have in landscape flammability
we use seasonal outlooks of the Fire Weather Index (FWI) using one of the models from the
aforementioned multi-model system, ECMWF-System 5. Leveraging seasonal predictions for
the FWI we generated probabilities of exceedance using a 51-member forecast ensemble and
a 24-year model climatological distribution (derived from a 25-member ensemble re-forecast)
covering the period 1993-2016. The probability of exceedance is determined based on the
proportion of forecast members meeting two anomaly criteria at any given geographical point.
We consider the 75th percentile indicative of moderate anomalous conditions and the 95th
percentile indicative of extreme anomalous conditions over a month for the next season.
### 2.5.2 Decadal Projections of Burned Area
In order to project future changes in BA, we utilised the same modelling approach detailed in
**Section 2.4.4.1**, following a similar protocol to UNEP et al. (2022). We drive the model with
ISIMIP3a and bias-corrected JULES-ES data. For predictive mode, we used bias-corrected
GCM model outputs from ISIMIP3b. ISIMIP3b provides four sets of driving data for 5 bias-
corrected models, including historical data up to 2014 and future scenarios from 2015-2100
(SSP126, SSP370, and SSP585). Each SSP represents future socio-economic pathways and
includes GHG emissions that feed each of the 5 GCMS: GFDL-ESM4 (Held et al., 2019),
IPSL-CM6A-LR (Boucher et al., 2020), MPI-ESM1-2-HR (Mauritsen et al., 2019), MRI-ESM2-
0 (Yukimoto et al., 2019), and UKESM1-0-LL (Tang et al., 2019; Sellar et al., 2019). As part
of ISIMIP3b, each GCM is bias-corrected as described in Lange (2019).
At present, future projections (aligned with shared socio-economic pathways; SSPs) for land
use and population density forcing were not available for ISIMIP3b, so we only considered the
climate's influence on fire distribution and intensity as well as available fuel from vegetation
cover, and not changes in ignitions or land use. We used JULES-ES land cover outputs as
per the previous section, but with JULES driven by each scenario-GCM combination in turn.
This was then bias-corrected in the same mapping procedure as **Section 2.4.4.1**, based on
biases between JULES-ES driven by ISIMIP3a and VCF observations to keep consistent with
GCM bias-correcting procedures. This mapping was subsequently applied to the surface
information outputs from JULES-ES driven by historical (1994-2014) and future (2015-2099)
model runs. To preserve the trend in the vegetation cover over the future periods, additive
detrending of the mean was applied. The results in **Section 3.5.3** are for the months June-
August for Canada, July-September or Greece and August-October for Western Amazonia,
corresponding to those regions' fire seasons today. See **"Data and Data Processing" in
Extended Methods Supplement** for details.
Our approach provides a probability distribution representing the uncertainty in future
emissions, climate, and vegetation responses for each scenario and year in the period 2010-





2100. For the first 4 years (2010-2014), we joined the historical experiment to each SSP in
turn.
For Western Amazonia, we additionally tested a 1-in-100 event under 2010-2020 climate,
defined as the BA at the 99th percentile ConFires distribution. We then calculated decadal
average likelihoods of the regions' event each decade up to 2100. Return times are simply
1/likelihood. The change in likelihood of an event occurring (of a given return time) was
calculated relative to the ISIMIP3b 2010-2020 baseline period.
We also calculated the integrated probability of a return of the 2023 Canadian event within the
expected lifespan of an individual. According to UN population statistics, the average life
expectancy of a Canadian citizen born today is 83 years (United Nations Population Division,
2022). In order to cover the 7-year period after 2100, we extrapolated the annual trend in
probabilities. The integrated probability therefore is calculated as the product of the annual
probability of not seeing a fire event like 2023, for each year between 2023 and 2106.
## 3  Results

### 3.1  Extreme Wildfire Events of 2023-24

#### 3.1.1  Highlights

- **Global:** A total of 3.9 million km² burned globally during the 2023-24 fire season,
slightly below the average of previous seasons (4.0 million km²) and ranking 13th since
2001. Despite the lower BA, fire C emissions were 16% above average, totaling 2.4
Pg C, ranking 7th highest since 2003. Global C emissions were pushed up by record
emissions in Canadian boreal forests and pulled down by below-average fire activity
in the African savannahs (the largest contributor to global mean annual totals). If global
savannah fire emissions had been in line with their average in 2023-24, global fire C
emissions would have been the highest on record.

- **North America:** Record fire activity in Canada's boreal forests, with BA reaching six
times the average and fire C emissions over nine times the average, contributing
significantly to global C emissions. Canada experienced extreme and widespread fires,
with over 150,000 km² burned, prompting evacuations of 232,000 people and eight
firefighters lost their lives. The United States saw generally below-average fire activity,
but the Lahaina wildfire in Maui, Hawai'i, resulted in 100 civilian deaths, destroyed
2,000 homes, and displaced 10,000 people. Texas recorded its largest ever single fire,
which destroyed 130 homes, killed two civilians, and caused significant livestock
losses. Mexico faced its highest BA in the last decade due to ongoing drought
conditions, with 10,000 km² burned.

- **South America:** South America experienced somewhat below-average fire extent
overall, but notable exceptions included significant anomalies in the northern and
western regions, linked to extended drought and heatwave conditions. In Brazil's
Amazonas state, fire counts reached record highs due to historic drought, severely
impacting air quality in Manaus and leading to deforestation and habitat loss. In Chile,
the Valparaíso wildfire in February 2024 resulted in at least 131 deaths and widespread
property destruction, highlighting severe societal impacts. Fires in Bolivia's La Paz and
Beni departments and Peru's Loreto region caused significant environmental damage
including loss of protected forest and biodiversity, and disruption of indigenous
communities' livelihoods.



- **Europe:** Low wildfire extent in general, however the Alexandroupolis fire in Greece set a new EU record for individual fire size. Individual fires in Greece, Spain, Italy, Portugal, France, and Scotland led to large-scale evacuations, significant suppression costs, disruption of water supplies, damage to infrastructure or agricultural lands, impacts on tourism and local economies, destruction of properties, or loss of life.

- **Oceania:** Above-average fire activity in northern Australia's savannahs, grasslands, and shrublands was linked to fuel growth from three years of La Niña. Although less impactful than the 2019-20 Black Summer fires, wildfires near Perth resulted in five homes lost and several injuries. The Tara and Mount Isa fires in Queensland destroyed 65 homes and claimed two lives. In Victoria, fires destroyed over 40 homes and injured five firefighters. In New South Wales, forest fires caused widespread smoke-related damages.

- **Asia:** The 2023-24 fire season in Asia saw generally low fire activity, with central Asia experiencing below-average burned areas. Lao PDR, Thailand, and Vietnam had high fire counts amidst heatwave conditions and a possible uptick in agricultural fire use, leading to severe regional haze and air quality issues. In Mongolia's Dornod Aimag province, wildfires burned large parts of the Daurian steppe and required extensive firefighting efforts to stop fires from crossing into Russia. Additionally, wildfires in Russia's southern borders, including Tyumenskaya, Omskaya, and Amurskaya Oblasts, caused evacuations, property damage, and at least one fatality.

- **Africa:** Low wildfire extent in general with BA 13% below average in the African grassland, savannah, and shrubland biome. However, extreme fires in Northern Africa, particularly Algeria and Tunisia, prompted significant emergency responses including assistance from the EU. In Algeria, wildfires occurring in temperatures around 50°C resulted in 34 deaths and over 1,500 evacuations, with over 8,000 personnel deployed to combat the fires. Tunisia faced similar challenges with strong winds exacerbating wildfires, leading to evacuations in the northwestern region. In coastal South Africa, fires in the Western Cape caused structure damage and evacuations.

### 3.1.2 Global Perspective from Earth Observations

#### 3.1.2.1 *Extreme Fire Seasons Evident in Earth Observations*

According to the MODIS BA product, 3.9 million km$^2$ burned globally during the 2023-24 global fire season (March 2023-February 2024), slightly below the average of previous fire seasons (4.0 million km$^2$) and overall ranking 13th of all fire seasons since 2001 (Jones et al., 2024). Despite this, fire C emissions were 16% above average at 2.4 Pg C during the 2023-24 global fire season, which ranks 7th amongst all fire seasons since 2003 (based on annual averages of GFED4.1s and GFAS estimates; see methods; Jones et al., 2024).

Stark regional contrasts in the anomalies in BA, fire C emissions and individual fire properties are visible in the Earth Observations on various regional scales (**Figure 1**, **Figure 2, Figure 3**). **Figure 1** shows the strongest BA and fire C emissions anomalies of 2023-24 at continental biome scale versus previous fire seasons. BA was around 300 thousand km2 (13%) below the average of previous fire seasons in the African grassland, savannah and shrubland biome, which is the largest contributor to global mean annual BA totals. BA was below average in the South American grassland, savannah and shrubland biome in 2023-24 and in Asian non-forest biomes. The global pattern of lower BA in savannah-like systems in 2023-24 was not observed in Australia, where several non-forest biomes experienced well above-average BA.

Earth System
Science
Data

The North American boreal forests experienced a record-breaking fire season, with BA
reaching six times the average since 2001 and fire C emissions reaching over nine times the
average since 2003 (**Figure 1**; Jones et al., 2024). This strong regional signal primarily
explains the above-average global C emissions total of 2023-24, with the high rates of fire
emissions per unit area in boreal forests aggregating to override the reduced emissions totals
in African and South American savannahs. Record levels of fire C emissions were seen also
across the global pan-boreal forest biome, with fire C emissions surpassing the pan-boreal
record set in 2021 by more than 60%. This is despite a below-average fire season for BA and
fire C emissions in boreal Asia during 2023-24, in contrast to the 2021-22 fire season when
there was a synchronous peak in BA in both the Eurasian and North American boreal regions
(Zheng et al., 2021). According to the Global Fire Atlas, new records for individual fire size
and rate of spread were also set in the North American boreal forests during 2023-24, while
the upper 5% of fires ranked by size and rate of growth in 2023-24 were in the top 2 and 3
years on record since 2002, respectively.

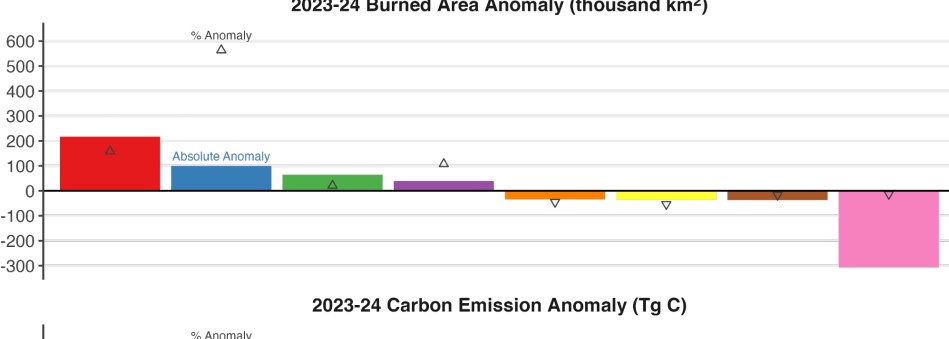

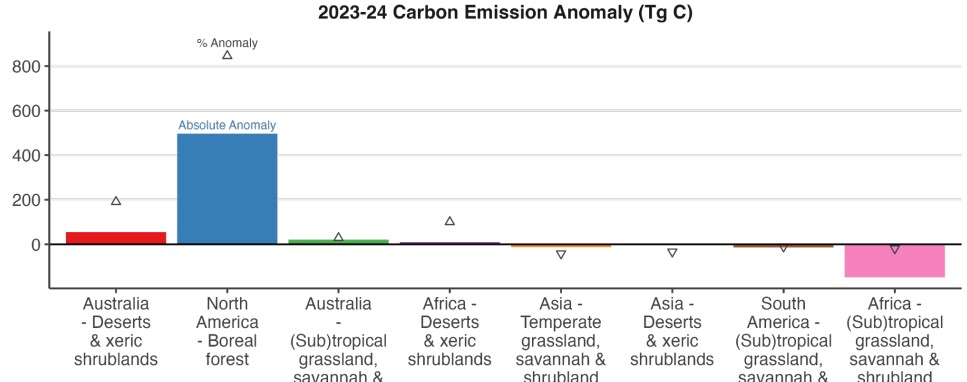

**Figure 1:** Anomalies in BA and C (C) emissions for selected continental biomes in the 2023-
24 global fire season (March 2023-February 2024), versus the average of prior fire seasons
since 2001. The selected regions all contribute at least 0.1% towards global mean annual BA
and experienced BA anomalies of over ±30,000 km$^2$ in the 2023-24 global fire season. Relative
changes (%) are also marked by triangular symbols and can be read on the same scale as
the absolute values.
Anomalies in the African (sub-)tropical grasslands, savannahs and shrublands strongly drive
inter-annual variability in global fire C emissions because this biome contributes on average
58% towards total global BA and 40% towards total global fire C emissions. If fire C emissions
from African (sub-)tropical grasslands, savannahs and shrublands had been around average





fire season in 2023-24, then global fire C emissions would have been the greatest of any fire
season on record since 2003.
Elsewhere at the biome scale, BA extent was in the top three years on record in the South
American broadleaf and mixed forests, the African xeric shrublands, and Australian xeric
shrublands, and the Australian (sub-)tropical grasslands, savannahs and shrublands (**Figure
1**). In contrast, BA or fire C emissions were the lowest on record in the European temperate
broadleaf and mixed forests and Asian xeric shrublands, and in the bottom three years on
record in the African savannahs, Asian montane grasslands and shrublands, and European
tropical grasslands and shrublands.
On national levels, the most prominent global anomaly of the 2023-24 fire season occurred in
Canada where BA reached six times the average of previous fire seasons and fire C emissions
reached nine times the average of previous fire seasons. Across the Canadian provinces and
territories, the highest BA or fire C emissions on record were observed in Northwest
Territories, British Columbia, Alberta, and Quebec while Yukon, New Brunswick, and Ontario
also experienced high-ranking years (**Figure 2, Figure 3**). The positive BA anomalies in
Canada were visible in the MODIS BA dataset from as early as April 2023 in most provinces
and persisted throughout summer through to October and even through to December 2023
and January 2024 in British Columbia and Alberta (**Figure S2**). Peak anomalies were
observed in Eastern Canada in June 2023, arriving later in western Canada (August-
September). Data on individual fire characteristics from the Global Fire Atlas further reveals
new record fire counts in many Canadian provinces, and high-ranking anomalies in fire count
and daily rate of growth across Canada, as well as new records for fire size and rate of spread
in provinces of both eastern and western Canada (**Figure 4**; Jones et al., 2024). **Section
3.1.3.4** discusses the unprecedented Canadian fire season of 2023-24 in greater detail,
including its impacts and regional context.
A second prominent regional feature of the 2023-24 global fire season, visible in Earth
observations, is a cluster of administrative regions with positive BA and C emissions
anomalies in the north and west of tropical South America (**Figure 2, Figure 3**). Bolivia,
Guyana, Suriname and French Guiana, Honduras, Nicaragua and Belize all experienced a
high-ranking fire season at national level in 2023-24. In addition, BA or fire emissions were
ranked in the top three years in western parts of Amazonia including in Amazonas state of
Brazil, the Loreto department of Peru, and the La Paz and Beni departments of Bolivia.
Anomalies in the western Amazon spanned June 2023-January 2024, peaking in August-
October 2023. In the north of South America, high-ranking fire seasons were seen in
Venezuela, various subdivisions of Guyana, Suriname, and French Guiana, and in Amapá
State in Brazil. The anomalies in northern South America spanned May 2023-February 2024,
peaking in November 2023-February 2024 (**Figure S2**). The Global Fire Atlas data suggest
that South American anomalies in BA during the 2023-24 fire season were principally driven
by a large number of fires, whereas anomalies in fire size or rate of growth were uncommon
in most of South America (**Figure 4**). **Section 3.1.3.6** discusses the 2023-24 fire season of
tropical South America and its impacts and regional context in greater detail.
Several parts of South and Southeast Asia experienced high-ranking anomalies in BA or fire
C emissions during the 2023-24 fire season, including various neighbouring administrative
zones of Lao People's Democratic Republic (PDR), Thailand and Vietnam. The temporal peak
of these anomalies was broadly in March-May 2023. Data on individual fire characteristics
indicates that high-ranking fire counts, rather than anomalies in fire size, were the primary
driver of the regional BA anomalies (**Figure 4**). **Section 3.1.3.2** discusses these anomalies
and their impacts in greater detail.
The anomalies observed in xeric biomes of Oceania are also apparent as high-ranking BA or
C emissions in the 2023-24 fire season in western parts of Australia, particularly in Western



Australia and the Northern Territory (**Figure 2, Figure 3**). Fires tended to affect more remote
areas and so the impacts on society were muted in comparison to the Black Summer events
affecting southeast Australia in 2019-20 (Abram et al., 2021); however, **Section 3.1.3.5**
discusses some notable exceptions.
Other regional pockets of high-ranking BA anomalies or C emissions anomalies were
observed in various dry zones of Africa and the Middle East, including the Sahel, Northern
Africa and the Horn of Africa, Southern Africa (specifically South Africa and Botswana where
three good rainfall years have resulted in grass fuel accumulation), parts of Iran, Iraq, parts of
the Levant region, and parts of the Arabian Peninsula (**Figure 2, Figure 3**). Although various
aspects of the fire season ranked highly in these regions, they are also fuel-limited with a
generally a low baseline for BA and fire C emissions and the wildfire season. Nonetheless,
regionally impactful wildfires were reported for Algeria, Tunisia and Morocco as well as coastal
regions of South Africa and in Pakistan and are discussed further in **Section 3.1.3.1** and
**Section 3.1.3.2**.






**Figure 2:** Ranks of BA during March 2023-February 2024 versus previous March-February periods (n = 23 global fire seasons) for three regional layers: **(top panel)** continental biomes; **(middle panel)** countries, and; **(bottom panel)** states or provinces. Results for regions with high-ranking (top 5 years) or low-ranking (bottom 5 years) events are highlighted. The timing of BA anomalies is shown in **Supplementary Figure 2**.

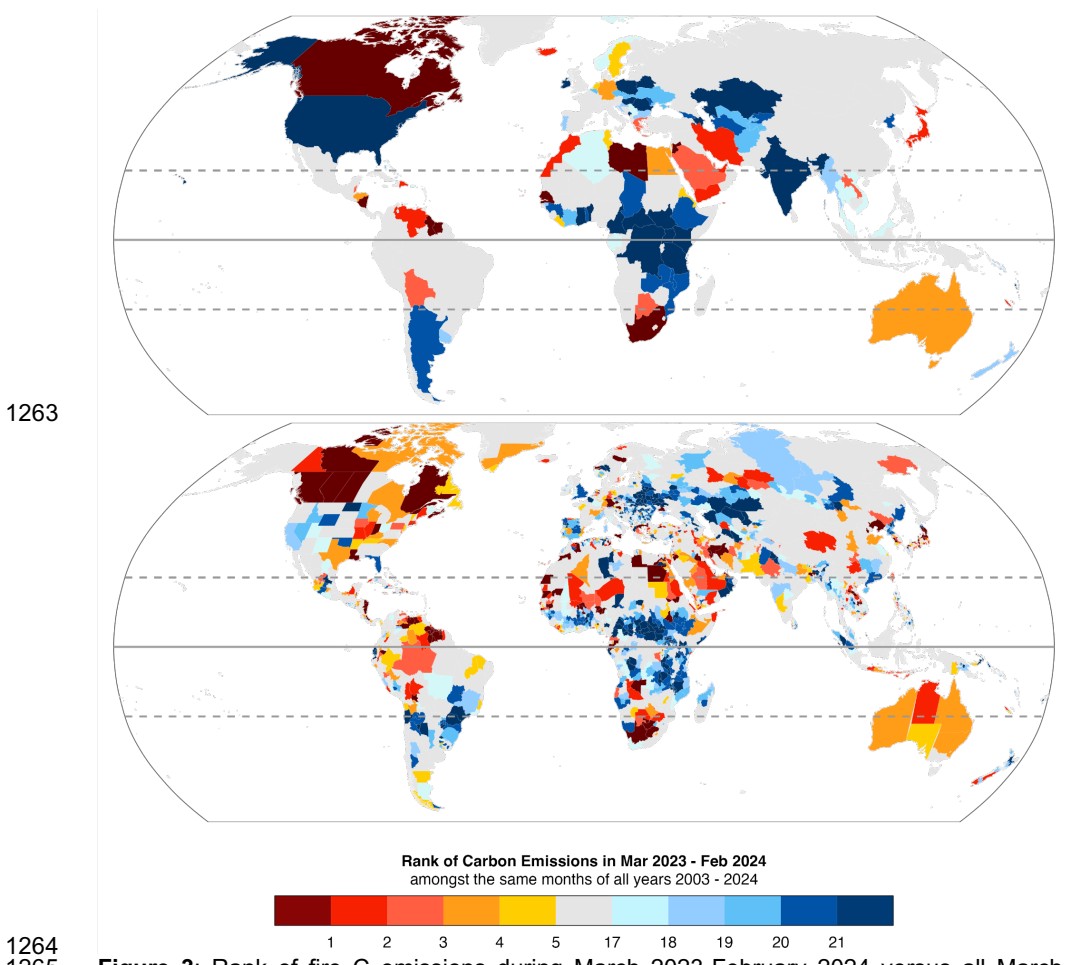


**Figure 3**: Rank of fire C emissions during March 2023-February 2024 versus all March-
January periods since 2003 (n = 21 global fire seasons), at the scales of **(top panel)** countries
and **(bottom panel)** level 1 administrative regions. We consider C emissions estimates from
two products (GFAS and GFED), first calculating the mean emissions value from the two
products, then ranking the values.

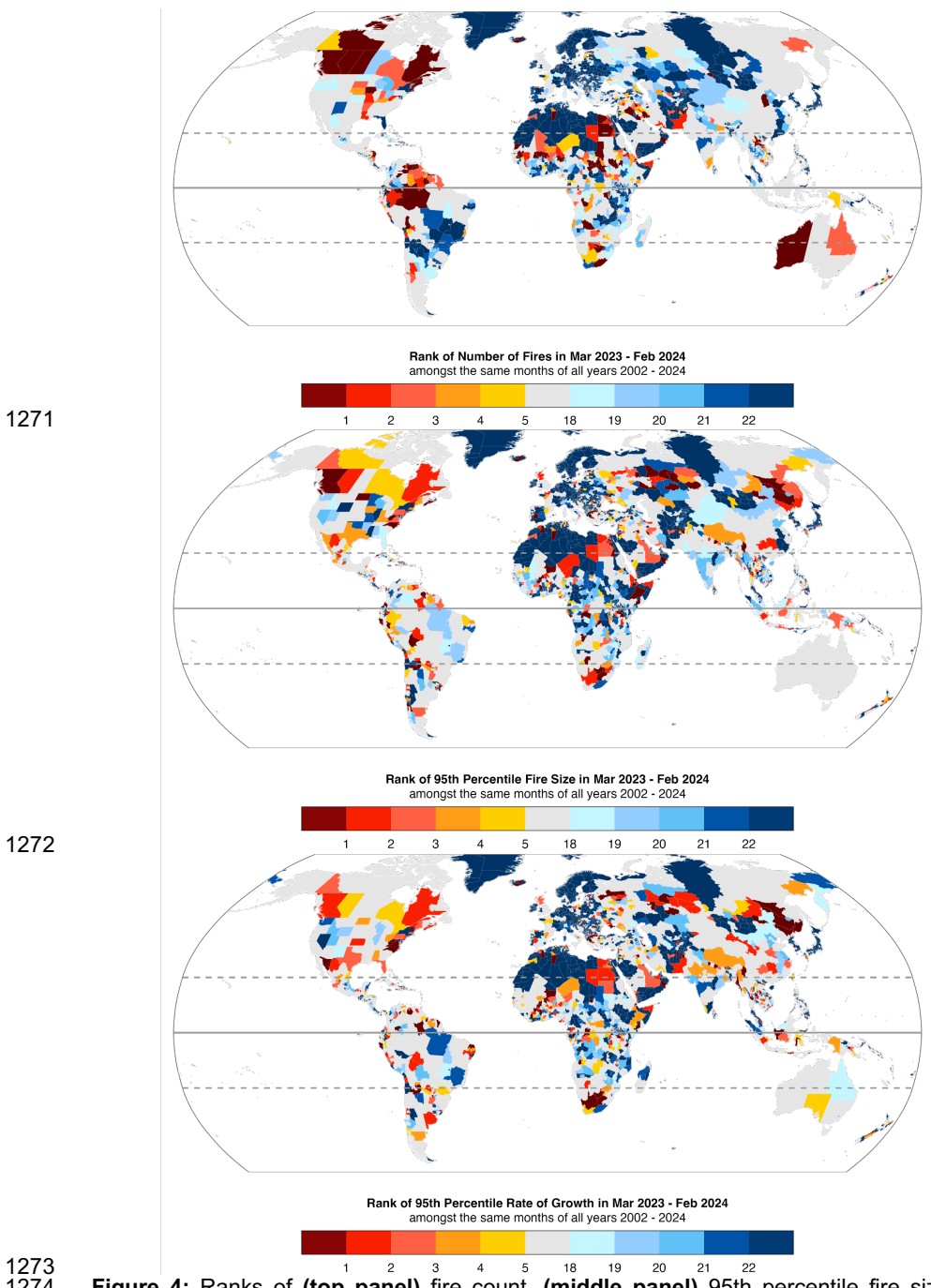



**Figure 4:** Ranks of **(top panel)** fire count, **(middle panel)** 95th percentile fire size, and
**(bottom panel)** 95th percentile daily rate of growth during March 2023-February 2024 versus
all March-February periods since 2002, at the scale of states or provinces (GADM
administrative level 1 regions).



### 3.1.2.2 Extreme Individual Fires in Earth Observations

The Global Fire Atlas (GFA) is one of several products tracking daily fire progression and identifying individual fires at global scale based on moderate resolution satellite data (Andela et al., 2019; Laurent et al., 2018; Artés et al., 2019). The product uses the MODIS BA product and the smallest unit of disaggregation is 500m and the shortest timestep on which the expansion of a fire can be observed is daily. The Global Fire Atlas is expected to represent the dynamics of large fires better than small, fast-moving fires.

The GFA represents some of the most impactful individual fire events in 2023-24 with varying skill (**Table 6; Figure S3, Figure S4, Figure S5**). For example, the Alexandroupolis fire that occurred in the decentralised administration of Macedonia and Thrace, Greece, in late August was captured reasonably well. The GFA identifies two fires that ignited on 19th August and merged to form one contiguous burned unit with an area of approximately 900 km$^2$. Alignment of the fire's timing, size and perimeter with high-resolution satellite imagery (**Figure S3**) and detailed reports (Xanthopoulos et al., 2024) suggest an overall reliable representation of this event. The impacts of this fire are discussed in detail below in **Section 3.1.3.3**. Notably, observations from the Visible Infrared Imaging Radiometer Suite (VIIRS) sensor, which features higher resolution and higher overpass frequency, did not improve the characterisation of this particular event (**Figure S3**).

A deadly fire near Valparaíso, Chile, is also captured with reasonable skill in the GFA (**Figure S4)**. Around 90 km$^2$ was burned, with the fire skirting the city of Placilla de Peñuelas and encroaching upon Viña del Mar and Quilpué (**Section 3.1.3.6**). The timing, extent and perimeter of the fire as recorded by the GFA compares well with those reported by other sources (**Table 6**).

Among the largest fires to occur in Canada during 2023-24 happened near the La Grande Reservoir in Quebec, Canada. According to both the Global Fire Atlas and a separate NASA fire tracking product based on the VIIRS sensor, the fire's extent was around 11 thousand km$^2$, whereas the National BA Composite (NBAC; Skakun et al. 2022) shows a slightly lower extent nearer 10 thousand km$^2$. The timing of the fire also showed high correspondence across the products.

The Lahaina wildfire in Maui, Hawai'i, is an example of an event that was captured poorly by the GFA. Issues relating to the small scale of this fire relative to the resolution of the MODIS BA data are evident in **Figure S4**. As the MODIS BA algorithm is focussed on the detection of wildland fire, its effectiveness in tracking fires at the wildland-urban interface is limited. In this case, burned areas were not detected in cells in urban areas or at the wildland-urban interface, and hence the size of the fire was under-estimated significantly (**Table 6**). The timing of the fire on vegetation land adjacent Lahaina was compatible with reference reports.

Another example of the challenges of defining individual fire extent and applying global algorithms to do so comes from Western Australia (**Figure S5**). Two fires recorded by the Department of Fire and Emergency Services, Western Australia (the Great Sandy Desert and Anna Plains fires) totalled 57 thousand km$^2$ in extent. In the Global Fire Atlas, the burned cells detected by MODIS were instead dissected into 53 separate fires with the largest unit burning 560 km$^2$. The date ranges were also rather different with the first record of fires logged in agency data one month later than in the Global Fire Atlas and the final record logged one month earlier.



**Table 6:** Representation of selected individual fire events in the MODIS BA product (Giglio et
al., 2018) and Global Fire Atlas (Andela et al., 2019).

| Event | GFA Fire Size (km²) | GFA Dates | Reported Area (km²) (Reference) | Dates (Reference) | Reference | Comment |
|---|---|---|---|---|---|---|
| Alexandroupolis Wildfire, Greece | 892 | 19/08/2023 to 30/08/2023 | 930 | 19/08/2023 to 31/08/2023 | Xanthopoulos et al. (2024) | Good characterisation of the event, with two fires merging in the date range and ultimate fire size comparable to reference. |
| Fire near La Grande Reservoir, Quebec, Canada | 10,725 | 29/05/2023 to 23/07/2023 | 11,400 (VIIRS) 9,694 (NBAC) | 01/06/2023 to 23/07/2023 | NASA Earth Observatory (2023; VIIRS); Jain et al. (2024; NBAC) | Reasonable characterisation of the fire's extent and timing. |
| Valparaíso Wildfire, Chile | 91.54 | 31/01/2024 to 10/02/2024 | 85 | 02/02/2024 to 05/02/2024 | NASA Earth Observatory (2024); Copernicus Emergency Management Service (2023a) | Good characterisation of the scale of the event and its perimeters at various wildland-urban interfaces, versus reference data. |
| Lahaina Wildfire, Maui, Hawaii | 1.50 | 08/08/2023 to 12/08/2023 | 8.49 | 08/08/2023 to 09/08/2023 | Fire Safety Research Institute (2024) | MODIS data has coarse spatial resolution relative to scale of event. Spread into urban areas not captured. |
| Western Australia (Great Sandy Desert and Anna Plains fires) | 45,544 | 31/08/2023 to 28/11/2023 | 56,561 | 27/09/2023 to 01/11/2023 | Department of Fire and Emergency Services, Western Australia (Shapefile for the Great Sandy Desert and Anna Plains fires; Agnes Kristina, pers. comm.) | GFA splits this event into 53 fires; we report their total combined area.<br><br>Largest individual fire area in GFA was 760 km² (ignited 06/09/2023).<br><br>Great Sandy Desert and Anna Plains fires merged on 25th October 2023. |

### 3.1.3 Year in Review by Continent
#### *3.1.3.1 Africa*
South Africa and Botswana experienced higher than average BA and fire size (**Figure 2,**
**Figure 4**) in 2023-24, linked to three consecutive years of above-average rainfall that
increased grassy fuel loads in the arid (fuel-limited) savannas and grasslands. This has
potentially been exacerbated by a lack of prescribed burning and active fire suppression in the
privately held land and conservancies in the region, that likewise would have resulted in fuel
build-up (*Atlas of Namibia*, 2021). The socio-economic impacts of these large fires were
minimal (extensive grassland fires linked to high rain years are expected due to periodic 7-20
year wet-dry cycles in these ecosystems). However, some farmers in western Botswana





reported increased livestock losses from predators such as lion, wild dog and hyena, with the
reduced vegetation cover after the fires driving predators to shift from wildlife prey to livestock.

In East Africa, the area burned was extremely low in 2023-24 due to the 2020-2023 triple la
Nina experienced in this region, which causes droughts in East Africa (while increasing rainfall
in southern Africa). This multi-year drought meant that there were no grass fuels to burn and
reduced the likelihood of spread of accidental ignitions in many of the East African rangelands.
However, the extreme fire weather enabled fires to burn through upland forests, which are not
normally flammable. The Aberdare Forest, in Nyeri county Kenya had a fire incident on the
17th of Feb 2023 that destroyed over 160 km$^2$, and was attributed to the dry conditions (*Citizen*
*Digital*, 2023). This also occurred at a time when the country was undergoing a heatwave.
This highlights the different responses of grassland and forest vegetation to the same climate
conditions.

The 2023 heatwave in North Africa exacerbated fire behaviour in the region (*Al Jazeera*,
2023a). Algeria recorded significant fires in the latter half of July, facilitated by high
temperatures that reached upwards of 48°C (*Al Jazeera*, 2023b). Over 8,000 personnel,
including firefighters and the military, were deployed to combat rapidly spreading fires across
15 provinces (*South African Broadcasting Corporation*, 2023a). These efforts were critical in
managing fires that forced over 1,500 people from their homes (*euronews*, 2023). Despite
these efforts, the wildfires claimed the lives of at least 34 individuals, including 10 soldiers (*Al*
*Jazeera*, 2023b).

Neighbouring Tunisia also faced wildfire outbreaks, exacerbated by strong winds that carried
fires across the national border from Algeria, leading to the closure of two border crossings
(*Reuters*, 2023a). The Tunisian wildfires prompted evacuations in the north-western region of
Tabarka, affecting at least 300 people and extending firefighting efforts to Bizerte, Siliana, and
Beja (Sullivan and Tondo, *The Guardian*, 2023). Resources such as firefighting aircraft and
personnel were sent from EU nations to help tackle the fires, despite the challenging
conditions imposed by near-record temperatures of 49°C (Gauldie, *AirMed&Rescue*, 2024).
In August, forest fires in mountainous regions of Morocco were also fanned by strong winds
and facilitated by protracted hot spring and summer temperatures (Erraji, *Morocco World*
*News*, 2023; Copernicus Climate Change Service, 2024a).

During December 2023-January 2024, the Western Cape of South Africa experienced
wildfires related to prolonged hot and windy conditions, causing substantial damage and
prompting widespread evacuations. In the Overstrand local municipality, which includes
coastal towns like Pringle Bay and Betty's Bay, multiple fires necessitated evacuations and
destroyed properties. The Hangklip area between Pringle Bay and Betty's Bay was particularly
affected with the fires destroying properties in the Sea Farm private nature reserve. On
January 29, a "code red" status was declared, indicating a serious threat to residential areas,
and evacuations were advised for communities including Silversands and Seafarms (*Crisis24*,
2024; *AfricaNews*, 2024). A wildfire swept from Simonstown to Scarborough in Cape Town,
necessitating large-scale evacuation (Hough, *IOL News*, 2023). This fire was challenging due
to its rapid spread fueled by strong south-easterly winds and high temperatures. The
firefighting efforts were supported by multiple helicopters and ground teams (*South African*
*Broadcasting Corporation*, 2023a). The most extensive damage was reported from the
Kluitjieskraal fire near Wolseley, where over 220 km$^2$ were burned, and more than 40
structures were destroyed. This fire also prompted evacuations and remained uncontained for
several days due to its size and complex terrain that hindered ground access (*Crisis24*, 2024)
. Despite these extreme wildfires, the plantation forestry industry was not affected, with
relatively low losses due to fire.

***3.1.3.2 Asia***



The 2023-24 fire season in Asia was generally not an extreme one, with much of central Asia experiencing low BA. Siberia, which has seen several record-breaking fire seasons since 2020 resulting in globally significant fire emissions (Zheng et al., 2021), also experienced a somewhat typical year for BA and fire C emissions. Likewise most provinces of China and states of India experience a fairly typical fire season.

Nonetheless, there were regional examples of high fire activity in the 2023-24 fire season. The Dornod Aimag province of eastern Mongolia, near the borders with Russia and China, experienced several extreme fires during April 2023 that are also visible as anomalies in the global fire observations (**Figure 2, Figure 4)**. Over 15% of the area of Dornod Aimag burned in 2023-24 in contrast to the 23-year average of below 5%. The province includes the Mongolian part of the Daurian steppe, notable for being one of the last remaining undisturbed steppes in the world (UNESCO World Heritage Centre, 2017). Unusually dry and warm conditions in eastern Mongolia during spring led to severe wildfires. A notable wildfire spread into Dornod from neighbouring Sukhbaatar province, fanned by windy, dry conditions (*Borneo Bulletin, 2023*). The National Emergency Management Agency mobilised over 250 individuals, including firefighters and local residents, and helicopters were deployed to manage the fast-spreading fires (*Borneo Bulletin, 2023*). The effects of these wildfires on herder and nomadic populations compounded the losses of 2 million livestock over the harsh winter in Mongolia, and the Mongolian Red Cross has provided aid to 4,800 herder households (International Federation of Red Cross and Red Crescent Societies, 2023).

Although BA extent and fire counts were overall below the 2001-2023 average along the southern border regions of Russia during 2023-24 (**Figure 2, Figure 4**), a number of disruptive wildfires fanned by strong winds broke out during April and May and affected regions bordering Kazakhstan, such as in the Tyumenskaya, Omskaya, and Amurskaya Oblasts where an emergency was declared, and Mongolia, such as in Irkutsk and Krasny Yar where at least one fatality was recorded (*Le Monde*, 2023). As well as detecting anomalies in fire size and rate of spread in these areas, the Global Fire Atlas also identified regionally large and fast-moving wildfires in the Russia-China border zone of Manchuria (**Figure 4**), however these were not widely reported on by media outlets or local authorities.

Lao People's Democratic Republic (PDR) experienced a notable fire season in 2023-24, marked by record-setting BA at national level since 2001 in the MODIS BA data (**Figure 2; Figure S6**). The fires were widespread, affecting various provinces from the south to the north, including Attapu, Khammouan, Louangphabang, Xaignabouli, and Bokeo. In Attapeu, BA in 2023-24 was over twice the average of prior fire seasons since 2001. The fires in 2023-24 were generally small in scale but anomalously numerous, consistent with the widespread use of slash-and-burn agricultural fires in these regions that have been problematic for regional air quality in this region during recent years (Meadley, *Laotian Times*, 2024). The uptick in fire counts in 2023-24 has been attributed in part to economic factors such as the high price of cassava and demand for greater corn supplies to supply animal feed, which act as incentives for farmers to clear forests for additional planting (Bhandari, *Radio Free Asia,* 2024). On top of economic factors, an enabling driver was an extreme heatwave that extended eastward from South Asia to Southeast Asia in April 2023 (Zachariah et al., 2023). The persistent smoke from these fires worsened air quality significantly in southeast Asia, where efforts to manage transboundary haze have been challenging during regional droughts, despite a new transboundary agreement being signed in 2023 (Antara News, 2023). Differences in fire management between Thailand and Lao PDR were evident during the 2023 event, with authorities intensifying patrols and seeking to control forest fires and agricultural burning for improved air quality in Thailand (Meadley, *Laotian Times*, 2024). Conversely, deforestation remains a critical issue in Lao PDR, with the Laotian government facing challenges in gaining local community support for the prevention of agricultural expansion and logging.



Earth Observations data showed high-ranking BA anomalies and fires with large size and rate of growth during 2023-24 in several regions of Pakistan, Iran, Iraq, and parts of the Levant region (**Figure 4**), consistent with reports of extreme drought-driven wildfires in some of these regions (*Reuters*, 2023b).

### 3.1.3.3  Europe

Overall, fires burned 8,400 km$^2$ in Europe from March 2023 to February 2024 according to the European Forest Fire Information System (EFFIS, 2024), of which 64.5% were from July to September and 18.1% were in March and April. Large fires (>5 km$^2$) amounted to 53.4% of the total BA, and those particularly large (>100 km$^2$, n=5) accounted for 17.7% of the total burned area. More than half (52.6%) of the BA corresponded to transitional woodland, with forests, shrublands and grasslands, and agriculture respectively amounting to 19.1%, 13.2%, and 14.4%. At least 44 people died as a direct result of wildfires (Copernicus Climate Change Service, 2024; Centre for Research on the Epidemiology of Disasters, 2024).

Most countries in the Mediterranean Basin experienced mild to typical fire seasons in general, with variable timing but affecting mostly non-forest (open) vegetation types (EFFIS, 2024). In the Balkans, fire activity varied among countries, but was mostly very low by historical patterns such as in Croatia, however, a major exception was Greece described in more detail below (**Figure 2, Figure 4, Figure 5**). The other exceptions were North Macedonia, with a typical fire year and Bulgaria, the worst year in a decade, with fire activity extending into October in both countries; and Bosnia and Herzegovina, Serbia and Montenegro, where collectively ~270 km$^2$ burned in January-February 2024.

Greece's 2023 fire season was reviewed at length by Xanthopoulos et al. (2024). It was the second worst on record regarding total area burned (1,727 km$^2$), despite the recent efforts to strengthen the firefighting mechanism of the country with more aerial resources and new personnel, after another challenging fire season in 2021. The situation was kept under control until mid-July, but in the period July 13-27, maximum temperature in many parts of the country exceeded the average for the 2010-2019 period by as much as 10°C, according to the records of the National Observatory of Athens. This resulted in multiple fire starts pushing the limits of firefighting, which relies heavily on the aerial resources. The fires starting 18th of July on the tourist island of Rhodes grew rapidly on the second day, finally burning 207 km$^2$ and stopping at the sea. About 20,000 tourists had to be evacuated from hotels along the coast. While the fire on Rhodes was still burning, three forest fires started on 3rd of July started near the city of Aigio, in North Peloponnese, on the island of Corfu and near the town of Karystos in the south of Evia island. On July 25th a Canadair CL-215 crashed near the village of Platanistos while fighting this last fire. Then, on July 26th, the tail of a cold front that passed over Greece, with the characteristic wind direction change that accompanies it, created further challenges, as a number of fire starts in central Greece and Thessaly spread fast, burning mostly in light fuels, challenged firefighters and threatened inhabited areas. One of the fires entered an Air Force base on the 27th, causing a powerful explosion of ammunition that resulted in damages to the town of Nea Aghialos 5 km away. By the end of July, the BA across the country had reached 550 km$^2$.

The next wave of multiple challenging fires in Greece began on 19th August. A lightning-caused fire that started before dawn NE of Alexandroupolis in the prefecture of Evros received limited attention at first and was destined to become the largest fire in recent European history. The fact that firefighting resources were focussed on evacuation of the villages in the path of the fire, rather than fire suppression may have contributed to its eventual size. On the 21st of August, a second fire started to the north of the first one, near the village of Dadia. Fanned by a strong NE wind, it spread quickly and within a few hours it reached the rear of the first one. On that day fire behaviour both in terms of spread and intensity was extreme (Athanasiou, 2024). Nineteen immigrants who had illegally entered the country and were trekking through





the forest, were trapped by the flames and were found dead on the 22nd of August. Another
group was saved by the firefighters at the last moment. The authorities emphasised safety
and evacuated the hospital in the outskirts of Alexandroupolis.
On Aug. 22, while the Evros fire was the focus of attention, a fire originating at more than one
point near the village of Phyli, south of mount Parnis in Attica, at the outskirts of Athens started
growing against the strong NE wind. Once more, many settlements were evacuated and
firefighting attention focused on protecting homes, as the fire moved slowly up the mountain
slopes finally burning 62 $km^2$ in three days. The Evros fire kept growing at various rates for
the next 15 days, finally reaching 938 $km^2$ and becoming the largest on record in recent history
in Europe. The simultaneous spread of the Evros fire, the fire in Attica and a number of smaller
fires, is likely to have increased the growth rate statistics (km day$^{-1}$) for fires in the region
(**Figure 4** and **Figure 5**).
The BA of the Evros fire included 258 $km^2$ of deciduous oak forest and 218 $km^2$ of oak forest
mixed with other species (Konstantinos Kaoukis, personal communication). The usually most
challenging forest types regarding fire behaviour, contributed less to burned area: 128 $km^2$
forest and 152 $km^2$ of evergreen shrubs. The fire was mostly brought into control only when it
reached agricultural areas and barren lands. The final size of the Evros fire may not be solely
attributed to adverse meteorological conditions. One aggregating factor may have been the
recent shift in directing firefighting personnel more strongly from suppression towards
evacuation, another the emphasis on aerial firefighting resources (Xanthopoulos et al., 2024).
The latter was not effective once the extreme fire behaviour commenced (21st to 23rd of
August). Deep forest litter layers further hampered fire suppression in some areas, although
a group of local forest workers working with handtools, were credited by the local forest service
officers with control of a large part of the fire perimeter to the north, saving an estimated 100
ha of forest (Athanasiou, 2024).
Italy was the second most affected country after Greece, with continuous fire activity from July
to October. More than 1,000 $km^2$ burned in the country, of which 69% were in Sicily (including
17 fires >10 $km^2$), although the largest fire (31 $km^2$) occurred in the nearby region of Calabria
(Istituto Superiore per la Protezione e la Ricerca Ambientale, 2023). A defining characteristic
of these large fires was the importance (42% overall) of agricultural land in the BA composition.
The outskirts of Palermo and the Madonie Natural Regional Park were impacted by multiple
wildfires in late September, causing one fatality and affecting wildland-urban interfaces, farms,
and tourism.
Fire activity was insignificant in France, except for benign mountain burning (175 $km^2$) in
March-April and then in January-February, mostly in the western Pyrenees. Like in France,
the north of Spain (Asturias-Cantabria) experienced unusual Spring burning activity,
amounting to 423 $km^2$ during late March and early April (Educación Forestal, 2023a). In
particular, the Foyedo wildfire (27th March 2023) was the largest on record for Asturias,
burning 101 $km^2$ across variable vegetation but with the predominance of conifer plantations,
mostly *Pinus pinaster*. It was a wind- and spot-driven fire but its soil and overstorey burn
severity were respectively low and mostly moderate, as slower-drying fuels were not available
to burn (Cátedra Cambio Climático de la Universidad de Oviedo, 2023).
Only two other notable large wildfires occurred in 2023 in continental Spain, and again were
unusual in that they happened in spring rather than summer. The Villanueva de Viver wildfire
(23rd March 2023, Castellón and Teruel) burned around 50 $km^2$ and was driven by abnormal
seasonally dry conditions, combined with a shift in wind direction. It mostly burned naturally-
regenerated continuous pine forest of *Pinus halepensis*. Canopy fire severity was
heterogeneous, with 39% of the wildfire area being classified as high to very high severity
(Mediterranean Center for Environmental Studies, 2023). The cost of fire suppression was
2M€ and 1,800 people were evacuated (*Las Provincias*, 2023).




At over 100 km$^2$, the Pinofranqueado wildfire (17th May 2023, Cáceres) was the largest fire in
the Iberian Peninsula in 2023 (Copernicus Emergency Management Service, 2023b). The BA
was 90% forest, mostly pine (*Pinus pinaster*; *Juntaex.es*, 2023). It was a wind-driven fire and
the Canadian FWI indexes indicate all fine fuel was available to burn and extreme fire
behaviour (FWI>50). The fire significantly impacted the nesting of protected bird species and
rainfall shortly after the wildfire caused important runoff, erosion, and disruption of water
supply to the local population (Armero, *Hoy*, 2023).

The two other significant fires in Spain happened in the Canary Islands, the Puntagorda
wildfire (14th July 2023, La Palma, 32 km$^2$) and the Arafo-Candelaria wildfire (15th August
2023, Tenerife, 123 km$^2$; Copernicus Emergency Management Service, 2023c). The latter
spread for 9 days and 94% of its area was forest under conservation status, mostly of Canary
pine (*Pinus canariensis*). While drought was advanced, atmospheric conditions were not
particularly severe. The fire was topographically driven and highly heterogeneous in severity,
but low to moderate severity prevailed (Educación Forestal, 2023b). Nonetheless, 26,000
people were evacuated, 364 farms and 246 buildings (none residential) were affected, smoke
impacts were substantial and damage was estimated at 80.4M€.

Like in Spain, winter shrubland burning was relevant in continental Portugal (~50 km$^2$ in
February) but subsequent significant wildfire activity was restricted to two fires. The Sarzedas
(66 km$^2$ ha) and Baiona (75 km$^2$) wildfires started on the 5th of August under extreme fire
weather (FWI>50), and burned mostly (~70%) forest, respectively of pine (*P. pinaster*) and
eucalypt (*Eucalyptus globulus*) stands (Direção Nacional de Gestão do Programa de Fogos
Rurais, 2023). Prevailing burn severity was moderate and damage to infrastructure and
emergency restoration amounted to 6.4 M€ cost for the Sarzedas fire and a forest value loss
of 1.4 M€ (Instituto da Conservação da Natureza e das Florestas, 2023). The major run of the
Baiona wildfire was on August 7, corresponding to 73% of the total BA, when it threatened
wildland-urban interfaces and damaged several buildings; one camping park and 20 small
villages were evacuated (*Economia Online*, 2023). Moderate to high burn severity classes
were dominant and costs were estimated at 2.7 M€ (tourism) and 7 M€ (houses), plus 1.4 M€
in forest values loss and 2.9 M€ for emergency stabilisation (*Rádio e Televisão de Portugal*,
2023; SIC Notícias, 2023). Finally, on 12th October, and under anomalously extreme fire
weather for the time of the year, the Ponta do Pargo wildfire burned 48 km$^2$ in Madeira island,
with an estimated agriculture-related cost of 3 M€ (*Rádio e Televisão de Portugal*, 2023).

The year was also mild in other European countries where burned areas can be extensive,
namely Romania, Hungary, and Poland, which collectively summed only ~210 km$^2$ burned.
EFFIS recorded 2461 km$^2$ burned in Ukraine, the largest fire attaining 42 km$^2$, but these figures
are far from those registered in recent years. In northern Europe, a notable fire occurred in
Scotland near Cannich, during the spring, the primary fire season in the humid Atlantic climate
of the UK (Belcher et al., 2021). It started on 19th of April and burned ~33 km$^2$ of mainly
moorland making it one of the largest fires in the UK in recent history (Sabljak, *The Herald*,
2023; personal communication Niall MacLennan, Scottish Fire and Rescue Service).


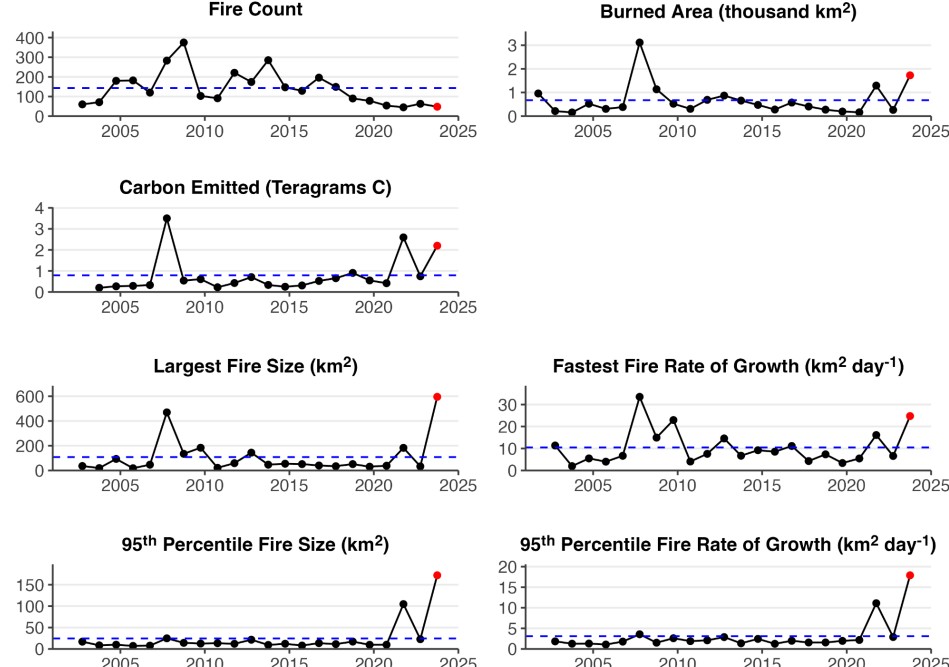

**Figure 5:** Summary of the 2023-2024 fire season in Greece. Time series of annual fire count, BA, C emissions, PM2.5 emissions, 95th percentile fire size, fastest daily rate of growth, and 95th percentile fire daily rate of growth. Black dots show annual values prior to the latest fire season, red dots the values during the latest fire season, and blue dashed lines the average values across all fire seasons.

### 3.1.3.4   North America

Wildfire across North America in 2023-2024 was characterised by record fire activity across Canada, lower than normal BA in the most flammable regions of the western US, above average fire activity across Mexico, and several extreme events that resulted in disastrous impacts to human communities. North America contributed the majority of the Northern Hemisphere area burned for the year, and two-thirds of the global C emissions from wildfire (Friedlingstein et al., 2023; Kolden et al., 2024). Canada burned over 150,000 km$^2$ in 2023, over twice the previous record and over seven times the annual average (Jain et al. 2024). The US burned 10,900 km$^2$ in 2023, well below the long-term annual average (National Interagency Fire Center [NIFC], 2024a). Mexico has a relatively short historical record of wildfires, but March 2023-February 2024 saw the highest area burned in the last decade (10,000 km$^2$) due to strong ongoing drought conditions (Comisión Nacional Forestal, 2024).

The fire season began earlier than normal in Canada, driven by early snowmelt across much of the country and persistent drought conditions in the west. Abnormally high temperatures and lack of rainfall also saw forested regions of eastern Canada, including Quebec, transition rapidly to drought conditions at the end of May. British Columbia province recorded its first wildfire evacuation in mid-April, and in late May, over 16,000 people were evacuated from Halifax, the capital city of Nova Scotia, which saw its largest ever wildfire in 2023 (Jain et al. 2024). In June, two lightning outbreaks in Quebec initiated several hundred new fires in what would eventually become a record BA year for the province (4,300 km$^2$) (Boulanger et al.,





2024). While the majority of the Quebec fires were in remote regions, the smoke they generated blanketed several major cities in eastern North America, including New York which experienced its worst air quality in half a century as the observed daily mean PM$_{2.5}$ concentration rose to 148.3 µg m$^{-3}$, over 4x the recommended daily limit (Wang et al. 2023). In total over 50 million people were exposed to high levels of PM2.5 for several days (Yu et al. 2024). This situation was further exacerbated in New York City by several wildfires in the nearby New Jersey pine barrens, a fire-prone dry pine forest that sees large fires during periods of drought.

The year started with low fire activity across the USA. In the high plains of the central US, an outbreak of large wildfires occurred coincident with dry conditions and strong winds in March-April 2023. One wind-driven wildfire started by power lines in Oklahoma destroyed several dozen homes (Oklahoma Department of Emergency Management, 2023). Outside of the high plains, dry conditions also elevated fire activity across the Southern, Eastern, and New England regions of the USA. Mexico saw slightly above average fire activity during spring, which is the peak period of the fire season as debris burning and field clearing activities provide ignitions for predominantly shrubland and grassland wildfires.

As summer arrived in Canada, the western and boreal provinces and territories saw extreme and widespread fire activity, even as Quebec continued to burn. By the end of the year, record area had burned in British Columbia (2,300 km$^2$), Alberta (2,700 km$^2$), and the Northwest Territories (3,500 km$^2$) accompanied evacuations of 232,000 people in numerous rural villages and large cities such as Yellowknife, NT and Kelowna/West Kelowna, BC, where a wildfire jumped the 2 km-wide Okanagan Lake (Jain et al., 2024; *CBC News*, 2023). The extreme behaviour of these fires not only shrouded large swaths of North America in smoke, but also generated an unprecedented 140 pyrocumulonimbus clouds (Jain et al. 2024). Eight firefighters were killed during summer 2023 in Canada (Jain et al. 2024), but miraculously no civilians died directly in the fires. Canada was at the highest National Preparedness Level 5 for an unprecedented 120 continuous days starting on May 11, indicative of the significant resource sharing required by fire management; in all, over 5500 international personnel from 12 countries and the EU were deployed to Canada during the 2023 fire season (Canadian Interagency Forest Fire Centre, 2023).

In the US, a relatively low activity fire season became deadly in August owing to unusual weather conditions facilitating extreme fire behaviour in multiple areas around the country. On 8$^{th}$ August, a pressure gradient-induced katabatic wind event fanned several small wildfires in Hawai'i, and 101 civilians died as the town of Lahaina on the island of Maui was consumed in the worst wildfire disaster in the US in a century (Pyne, 2017). Over 2,000 homes were destroyed and over 10,000 people were displaced as a result. Extreme heat and flash drought produced fire behaviour that killed five additional civilians in the US states of Washington (2 fatalities), Louisiana (2 fatalities) and California (C. Kolden, unpublished data). These extreme events stood in contrast to overall low fire activity and were notable for where they occurred. Washington does not typically see many extreme, wind-driven wildfires, and Louisiana is one of the wettest states in the US. By the end of August, the US BA was only 40 percent of normal and the lowest since at least 2000 (NIFC, 2023; National Oceanic and Atmospheric Administration [NOAA], 2023).

As North America transitioned to fall and then winter, Canada continued to burn nearly a month longer than normal, with the last large fires not controlled until late October. On 22$^{nd}$ September, remarkably late in the fire season, Canada saw its largest ever one-day total for BA at approximately 4,400 km$^2$ (Jain et al., 2024). While many of the Canadian fires were fully extinguished by winter, others simply smouldered into the deep peat layers, aided by a warmer-than-normal winter with a reduced snowpack. At the end of February 2024, spaceborne thermal sensors detected several dozen fires in northern Alberta and British

Earth System
Science
Data

Columbia that were overwintering fires, likely sustained by peat smouldering (Shingler, *CBC*
*News*, 2024; Scholten et al., 2021).

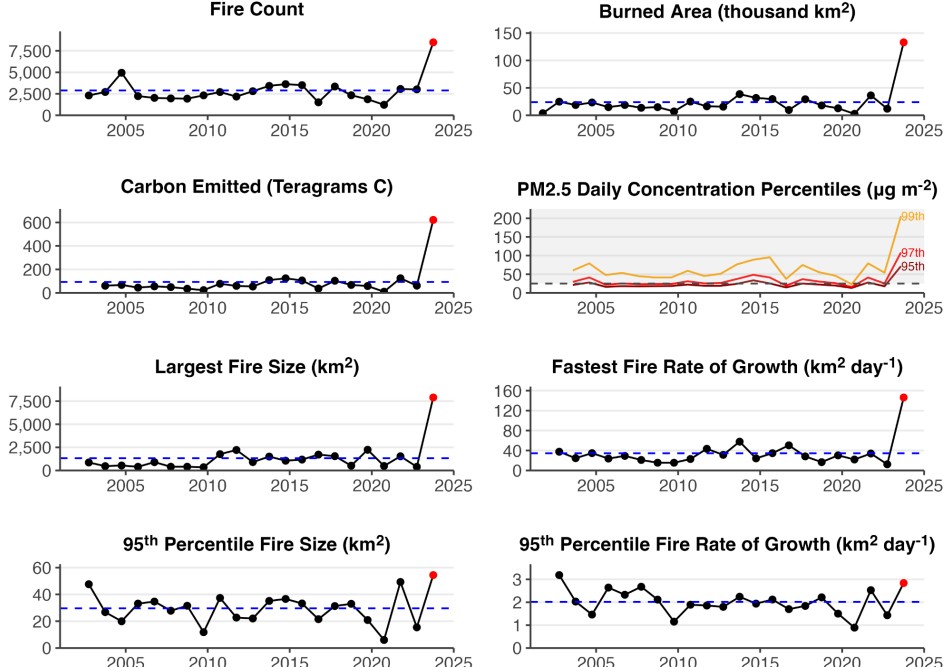

**Figure 6:** Summary of the 2023-2024 fire season in Canada. Time series of annual fire count,
BA, C emissions, PM2.5 daily concentration percentiles (95th, 97th, and 99th), 95th percentile
fire size, fastest daily rate of growth, and 95th percentile fire daily rate of growth. Black dots
show annual values prior to the latest fire season, red dots the values during the latest fire
season, and blue dashed lines the average values across all fire seasons. The PM2.5 daily
concentration percentiles are based on area-weighted daily mean PM2.5 concentrations
across Canada (see Methods). The 95th, 97th, and 99th percentiles correspond to PM2.5
concentrations that are breached on 18, 10, and 3 days of the year, respectively. Grey shading
represents concentrations that exceed the reference levels for 24-hour mean PM2.5
concentration in Canada, statistically derived on the basis of several key epidemiological
studies (Canadian Environmental Protection Act Federal-Provincial Working Group on Air
Quality, 1998).
US fire agencies recorded just over 10,900 km$^2$ burned in 2023, just over half of the 20-year
mean of 29,000 km$^2$ (NIFC, 2024a). Notably, over half of the BA was associated with higher
fire activity in the central plains grasslands and the southeastern US, while below normal fire
activity characterised California and the western US throughout 2023 as the region exited a
multi-year drought. However, the number of fires recorded was only slightly smaller than
average. This quiet pattern broke in February 2024, however, when drought conditions from
the high plains region of the US down into north central Mexico coupled with strong winds to
produce massive, fast moving wildfires across multiple states on both sides of the US-Mexico
border. The US state of Texas recorded its largest ever single fire at over 4,000 km$^2$
(Smokehouse Creek fire) in late February and early March that destroyed 130 homes across
the high plains region of the central US (NIFC, 2024b). Two civilians were killed by the flames

Earth System
Science
Data

in the relatively rural area dominated by ranching, over 10,000 head of cattle died, and
damages are estimated to be at least $4.6 million (NOAA, 2024).

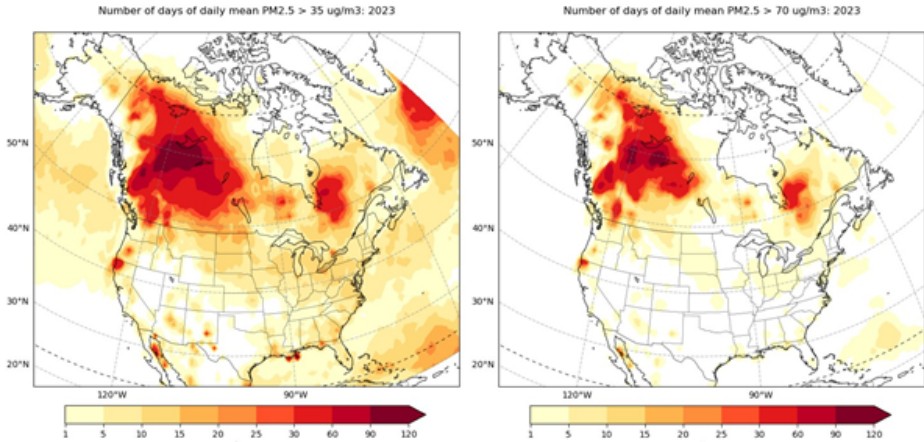

**Figure 7**: Impact of Canadian wildfires visible in air quality metrics for North America. Panels
show the number of days in 2023 with mean PM2.5 concentration over a threshold of **(left
panel)** 35 µg m$^{-3}$ and **(right panel)** 70 µg m$^{-3}$. Both the National Ambient Air Quality Standards
(NAAQS) in the USA and the Canadian Ambient Air Quality Standards (CAAQS) have
exposure targets of 35 µg/m³ on average within a single day.

The impact of North American fires on air quality is significant with half of the PM2.5 in America
suggested to originate from fires (O'Dell et al. 2019). An exceptional fire season, such as seen
in Canada in 2023, therefore poses an elevated level of health risk. Canada's wildfires
produced levels of PM2.5 across the Country that were well in excess of the last 20 years
(**Figure 6**). Additionally, long-range transport of pollution from Canada affected the Pacific
Northwest, northern Midwest, and many eastern states (**Figure 7**). According to the National
Ambient Air Quality Standards (NAAQS) in the USA, the threshold for PM2.5 exposure is not
to exceed 35 µg/m³ on average within a single day. CAMS analysis suggests that people living
within over half of US states experienced up to 2 weeks of exposure at or above this level. In
Canada, the safe limit for PM2.5 exposure, as defined by the Canadian Ambient Air Quality
Standards (CAAQS), is also 35 µg/m³ over a 24-hour period. However, the situation was worse
in Canada due to closer proximity to fires, with many territories along the border experiencing
up to a month of degraded air quality that exceeded national recommended exposure limits,
British Columbia possibly facing twice the number of days at up to 2 months, and the Northern
Territories potentially with 3 to 4 months of exposure.
The scale of the impact becomes particularly evident when comparing the number of days at
double the exceedance level (70 µg/m³) due to short-range versus long-range transport of
pollutants. In Canada, where the pollution sources were more localised, the number of days
above this higher level remains substantial, ranging from a week along the border to months
still at the fire epicentres. In contrast, the USA, affected primarily by long-range transport from
Canada, experienced approximately half the number of such high-pollution days. It is also
important to consider the context of interannual variability in fire occurrence. Last year the
USA experienced its lowest number of fires in 2 decades (**Figure 4**), so most of the pollution
impacts came from Canada. Consequently, if the USA had also experienced a high fire year,
the air quality would have worsened in many states.





### 3.1.3.5 Oceania

As is commonly the case, there was a marked latitudinal difference in wildfire patterns in Oceania in 2023-24. Fire activity was above average in the savannahs, grasslands and shrublands of tropical, subtropical and arid northern Australia. In contrast, fire activity in the southern states of Australia was generally below average, and well below the levels seen during the high impacting 'Black Summer' fires of 2019-20. In New Zealand and the Pacific Islands, fire activity was relatively low compared to the preceding two decades.

Given the vast scale of savannah fires, 2023-24 ranked among the top five years in BA for Australia as a whole since 2001 (Fisher, 2024; **Figure 2**). Fire in tropical and arid areas is tightly linked to rainfall in the preceding season (Alvarado et al., 2020). The above average fire seasons in the Northern Territory and northern Western Australia were driven to a large extent by elevated fuel growth associated with the La Nina conditions of the previous three years. These fires represented the vast majority of areas burned across the country in 2023-24 (**Figure S7)**.

In the monsoonal north, savannah fires follow a strong seasonal pattern, with regular summer rain predictably followed by fire in the dry winter and spring months. In arid regions further south, fire remains tightly coupled to rain, but the seasonality is less pronounced. Anomalously large fires began as early as May and June in Western Australia and the Northern Territory respectively, continuing to as late as January.

The year was also marked by a series of early season, high impact fires in populated areas of southwestern Western Australia, southeast Queensland, NSW, Victoria and Tasmania. Hot, dry, windy conditions, and extended dry periods, are a major driver of forest, woodland and shrubland fires of the subtropical and temperate south of Australia (Collins et al., 2022). In addition to 2023 being the eighth-warmest year on record, the three months from August to October were the driest in over 100 years of records (Bureau of Meteorology).

From October to January a string of fire events led to loss of life, property and a range of other human and environmental impacts throughout the country's southeast and southwest. In some cases, significant fire activity was observed in areas impacted by the 2019-20 fire season. Despite these impacts, average rain in southern and eastern parts of Australia tempered fire activity for the austral summer. In Queensland and NSW, large fires in remote areas pushed the total BA in line with the long term mean, but this figure was well below average in Victoria, the Australian Capital Territory and Tasmania.

In the southwest of Australia, a volunteer firefighter was killed while responding to a fire near Esperance. The Kings Park fire in October occurred in a popular tourist area containing vulnerable flora and threatened Perth Children's Hospital. Perth was again affected by fires in December, with several injured and five homes lost. A further two homes were lost in the region in mid-January from fires that burned 60 km². A similar sized fire burned through rugged terrain in the Gammon Ranges 600 km north of Adelaide, threatening highly significant cultural and ecological values.

A large number of significant fires affected the eastern States of Australia in October. The Tara and Mount Isa fires in Queensland burned well over 400 km² combined, destroying 65 homes and claiming the lives of two people. International and interstate support were deployed from New Zealand and Victoria to respond to the fires. Further south in New South Wales, significant fires included the Coolagolite Rd fire in Bega (over 70 km², 2 houses destroyed), the Willi Willi Rd fire in Kempsey (over 290 km$^2$, 8 houses destroyed, one person killed) and large fires around Tenterfield (approximately 300 km², four homes destroyed). In November the Hudson Fire burned 228 km², destroyed 4 properties and led to the death of a volunteer fire fighter, who was killed by a falling tree while fighting the fire. In December the



Duck Creek Pilliga Forest fire burned 1,385 km² and initiated 3 documented fire-generated
thunderstorms, with smoke impacts extending 500 km away and reaching Sydney.
In neighbouring Victoria the fire season was bookended by high impact events in October and
then in February and March. Fires in Gippsland during October totalled 120 km$^2$ and exhibited
some overnight fire runs that were regarded as unusual (Mills et al., 2022). An extended dry
period saw fires impacting towns in central and western parts of the state in late February and
early March. Over 40 homes were lost and five firefighters were injured fighting two fires that
originated in the Grampians National Park and burned 60 km$^2$. Interactions with the
atmosphere and topography were suggested to explain extreme behaviour that was reported.
This fire was followed by another near Ballarat affecting grass, forest and a pine plantation.
Despite several extreme fire weather days and evacuation advice, a significant suppression
effort aided by interstate deployments minimised impacts. The fire burned over 200 km$^2$.
In the island state of Tasmania the fire season began early with the Coles Bay Bushfire burning
27 km$^2$ of both private land and national park in September and then fires on Flinders Island
in October. Other impactful fires that occurred during the season include the Dolphin Sands
fire on the east coast of Tasmania that destroyed two homes and burned 2.5 km$^2$ and the
multiple fires in the Brady lake area (Tasmania's central highlands) in February that destroyed
two homes and burned up to 100 km$^2$ and a fire in the Waterhouse Conservation areas that
required campers to evacuate.
New Zealand experienced a normal fire season after three well below average seasons prior
under La Nina conditions. The fire season began early with a relatively large fire in September
on the western side of Lake Pukaki in the central South Island in wilding pines. This fire totalled
29 km$^2$ and this was the third major wildfire event in recent years in this area at an earlier than
normal stage of the fire season, following the 2020 Pukaki (Aug.) and Ohau (Oct.) fires. New
Zealand then experienced a spate of fires around Canterbury, on the South Island between
late Jan. and mid Feb. 2024 with several houses burned and farmlands affected.
**3.1.3.6 South America**
The 2023-24 fire season in South America was characterised by a moderately below average
fire activity but with positive wildfire anomalies in specific regions, which were exacerbated by
extended periods of drought and heatwave across the continent (Clarke et al., 2024; **Figure**
**2, Figure 3, Figure 4**). In the Brazilian State of Amazonas, which features the largest extent
of preserved old growth forests in Amazonia, June and October 2023 saw the highest fire
counts since records began in 1998 (National Institute for Space Research, 2024; see also
**Figure 8**). This continues a recent trend towards record-setting months for fire in Amazonas
state, with new maxima being set in 7 months of the year since 2019 (National Institute for
Space Research, 2024). Recent changes in deforestation and land use patterns are
contributing to elevated fire ignitions in the state, amplified by climatic extremes in 2023 by a
historic drought and heatwave driven by El Niño (Espinoza et al., 2024, Clarke et al., 2024).
Due to emissions of wildfire smoke, many areas of Amazonas experienced poor air quality
from September to December 2023, including in the state capital, Manaus, where over 2
million people were exposed to the second-worst air quality in the world in October (Ministério
Público Federal, 2023). The event was so severe that, in November 2023, the Federal Public
Ministry opened a Civil Action case against the State of Amazonas, demanding evidence that
the State was investing in fire prevention and combat in line with the Plan for the Prevention
and Control of Deforestation and Fires (Estado do Amazonas, 2020). This procedure
evaluates whether Amazonas authorities are accountable for environmental damage causing
severe air pollution, reflecting the Public Ministry's growing involvement at both federal and
state levels in monitoring environmental degradation and seeking to make authority figures
accountable (Ministério Público Federal, 2023).



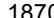


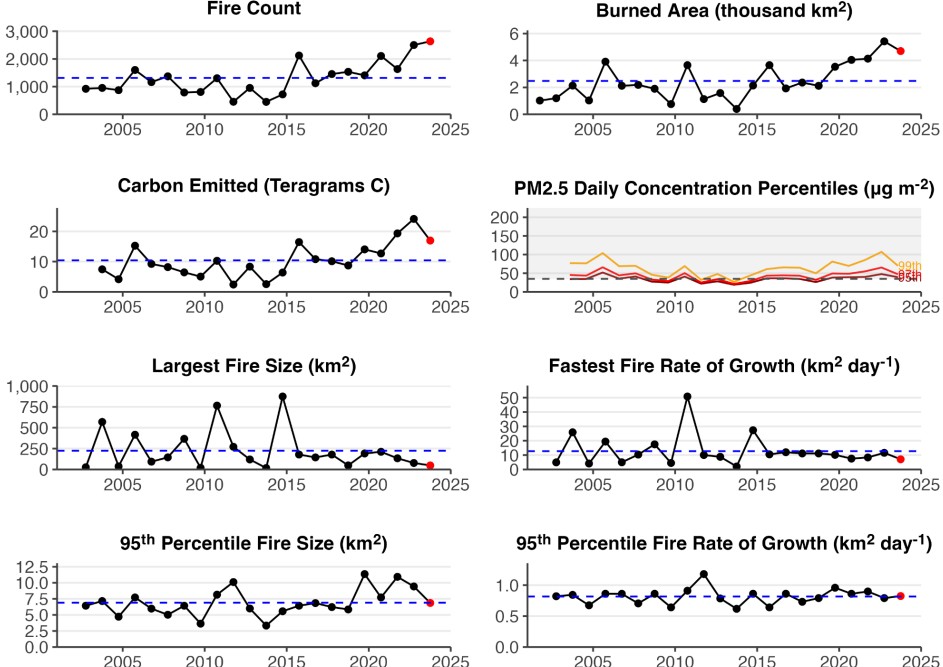

**Figure 8:** Summary of the 2023-2024 fire season in the Brazilian state of Amazonas. Time
series of annual fire count, BA, C emissions, PM2.5 emissions, 95th percentile fire size, fastest
daily rate of growth, and 95th percentile fire daily rate of growth. Black dots show annual
values prior to the latest fire season, red dots the values during the latest fire season, and blue
dashed lines the average values across all fire seasons.
National fire monitoring systems in Brazil indicate that some areas of Amazonia experienced
anomalies in BA at the sub-state level. For example, BA in the municipality of Santarém in the
State of Pará rose from an average of 70 km$^2$ in 2019 to 2022 to over 1,000 km$^2$ in 2023, and
has already exceeded 250 km$^2$ in 2024 (MapBiomas Brasil, 2024). Similarly, in neighbouring
Belterra municipality, BA extent was more than 3 times greater during the year 2023 than in
2019-2022 (MapBiomas Brasil, 2024). In Floresta Nacional do Tapajós, one of the most
studied forest sites in the Amazon which spans Satarém and Belterra, forest fires accounted
for more than 60% of the burned areas (MapBiomas Brasil, 2024). 4 thousand people live in
24 communities of traditional and Indigenous populations in the region and depend on
protected forest resources for their cultural heritage, food security, economy and livelihood in
Floresta Nacional do Tapajós and Reserva Extrativista Tapajós-Arapiuns (Instituto Chico
Mendes de Conservação da Biodiversidade, 2019). Fires in 2023 compounded the challenges
faced by these communities, who were already isolated by low river levels resulting from the
drought that severely reduced their mobility and fishing, impacting on food security and
enhancing socio-economic vulnerabilities.
In Chile, the 2023-24 fire season was marked by a significant escalation in both the number
and size of wildfires, especially in the central and southern regions (**Figure 4**). Chile
experienced its second-highest BA in the past 20 years (>4,000 km$^2$; Jones et al., 2024). The
peaks in BA at national scale were accompanied by peaks in the 95th percentile of fire size



and daily rate of growth in highly-populated regions such as Valparaíso (**Figure 2, Figure 4**),
indicating unusually large and fast-moving fires. These fires drew international attention due
to their deadly impacts on society. In February 2024, severe wildfires struck the Valparaíso
region in Chile, particularly affecting Viña del Mar and other surrounding areas (NASA Earth
Observatory, 2024). These fires, fueled by a prolonged heatwave, resulted in the deaths of at
least 131 people and destroyed thousands of homes, leaving at least 1,600 people homeless
(UN Resident Coordinator in Chile, 2024; *Al Jazeera*, 2024; El *Disconcierto*, 2024). The
emergency impacted the Lago Peñuelas National Reserve where more than 60 km$^2$ of forest
were affected (Oberholtz, *Fox Weather*, 2024). The National System for Disaster Prevention,
Mitigation and Attention (SENAPRED) issued a red alert and ordered the evacuation of over
18 nearby towns (Oberholtz, *Fox Weather*, 2024). The February 2024 wildfires in Valparaíso
followed other major disruptive wildfires in February 2023, which affected nearby regions of
central Chile including Maule, Nūble, Bio bío, La Araucanía and Los Rios.
Several countries with land in the west of Amazonia experienced anomalies in BA and fire
behaviour during the 2023-24 fire season, which correlated with specific regional climatic and
environmental conditions. Bolivia, Peru, Ecuador and Colombia did not experience particularly
extreme levels of BA at national scale, possibly influenced by above-average rainfall over
three consecutive years of La Niña between 2020 and 2022 (iMMAP Inc., 2023). However,
Peru's Loreto region, which neighbours the Brazilian state of Amazonas, faced its highest BA
on record in the 2023-2024 fire season signalling the wider impacts of drought conditions in
western Amazonia (**Figure 2**). The timing of peak anomalies in BA also coincided with those
in Amazonas, around September-October 2023 (**Figure S2**). The northern Bolivian
departments of La Paz and Beni experienced similarly-timed anomalies in BA. Remote parts
of the Colombian Amazon also saw a significant uptick in BA since November 2023 which
peaked in January 2024 (Mongabay, 2024). As a result of months of record-high temperatures
and drought conditions since the beginning of El Niño, the region recorded higher C emissions,
reflecting the severity of the burning at the end of the studied fire season. While the direct
impacts on society throughout these regions was not as pointed as in the case of fires in Chile,
the events are likely to have contributed to reductions in regional air quality and also impacted
forest ecosystems and raised C emissions from fires in South America. At the end of January
there was a wildfire in the mountains surrounding Bogota that affected the air quality that
affected thousands of citizens of the capital of Colombia (France-Presse, *VOA News,* 2024).
In 2023-2024, Venezuela experienced its highest level of fire activity on record, particularly in
January and February 2024 (ALER, 2024; *Tiempo*, 2024; **Figure 4, Figure S2, Figure S8**),
notably affecting the states of Anzoátegui, Cojedes, Guárico, and Monagas, areas which
dominant land cover primarily consists of savannas and extensive grasslands. This surge in
fires during the dry season was intensified by unusually warm and dry conditions in the
preceding months. These conditions, likely a result of global warming and changes in
circulation and rainfall patterns associated with El Niño, making the landscapes more
vulnerable to fires.
**3.1.4   Context of Recent Extremes**
The anomalies of 2023-24 occur against a backdrop of trends in BA this century that point
towards shifts in fire regime. **Figure 9** shows significant trends in BA and forest BA across the
fire seasons in the period March 2001-February 2024 derived from MODIS BA data (see
Methods). While many world regions are experiencing declines in total BA, increases in forest
BA are far more prevalent than declines at the scale of continental biomes, countries, and
administrative regions.
Northern hemisphere extratropical biomes in North America and Asia show a clear signal
towards increased forest BA since 2001, which are also visible on national and provincial
scales in Canada, the US and Russia and on provincial scales in various states of western



and eastern Canada, the western US, and Siberia. These trends occasionally propagate to
trends in total BA, for example in western and northern Canada and in the Sakha Republic
(eastern Siberia). The large 2023-24 anomalies in BA in Canada align with the doubling of
forest BA in Canada across fire seasons since 2001 (a significant trend, $p < 0.05$) and a 23%
increase in total BA in Canada (marginally significant at $p < 0.1$). Three Canadian provinces
showed significant increases in both total and forest BA this century: British Columbia (+35-
42%); Northwest Territories (+55-68%), and; Yukon (+60-135%). No Canadian provinces
experienced a significant decline in forest BA or total BA. More widely, there was a 58%
increase in forest BA in the North American boreal forest biome since 2001, and a 134%
increase across the pan-boreal forest biome of North America and Eurasia. The succession
of events affecting boreal forests in Canada in 2023, Siberia in 2020, and both North America
and Asia during 2021 appear to be part of a continued trend towards rising fire extent in the
high latitudes this century.
Elsewhere in the southern hemisphere extratropics, significant rises in forest BA are seen in
Chile since 2001 (+87%), including in the central regions of Araucanía, Bio bío, Maule, Ñuble,
and Valparaíso ranging from 35 to 109%. Extreme fires in Valparaíso during 2023-24 and in
Araucanía, Biobío and Ñuble in the 2022-23 fire season Maule, Nũble, Bio bío, Araucanía
(**Section 3.1.3.6**) are also consistent with a longer-term rise BA in central Chile (**Figure 9**).
Increases in BA are not generally significant in fire-prone parts of the southern hemisphere
extratropics, such as Africa or Australia.
In the tropics, trends in total and forest BA show a variety of patterns. While total BA has
reduced across much of the savannah-occupied regions of South America, Africa and northern
Australia, trends in forest BA (>30% tree cover) are far more varied (**Figure 9**). Hence, fires
in woody tropical vegetation show a less consistent global trend. In addition, exceptions to the
general decline in total BA across the tropics are seen in the Brazilian state of Amazonas, the
Congo basin, and across much of India (**Figure 9**). The trend in Amazonas, among the most
pristine parts of Amazonia, contrasts with other states of Brazil such as Mato Grosso and
Pará, where declines in deforestation rates and deforestation-related fires have fallen since
their peak during the early 2000s (Silva Junior, 2020). The anomalous fire activity and C
emissions in Amazonas state during the 2023-24 fire season (but not other states of Brazil)
thus appear to be consistent with the emerging pattern of increased fire in the region.
Meanwhile, the 2023-24 anomalies in BA and other fire properties in the Bolivian, Peruvian,
Colombian and Venezuelan parts of Amazonia (**Section 3.1.3.6**) typically occurred against a
backdrop of reduced BA or no significant trend in recent decades (**Figure 9**).

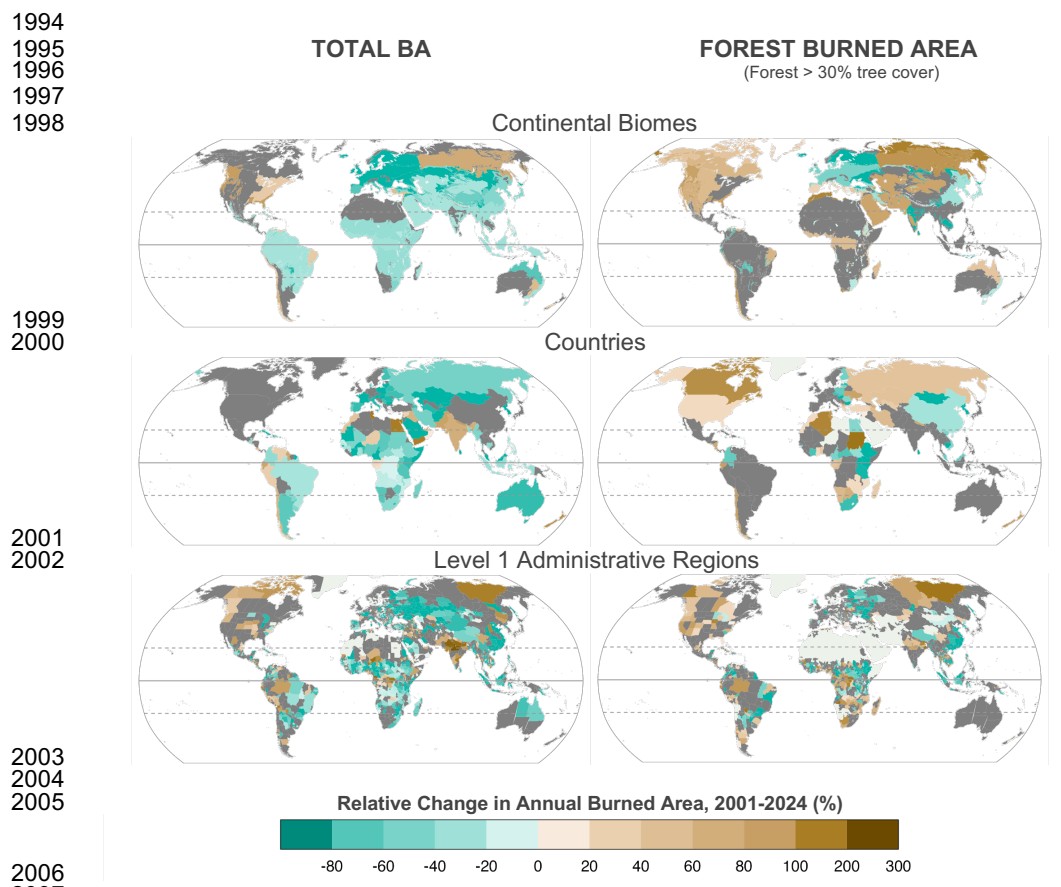

**Figure 9:** Relative changes (%) in **(left panels)** total annual BA and **(right panels)** forest BA across March-February fire seasons during 2001-2024 for three regional layers: **(top panels)** continental biomes; **(middle panels)** countries, and; **(bottom panels)** level 1 administrative regions (e.g. states or provinces). Forest BA considers only areas with tree cover over 30% at the native (500 m) resolution of the BA observations. Relative changes are calculated as the trend in BA across fire seasons March 2001-February 2022 through March 2023-February 2024 multiplied by the number of years in the time series and divided by the mean annual BA during the period. Trends in BA are derived using the Theil-Sen slope estimator. Only significant trends in BA are shown (dark grey fill signifies no significant trend).

## 3.2   Focal Events of this Report

### 3.2.1.1  *Canada*

In this year's report, the extreme wildfire season in Canada is selected as one of our focal events. It emerges as a major event of global relevance for the following reasons:

- **Record-Breaking Burned Area:** The North American boreal forests, particularly in Canada, experienced an unprecedented fire season. The BA reached six times the average since 2001.



- **High C Emissions:** Fire C emissions in Canada were over nine times the average since 2003, contributing significantly to global C emission totals for the year.
- **Early and Persistent Fires:** Positive BA anomalies were visible from April 2023 (**Figure S9**) and persisted through to October, with some regions experiencing fires until January 2024. The fire season lasted nearly a month longer than normal, with the largest one-day BA total ever recorded in Canada occurring on 22nd September 2023.
- **Regional Anomalies:** Peak fire anomalies were observed in Eastern Canada in June 2023 and later in Western Canada (August-September), indicating widespread and prolonged fire activity across the country.
- **Record Fire Size and Spread:** New records for individual fire size and rate of spread were set, with many provinces experiencing high-ranking anomalies in fire count and daily growth rates.
- **Extensive Impact Across Provinces:** The highest BA or fire C emissions on record were observed in Northwest Territories, British Columbia, Alberta, and Quebec, with other provinces like Yukon, New Brunswick, and Ontario also experiencing significant fire activity.
- **Air Quality Impact:** Smoke from these fires led to severe air quality issues, affecting major cities in North America, including New York, which experienced its worst air quality in half a century.
- **Firefighting Challenges:** Canada was at its highest National Preparedness Level for an unprecedented 120 continuous days, indicating the significant resource sharing and international assistance required to manage the fires.
- **Human and Economic Toll:** Over 232,000 people were evacuated across various regions, and despite the extreme fire activity, no civilian deaths were directly attributed to the fires, showcasing the effective, albeit strained, emergency response efforts.
- **Global Significance:** The extreme fire season in Canada was a major contributor to the overall above-average global fire C emissions in 2023-24, highlighting its global environmental impact.

To assess the causes of specific regional BA anomalies, four anomalous BA regions/month combinations were chosen across Canada: Western Taiga Shield and Taiga Boreal Plains for May and June (includes Alberta and British Columbia Boreal plains, and the Mountain Cordillera); and Eastern Taiga Shield in Quebec for June and July. **Figure 10** maps the magnitude of anomalies in these regions and months. Though note the size and long period this protracted event means that even these regions/month to not capture all the anomalous BA over Canada in 2023 (**Figure S9**).

### 3.2.1.2  *Greece*

In this year's report, the extreme wildfire season in Greece is selected as one of our focal events. It emerges as a major event of global relevance for the following reasons:

- **Second-Highest BA on Record:** Greece experienced its second worst fire season in terms of total area burned, with 1,727 km$^2$ affected, despite recent efforts to strengthen firefighting mechanisms. The 2023 fire season was notably more severe than typical years, with the total BA significantly exceeding the country's historical averages and recent challenging fire seasons.
- **Multiple Large Fires:** From mid-July to late August, Greece faced numerous large fires that overwhelmed firefighting capabilities. Key fires included those on the island of Rhodes, which burned 207 km$^2$, and the massive Evros fire, which reached 938 km$^2$.
- **Evros Fire Disaster:** The Evros fire became the largest on record in recent European history, significantly impacting both forested and agricultural areas. It also led to the tragic deaths of 19 immigrants who were trapped by the flames.
- **Urban and Infrastructural Impact:** Fires near populated areas necessitated large-scale evacuations, including 20,000 tourists on Rhodes and multiple settlements





around Mount Parnis in Attica. The Evros fire also caused a powerful explosion at an
Air Force base, resulting in damage to the town of Nea Aghialos.
● **Significant Evacuations:** Numerous evacuations took place, highlighting the severity
of the situation. These included evacuations in Alexandroupolis and its surrounding
villages due to the Evros fire.
● **Economic and Environmental Damage:** The fires caused extensive damage to
properties, infrastructure, and natural reserves, with significant impacts on biodiversity
and local economies.
● **Firefighting Challenges:** The simultaneous spread of multiple fires stretched
firefighting resources to their limits, with a notable focus on evacuations rather than
fire suppression in some instances.
Abnormally high burned areas were reported around Alexandroupolis in August and extended
further west across the administrative region of Macedonia and Thrace. Anomalies were also
present in central Greece and around Athens in July and August (**Figure S10**). **Figure 10**
maps the magnitude of the anomalies for August.

### 3.2.1.3  Western Amazonia
Our final focal event of 2023-24 is a box drawn in western Amazonia with bounding
coordinates 2.25° N, -56.00° E, -9.75° S, -77.75° W. It includes Amazonas (Brazil), Loreto
(Peru), and La Paz and Beni (Bolivia) where peak fire anomalies occurred simultaneously and
coincided with a historic drought and heatwave. It emerges as a major event of global
relevance for the following reasons:
● **Record-Setting Fire Activity:** The 2023 fire season in Western Amazonia saw
unprecedented fire counts, with new records set across Amazonas state in Brazil,
Loreto department in Peru, and La Paz and Beni departments in Bolivia.
● **Severe Air Quality Degradation:** Smoke from widespread fires led to significantly
degraded air quality across the region, impacting millions of people and posing serious
public health risks.
● **Broad Socio-Economic and Health Impacts:** The fires caused extensive socio-
economic disruptions, including health issues from poor air quality, legal actions for
inadequate fire prevention, and impacts on livelihoods, particularly for Indigenous and
traditional communities.
● **Widespread Environmental Degradation:** The fires contributed to significant C
emissions and environmental degradation, affecting forest ecosystems and increasing
the region's vulnerability to future climatic extremes. Western Amazonia has global
significance due to its critical role in C storage and biodiversity and relatively low levels
of disturbance.
● **Impact on Indigenous and Traditional Communities:** Fires in key areas disrupted
the lives of Indigenous and traditional populations, exacerbating their vulnerabilities
due to isolation and reduced access to resources.
Abnormally high burned areas were reported in Western Amazonia during September and
October. **Figure 10** maps the magnitude of these anomalies. Some anomalous BA starting in
August and extending to November (**Figure S11**).




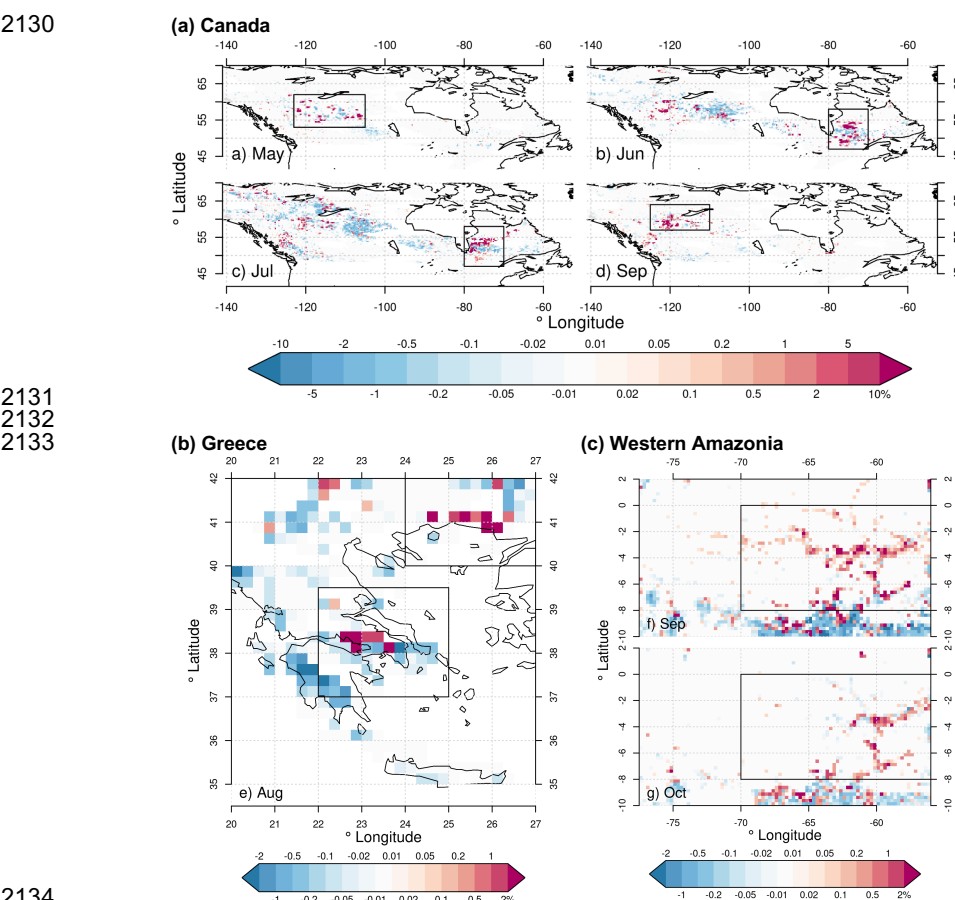


**Figure 10:** Spatially explicit anomalies in BA fraction (%) during key months for focal events
in **(a)** Canada, **(b)** Greece and **(c)** Western Amazonia. Plotted data are the absolute change
from the climatological mean BA fraction for the month (%), based on MODIS MCD64A1
aggregated to 0.25°. Red indicates higher BA in that month of 2023 vs the 2002-2022
climatological average for that month. Boxes indicate focus events for our analyses in this
report. The top panels show anomalies in Canada for various months, the lower-left panel
shows anomalies in Greece for August, and the lower-right panels show anomalies in Western
Amazonia during September and August.

## 3.3   Diagnosing Drivers and Assessing Predictability
### 3.3.1   Highlights
**In Canada, fire weather conditions were the primary drivers of the extent of most fire**
**activity.** However, instances of intense burning events, particularly in mid-July, early August,
and late September were missed by the driver attribution analysis, resulting in 'others'. This,
highlighted potential inadequacies in predicting certain ignition sources or accurately
representing fire propagation across vast landscapes in current forecasting systems.



**The exceptional nature of the event in Alexandroupolis, Greece, could not have been predicted using fire weather forecasts.** While there were discernible indications in the Fire Weather Index (FWI) records that the days around the event were more extreme than most days in the fire season, similar conditions were also observed in the last 10 days of July without resulting in the same catastrophic impacts. This underscores the importance of expanding early warning beyond fire weather, considering fuel availability and ignition variability, to enhance reliability in fire risk assessment.

**The extreme fire season in South America was driven by prolonged drought conditions linked to the strong El Niño**. Many fires developed, triggered by lightning ignitions early in the season amidst high FWI values. Other than weather conditions that acted as persistent controls for fire activity, several intense active fire periods were not predicted in late August and throughout September likely due to the result of unrepresented ignition sources.

**Extreme burned areas were driven by anomalies in multiple controls, with weather, fuel abundance, and moisture being the most relevant factors.** The synchrony of all three factors resulted in the most severe BA anomalies in the three highlighted events. This underscores that no single factor can explain the most severe fires. Instead, multiple contributing controls must often coincide for the most extreme events to occur.

**Fuel load is a critical modulator for burned extent**. Higher and.or dries fuel loads combined with high fire weather conditions determined the unprecedented extent of burning in Canada and Western Amazonia. Conversely, the boundaries of extreme fires in Canada and Greece often corresponded to areas with lower fuel loads, demonstrating that discontinuity in fuel availability influenced fire behaviour and may have created firebreaks.

### 3.3.2  Predictability of Focal Events using Fire Weather Forecasts

#### 3.3.2.1  Canada

The early establishment of fire weather conditions as well as the late cessation of the fire season are well captured in the FWI reanalysis in Canada (**Figure 11**). The FWI also captures the intermittent pattern of fire danger and its correlation to actual fire activity. However, at the seasonal time scale, the signal is weakened, and there were no prior indications that the upcoming season would have been as extreme as it was with respect to fire activity (**Figure 11)**. For most part of 2023 Canada was in drought. The analysis of the other subindices of the FWI system, notably the drought code shows more persistent patterns in fire weather with longer predictability lead times (not-shown). The weather-limited nature of the Canadian fire season means that the FWI modelling framework serves as an essential indicator of anomalous conditions, acting as a prerequisite for the intensity and spread of fires. It provides valuable insights into the sequence and extent of extreme fire weather days during such events.

# Fire Prediction & Activity

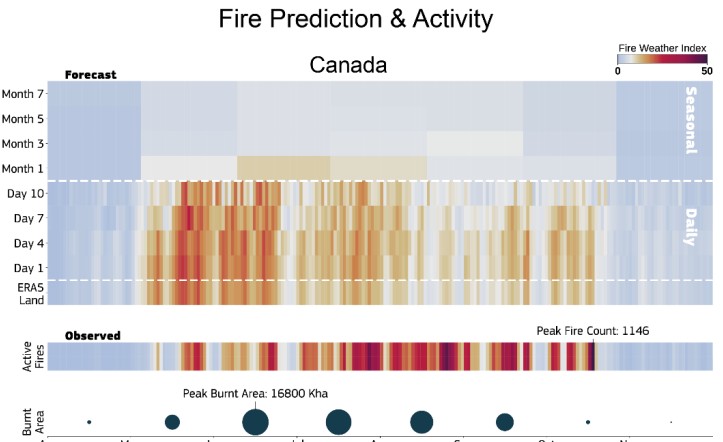

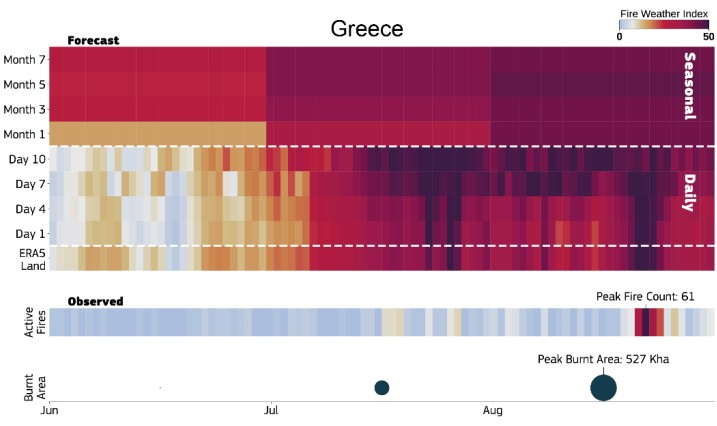

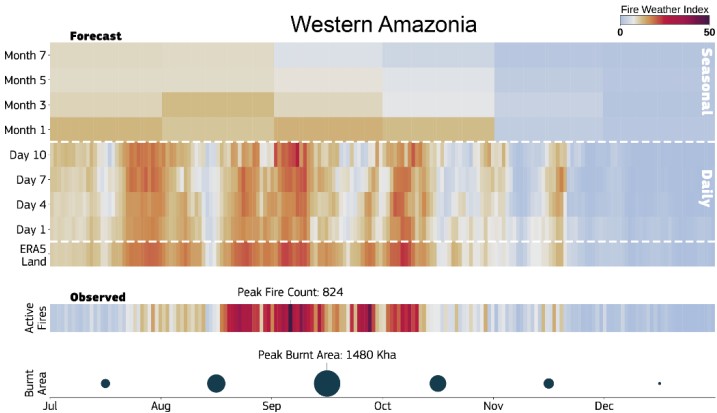

**Figure 11:** Chicklet plots displaying seamless FWI predictions over time from various
forecasting systems of the ECMWF (see Methods). The x-axis corresponds to specific dates
throughout the year, while the y-axis denotes either observations or the time leading up to the





date when a forecast was generated. The vertical colour coherence allows for quick identification of the time windows of predictability associated to the observed fire activity both provided in terms of number of detected active in a day fires and total burned areas in a month (circles).

### 3.3.2.2 *Greece*

The establishment of fire-prone conditions in the Mediterranean, particularly in Greece, is part of the region's seasonal weather cycle (**Figure 11**). In 2023, this pattern persisted, and extreme landscape flammability could be forecasted well in advance. In arid and semi-arid regions fire occurrence is driven not solely by weather but also by fuel availability and its intermittent short-term drying. In these regions the FWI often reaches extreme levels for much of the summer. However, fires do not always occur even when the FWI is extreme, as ignition and early suppression play a crucial role. The anomalous event in Alexandroupolis, highlights the limitations of relying solely on fire weather indices in these areas. There were no discernible indications in the FWI records that the particular day was more extreme than the day before or the one afterward, emphasising the need for a more holistic approach to fire risk assessment in regions where fuel is a limiting factor or live fuel moisture plays a crucial role in the extent of the burnings (Di Giuseppe, 2023).

### 3.3.2.3 *Western Amazonia*

The correlation between FWI and fire activity in the South America region at the shorter lead times is generally poor, primarily due to the strong dependency on either lightning or human ignitions (Kelley et al., 2021). In 2023, this pattern persisted (**Figure 11)**, with the onset of fire weather following the seasonal pattern well ahead of the time where fires were triggered. Seasonal predictions indicated high fire danger during the summer period probably driven by El Niño conditions.

### 3.3.3  Identifying Key Drivers of Focal Events

Weather, fuel moisture, fuel abundance, and ignitions are the four primary controls identified as influencing the occurrence and intensity of the focal fire events. Anomalies in individual drivers of these controls, such as temperature or soil moisture, are calculated by comparing regional daily 2023 averages with the average for 2003-2022.

Analysing the time series of a few key drivers helps contextualise the conditions under which the examined events took place (see **Figure 12**). However, it is by leveraging the infrastructure provided by PoF and ConFire models that we can perform a statistical causality attribution of the four controls for the observed fire occurrence (see **Figure 13**). By design, the PoF and ConFire models provide control attribution even if a fire event is not recorded. In such cases, a low probability across all controls indicates an accurate prediction. However, even in instances where no fire event is recorded, the models enable us to gain insight into the complex interactions among weather conditions, fuel characteristics, ignition sources, and other environmental factors. For example, a high probability of fire not associated with any AF could indicate successful suppression or fire-prevention policies implemented at a local level. Moreover, as our driver selection may not fully account for human influence, it is important to note that the category of "other" also encompasses unaccounted-for variables, as well as a measure of observed active fires and partially burned areas that were not forecasted by the models. Therefore, this analysis not only enhances our ability to understand the main controls on fire activity, but it also helps identify missing pieces of information needed to fully comprehend the events. It is important to note there is not a one-to-one correspondence




between observed fire activity and the attribution analysis, as the analysis is based on model
predictions alone.

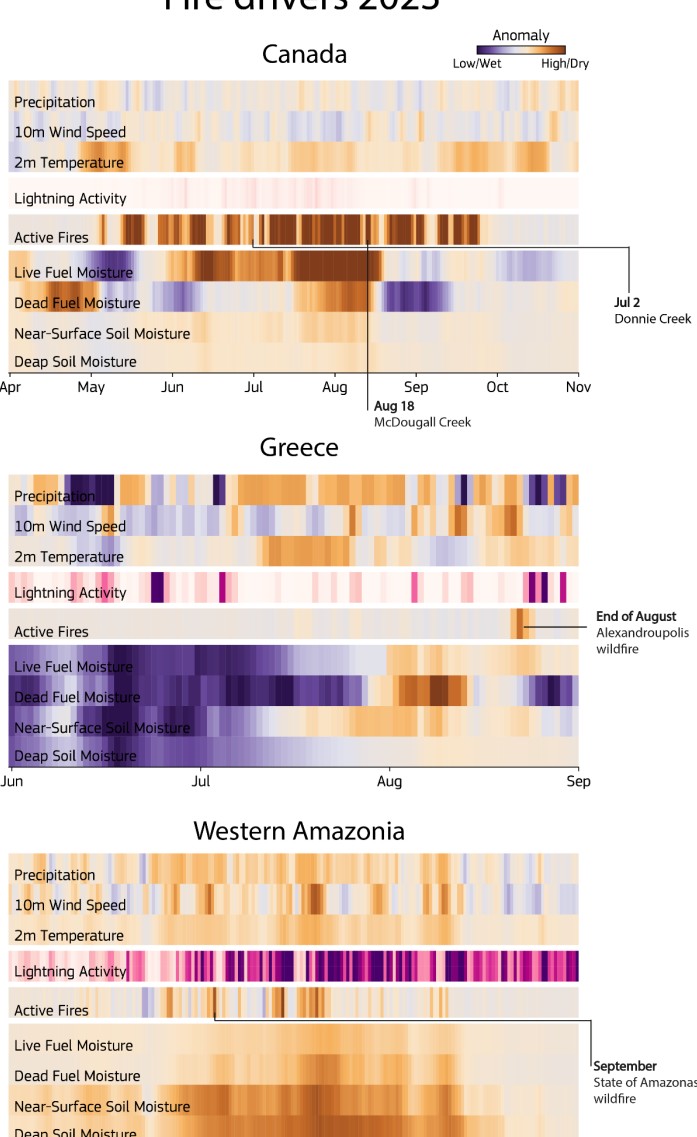

**Figure 12:** Anomaly driver stripes for the three focal events. The drivers are selected to
contextualise the conditions under which the examined events took place. All values are
expressed as anomalies compared to the 2003-2021 climatology with the exception of lighting
activity which is expressed as absolute flash density.





### 3.3.4 Drivers of Active Fire Extremes

#### 3.3.4.1 *Canada*

Persistent fire-favourable weather conditions played a crucial role in controlling the extent of active fires in Canada during the summer of 2023. Dry weather contributed to extensive drying of both live and dead vegetation, further exacerbating fire risk (**Figure 12, 13**). Most of the explainability of the Canada event comes from anomalous weather conditions. Increased lightning activity often coincides with or precede significant fire periods, indicating lightning as a key source of ignitions in the region, in agreement with the attribution of 59% of wildfires and 93% of total BA to lightning ignition sources in Canada during 2023 (Jain et al., 2024). Adverse weather conditions in mid-May in western Canada were identified as influential factors in shaping fire events. However, multiple instances of intense burning events, notably in mid-July, early August, and late September, fall into the 'Other' category, heavily contributing to the total number of events for which there is no attribution among the controls. The fact that clusters of events were not predicted, suggests potential inadequacies in accounting for some ignition sources or accurately representing fire propagation across vast, densely vegetated landscapes.

#### 3.3.4.2 *Greece*

The driver anomalies (**Figure 12**) and control attribution (**Figure 13**) do not suggest an abnormally fire-prone year in Greece, failing to justify the extent and severity of the Alexandroupolis fire. An anomalously wet spring may have led to increased foliage and subsequently quick drying of plant material. A sustained dry period in late July and August further dried out new foliage, creating favourable conditions for fire activity, as indicated by the anomalously dry live and dead fuel moisture content in August. Despite these conditions, the unexpected extent and severity of the Alexandroupolis fire were not predicted, highlighting the intrinsic difficulties in forecasting isolated extreme events even when most predictors are included. Additionally, the high wind speeds at the time partially contributed to the extensive BA during the fire.

#### 3.3.4.3 *Western Amazonia*

Prolonged drought conditions, stemming from anomalously low rainfall and high temperatures, created favourable conditions for an active fire season in Western Amazonia (**Figure 12**, **Figure 13**). These conditions had a significant impact on the typically wet ecosystem, affecting soil moisture as well as live and dead fuel moisture. Despite weather conditions serving as a persistent control for fire activity, several intense active fire periods in late August and throughout September were not predicted, possibly due to unrepresented ignition sources. Additionally, fire activity from September onwards was intensified by intense lightning activity, characteristic of the region, which substantially contributed to ignitions.



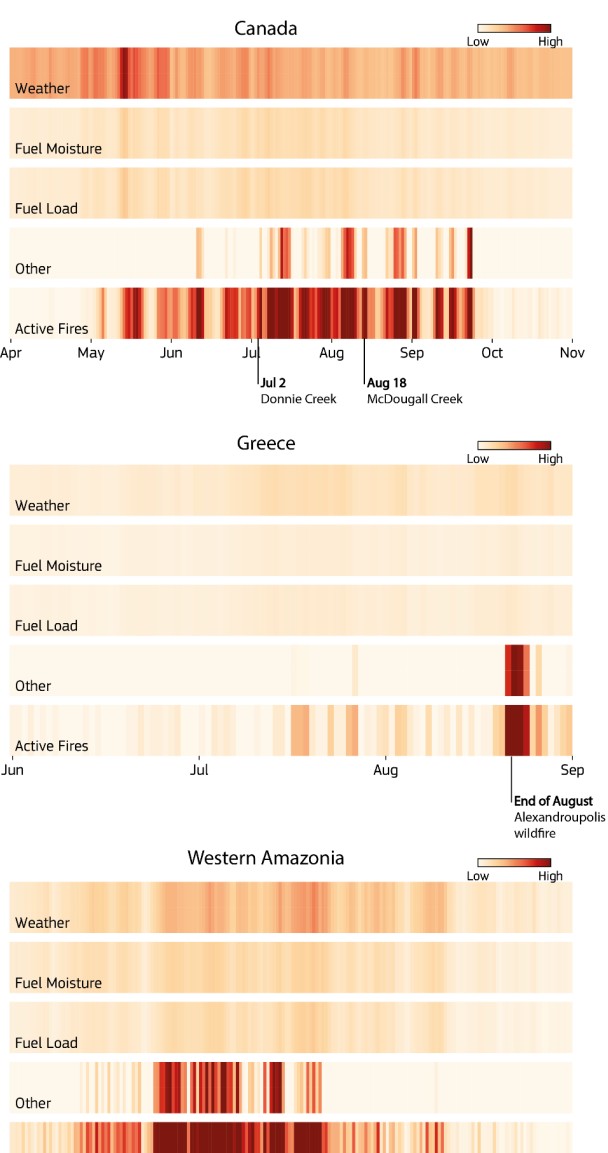

**Figure 13:** Contributions of different fire controls to daily active fire anomalies in our focal events. All values are scaled to the observed daily fire anomaly such that the sum of the 4 daily control values matches the total observed anomaly.



### 3.3.5 Drivers of Regional Burned Area Extremes

#### 3.3.5.1 Canada

ConFire detected a significant anomaly in BA starting in late April, as seen in the observations (**Figure S9**) with very high confidence between May (99.2% likelihood) and June 2023 (>99.9% likelihood; **Figure 14**). While less confident in a positive anomaly in September and August (71% likelihood), ConFire detects the possibility of much higher burning in August and September, corresponding with the increase in burning in the Western Shield (**Figure S9**). **Figure 14** shows the controls that contribute to these anomalies. Our analysis indicates a >99.9% likelihood that elevated fire weather conditions persisting throughout the 2023 fire season, led to a notable increase in burning, explaining between 4.6-45% (based on 10 to 90th percentile confidence range) of the BA anomaly in May and 1-110% in June. If fire weather did explain more than 100% of the BA anomaly, then other controls must have acted to suppress BA. However, the anomalous weather conditions subsided later in the fire season. Drier fuel conditions could have contributed significantly to the increase in BA (up to 65% of Mays brunt area anomaly and 45% of Junes), though with low confidence (60.5% in May and 61.3% in June), and wetter fuels exerting a suppressive effect on fire spread was also possible, suggesting their potential role in mitigating fire severity. A small but confident suppressive effect (100% likelihood in May, 64.8% likelihood in June) from fuel load was observed, reducing relative increases in BA between 1.7-7.1% in May. Direct human-induced landscape changes exhibited a small impact on the extent of burned areas (likelihood of 97.4% in May), explaining between 1.2-22% of Mays increased burning and 0.6-24% of Junes. The analysis revealed relatively low noise levels in the results ("uncertainty measure" in **Figure 14**), indicating a robust signal despite uncertainties associated with the controls. This is shows that the higher burnt arae anomalies have the most confident signal. This robustness is partly attributed to the large spatial coverage of Canada's grid cells, enhancing the signal-to-noise ratio.

#### 3.3.5.2 Greece

The analysis reveals an anomalously high BA, particularly from mid-August onwards, though with a lower confidence level compared to the Canadian case (69.9% likelihood; **Figure 14**). **Figure 14** shows the controls that contribute to these anomalies. Our findings demonstrate with very high confidence the presence of anomalously high fire weather conditions during the 2023 fire season in Greece (98.3% in July, and >99.9% in August and September). In August, these conditions explained between 4.3-140% of the increased BA, escalating to 5.6-170% by September. Assessing the impact of fuel moisture on BA, our analysis shows a wide range of possibilities, with confidence ranging from a 21% increase to 180% decrease in relative BA extent, which would have offset some of the increases from fire weather. This uncertainty underscores the complexity of the interactions between fuel moisture and fire behaviour in Greece. Similar to the findings in Canada, direct human-induced landscape changes exerted minimal influence on the extent of burned areas in Greece during the analysed period. While the analysis indicates a slightly higher than normal fuel load

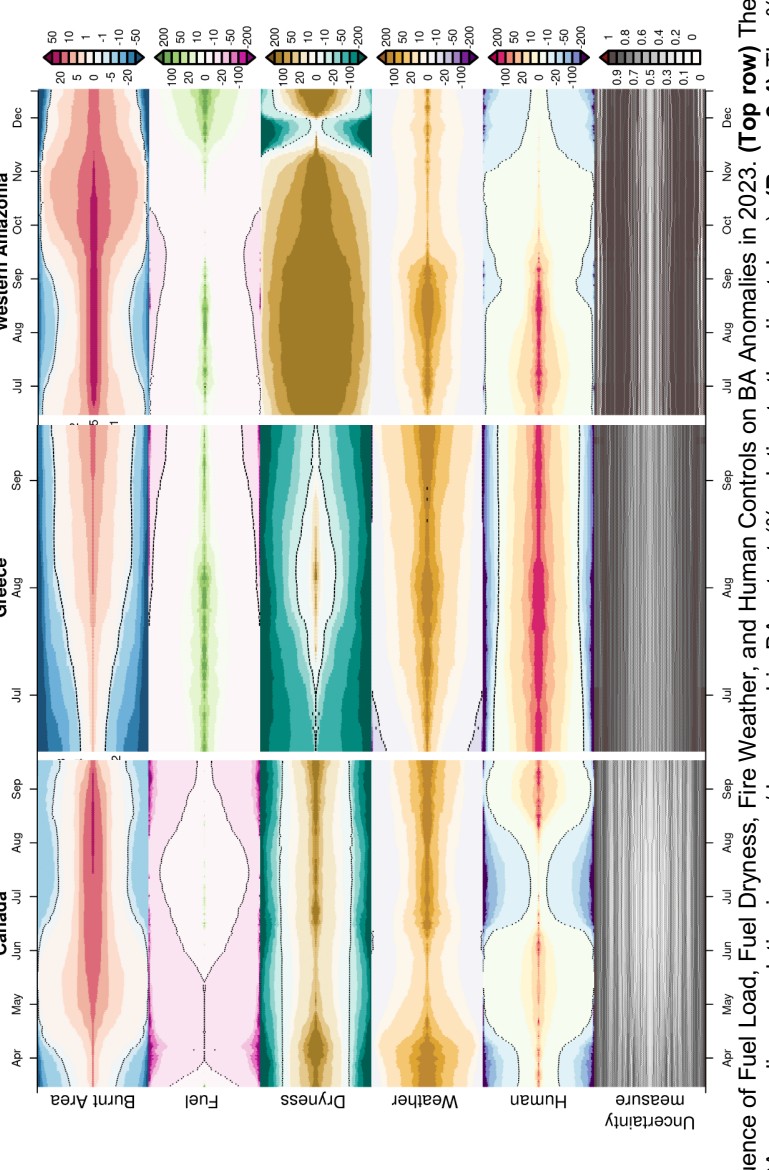

**Figure 14**: Influence of Fuel Load, Fuel Dryness, Fire Weather, and Human Controls on BA Anomalies in 2023. **(Top row)** The colour indicates the modelled BA anomalies as relative increases/decreased in BA extent (% relative to the climatology). **(Rows 2-4)** The % of the anomaly explained by each control. At each timestep, the gradient of colours along the y-axis represent the range of modelled outcomes: high diversity in the colours indicates high uncertainty, whereas a consistent colour (e.g., red for positive BA anomalies) indicates greater confidence in the projected direction and magnitude of change. Dotted black line indicated threshold between positive and negative anomaly. **(Bottom row)** Uncertainty measure representing potential variations around our simulated BA anomaly as a result of factors not considered by the modelling framework, as a fraction of land area. Lower numbers (lighter colours) indicate higher confidence in the BA anomaly in the top row.



### 3.3.5.3 Western Amazonia

Our analysis indicates a reasonably high confidence that the considered drivers contributed to anomalous high burning during September (73.8% likelihood), October (94.9% likelihood) and November (96.1% likelihood; **Figure 14**). **Figure 14** shows the controls that contribute to these anomalies. The primary driver of the observed BA anomaly appears to be dry fuel conditions, with a very high likelihood (99.7%) of drier-than-normal conditions persisting through November. This led to a substantial increase in BA, explaining at least 57% of the increase in BA. While fire weather conditions were also elevated (likelihood >99.9%), their impact on BA was comparatively lower, resulting in at least 2% increase in BA during October and November, though with a small probability of contributing much more. Direct human influence was identified as a contributing factor, with a high likelihood (92.9% in September, 94.5% in October) of increasing burned area. However, the magnitude of this influence was approximately one-tenth of that attributed to fuel dryness. The influence of fuel load on BA was found to be small and statistically insignificant, with likely influences ranging from a small suppressive effect to explaining a maximum of the increased BA in September, to virtually no influence in October, suggesting that fuel dynamics played a minor role in driving the observed fire activity. The analysis reveals a higher confidence in the simulations indicating positive anomalies, indicating a robust signal in the attribution of drivers to observed BA anomalies.

### 3.3.6 Spatial Variation in Drivers of Burned Area Extremes

#### 3.3.6.1 Canada

In Canada, most BA anomalies were linked to widespread high fire weather, with 95% of the country being influenced by higher-than-normal fire weather (**Figure S12**). There was a tendency for fuels to dry out, although this was not as widespread. Fuel load anomalies were more scattered, but areas of low fuel anomaly did correspond to some boundaries in fire extremes (**Figure 15, S12**). Increased human influence may have had some influence at suppressing fires, but this is not significant, and in some places, the model indicates a small possibility of increased fire from human activity. In the Eastern Shield, fire extremes in some areas were driven by high fire weather and dry fuel, compounded with more vegetation cover and hence higher-than-normal fuel load in some places (**Figure 15**). However, the borders of extreme fires corresponded to a suppressive effect from decreased fuel load. In the Western Shield, dry fuel and high fire weather drove fire incidents, with high fire weather dominating in some areas. Increased suppression may have had some influence at suppressing fires, but this is not significant, and in some places, the model indicates a small possibility of increased fire from human activity.

In June, there were high anomalous burned areas in Quebec's Eastern Shield, which were divided into two major fire components with a slightly reduced BA in between (**Figure 15**). Both components were associated with high fire weather, but in areas where high fire weather occurred without the contribution of other controls, it tended not to cause high levels of burning. The highest burned areas were mainly found in the northern component and were associated with anomalously low levels of moisture and high fire weather. Some cells with the very highest BA also showed anomalously high fuel load (**Figure 15**). In a region further north, around 56-57 degrees north and 72-80 degrees west, there was high fuel load, dry conditions, and high fire weather, but fire in the area was found to be highly fuel limited and largely insensitive to even large changes in controls (Kelley et al. 2019). The southern component of high burning corresponded with high fire weather and either fire fuel or high fuel load. Additionally, any boundaries between higher and normal/lower levels of BA also saw lower than average fuel loads, which may have inhibited fire spread (**Figure 15**).



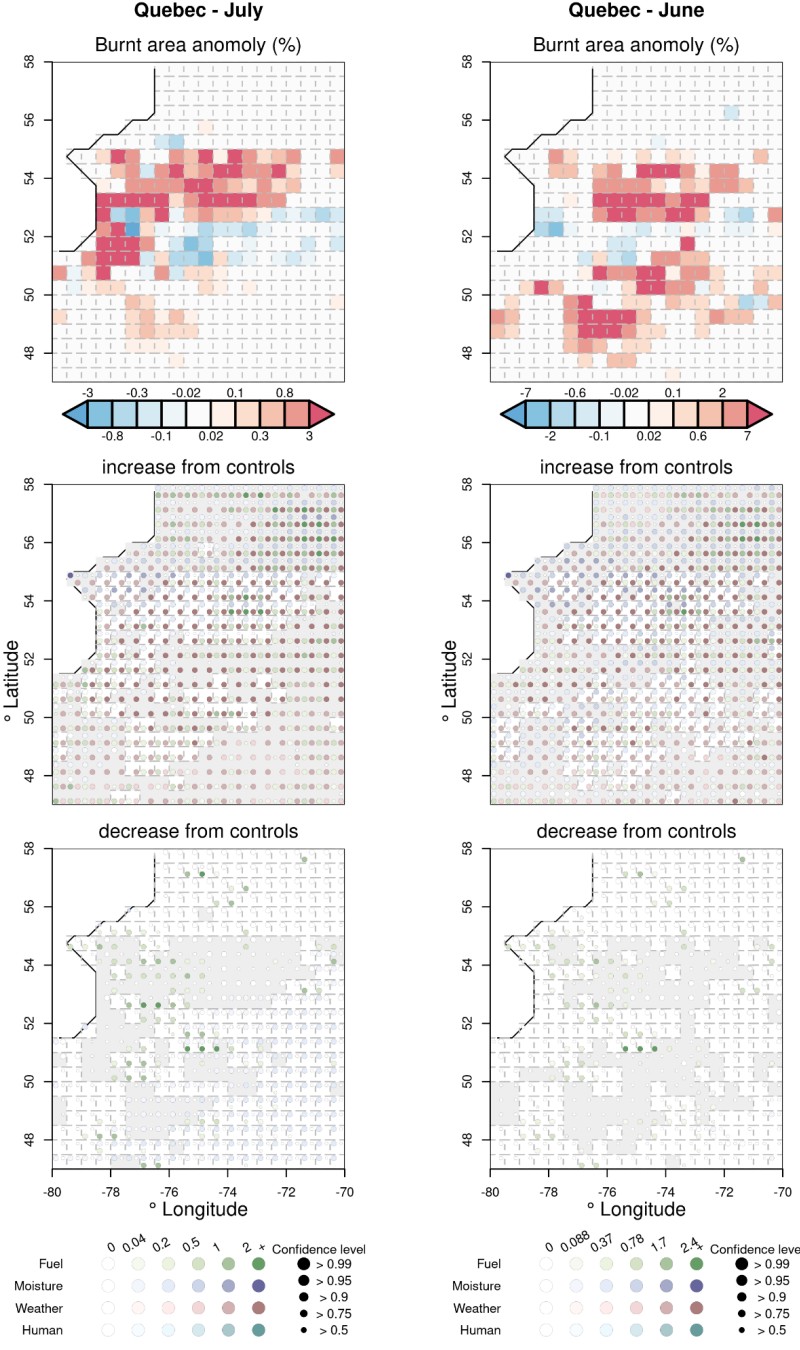

**Figure 15:** Anomalies in controls during months and regions of high BA in Quebec. The top
left map of each region shows BA anomalies at 0.5 degrees for that month in 2023 versus the
2014-2023 monthly average. The left of the other two maps looks at anomalies in controls that
would cause higher BA, with areas not greyed out representing regions with greater than



monthly average BA in 2023. The right map shows drivers that would have led to lower than normal levels of burning, with areas not greyed out showing lower or non-change from monthly average BA in 2023. Each grid cell has four points: green points show anomalies in fuel load, purple in fuel moisture, red in fire weather, and cyan in humans. This way, we can see if controls acted in unison to cause extreme levels of burning or prevent extreme fires from extending further. The shade of the point shows the most likely expected level of anomaly in that control, while the size shows how confident we are in the direction of the anomaly.

In May, the Western Shield and Taiga/Boreal plains experienced higher-than-normal fire weather across the region (**Figure S13**). The increased burned areas in the west were due to extreme low fuel moisture and high fire weather, as well as higher fuel loads. In contrast, areas to the south experienced high fire weather without the extreme burned areas. Additionally, anomalies in fuel loads and burning levels became more evident in September, with some areas displaying lower-than-average fuel and burning (**Figure S13**). These anomalies persisted, with regions still experiencing high fire weather and variations in fuel moisture levels. Furthermore, the eastern areas with higher fire weather also showed higher fuel loads, while drier fuel moisture was observed in less extreme regions to the east. Additionally, specific locations saw higher fire weather and above-average fuel moisture, while areas just north of the extreme fires experienced wetter-than-normal fuel moisture (**Figure S13**).

### 3.3.6.2 *Greece*

Interestingly, most of Greece showed a tendency towards less suppression from people (**Figure S12**). However, the dominant driver over most (73%) of Greece was high fire weather, with some areas in central Greece showing notably low fire weather. These areas do not correspond to a fire anomaly, though higher fuel loads were detected. Except for central Greece, other areas of lower than average BA correspond to lower than average fuel loads (**Figure 16**). There was no significant anomaly in fuel moisture across Greece, though the northeastern fire extreme does correspond to a joint increase in fire weather and decrease in fuel moisture. Extremes in Eastern Coastal Greece correspond to anomalies in fuel load, fuel dryness, and heightened fire weather (**Figure 16**).

In August, Northern Greece experienced high fire weather and low fuel moisture, particularly around the Alexandroupolis fires and Macedonia and Thrace (**Figure 16**). The region extended further east, experiencing extreme fires that reached into Central Macedonia. Unlike Canada, the framework in North Greece did not show the same level of detail in the boundaries around extreme levels of burning. However, the transition to less burning in the South of West Macedonia did correspond to reduced fuel load. Areas around Athens and Central Greece that saw unusually high levels of burning also experienced decreased fuel moisture and increased fuel load (**Figure 16**). In contrast, areas in Southern Thessaly and Central Greece that did not experience high burned areas saw lower than normal fire weather. In the Peloponnese region, there was either high fire weather or reduced fuel moisture, but these conditions rarely occurred together, which might explain the lack of increased BA throughout the region. (**Figure 16**)

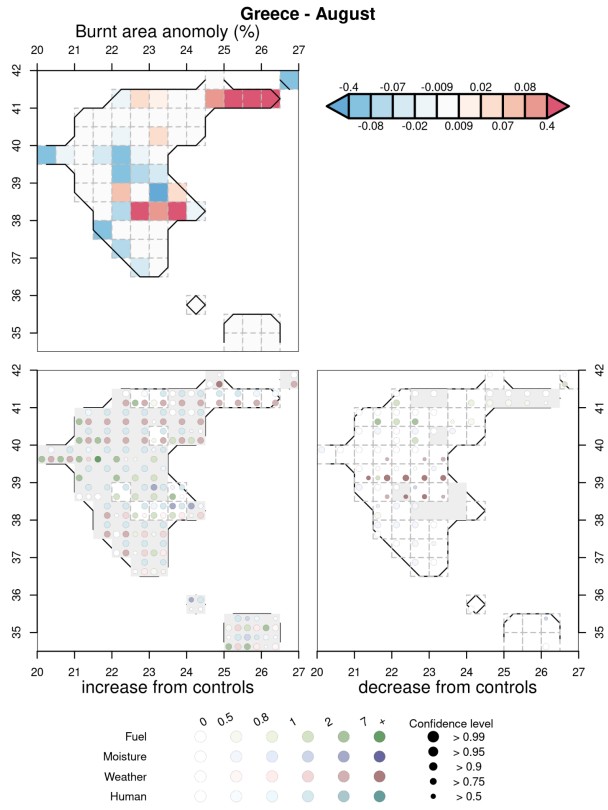


**Figure 16:** Same as Figure **15** but for Greece.


### 3.3.6.3  *Western Amazonia*


High fire weather anomalies were almost universal across our Western Amazonia region
(**Figure S12**). However, anomalies in other controls varied widely across the region, which
appears to modulate the occurrence of areas with higher than average burning (**Figure S12,**
**S14**). In September, the regions of high burning in the Western Amazon all exhibited higher
than average fire weather, with the highest BA anomalies associated with the highest fire
weathers. The areas with the highest BA anomalies along the Amazon River were located
around Manaus and also showed lower than average fuel moisture and higher than average
fuel load. The very highest pixels exhibited the anomaly in increased fuel loads. In the southern
and central part of the region (65 to 62 west/7/8 south), near Porto Velho, BA anomalies were
associated with extremely high fire weather and higher fuel loads. The areas of highest burning
also showed an increase in human-driven burning. Further east (59 west, 7 south), the highest
fire weather and decreased fuel moisture occurred alongside higher burning. Areas of higher
BA in the north (63 west, 0 north) were associated with extreme fire weather, though offset by
lower than normal fuel loads. This may explain why they were not as extreme as the fires
around Manaus, though the area is not generally considered fuel-limited (Kelley et al. 2019),
and fuel had little overall impact across the region **Figure 14)**.





## 3.4    Attribution to Global Change

### 3.4.1    Highlights

**In Canada, anthropogenic forcing increased the chance of high fire weather in 2023, total climate forcing has led to higher present-day BA and socio-economic factors have likely decreased burning.** Human influence at least doubled the probability of experiencing high fire weather in June 2023. It is virtually certain that total climate forcing increased recent high BA in the region by up to 38%, but it is less certain that socioeconomic factors decreased high burning. All forcings combined have led to an overall reduction in today's average BA across Canada.

**In Greece, anthropogenic forcing increased the chance of high fire weather in 2023. total climate forcing has likely led to higher present-day BA and socio-economic factors have likely decreased burning.** Human influence increased the probability of experiencing high fire weather in August 2023 by 1.9-4.1 times. It is likely that total climate forcing increased recent high BA in the region by up to 30%, but socio-economic factors could have led to an increase or decrease. Climate change has increased today's average BA in the Mediterranean region, but this has been mainly offset by socio-economic factors.

**In western Amazonia, anthropogenic forcing has greatly increased the chance of high fire weather in 2023. It is virtually certain that total climate forcing has led to higher BA, and very likely that socio-economic factors also contributed.** Human influence increased the probability of experiencing high fire weather by 20.0-28.5 times. It is virtually certain that total climate forcing increased recent high BA in the region by up to 47%, and very likely that socio-economic factors exacerbated the increase. Climate change has increased today's average BA in the Northwest region, and all forcings have led to an overall increase in burning.

### 3.4.2    Change in Likelihood of High Fire Weather due to Anthropogenic Forcing

#### 3.4.2.1    *Canada*

The fire weather conditions we saw in Canada during June 2023 were 2.9-3.6 times more likely due to anthropogenic forcing. Here we assess the 95th percentile of FWI over the country during the month of peak anomaly in BA (June) in the ALL and NAT HadGEM3 simulations. More of the ALL distribution lies above the observed 95th percentile of FWI from ERA5 compared to the NAT distribution (**Figure 17**), and we therefore conclude that the probability of experiencing the high fire weather observed during June 2023 is more likely in a climate forced with anthropogenic emissions.

#### 3.4.2.2    *Greece*

The high fire weather conditions experienced during the peak anomaly in BA in August 2023 were 1.9-4.1 times more likely due to anthropogenic forcing (**Figure 17)**. In this case the 95th percentile of FWI is outside of the distribution so we instead assess the 90th percentile of FWI over the country. This is likely a result of our linear inference of 2023 for the bias correction based on the 1960-2013 period, where in fact 2023 was so anomalous that it doesn't fit this trend. The 2023 event threshold here also lies at the very high end of simulated fire weather, meaning it was very unusual in the model simulations. The result range here is also larger than for Canada, meaning there is less certainty about how much human influence has increased the probability, although it does highlight at least a 50% increase in likelihood of high fire weather.



### 3.4.2.3 Western Amazonia

High fire weather in western Amazonia during Sept-Oct 2023 was 20.0-28.5 times more likely due to anthropogenic forcing (**Figure 17)**. In this region there is a large shift in the ALL forcing distribution compared to the NAT only forcing for the 95th percentile of FWI, and the high risk ratio shows a strong anthropogenic signal in driving the meteorological conditions that led to high fires over this period.

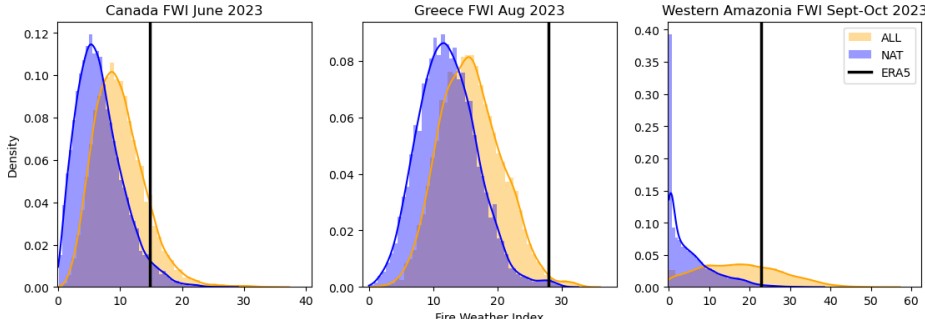

**Figure 17:** 95th percentile of FWI for June 2023 over **(left)** Canada, **(middle)** Greece and **(right)** western Amazonia, in the HadGEM3 ensemble of ALL (anthropogenic plus natural forcing, orange) and NAT (natural-only forcing, blue) bias-corrected simulations, and ERA5 reanalysis (black line).

### 3.4.3 Change in Likelihood of High Burned Area in 2023 due to Total Climate Forcing and Socioeconomic factors

### 3.4.3.1 Canada

We show that total climate forcing was virtually certain (99.9% likelihood) to have led to more burning in events such as those observed in Canada in June 2023 (**Figure 18**). In the cells of Canada that burn most regularly (95th percentile burned area), our attribution results indicate that total climate forcing increased the extent of area burned in Canada by between 0.7 - 6.22% during the month of June in the period 2003-2019. This estimate is almost certainly conservative, as we are testing the influence of climate change on the likelihood of equivalent levels of burning in years preceding 2023, and these years have experienced less warming. Additionally in **Figure 14**, considering the anomalies of 2023, we show between 1-30% additional increase in BA extent caused by anomalous weather conditions during this heatwave on top of the 2003-2019 period. The impact of socio-economic factors is less certain, with only a 54.77% likelihood of decreasing burning, affecting BA by between -10.12 and 3.03% during 2003-2019.

Overall, we estimate that BA in Canada in June 2023 was 0.8-38.0% greater due to total climate forcing in the 2003-2019 period combined with this year's anomaly in the climatic variables. As a caveat, we note that this is not a formal attribution of the 2023 anomaly because no counterfactual exists for the year, but rather an attribution of the change in BA in 2003-2019 with the additional influence of climate factors on BA in 2023 superimposed.



### 3.4.3.2  Greece

We show total climate forcing caused a change in high BA in Greece of between -0.66 and 18.58% in the period 2003-2019, with a likelihood of an increase of 84.92% (**Figure 18**). In this case we use the 90th percentile (gridcell with the 10% highest burned areas) to represent high BA over the region for the month of August, over 2003-2019. Again this increase is likely a conservative figure given the additional warming since 2019, and estimating the additional burning that might have been experienced during the anomalous conditions of 2023, we find an additional change of -9.3-11%. Socioeconomic factors likely (79.9%) caused a decrease in burning, though could have caused an increase, affecting BA extent by -36.58 to 5.4%.

Overall we estimate that BA in Greece in August 2023 was influenced by -15.2-29.6% due to total climate forcing in the 2003-2019 period combined with this year's anomaly in the climatic variables. As per the results for Canada, we note that this is not a formal attribution of the 2023 anomaly, but an attribution of the change in BA in 2003-2019 with the additional influence of climate factors on BA in 2023 superimposed. In the case of Greece, uncertainties around whether total climate forcing and socioeconomic factors caused an increase or decrease in BA are higher, and the smaller region size makes detecting a strong signal of change more challenging.

### 3.4.3.3  Western Amazonia

Over the period 2003-2019, total climate forcing was virtually certain to have caused an increase in burned areas like the one experienced in western Amazonia in September and Oct 2023 (99% likelihood), with a likely range of increase in extent of 1.53-7.66% (**Figure 18**). Here we assess the change in BA due to total climate forcing in the cells that burn most regularly (95th percentile) of our defined region of western Amazonia over Sept and Oct 2003-2019. Extending our analysis to the 2023 anomaly, we estimate that additional burning could have been up to 0.8-36.2% on top of the 2003-2019 levels. Despite finding little influence of humans specifically in 2023 compared to the previous 10 years in (**Figure 14**), since early-industrial, we show socioeconomic factors have had a large influence on the occurrence of extreme levels of burning. Events similar to 2023 were very likely exacerbated by socioeconomic conditions (95.98% likelihood) increasing BA by 0.18-5.32%.

Overall we estimate that BA in western Amazonia in September-October 2023 was increased by 2.3-46.7% due to total climate forcing in the 2003-2019 period combined with this year's anomaly in the climatic variables. Though again, we note that we determine this by convoluting the change in BA in 2003-2019 with the additional influence of climate factors on BA in 2023 superimposed, and therefore it is not a formal attribution statement of the 2023 anomaly.

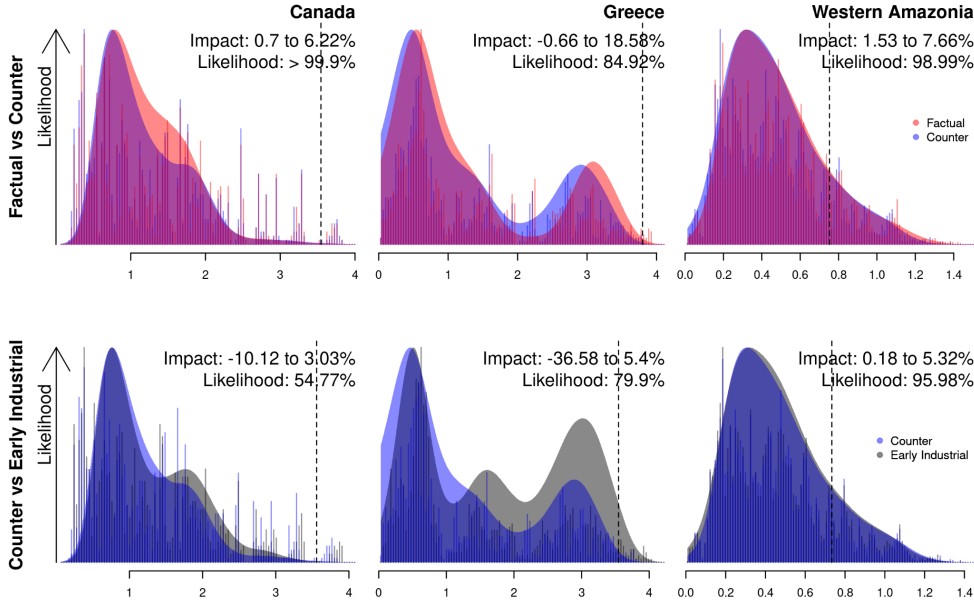

**Figure 18:** Likelihood of high BA from ConFire ISIMIP3a. Panels show **(top row)** the likelihood (probability) of an extreme event such as experienced in the month of peak burning in 2023 with (red) and without (blue) total climate forcing, and **(bottom row)** likelihood of the same extreme event with (blue) and without (grey) socioeconomic forcing for **(left)** Canada, **(middle)** Greece (Centre) and **(right)** Western Amazonia. Likelihood shown on each plot, and impact refers to the increase in BA due to the forcing.

### 3.4.4 Change in Burned Area Anomaly due to Total Climate Forcing, Socioeconomic Factors, and All Forcing

#### 3.4.4.1 Canada

As reported in Burton, Lampe et al. (2023), we also show how median BA for months of interest has changed overall in the region today due to total climate forcing (**Figure 19**), socioeconomic forcing (**Figure S15**), and all forcings (**Figure S16**). Using AR6 regions that best match our focus areas, we show that BA has increased by 1.9% [0.1, 3.6] in North West North America (NWN) due to total climate forcing, but reduced by -0.2% [-1.7, 1.3] in North East North America (NEN) (**Figure 19**). In these regions, socioeconomic forcing has dampened the effects of climate change, by reducing BA by -9.5% [-13.6, -6.3] in NWN, and -8.5% [-12.5, -5.7] in NEN (see Supplement). All forcings combined have led to an overall reduction in BA of -8.3% [-12.5, -4.9] in NWN and -8.7% [-12.8, -5.8] in NEN (**Figure S16**).

#### 3.4.4.2 Greece

Burton, Lampe et al. (2023) find a larger increase in median BA for months of interest due to total climate forcing in the Mediterranean region (MED), with an increase of 14.5% [11.5, 18.1] today compared to the counterfactual (**Figure 19**). This is particularly the case for the high burned areas, where the increase is larger compared to the lower end of the distribution. However, socioeconomic factors have largely offset this by reducing BA by -10.2% [-13.6, -



6.6]. However all forcings combined have led to an overall regional increase in BA of 0.5% [-
3.5, 5.5] (see **Figure S16**).

### 2656  3.4.4.3  *Western Amazonia*


As per Burton, Lampe et al. (2023), total climate forcing has increased median BA for months
of interest by 11.5% [5.4, 18.4] in Northwest South America (NWS) today compared to the
counterfactual (**Figure 19**). Again, this increase is mostly impacting the BA at the higher end
of the distribution. This is mostly offset by socioeconomic factors (-9.0% [-18.9, 1.2]), although
all forcings combined have still led to an overall increase in BA of 1.5% [-6.9, 10.5] in the
region (see Supplement).

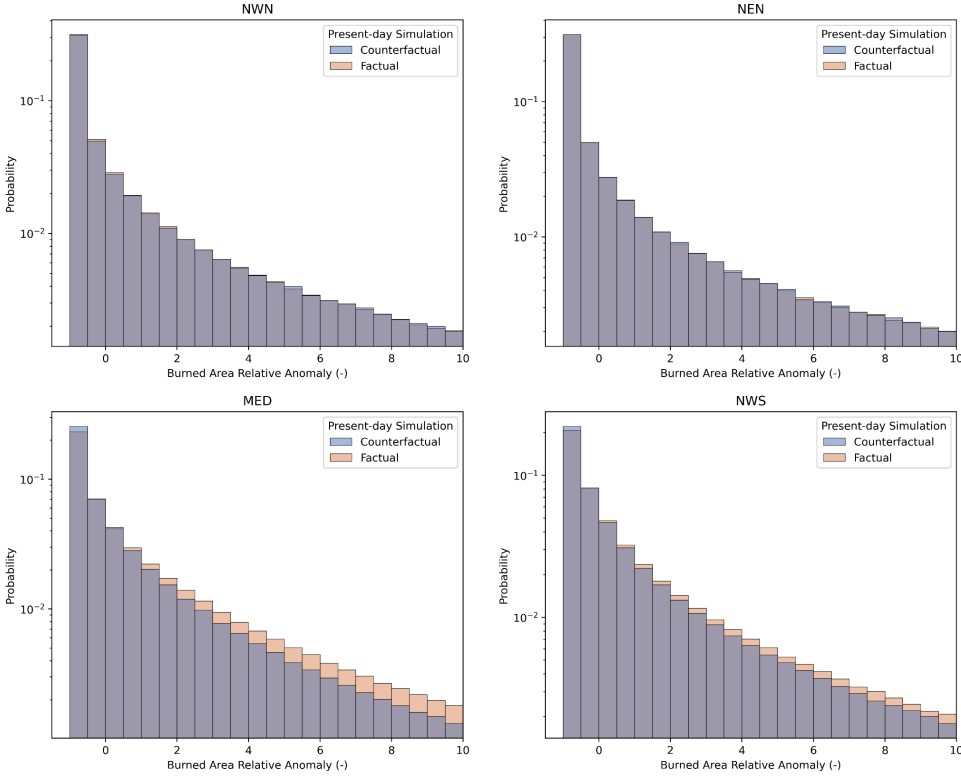

**Figure 19:** Change in median BA due to total climate forcing from FireMIP. Present day BA
(2003-2019) for factual (historical forcing, orange) and counterfactual (detrended climate,
blue), for AR6 regions. Panels show **(top left)** North West North America (NWN), **(top right)**
North East North America NEN, **(bottom left)** Mediterranean (MED), and **(bottom right)** North
West South America (NWS). Probability is shown on a log scale.





## 3.5 Seasonal and Decadal Outlook

### 3.5.1 Highlights

- **Predictions see the Indian Ocean Dipole persisting in its positive phase well into 2024.** The 2023–2024 El Niño event emerged as the fourth-most powerful on record. As 2024 comes to an end most simulations forecast a transition to ENSO-neutral conditions. Positive IOD phases are associated with elevated BA Amazonia, Indonesia, and parts of Australia though these tend to depend on interactions with ENSO.

- **Seasonal predictions of fire weather through August 2024 highlight moderate positive anomalies in Western Canada, South America (including southern Amazonia)** but no clear signal for extreme anomalies is present in the forecast. These forecasts have limited skills beyond month 1-2 (e.g., **Section 3.3.2**).

- **Effective strong mitigation efforts could limit increases in future fire extremes.** Canada in particular is very likely to see substantial increases in BA under high emissions scenarios (SSP370, SSP585) compared to high mitigation (SSP126), driven by dry conditions and climate and CO2 driven increase in fuel load. However the lack of significant difference between middle-of-the-road (some mitigation e.g. SSP370) and no mitigation (SSP585) scenarios in this region highlights the need to constrain uncertainties in subsequent studies. In Western Amazonia, while SSP585 predicts significant increases in fire likelihood by 2100, the SSP126 scenario suggests that proactive climate strategies could maintain conditions that largely remain unchanged, underscoring the importance of mitigation actions for this region.

- **However, even with mitigation, we will still see an increase of extremes in Canada.** This necessitates robust adaptation strategies to manage and prepare for the expected rise in extreme fire events as global mean temperature continues to rise towards 1.5C above PI, ensuring that communities and ecosystems are better equipped to handle future challenges.

- **There is still a lot of uncertainty regarding how extreme fire events will change with current efforts to reduce emissions.** Our projections show a high level of uncertainty in all regions for the expected increase in extreme events under moderate mitigation efforts (SSP370 scenario) and without any mitigation (SSP585 scenario). The wide range of projections means that we cannot determine if current mitigation efforts are effective, and more research is needed to reduce uncertainties in future projections. However, even in the best-case scenarios, both SSP370 and SSP585 show significant increases in the frequency of extreme events in Canada and Western Amazonia. While all regions show, with confidence, that strong mitigation (SSP126) will reduce the occurrence of extreme events, there is still a large plausible set of future increases in extremes. Combined with the lack of understanding in future land use and direct human influence on fire, this makes it hard to inform adaptation efforts.

- **Greece's smaller domain size means that Global Climate Models (GCMs) introduce biases and uncertainties making future projections of extremes difficult.** This highlights the need to improve model projections of future climate or incorporate other information, observations or model/statistical approaches to constrain uncertainties on small scales.

### 3.5.2 Seasonal Outlook for 2024

The 2023–2024 El Niño event emerged as the fourth-most powerful on record, unleashing widespread droughts, as well as floods and other anomalous conditions worldwide. Officially declared by the World Meteorological Organization (WMO) on July 4, 2023, its meteorological impacts have unfolded between November 2023 and April 2024 (Joshi, 2023). Climate scientists have found that the 2023–24 El Niño event, compounded by the climate change



crisis, has elevated global temperatures beyond the records set during the 2016 El Niño event.
This has caused a 1.5°C increase in global temperature compared to pre-industrial times for
most of the year and established new temperature records in 2024 (McCulloch et al., 2024).
As of mid-April 2024, El Niño conditions persist in the central-eastern equatorial Pacific, with
important oceanic and atmospheric indicators aligning with an ongoing El Niño event that is
gradually diminishing (**Figure 20)**. Most simulation prediction plumes forecast a transition of
the El Niño event to ENSO-neutral in Apr-Jun, 2024, which then persists during the boreal
summer. The Indian Ocean Dipole (IOD) index is currently positive with the most recent value
of +0.95 °C and forecasts indicate that it will remain in a positive state for the next season.
Connections are established between a positive IOD phase and fire risk in Indonesia and parts
of Australia, though outcomes generally depend on interactions with the ENSO phase (Pan et
al., 2018; Ren et al., 2024; Abram et al., 2021). Similarly, the positive phase of the IOD is
linked with heightened fire risk in South America, particularly in the Amazon basin, where it
can interact with other climate teleconnections to exacerbate droughts and increase wildfire
risk (Cardil et al., 2023; Dong et al., 2021).
**Figure 21** shows the predicted probabilities of the monthly average FWI exceeding moderate
(75th percentile) or high (95th percentile) thresholds of the monthly climatology. As the
strength of a positive ENSO diminishes, most areas in Southeast Asia and South America are
predicted to experience a decrease in the likelihood of anomalous conditions over the next
three months. Parts of Western Canada are predicted to reach moderate anomalous
conditions once again in early summer, and this combined with overwintering fires could
promote a second consecutive high fire season as has already been reported in the media
(*BBC News*, 2024, *New York Times*, 2024). Predictions also suggest that the moderate FWI
threshold will be exceeded in southeast, central, and western Brazil with the high threshold
exceeded in southern and western parts. In parts of Africa, moderate FWI anomalies may be
experienced throughout June-August.

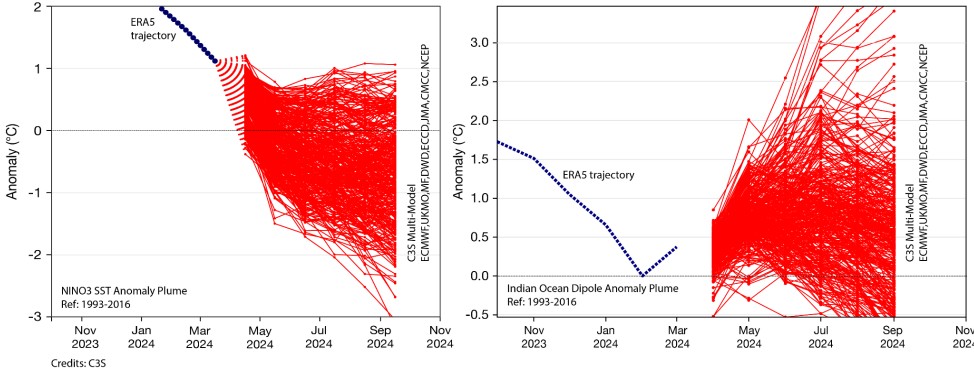

**Figure 20:** Anomaly plume for the ENSO and IOD as forecasted by the multimodal C3S
ensemble system for April 2024. Ongoing positive ENSO event is gradually diminishing while
Indian Ocean dipole is likely to remain in a positive state until September 2024 (Copernicus
Climate Change Service, 2024b).

Earth System
Science
Data

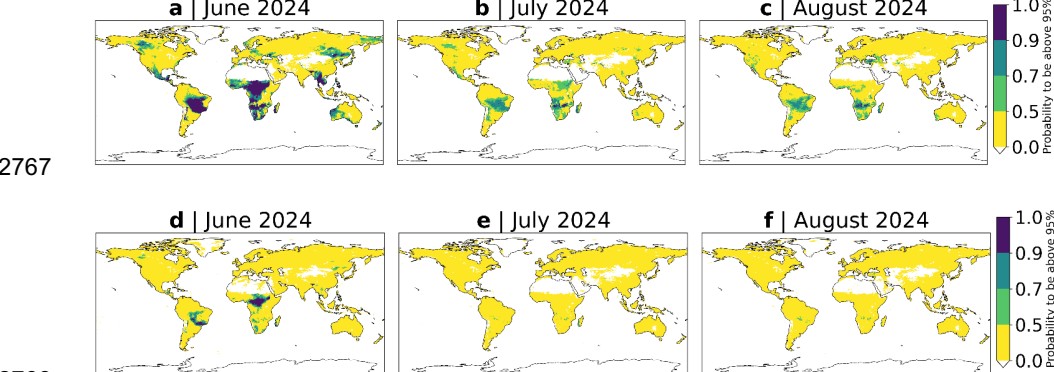

**Figure 21:** Probability for the monthly average FWI exceeding the **(top row)** 75th percentile threshold (anomalous conditions) and **(bottom row)** 95th percentile threshold (extremely anomalous conditions) of the monthly climatological distributions. The probability is calculated using the 51 ensemble member realisation from ECMWF's long-range forecasting system, SEAS5 FWI, and comparing them with the 1991-2016 climatology (Copernicus Emergency Management Service, 2019).

### 3.5.3 Future Changes in Likelihood of Extreme Fire Events

#### *3.5.3.1 Canada*

The probability of Canada experiencing a fire event similar to June 2023 (measured in terms of BA in the cells with the highest 5% burned areas) is estimated to be 0.14% under the climate conditions of 2010-2020 (**Table 7**; **Figure 22**). This corresponds to a return time of approximately once every 700 years. Even though the GCMs were bias-corrected to 1979-2014 values as part of the ISIMIP3b protocol, internal variability and trends were maintained. This difference can lead to discrepancies between models due to the non-linear response of fire to small changes in climate variables and simulated vegetation between ISIMIP3a and ISIMIP3b, which substantially altered the results over our baseline periods. When using bias-corrected GCM model output instead of reanalysis data, the likelihood ranges from 0.08% to 0.31%, indicating a slight bias of the probability of the event introduced by using GCM historical simulations. We describe future changes as significant if the range across GCM projects does not overlap with this range.

Over the next 10 years, the likelihood of a similar fire event occurring increases significantly to between 0.14% and 0.48%, with no significant difference across scenarios. This suggests that a 2023 fire event could be up to three times more likely by the 2030s than in the last decade. By the middle of the century, the likelihood of a 2023 fire event recurrence increases to 0.31% - 0.9% - two to almost four times as likely as today (**Table 7**; **Figure 22**). The different SSP scenarios diverge significantly by 2070. Under SSP126, the likelihood of a 2023 fire event recurring stabilises and remains largely unchanged until 2100, with a range of 3.07% to 3.5%. This means the recurrence of a 2023-like event would be around 2.8% to 8.3% in the last decade. Integrating between today and 2100, under SSP126, the probability of at least one event similar to or worse than the 2023 event occurring again is estimated to be between 16% and 25%.





**Table 7:** Summary of the likelihood of extreme events today using reanalysis 'factual' and today and into the future using bias-corrected GCMs for our three focal regions. '2023' events focuses on the BA extreme identified in **Section 3.4.3**. 1-in-100 for Western Amazonia additionally looks at the likelihood of a 1-in-100 event under present day climate conditions, following the definition of extreme in (UNEP, 2024). We also determine how much more frequent the events will be at two different time horizons based on each models likelihood in the future projections over likelihood during 2010-2020. *indicates non-significant result.

| Region | Event | SSP | Represents | Likelihood (%/year) 2010-2020 min | max | 2040-2050 min | max | 2090-2100 min | max | How much more frequent 2040-2050 min | max | 2090-2100 min | max |
|---|---|---|---|---|---|---|---|---|---|---|---|---|---|
| Canada | 2023 (~1-in-700) | ~1-Factual | observed | 0.14 | | | | | | | | | |
| | | SSP126 | strong mitigation | 0.08 | 0.27 | 0.31 | 0.6 | 0.28 | 0.83 | 2.22 | 3.88 | 3.07 | 3.5 |
| | | SSP370 | middle of the road | 0.13 | 0.31 | 0.36 | 0.69 | 0.93 | 1.91 | 2.23 | 2.77 | 6.16 | 7.15 |
| | | SSP585 | no mitigation | 0.12 | 0.29 | 0.46 | 0.9 | 1.15 | 2.2 | 3.1 | 3.82 | 7.59 | 9.58 |
| Greece | 2023 (~1-in-80) | ~1-Factual | observed | 1.3 | | | | | | | | | |
| | | SSP126 | strong mitigation | 0.91 | 1.65 | 1.36* | 1.8* | 1.42* | 1.94* | 1.09* | 1.49* | 1.18* | 1.56* |
| | | SSP370 | middle of the road | 0.99 | 1.46 | 0.87* | 2.2* | 2.54 | 3.11 | 0.88* | 1.51* | 2.13 | 2.57 |
| | | SSP585 | no mitigation | 0.67 | 1.78 | 1.16* | 2.39* | 2.87 | 3.26 | 1.34* | 1.73* | 1.83 | 4.28 |
| Western Amazonia | 2023 (~1-in-6) | ~1-Factual | observed | 16.58 | | | | | | | | | |
| | | SSP126 | strong mitigation | 15.13 | 16.57 | 15.83* | 17.92* | 15.89* | 17.61* | 1.05* | 1.08* | 1.05* | 1.06* |
| | | SSP370 | middle of the road | 15.22 | 16.29 | 16.16* | 18.07* | 17.65 | 20.36 | 1.06* | 1.11* | 1.16 | 1.25 |
| | | SSP585 | no mitigation | 15.13 | 16.5 | 16.39* | 18.34* | 18.21 | 21 | 1.08* | 1.11* | 1.2 | 1.27 |
| | 1-in-100 | Factual | observed | 1.0 | | | | | | | | | |
| | | SSP126 | strong mitigation | 0.82 | 1.53 | 1.28* | 2.16* | 1.23* | 2.01* | 1.41* | 1.46* | 1.31* | 1.5* |
| | | SSP370 | middle of the road | 0.81 | 1.46 | 1.29* | 2.23* | 1.97 | 3.14 | 1.53* | 1.59* | 2.15 | 2.43 |
| | | SSP585 | no mitigation | 0.8 | 1.49 | 1.43* | 2.36* | 2.32 | 3.34 | 1.58* | 1.79* | 2.24 | 2.9 |



Mitigation efforts under SSP370 have some impact, lowering the likelihood to 0.93% to 1.91%
by 2100 compared to 1.15% to 2.2% in SSP585 (no mitigation; **Table 7**; **Figure 22**). The
probability of an event like 2023 occurring at least once by 2100 is estimated to be between
30% and 52% under SSP370 and between 35% and 56% under SSP585. Someone born in
Canada in the current decade, with a life expectancy of 83 years (United Nations Population
Division, 2022), has a 43-66% probability of seeing a similar event in their lifetimes under
SSP585, compared with only an 11% likelihood of someone who was 83 years old in the
2010s. Someone born in Canada today would also have a 18-45% probability of seeing an
event of similar magnitude *twice* under SSP585. This reduces substantially to 20-31% for one
occurrence and 4-10% for two occurrences under SSP126.
The divergence between shared socioeconomic pathways (SSPs; **Table 7**; **Figure 22**) is
primarily driven by variations in the increase in fuel load, although an increase in fuel dryness
is a major driver of increased burning across SSPs, particularly in SSP585. The worst-case
scenario within SSP585 sees a substantial increase in both and changes in both controls are
needed to explain divergence between SSPs.

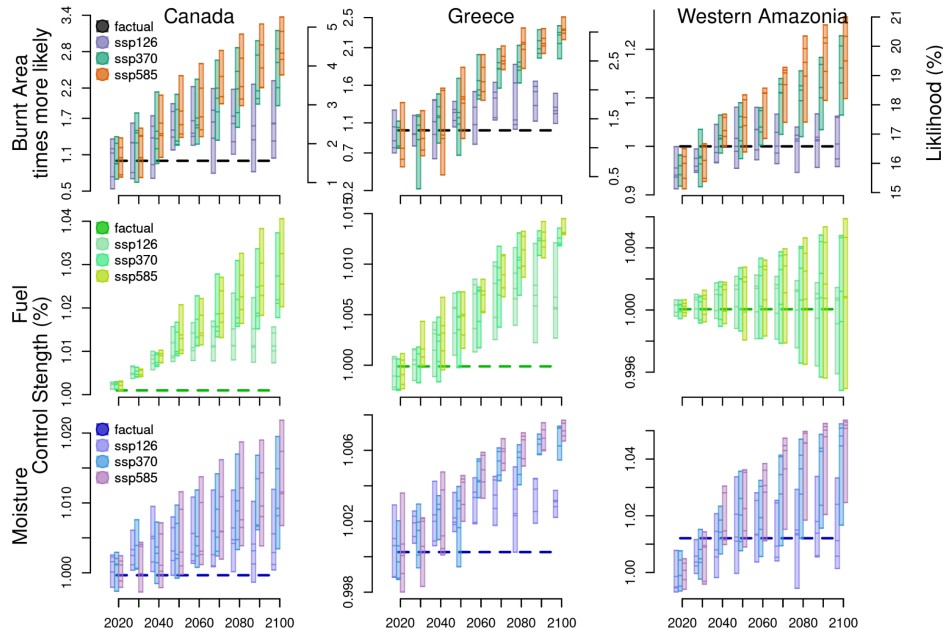

**Figure 22:** Future projections from ConFire of the likelihood of BA extremes and their
corresponding fuel and moisture controls. Each set of bars shows changes in each decade,
with each bar representing a different SSP scenario and the spread of bars indicating the
variation across GCMs. The BA data indicates **(right axis)** the likelihood of the defined event
occurring in any given year for future decades and **(left axis)** the frequency of occurrence for
relative to reanalysis-driven simulations (factual ISIMIP3a – dashed line) of the years 2010-
2020. The fuel and moisture rows illustrate the change in the maximum burning allowed by
that control relative to reanalysis-driven simulations for the years 2010-2020.



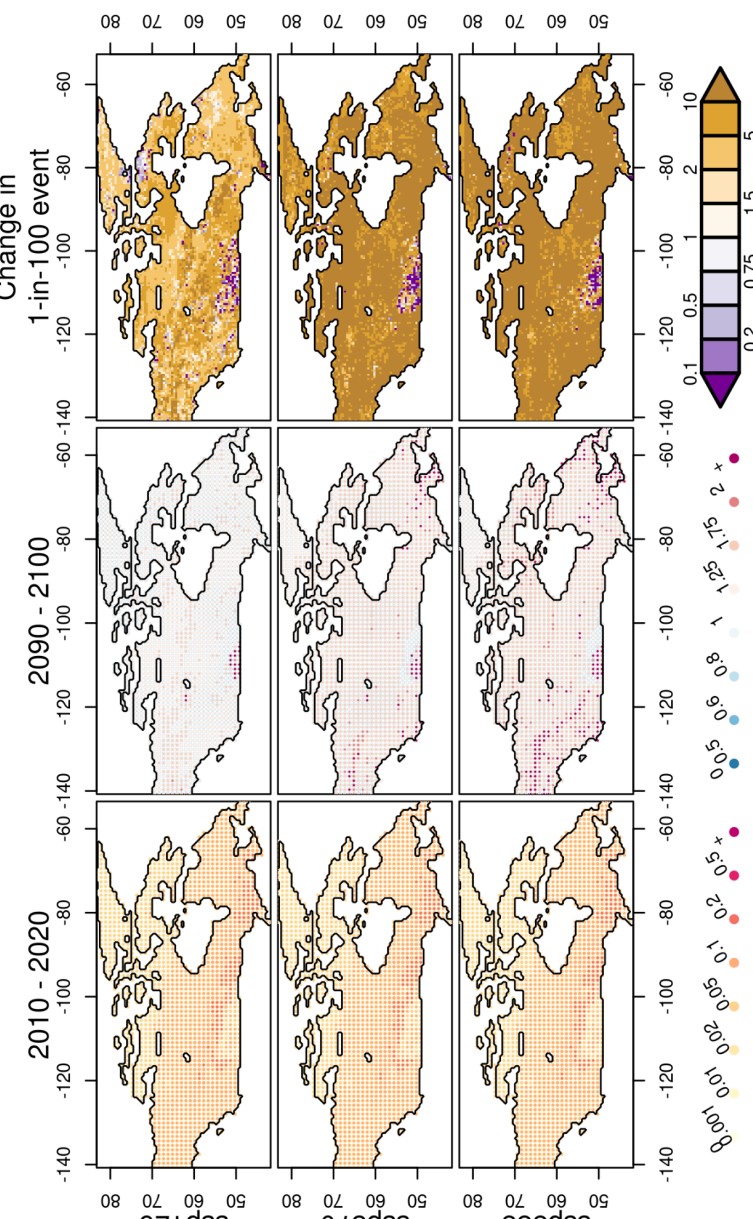

**Figure 23:** Projected changes in June-August BA over Canada by 2090-2100 under three SSP scenarios, with BA simulated by ConFire. (**Left**) Average June-August BA fraction (%) for 2010-2020. (**Middle**) Relative change in June-August BA extent projected for 2090-2100 expressed as a ratio of 2010-2020 values. (**Right**) Increased (or decreased) frequency in 2090-2100 of a 1-in-100 year event defined for the period 2010-2020, expressed as a ratio of 2010-2020 values. In the left column, the size of the dot in each grid cell indicates the likelihood (larger = higher likelihood) of a BA fraction (or being greater than the threshold indicated by the coloured dot (see legend at the base). Likewise, the size of the dot varies with likelihood that the BA fraction exceeds the threshold indicated by the coloured dot (see legend at the base). For example, a large pale orange dot in the left column indicates a high likelihood of the BA fraction exceeding 0.05%, whereas a small dark red dot indicates a small (but non-zero) likelihood of the BA fraction exceeding 0.5%.





While some areas of Canada see increases in BA and fire extremes (spatially measured as
the proportional change in probability of a 2010-2020 1-in-100 annual event, as per UNEP
(2024) almost everywhere (**Figure 23**), increases in BA are likely to be particularly strong in
the critical regions of agriculture in Southern Alberta and Saskatchewan, starting from 2030
(**Figure S17**) and becoming particularly strong by 2090s (**Figure 23**). However, some regions
just to the North of this region may see a decrease in future extremes. While Yukon and
Northwest Territories will see an increase in BA and fire extremes from 2040 (**Figure S18**),
the increase in extremes remains similar to the rest of Canada, at around a doubling in
occurrence. By the end of the century, however, SSP370 and SSP585 diverged from the high
mitigation SSP126, and both scenarios see a much larger increase in burned areas and some
areas with extreme occurrences becoming 5 times more common. SSP585 sees these
increase in fire risk extending down through British Columbia.

### 3.5.3.2 Greece
Using the ISIMIP3b GCMs for Greece introduced a strong bias when using the 95% of burned
areas. We therefore used the 90% percentile (i.e BA in the 10% cells that see the most BA) to
measure areas showing extreme burning. Here, using GCMs instead of reanalysis changed
the likelihood of Greece's August 2023 event from 1.3% to 0.67-1.78% for 2010-2020 (**Table
7**; **Figure 22**). While there is an indication of increased likelihood of extensive BA by the middle
of the century, the projected range of changes in future extremes remains smaller than the
introduced historical bias and is, therefore, insignificant. For SSP126, this small but
insignificant increase remains until 2100, though at times, SSP126 shows the possibility of a
substantial increase – up to 2.3 times more likely than today. This shows that Greece may still
see large increases in the occurrence of extreme BAs even under strong mitigation, though
uncertainties are too large to state this confidently. This may be due to Greece's smaller region
and the need for a larger sample of points to overcome the inherent noise in simulating BA
extremes. Our uncertainty quantification was necessary to determine these significant/non-
significant results. SSP350 and SSP585 show significant increases by 2070, and significantly
diverge from SSP126 by 2080. There is no significant difference between the scenarios by
2100, which indicate a likelihood of seeing an annual recurrence of 2023's event of between
2-54-3.26% by 2100 – a 1.83-to-4.28-fold increase today and an average return time of every
39-31 years. While fuel loads show a larger increase than fuel moisture in future projections,
the regions are less sensitive to fuel (Kelley et al. 2019; **Figure 14**), and differential increases
in fuel dryness dominate the variations between SSPs by 2100.
While it is hard to ascertain a clearer climate signal on area extremes across Greece, there
are some spatial patterns in likelihood of extreme occurrence with Greece (**Figure 24**). The
middle of Greece may see a decrease in extremes in the future as early as the 2030s and
2040s (**Figure S20, S21**), with Central Greece and Macedonia seeing that large decreases
by 2090 (**Figure 24**), though some areas of Central Greece, West Greece and Peloponnese
seeing a potential rise in extremes. Changes in BA are less pronounced and future projections
suggest either decreases or increases in summer burning is possible.

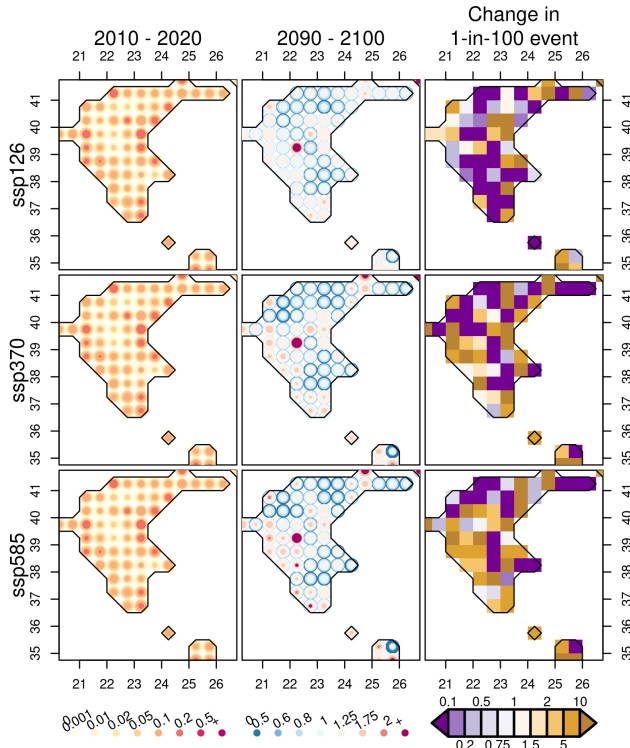

**Figure 24:** Same as **Figure 23**, but for Greece and months July-September.

### 3.5.3.3 Western Amazonia
The analysis of the South American region suggests a 16.58% chance of experiencing high
burned areas (measured in terms of BA in the cells with the highest 5% burned areas) in
Sep/Oct 2023, based on data from 2010-2020 using ISIMIP3a (**Table 7**; **Figure 22**). Replacing
this with ISIMIP3b introduces a small bias, with a likelihood of 15.13-16.57%. These events,
while higher than average, are not as extreme as those in Canada and Greece, with a likely
return time of 6 years in ISIMIP3a and 6-7 years in ISIMIP3a. However, this may hide the
spatial variations and shift in locations of extremes, particularly along the deforestation front,
as highlighted in **Figure S14**.
There are only small increases in projected future extremes, noting that we do not consider
land use change, which can be a major driver of BA in some regions (Lapola et al., 2023;
Kelley et al., 2021; Silveira et al., 2020). Under the SSP585 scenario, there is a notable and
significant increase in future burning, estimated to be 1.2-1.27 times more likely than current
levels by 2100. This increase is primarily attributed to drier fuel moisture, in line with the trend
suggested by the FWI analysis in **Section 3.4.2**. There is potential for mitigation, particularly
under the SSP126 scenario, which converges with the other two SSPs. This suggests that the
levels of extreme fire activity may largely remain unchanged by 2100.
While fuel load could potentially have a significant impact, it remains uncertain whether it
would lead to an increase or decrease in burning. **Section 3.3** emphasises the positive
influence of land use on the 2023 burning levels in some locations. However, despite the likely





substantial impact inferred from results in these sections, future land use projections were not available at the time of writing and should be prioritised for next year's report.

The likelihood of the occurrence of the 2023 event increases substantially in the centre of the region by the 2030s (**Figure S21**) and is almost universal across the region by 2024 (**Figure S22**), though small pockets around Manaus and roads in the south may see either not much change or the chance of a decrease. Perhaps more important for the region's fire-sensitive forests is the projected increase in BA in SSP370 and SSP585 across forests in the north of the region could have a significant impact on local vital ecosystems within the next few decades (**Figure S23**).

When testing more extreme levels of burning of a 1% likelihood in ISIMIP3a, there is a noticeable divergence compared to present conditions and between different SSP scenarios in future projections (**Table 7**). Introducing a minimal bias using ISIMIP3b changes the likelihood to between 0.8-1.53. Across all scenarios, there is an increase in extreme events throughout the 21st century, although the significance is only reached in the last couple of decades in SSP370 and SSP585. Without mitigation, there is a likelihood of 2.32-3.34 of a 1-in-100 event by 2100, representing an increase of 2.24-2.9 times. However, stringent mitigation measures show a decrease in extremes, with a 1-in-100 event only increasing to 1.31-1.5. This suggests that strong mitigation measures may offset the risk of extreme burned areas in the region in the future.

## 4 Discussion and Conclusions

### 4.1 Summary of the State of Wildfires in 2023-24

**Objective 1: Regionally identify extreme individual wildfires or extreme wildfire seasons of the past 12 months, and place them in context of recent trends**

In this report, we used a combination of data from global Earth Observation products and data-driven models (Jones et al., 2024; Giglio et al., 2018; van der Werf et al. 2017; Kaiser et al. 2012), regional products (e.g. EFFIS), and insights from regional experts to identify key events of the period March 2023-February 2024. Highlighted results are provided in **Section 3.1.1** and summarised here. The 2023-24 fire season saw 3.9 million km² burned globally, slightly below the previous average, yet carbon emissions were 16% above average due to severe fires in Canada's boreal forests and despite low fire activity in African savannahs. In North America, Canada experienced record-breaking fires, with extensive damage and significant evacuations, while there were also localised yet severe incidents in the US (notably in Lahaina, Hawai'i, where 100 people lost their lives). South America saw below-average fire activity overall, but with an anomalous high fire season in Western Amazonia in 2023 and northern parts of the continent so far in 2024. Severe impacts of individual fires were also seen, notably in Paraíso, Chile, where 131 people lost their lives. Europe had generally low fire activity, except for significant incidents in Greece which included the largest fire on record in the EU. Oceania saw above-average fire activity focussed on Western and Northern parts of Australia. Asia experienced low fire activity overall, but with some significant regional impacts in Southeast Asia and parts of Russia. Africa's fire extent was low, although notable fire-related emergencies occurred in Northern and Southern Africa.

**Objective 2: Shortlist a selective number of focal events (extreme individual wildfires of extreme wildfire seasons) with notable impacts on society or the environment.**

While identifying events to become foci of the report is not trivial due to the diversity of impacts that wildfires present, we selected a pragmatic set of three events for further analysis. In



**Canada**, the 2023-24 fire season was marked by an unprecedented burned area, high carbon emissions, and a prolonged fire season with severe air quality impacts and significant human and economic tolls. **Greece** experienced its second-worst fire season on record, with extensive damage, large-scale evacuations, and significant urban and infrastructural impacts, particularly from the massive Evros fire. **Western Amazonia** faced unprecedented fire counts, severe air quality degradation, and broad socio-economic and environmental impacts, highlighting the critical global significance of the region's fire activity.

**Objective 3: Diagnose the contributions of fuel dryness, fuel load, ignitions and suppression to the occurrence of each focal event.**

Using Bayesian inference and data-driven methods, we attributed observed fire activity and resulting burned areas to four controls: weather, fuel abundance and status, and others. The "others" category includes human influences, ignition sources like lightning, and residuals of all missing predictions. For BA, we can still include long-term variations in "other" human drivers to test anomalies in direct human influences on fire. We found that fire weather conditions alone do not entirely drive the occurrence of significant fire events in 2023-24. Even in Canada, where fires are primarily weather-driven, the most intense burning events in mid-July, early August, and late September were not predicted, highlighting potential shortcomings in forecasting specific ignition sources and accurately depicting fire spread across extensive landscapes. By incorporating variables beyond fire weather, however, we successfully attributed drivers of much of the fire activity and BA with reanalysis meteorology and remote sensed land data. The event in Greece could not be uniquely attributed to one source; instead, all elements, including human influence, strongly contributed to the prediction. Availability of input data, related to the complexity of fire at the urban interface, limited the ability of the modelling frameworks to accurately diagnose contributions to the Alexandroupolis fire. Future developments aim to further expand on the variables related to manmade ignition sources and suppression efforts (e.g. urbanisation). A similar situation occurred in the Amazonian domain, where a combination of extended drought, human-induced practices, and weather explained fire activity. Additionally, changing patterns of human activity and fuel loads explained fire anomaly edges. It was found that the most severe burning could not be explained by one single control and often multiple factors were concomitant. Across all three regions, the extent of fire anomalies and the boundaries between extreme and non-extreme burned areas were often influenced by lack of anomalous control or by lower-than-average fuel anomalies.

**Objective 4: Assess the capacity of operational predictive systems to forecast each focal event.**

Employing fire weather indices, we have analysed current capability to pre-alert for the extreme events highlighted in this report in line with fire agency practices. Fire agencies typically use local fuel condition data to interpret FWI values, which reduces intervention on high FWI values when fuel conditions aren't hazardous. The analysis mostly concentrated on the ability of metrics such as the FWI to indicate anomalous conditions rather than their reliability in prediction. In Canada, conditions were well predicted, and FWI could even capture the timing of the release break. However, this metric is less successful in the Amazon domain, where weather is not the primary driver of fire activity. Of particular concern is our current poor capacity to pre-alert for extreme fires such as the one in Alexandroupolis, Greece. Despite indications of extreme conditions in FWI records surrounding the event, similar circumstances in late July did not yield catastrophic impacts. Comparable findings in South America further emphasise that despite weather conditions serving as persistent controls, several intense fire periods in late August and September went unpredicted, likely due to unrepresented ignition sources. This highlights the necessity of broadening early warning systems beyond fire weather, taking into account fuel availability, fuel types, and ignition variability, to enhance the reliability of fire risk assessment. The primary goal and philosophy of fire danger rating is to



indicate potential fire behaviour in presence of ignitions, and so our results are consistent with
and emphasise the caveats of FWI.

**Objective 5: Attribute each focal event to anthropogenic factors including climate**
**change and land use.**
Using three methods of fire attribution, we find that the probability of high fire weather has
been increased in all of the focal regions due to anthropogenic forcing, extreme fires have
increased due to climate change, but average BA has been largely offset by socioeconomic
factors. The highest increase in FWI has been in western Amazonia, where the probability of
high fire weather has increased by 20.0-28.5 times due to human influence. In Canada we
find that the probability of high FWI at least doubled as a result of anthropogenic forcing, in
line with findings from the World Weather Attribution (Barnes et al., 2023). In both Canada and
western Amazonia it is virtually certain that climate change has increased the extreme BA
experienced in the region (by up to an estimated 38% and 47% respectively), such as the
events witnessed in 2023. The influence of socioeconomic factors on extreme burning is less
certain for each region, except for western Amazonia where it is very likely that they
exacerbated the fires. When considering how median BA has changed in the present day
period, results show that climate change has led to an increase in most of the AR6 regions
related to our case study areas. However in all three areas, socioeconomic factors have
reduced BA, dampening the effects of climate change that would otherwise have been
experienced. When the impact of all forcings is assessed, we find there is a net increase in
mean BA in Northwest South America and the Mediterranean, but an overall reduction in mean
BA in North West and East North America, signalling that socioeconomic factors outweigh the
increase from climate change in these areas.
**Objective 6: Provide an outlook on the probability of extreme events in the coming fire**
**season**
With the weakening of a positive ENSO, it is anticipated that most regions in Southeast Asia
and South America will witness a decrease in the likelihood of anomalous conditions persisting
over the next three months. However, the ongoing Indian Ocean Dipole (IOD) is expected to
elevate landscape flammability across South America and West Africa.We provide the
expected anomaly in the next 3 months for the FWI, highlighting the likelihood of moderate
anomalous conditions in western Canada in early summer. Given the severity of the previous
fire season and the possibility of overwintering fires, this could once again lead to an early
start of the Canadian fire season. At extended lead times (greater than 2 months), forecast
skills are limited, and the model is likely to represent climatological values, so no discernible
signal about moderate or anomalous conditions is identified (Di Giuseppe et al., 2024).
**Objective 7: Project future changes in the probability of each focal event under future**
**climate scenarios.**
We projected future changes in the annual likelihood of each focal event under different
climate scenarios. In Canada, we project significant increases in the likelihood of burned areas
over the coming decades, driven by dry conditions and a climate- and CO2-driven increase in
fuel load. Climate-change mitigation efforts, particularly low emissions scenarios such as
SSP126, offer potential to limit future fire extremes. Similarly, projections for Western
Amazonia indicate a notable increase in future burning, primarily attributed to drier fuel trends.
Using a more extreme baseline than the observed event - a 1-in-100 year event under recent
climate, shows that likelihood of extremes increases with the magnitude of the extreme.
Effective mitigation actions are crucial in keeping existing conditions largely unchanged,
emphasising the importance of limiting temperature rise to well below 2.0°C. However, even
with mitigation efforts, an increase in extremes is expected in Canada as global temperatures
continue to rise towards 1.5°C, necessitating robust adaptation strategies to manage and





prepare for the rise in extreme fire events. Meanwhile, Greece's smaller domain size makes
confident projections difficult, given remaining biases and uncertainties in future projections,
highlighting the necessity of improving model projections or incorporating additional
information to constrain uncertainties on smaller scales.

## 4.2 Frontiers in Observing and Modelling Extreme Fire Occurrence



As part of this inaugural edition of the State of Wildfires report, we review current capabilities
in the observation and modelling of extreme fire. This section will not be revised annually but
will be revisited on longer intervals at times that are appropriate given progress in the
observational and modelling fields. The review serves to define the current challenges that
studies such as our face and identify pathways to overcoming those challenges harnessing
new capabilities and technologies.

### 4.2.1 Defining of Extreme Fire Events



While the study of impactful fires or unusual fire seasons is critical for understanding changes
in the exposure and vulnerability of people and the environment to fire, a number of challenges
hinder our ability to arrive at a clear and universally-accepted definition of "extreme" event.

Key challenges to the data-oriented identification of extreme events include:

- ***Lack of Consensus on Quantitative Criteria:*** There is considerable variability in the
measurable criteria used to define "extreme fire" across different studies, such as fire
size and area affected. Additionally, there is no well-defined statistical threshold
established for what constitutes an outlier event, which is crucial for consistent data
analysis and classification.
- ***Geographic Variability:*** Definitions of what constitutes an extreme fire vary
significantly by region, complicating the establishment of a universally accepted
definition. The size threshold can range from >100 hectares to >100,000 hectares
depending on geographic location, influenced by local fire regimes.
- ***Evolving Definitions:*** Over time, the definition of extreme fire has evolved and
expanded, leading to ambiguity. This evolution means the term now encompasses a
wider range of fire types and behaviours. Climate change is imposing an increasing
trend in fire severity in many regions, suggesting that definitions should have the
flexibility to evolve.
- ***Context Dependence:*** Definitions of extreme fire are often context-dependent,
varying with ecosystem types and fire histories. The baseline for defining what is
considered extreme is not standardised, such as whether it should be based on fire
return intervals or on the quantified damage to ecosystems and societies.

Key challenges to the knowledge-oriented identification of extreme events include:

- ***Lack of Consensus on Qualitative Criteria:*** There is significant variability in how
criteria such as fire behaviour, resistance to control, and ecosystem and socio-
economic impacts are interpreted and integrated into definitions of "extreme fire." This
variability reflects differing expert opinions and knowledge bases. The challenges here
principally relate to the subjective nature of what constitutes significant impact, whether
ecological, economic, or societal, further complicating the establishment of a clear and
universally accepted definition.
- ***Terminological Overlap and Redundancy:*** The term "extreme fire" overlaps with
other fire-related terms such as "catastrophic fire" and "megafire", which may also lack
clear definitions or be used interchangeably, leading to confusion.





- ***Influence of Language and Culture:*** The interpretation and usage of the term can differ across languages and cultures, influencing how extreme fires are defined and reported in the global scientific community.
- ***Societal Influence on Scientific Terminology:*** Scientific terminology often evolves in response to its usage in broader societal contexts, such as media or public discourse. As the societal challenges arising from wildfires ultimately motivate scientific inquiry, the language used in scientific communication must be adaptable and responsive to ensure it remains relevant and accessible to non-scientific audiences.
- ***Scientific Rigour and Clarity:*** There is a need for a definition that is clear, consistent, and scientifically rigorous to allow for standardised and repeatable measurements across studies. Existing definitions often fail to meet these criteria comprehensively.

Defining the term 'extreme fire' and how to address it requires a significant and inclusive effort across the fire science community. We actively avoid a strict definition of 'extreme fire' here due to the risk that the wider fire science community rejects our proposed solution, prompting confusion. Instead, we adopt a broad definition of 'extreme fire' as discussed in **Section 2.1.2**. Both of the challenges above can be addressed by developing standardised criteria and protocols for defining extreme fires (Chu et al., 2023). This task has been approached with respect to the term "megafire" although it remains to be seen if that intervention will be effective (Linley et al., 2022). Doing so requires a transdisciplinary approach, including input from the science community, fire practitioners, legislators, and impacted communities at a minimum (Shuman et al., 2022).

### 4.2.2 Observing Extreme Fire Events

Global-scale data for characterising extreme fire events are primarily sourced from satellite observations, notably active fire detections, BA maps, and tracking of smoke plumes. However, to accurately define how extreme a fire event is, it is crucial to contextualise present-day observations within historical data. Unfortunately, the historical records of satellite-derived active fire and BA products are relatively short. The longest coherent observations on a global scale are derived from the MODIS instruments onboard the Aqua and Terra satellites, launched in 1999 and 2002 respectively. Various global BA products, such as MCD64 product family (Giglio et al., 2018) and FireCCI51 (Lizundia-Loiola et al., 2020), as well as active fire data like the MCD14 product family (Giglio et al., 2016), have been generated based on imagery acquired by MODIS. Although these time-series now span over two decades, they are still relatively short when compared to the decadal to centurial fire return intervals observed in many ecosystems.

Pre-MODIS satellite data, like that from the AVHRR program, exists and provides a continuous imagery archive from 1982 onwards. Although efforts are ongoing to generate a coherent BA product from AVHRR data (Otón et al., 2021), there are limits to the global applicability of these products. For example, unresolved challenges stemming from coherence issues between imagery from different AVHRR sensors result in artefacts and spurious trends in various regions worldwide (Giglio and Roy, 2022), although this has been debated by other authors (Pullabhotla et al., 2023). Other pre-MODIS data available are the Along-Track Scanning Radiometer (ATSR) and Visible and Infrared Scanner (VIRS) active fire data (Giglio et al., 2013; Arino et al., 1999), but with important limitations concerning detection sensitivity, missing data and only allow for the expansion of existing time-series by 3 years (start in 1997).

Efforts are ongoing to extend the MODIS time-series by incorporating active fire data from ATSR and VIRS with BA data (e.g. Chen et al. 2023). However, due to the different characteristics of these data, creating a coherent, multi-satellite time-series of active fire data and/or BA is not straightforward. Concerns also arise with the impending decommissioning of



MODIS-Terra, raising doubts about the continuity of existing long-term fire records. However,
operational satellite sensors such as VIIRS onboard NOAA's series of satellites and SLSTR
onboard the Sentinel-3 satellites offer promising capabilities for medium-resolution BA
mapping (e.g. Román et al., 2024; Lizundia-Loiola et al., 2022). Urgent attention should be
directed towards developing methodologies to integrate these new datasets into a coherent,
long-term BA dataset. Furthermore, advancements in medium-resolution satellite data
availability and revisit times, particularly from Landsat and Sentinel-2, now enable global BA
mapping at spatial resolutions as fine as 20-30m (e.g., Roteta et al., 2021; Chuvieco et al.,
2022), suggesting a potential future direction for coherent long-term global BA monitoring.
While BA is a key variable to characterise extreme fire occurrence, multiple other aspects of
extreme fires can be characterised using satellite data. BA products can be used to cluster
burned pixels in burned patches to obtain the number and size of individual fires (Archibald
and Roy, 2009). Furthermore, the daily fire rate-of -spread, length of the active fire line and
spread direction can be extracted from the daily fire expansion (Andela et al., 2019). These
algorithms, such as the Global Fire Atlas used in this study, give global scale, coherent
estimates of patterns and trends in fire number, fire size and rate of spread. However, these
algorithms are sensitive to the temporal accuracy of the per-pixel burned date detection, the
spatial resolution of the BA product, and any errors within each product. Recent advances
focusing on clustering VIIRS thermal anomalies and extracting fire rate of spread, fire
expansion and length of the active fire line show promising results (Andela et al., 2022,
Hantson et al., 2022, Chen et al., 2022) but have so far not been developed globally. Future
development towards a global product should allow for a more detailed characterization of fire
characteristics in near real-time, well-suited for detection and quantification of fire extremes.
Active fire detections also record the amount of radiation emitted by the fire at the moment of
satellite overpass (Fire Radiative Power, FRP), within the pixel detected by the satellite. While
this information is related to the intensity of the fire, the usage of FRP has been difficult as a
low-intensity fire burning a large extent of the pixel can have a higher FRP than a high-intensity
fire burning a small fraction of a pixel. These complications have limited a more standardised
and operational usage of FRP for quantifying fire extremes. Advances in active fire detection
from higher resolution sensors may allow for a more comprehensive estimate of fire intensity
when combined with FRP estimates from coarse resolution sensors (Schroeder et al., 2014,
3211 2016).

### 4.2.3   Predicting Extreme Fire Events

Since the 1970s, fire predictions have relied on empirical fire behaviour models tailored to
specific ecosystems (Bradshaw et al., 1983; Noble et al., 1980; Stocks et al., 1980; van
Wagner, 1987), becoming pivotal tools for fire management agencies (San-Miguel-Ayanz et
al., 2013). The ease of implementation and the availability of weather data have contributed
significantly to their widespread adoption. However, despite their utility, several studies have
highlighted the limited effectiveness of the FWI and similar metrics in fuel-limited ecosystems,
where fires are driven by the short-term superficial drying of intermittently available biomass
(Yebra et al., 2013). The absence of consideration for actual fuel availability presents a
constraint to the meaningful application of the FWI in savanna-type ecosystems. Beyond
weather conditions, the remaining prerequisites for fire activity—namely, fuel and ignitions—
are intricately linked to vegetation state, lightning activity, and human behaviours. Improving
fire forecasts beyond solely considering fire weather could be achievable by accurately
describing these components. This has been widely recognised in the global vegetation-fire
community for several decades (Hantson et al., 2016), and consequently great advances have
been made to address this through the development of fire-enabled DGVMs, as used in this
report. However, explicit representation of these processes introduces biases and instabilities
that, when used in isolation, limits their utilage for assessing climate and human drivers of BA
extremes (Hantson et al., 2020; Burton, Lampe et al., 2023).



The availability of remote observations for fuel, either independently (Yebra et al., 2018) or supported by modelling frameworks (McNorton and Di Giuseppe, 2024), has demonstrated potential in aiding the development of new fire models and indices that partially incorporate fuel considerations into their formulation (Di Giuseppe, 2023, Hantson et al., 2016). However, it is the emergence of the data-driven revolution that holds the promise of significantly enhancing our predictive capabilities for extreme fires (McNorton et al., 2024), and has driven the development of the semi-empirical tools used in this report. Not only can these tools enhance predictive capability for extreme fires, they also present an opportunity to disentangle the drivers of the prediction, giving us the capability to address or at least understand the causes of the event, as demonstrated by the POF and ConFire frameworks used here. The coupling of FireMIP models with observational data (Burton, Lampe et al., 2023 and used here), also showcases the potential to bridge the advanced modelling capabilities of FireMIP with application-specific approaches such as ConFire and POF.

Despite these technological advancements, widespread adoption is unlikely to occur suddenly, as there typically exists a delay between the creation of new indices, their operationalization, and global acceptance by those responsible for fire prevention and control.

Another emerging element is the recent availability of fire danger predictive systems at the seasonal and subseasonal timescales (Di Giuseppe et al., 2024). Currently, there is limited evidence on how these longer-range tools could contribute to prevention planning and adaptation strategies. While they exhibit minimal skill beyond two months, they may offer valuable pre-seasonal warnings under specific conditions established during important atmospheric modes of variability. Certainly, this aspect will need to be further investigated in the following issues of this report.

### 4.2.4   Attributing Extreme Fires to Global Change

The prediction and management of extreme fire events have become increasingly complex due to the multifaceted impacts of global change. Climate change exacerbates fire risks through rising temperatures, altered precipitation patterns, and more frequent and severe droughts, as shown in Canada and NW Amazon in this report. These climatic shifts affect vegetation productivity, with elevated $CO_2$ levels potentially increasing biomass and thereby providing more fuel for fires. Nutrient deposition and other environmental changes influence ecosystem responses, further altering fire potential. Land use changes and management practices also significantly influence fire dynamics. For example, human activities such as deforestation, urban expansion, and agricultural practices can both mitigate and exacerbate fire risks, with socioeconomic factors shown to have a strong influence on overall extreme fire likelihood in NW Amazon, and potentially contributing to increases in BA in 2023 in some areas of Greece. Effective land management strategies, including prescribed burns and forest thinning, are crucial for reducing fuel loads and minimising fire impacts. While climate driven estimates of extreme behaviour are plentiful, few modelling frameworks take into account most of these dynamic factors and their interactions (Rabin et al., 2017).

We have used model-data fusion techniques that account for these factors in this report, and have been able to attribute some of their influences in certain places. This report utilises semi-empirical models that blend empirical data with process-based understanding to better predict fire behaviour. Quantifying uncertainty in these models is essential, especially when dealing with extremes. By generating probability distributions, researchers can better understand the likelihood of various fire scenarios, informing more effective management and policy decisions.

Thanks to the uncertainty quantification techniques, we have been able to ascertain where we are confident in our attributions. However, uncertainties still remain, many from not considering



the complex interactions and feedback onto fire, some from fire itself as it consumes fuel, and affects from weather. Coupled vegetation-fire models explicitly represent many of these feedbacks. However, current FireMIP models struggle to accurately simulate extreme fire events (Hantson et al., 2020). One key factor hampering improvements in model development is our limited understanding of factors driving fire extinguishers in a natural setting. While much process based knowledge exists on the factors influencing fire start and fire spread, only limited knowledge exists on the myriad of factors that can stop a fire, from changes in fuel moisture, structure and heterogeneity to landscape fragmentation and fire fighting, and how these interact (e.g. Finney et al., 2012). Without a strong theoretical understanding of these factors, process based modelling of extremes at a global scale might be limited in the near future. For the 2023 focal events, we have shown that low fuel loads and variations in human modification of the landscape can limit fuel spread (**Figure 15, Figure 16, Figure S13, Figure S14**). However, we only look at a handful of events and further examples are required at larger scales to inform improvement of process-based rates of spread in fire models.

To move forward, we need to combine these concepts in attribution techniques and quantifying uncertainty with coupled vegetation fire models, such as in FireMIP. Early attempts of this are promising – ConFire (used in **Section 3.3**, **Section 3.4** and **Section 3.5**) borrows many of the FireMIP-style model concepts, though it still lacks many feedbacks from fire itself. We have also used the latest FireMIP models coupled to an uncertainty framework for broad-scale, uncertainty-based attribution (obtained from Burton, Lampe et al., 2024). But they struggle at reproducing the tails of distributions where extreme events are found. Another way to develop these techniques is to move towards an integrated system that would inform both attribution and future projections in a seamless way. We make some progress in this direction here using tools such as ConFire, by using the information gained from fire drivers to build future projections, however there is more work to do to link statistical approaches for today's fires to future projections.

The human role in driving fire and extremes is hard to represent. Despite the often reported influences people have on both increasing extreme burning or causing the observed decline in global BA, the role of humans in the landscape remains hard to capture and on the whole, remains one of the most uncertain aspects of this report. Agent-based modelling (ABM) is trying to address this by simulating the behaviours and interactions of individual human entities (e.g., deforestation, crop residue burning, suppression, etc.) within a given environment (Ford et al., 2021). This approach provides a dynamic representation of how different factors contribute to fire risk and links well with subsequent sections of the report. These approaches could be a major contributor to subsequent issues of this report. However, the integration of these advanced modelling techniques into operational use faces challenges, as there is often a delay between the development of new approaches and their widespread adoption by fire management agencies. This underscores the need for continuous improvement and adoption of innovative modelling approaches to address the growing threat of extreme fire events effectively.

In addition to fire weather index (FWI) and BA projections, it is crucial to go beyond these metrics to consider wider impacts such as intensity, and emissions. Understanding the intensity of fires helps in assessing their destructive potential and the severity of their ecological and societal impacts. Emissions from fires, including C dioxide and other greenhouse gases, contribute to climate change and air quality issues. Finally, evaluating the broader impacts of fires, such as on biodiversity, human health, and economic stability, is essential for developing comprehensive adaptation and mitigation strategies. Quantifying and understanding the uncertainty in these projections is crucial for developing adaptive strategies that can effectively respond to the evolving fire risks posed by global change.

### 4.2.5 Modelling Future Projections of Fire Extremes





Projections of extreme fire events under future climate scenarios indicate a significant increase
in their frequency and intensity. Semi-empirical models used in this report project that extreme
BA events, currently rare, are likely to become more common by the end of the century. These
projections highlight the urgent need for robust fire management strategies and policies to
mitigate the impacts of these increasingly severe fire events on ecosystems, communities,
and global C dynamics. Quantifying and understanding the uncertainty in these projections is
crucial for developing adaptive strategies that can effectively respond to the evolving fire risks
posed by global change. In our ConFire uncertainty quantification framework, we have been
able to make some confident inferences about the potential state of wildfires in the coming
decades. However, we have also identified that there is a significant amount of crucial
information that is currently beyond our reach due to the uncertainties involved. Our ability to
forecast for the upcoming season, as well as for the next 2-3 decades, requires further
refinement as we are observing mixed and uncertain responses. Beyond that, we still show
similar uncertainties in responses of Canada and Western Amazonia under different scenarios
as highlighted by UNEP (2022), which uses the previous generation of climate models – shows
a large overlap in the potential range of changes in the occurrence of fire extremes between
SSP370 and SSP585. This does not imply that mitigation efforts for one scenario will be
ineffective compared to another, but rather indicates a lack of understanding regarding the
response of extremes to these scenarios. By narrowing down the uncertainty ranges, we can
better target adaptation efforts and evaluate the effectiveness of mitigation strategies. The
reduced likelihood of extreme event recurrence in our high mitigation, however, does show
that we can start separating out how mitigation efforts might affect fire extremism, though not
in the level of detail needed for policy.
There are three main ways we may be able to constrain uncertainties in the coming reports.
The first is development of the underlying GCMs that project future change in the drivers of
BA. For individual models, this is a slow process and, beyond informing CMIP model
development, is outside what the State of Wildfires can contribute to. However, bringing in
more models, including having another model to incorporate any remaining biases in
simulated fire from the correct models into our uncertainty projection, will help us constrain
uncertainties more (Kelley et al., 2023) The second is obtaining more information and
understanding of how fire drivers relate to fire extreme's as outlined in the previous section.
Better ways of describing the statistical relationship between observed and modelled climate,
land surface and fire today is a third approach. Investigating the dynamical climatic drivers of
extreme fire conditions in different regions can help to physically disentangle and potentially
constrain sources of uncertainty in future climate projections, for example by constructing
physical storylines (Shepherd et al., 2018; Mindlin et al., 2023). These storylines of plausible
future change, or other similar approaches to quantify and explain uncertainty in projections,
provide critical information for communities to develop robust adaptation strategies (Lemos et
al., 2012) and prepare for future losses and damage caused by evolving fire risks posed by
global change. Next to understanding future uncertainty, further insights into these dynamical
drivers can support the development of improved physics-informed bias adjustment of climate
models (Maraun et al. 2017). Currently available methods to bias adjust climate models for
their use in fire models, such as the ISIMIP3BASD method, have been shown to modify the
climate change trend, particularly in extreme threshold indices (Casanueva et al 2020, Spuler
et al 2024) or increase spread in climate model projections (Lafferty and Sriver, 2023). Bias
adjustment methods should therefore be evaluated carefully and leave scope for future
method development that physically links present-day biases to future uncertainty.
**4.3   Frontiers in Studying Extreme Fire Impacts**





As above, this section will not be revised annually but will be revisited on longer intervals at
times consistent with progress in the fields of extreme fire impact assessment. This section
identifies current challenges and emerging opportunities for overcoming them.
**4.3.1    Direct Exposure of People and the Built Environment**
The wildland urban interface (WUI) has been the focus of much of the direct exposure of
populations to fire (i.e., populations residing within the footprint of a fire). While extreme fires
can become urban conflagrations – including the Lahaina fire in August 2023 – there has been
a focus on reducing fire losses in WUI through fire prevention, fuel reduction, and other
mitigation efforts (Calkin et al., 2023). The dual roles of increased population living in the WUI
in some areas (Radeloff et al., 2018) and increased fire potential has contributed to
accelerated community impacts (Higuera et al., 2023). Recent studies in the US have shown
a doubling in the direct population exposure to large fires during 2000-2019 (Modaresi Rad et
al., 2023), yet with a majority of the increase attributable to fires encroaching on the WUI rather
than recent increases in human settlement in such landscapes. They showed similar increases
in wildfires intersecting roads and energy infrastructure, thereby causing additional
complexities to transportation and energy reliability. In the US, most of the structures lost in
fires occurred in grasslands and shrublands, rather than forested environments, underscoring
the importance of looking beyond traditional forest-centric fire risk assessments. Recently
developed efforts to map the WUI at global scales (Schug et al., 2023) present opportunities
for identifying areas most vulnerable to direct fire impacts, and characterise trends in fire
exposure more broadly (Chen et al., 2024; Tang et al., 2024).
Moreover, efforts are needed to better understand characteristics of fires that result in direct
exposure of people, loss of structures, and loss of life. For example, Abatzoglou et al. (2023)
showed that fires coincident with strong downslope winds in the western US were responsible
for a majority of structure loss and fatalities during 1999-2020 despite accounting for only 12%
of all fires and burned areas. Downslope wind driven fires defy typical terrain-driven fire
behaviour by pushing fire downhill from wildlands and into communities in the WUI – often
overwhelming both fire suppression efforts and fuel treatments. During the 2023 fire year,
many of the large loss fires in Hawaii, Greece, and Chile were driven by such conditions
(Synolakis and Karagiannis, 2024). Focused efforts to diagnose characteristics accompanying
other extreme fires, including meteorological conditions such as dry cold frontal passage (Van
Wagtendonk, 2006) and pyrocumulonimbus (Lareau et al., 2018), as well as pre-existing
vegetation conditions such as extensive tree mortality (Stephens et al., 2022), can lead to
additional insight on fires most likely to result in impacts. Highlighting geographic areas prone
to such conditions and how they may change into the future can help prioritise mitigation
efforts.
One of the current gaps inhibiting the characterisation of extreme fires and understanding
drivers of fire impacts on humans, such as fatalities, evacuations, structure loss, secondary
morbidity from toxic smoke, economic losses, and impacts to food, water, energy, and
transportation systems, is the lack of comprehensive data at national-to-global scales
quantifying such outcomes. For example, wildfire morbidity data are collected in only a few
countries due to the relatively infrequent occurrence of wildfires that incur fatalities and the
lack of clear responsibility for cataloguing such information, particularly for civilians (Haynes
et al. 2019). Estimates of smoke-induced morbidities rely on identifying spikes in hospital visits
above a baseline and then extrapolating, nearly always associated with a specific event
(Johnston et al. 2021). The US state of California pioneered a systematic cataloguing of
structure losses to wildfire beginning in 2013 (California Department of Forestry and Fire
Protection, 2024). However, this initiative required a commitment of fiscal and personnel
resources from the state wildfire agency and has not been adopted elsewhere in the US,
despite the programme's success in yielding substantial additional insights about wildfire
structure loss mechanisms (Kolden and Henson, 2019; Syphard and Keeley, 2019). Canada



now catalogues wildfire evacuations, but there is considerable complexity in characterising
initiation responsibility, duration, and number of people impacted (Beverly and Bothell, 2011).
Global insurance companies document insured losses as one mechanism of economic
impacts from wildfires, but such records are neither representative of the broader losses due
to high rates of uninsured or under-insured property ownership (Hazra and Gallagher, 2022)
nor are they publicly available from private companies, leaving researchers to attempt
imperfect estimations from websites that model property prices (e.g., Conlisk et al., 2024).
While the global disaster database EM-DAT and others like it have attempted to rectify this
gap in recent years, such data often overgeneralize both the spatial extent and the impacts of
wildfire disasters (e.g., reporting all of the fires across a country for an entire summer as a
single event) because they rely on news reporting that varies widely in accuracy, rather than
official record-keeping. Expanding and improving quantification of wildfire impacts on humans
is critical to overcoming the BA fallacy that fire size is a proxy for impact and aids in developing
models that can point to the most effective mitigation solutions (Kolden, 2020). This is
challenging for many metrics, but remote-sensing based documenting of structure loss and
fire incursion into the WUI is now possible with the availability of high-resolution sensor arrays
(Frazier and Hemingway, 2021), and the widespread distribution of air quality monitoring
sensors should enable an increase in more interdisciplinary research on relationships between
wildfire smoke events and medical morbidity outcomes (Liang et al., 2021).
**4.3.2 Air Quality and Health Impacts**
Exposure to outdoor pollution is recognized as a major global risk to human health (Murray et
al., 2020). The abundance of fine-sized particles with a diameter less than 2.5 μm (PM2.5) are
of particular concern due to their ability to cause a range of cardiovascular diseases. A growing
concern is the contribution of fire smoke to air quality, not only due to increasingly large and
severe fire events across the globe degrading air quality (**Figure 7**) but also because fire has
been suggested to be more toxic per unit of PM exposure than other pollution sources
(Aguilera et al., 2021). The World Health Organization (WHO) sets global Air Quality Guidelines
on the threshold for PM2.5 exposure. Thresholds are continually revised and in September 2021
they reduced guidelines for an annual mean exposure limit for PM 2.5 from 10 μg/m3 to 5 μg/m3.
Given that 95% of the world's population is estimated to be exposed to annual mean PM2.5
concentrations of at least 10 μg/m3 (Shaddick et al., 2018), additional pollution exposure due to
increasing or severe wildfire pollution events creates an elevated health risk that also goes beyond
both current and past guideline limits.
Here we discuss three of the major challenges in quantifying the impact of extreme fire
pollution on human health. The first is accurately measuring the amount of pollution that a
wide variety of communities are exposed to and then attributing the contribution of a wildfire
event, that could be 100s or 1000s of km away, to the measured concentration. The second
is that PM2.5 is not the only pollutant of concern, the EPA regulates six pollutants of concern
for American citizens and a wildfire produces them all. The third is accurately linking exposure
to a wide range of pollutants to their associated short-term and long-term health impacts.
Tools to assess air quality primarily consist of ground-based measurements and modelling.
Ground based observations provide an accurate measurement of pollution at their location.
However, measurement locations are spatially sparse. Ongoing efforts to increase spatial
coverage include deployment of small relatively affordable particulate matter sensors, such as
the PurpleAir network, by a wide range of communities (not just scientists), and efforts to relate
surface PM concentrations to measured aerosol optical depth from satellites (Li et al., 2021).
One additional constraint of observations is that they cannot differentiate pollution sources,
but modelling can. Dispersion modelling uses emission estimates, reanalysis meteorology,
and topography to provide estimates of ambient pollutant concentrations at varying
spatiotemporal scales.



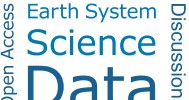

A challenge for models currently lies in the large uncertainty in fire emissions (Reddington et al., 2016; Carter et al., 2020; Pan et al., 2020). Emission data will therefore require calibration against observations and adjusting before the contribution of fire to pollution can be quantified. Improved emission datasets will increase confidence in these assessments. Given the complexity of smoke composition, other pollutants, such as gases and metals, are also present and have deleterious health implications. Increased study of the chemical composition of fire smoke plumes is therefore needed alongside monitoring of short and long range transport of these pollutants.

Environmental and personal factors both influence cardiovascular health, making it challenging to isolate the effects of fire smoke. Impacts may also not be immediate; some effects can be acute, such as exacerbation of asthma, while others emerge over a longer period, like the development of cardiovascular disease, and are thus much more difficult to directly connect to a specific extreme fire event. Conducting epidemiological studies that link fire smoke exposure to specific health outcomes requires comprehensive data collection and follow-up. These studies are resource-intensive, time consuming, and subject to potential limitations in data. In the shorter-term development of effective response planning that includes coordinated efforts between public health officials, environmental agencies, and emergency services could help in mitigating some of the health impacts of fire smoke.

### 4.3.3 Impacts on Indigenous and Traditional Communities

Indigenous peoples and local communities (IP&LC) are disproportionately exposed to extreme fire impacts because of their proximity to the land and resources from which their cultures, livelihoods and often food and medicines derive. Once landscapes are degraded through fire the access and abundance of various resources can be shifted. At the same time these communities are often less supported by the state due to access and their political and economic marginalisation linked to systemic socioeconomic disadvantages. These communities suffer not only post-fire impacts, but can be disincentivized from particular land and resource use activities because of the increasing threat of forthcoming wildfires. Whilst the multiple important values (e.g. instrumental, intrinsic and relational) associated with landscapes by IP&LCs are threatened by fire, there is a lack of systematic pre- and post-fire assessment of these impacts (van Leeuwen and Miller-Sabbioni, 2023).

Historically fire governance has added an additional burden to IP&LC, often prohibiting cultural fire use and management to the detriment of local knowledge and values, and in some cases also increasing the propensity of wildfire in tropical systems as well as savannahs (Carmenta et al., 2019; Daeli et al., 2021; Croker et al., 2023). In some contexts there is a shift towards correcting these issues and renewed interest in and support for cultural burning and Indigenous approaches to land management. For example, integrated fire management is gaining traction globally and sits at the heart of a number of interventions and international policy efforts (e.g. Fire Hub, 2023). The premise of IFM is to maximise the 'good' fire and minimise the occurrence of wildfire often through an approach of connecting knowledges (i.e. expert and place-based or Indigenous). Whilst nations sit at different stages of development in respect to IFM, and many IP&LC feel there is a long way to go, the growing interest is promising (Bilbao et al., 2019; Luque et al., 2020; Rodríguez-Trejo et al., 2022). Research is needed to better understand the effectiveness of IFM and what mechanisms and processes work best for rebalancing the influence of various forms of knowledge on fire management. For instance in North America where fire use is considered a form of medicine to the land, and anointed to particular patches of the landscape for care (Palmer, 2021). These approaches are perceived as potentially more just as they allow for the many meanings and uses of fire to exist and persist. For example, In Australia, the term 'Country' is used to convey the cultural and spiritual connection of Aboriginal peoples to the land and water in specific regions. This





link profoundly colours Indigenous peoples' experience of extreme fire events such as the
Black Summer fires of 2019-20 (Nolan et al., 2021b).

### 4.3.4 Economic Impacts

Extreme wildfires cause economic disturbance worldwide, with the nature of their impacts
varying across regions due to distinct economic structures, environmental conditions, and
mitigation and response strategies. These economic impacts arise through a myriad of
channels including property and infrastructure loss, business downtimes, supply chain
disruptions, decreased tourism revenues, health costs, reduced economic productivity, and
damages to ecosystem services. While tangible costs (e.g., insured losses such as property
or infrastructure) are relatively straightforward to measure, intangible costs (e.g., lives lost,
shortened lifespans and impaired performance/productivity due to smoke exposure, damage
to species and habitats) are more challenging to quantify owing to issues of data availability
and their occurrence over varying temporal and spatial (geopolitical) scales. Consequently,
assessing the true economic costs of an extreme wildfire event or season is inherently
challenging. Furthermore, while wildfires are predominantly destructive, certain economic
sectors may experience positive effects such as those arising from reconstruction and
rehabilitation in the aftermath, or even from the fire suppression efforts during the event
(Nielsen-Pincus et al., 2014, Meier et al., 2023a).

Building upon these challenges, existing work on quantifying the economic impacts of wildfires
has largely focussed on developed economies in Europe (World Bank, 2023), the US, and
Australia. For example, modelling extreme wildfires in Mediterranean Europe, Meier et al.
(2023b) estimate that economic losses caused by a single 1-in-10 year extreme wildfire event
amount to €162–439 million in Portugal, €81–219 million in Spain, €41–290 million in Greece,
and €18–78 million in Italy. The unprecedented 2018 and 2020 wildfire seasons in California
are estimated to have resulted in approximately $150 billion (including indirect costs beyond
the immediate burn zone) and $19 billion in economic damages, respectively (Wang et al.,
2021; Safford et al. 2022). Similarly, Australia's 2019-2020 fire season has been documented
as the country's costliest natural disaster, with economic damages, measured as a decrease
in GDP, estimated at $10 billion (Wittwer and Waschik, 2021).

While proxies of economic data can be publicly accessible from satellite data shortly after
events, this is often not the case for more traditional economic indicators such as sectoral
GDP (or alternative wealth measures), employment, hospitalisations, fire prevention and
suppression spending, house prices, or tourism revenues. These data are often not publicly
available, not harmonised across larger geographic areas, or unable to capture medium and
long-term effects. Consequently, the full economic costs of such events may only become
apparent years, or even decades, after they occur.

In addition to these data challenges, the econometric analysis of wildfires is arguably more
challenging than for other natural hazards such as earthquakes or hurricanes, because wildfire
occurrence is largely shaped by human activity through land-use choices, fire prevention and
suppression policies, and socioeconomic factors. While earthquakes can be seen as non-
human induced and therefore it is easier to find a counterfactual for a causal analysis, wildfire
occurrence correlates with many economic outcome variables of interest. Despite these
drawbacks, carefully constructed counterfactual analyses or other econometric approaches,
such as instrumental variables, can provide reliable causal estimates of different aspects of
economic impact estimations of extreme wildfires. These methods hold significant promise for
future research, offering the potential to better understand and mitigate the economic
consequences of wildfires through more accurate and comprehensive analyses, while also
helping to target suppression policies and allocate resources effectively.





### 4.3.5 Loss of Biodiversity, Ecosystem Function and Carbon Storage

The reshaping of fire regimes and ecosystem functioning by climate change and altered ignition and suppression patterns has a range of consequences for biodiversity, ecosystem function, and ecosystem services such as C storage. Mechanisms such as reduced seed quantity or quality, resprouting exhaustion, organic soil burning, and misalignment of key reproductive processes with optimal conditions under intensified fire regimes can hamper post-fire regeneration (Burell et al., 2022; Nolan et al., 2021a; Johnstone et al., 2016), Interaction with other disturbance mechanisms (e.g. drought and insect infestation) can further compound these effects. For example, in boreal forests, rising fire activity has reduced the prevalence of fire-adapted species like black spruce, with more frequent and severe fires hindering their regeneration and favouring deciduous species such as birch and aspen, thus altering forest structure and C dynamics (Baltzer et al., 2021). In Western North America, the growing frequency of high-severity fires coupled with warmer and drier post-fire conditions has been linked with poor forest recovery and transition to non-forest vegetation types such as shrublands or grasslands, impacting ecosystem services like habitat provision, water regulation, and C storage (Coop et al., 2020). Likewise, in transitional systems where tropical savannas and tropical forests can exist under the same climatic conditions depending on fire history, increased frequency can favour savannah establishment and reduced frequency can promote forest establishment (Staver et al., 2011).

There are a range of significant challenges to observing and modelling the impacts of shifting fire regimes on ecosystems and C storage. Fire regimes vary widely, leading to inconsistencies in historical data and complicating model development. Short-term observations often miss long-term trends, especially in areas with episodic recruitment (Nolan et al., 2021a; Coop et al., 2020). Ecosystem responses are complex, with species-specific reactions influenced by fire-adaptive traits and genetic variability (Grau-Andrés et al., 2024). Fire regimes also respond to multiple disturbances that can be difficult to disentangle, such as climate change and extreme events alongside land use change or fire suppression (Nolan et al., 2021a; Coop et al., 2020).

To overcome these challenges, researchers are leveraging new technologies. Advanced remote sensing, including satellite imagery and drones, provides detailed data on fire extent and severity, enabling real-time and long-term monitoring of outcomes (Burell et al., 2022; Baltzer et al., 2021). Improved modelling techniques, such as process-based simulation models and the integration of big data with machine learning, enhance predictive capabilities (Coop et al., 2020; Nolan et al., 2021a). Enhanced field data collection through long-term monitoring networks and citizen science initiatives also contributes to comprehensive datasets (Baltzer et al., 2021; Coop et al., 2020). Other opportunities include advances in genomic tools for understanding species-specific fire adaptations, climate-adaptive management frameworks incorporating real-time data and predictive models, and the use of AI and machine learning for better fire impact predictions (Nolan et al., 2021a; Coop et al., 2020; Grau-Andrés et al., 2024).

A good example of emerging opportunities stemming from the assemblage of large datasets and application of AI is the comprehensive meta-analysis by Grau-Andrés et al. (2024). By analysing data from published studies, the meta-analysis detected systematic tendencies for intensified fire regimes to reduce plant abundance, diversity, and overall health globally, with stronger effects from increased fire severity than frequency. Forests, particularly conifer and mixed forests, are more negatively impacted than open-canopy ecosystems. Woody plants are more susceptible than herbs to frequent fires due to their slower growth and recovery rates. These impacts are more severe in ecosystems with arid and cold climates, where plant recovery is further hindered by limited water and lower temperatures. Additionally, step-changes from surface to crown fires in forests can significantly reduce plant abundance, diversity, and health.





### 4.3.6 Nature-based Solutions and Net Zero

Terrestrial ecosystems remove about 30% of annual anthropogenic C emissions through enhanced growth and ecosystem recovery, but this is partly offset by land use change that increases annual C emissions by about 12% (Friedlingstein et al., 2023). Emissions removal or avoidance through nature-based solutions has therefore emerged as a key component of regional, national and corporate net zero strategies (Seddon et al., 2020). Forestry projects, focused on planting, restoration, or conservation of forests, are one sector of nature based solutions that are at particular risk of fire, which can result in inaccurate C accounting and reversal risks. Spatial clustering of C projects in specific geographic areas may compound risk from climate variability or change, like the impacts of El Niño. In C markets, projects often account for reversal risks by contributing a proportion of their credits to a buffer pool. Nevertheless, such allocations may not be adequate in regions exposed to increasing wildfire extremes (Badgley et al., 2022; Anderegg et al., 2024).

Despite these challenges, C markets also provide opportunities to unlock the resources required to improve management of fire prone ecosystems, like savannas (Russel-Smith et al., 2015) and temperate forests (Nikolakis et al., 2022) with benefits for ecosystems, climate, and local communities. For these markets to be effective and scale, they require accurate and transparent monitoring systems, enabling science based fire management, accurate accounting of C losses from fire, and assessment of reversal risk from fire. Novel satellite based monitoring systems can provide early warning and response to wildfire activity and improved estimates of ecosystem C stocks. Nevertheless, field based studies remain critical for understanding both immediate and long-term impacts of fire on forest ecosystems (Silva et al., 2020) and novel models forecasting evolving fire risk at decadal time scales are required to improve management strategies and making credible C claims.

### 4.3.7 Water Quality and other Aquatic Impacts

Fire impacts freshwater ecosystems mainly through (i) the loss of vegetation and litter cover and (ii) the enhanced input of soil, sediment and residual ash. The former leads to a reduction in rainfall interception and evapotranspiration from plants, and thus a greater proportion of rainfall reaching streams, lakes and reservoirs (Smith et al., 2011). This increased runoff from burned hillslopes can lead to accelerated erosion, debris flows and localised flooding (Shakesby and Doerr, 2006).

Wildfire ash is typically enriched in nutrients and contaminants (e.g. metals, polycyclic aromatic hydrocarbons) compared to vegetation and soil (Bodi et al., 2014; Sánchez-García et al., 2023) and together with soil and organic debris can directly and indirectly affect water quality. These include enhanced turbidity, temperature, nutrient and toxin content, and decreased dissolved oxygen. Associated effects can be increased mortality in freshwater ecosystems, algal blooms, as well as water quality impacts for water supply catchments (Smith et al., 2011). For example, increased organic matter in water can be costly for water treatment plants to remove. The 2016 Horse River Fire affecting Fort McMurray, Canada, caused $9 M in additional water treatment expenditures (Pomeroy et al., 2019). The magnitude of such impacts will depend on the type of ecosystem burned (including the presence of any legacy contamination), the size and severity of the fire, the rate of vegetation recovery and critically also the intensity and patterns of rainfall events following the fire (Shakesby and Doerr, 2006; Nunes et al., 2018). The direct hydrological impacts of fire are relatively well understood, and models have been developed to support catchment managers in assessing risks to water supply (Nunes et al., 2018; Neris et al., 2021), however, given the wide range of potential post-fire rainfall events, large uncertainties remain in predicting actual outcomes.



Fire emissions to the atmosphere also contain compounds that can be either nutrient-enriching
or toxic in aquatic ecosystems at high concentrations (Hamilton et al. 2022; Perron et al.,
2022). While the impact of 2023 fire events on marine ecosystems has not yet been
extensively evaluated in the literature, past extreme fire events were linked to disturbance of
the open ocean productivity, sometimes thousands of kilometres down the fire plume pathway.
In the Northern Hemisphere large fires across Siberian forests and peatlands during the
summer of 2014 were suggested to contribute significant reactive nitrogen inputs to the
surface Arctic ocean, helping boost and sustain an unusually large phytoplankton bloom
(Ardyna et al., 2022). A similar increase in East Siberian Sea productivity (by over 200%) was
attributed to direct nutrient (nitrogen) deposition and black carbon-induced fast-ice melting
from increased fires on the Eurasian continent in 2019-2020 (Seok et al., 2024). Coastal
community shifts were observed in the Santa Barbara Channel (California, USA) during the
2017 record-breaking Thomas fires (Kramer et al., 2020; Ladd et al., 2023). In the Southern
Hemisphere, the unprecedented Black Summer fires which occurred in southeastern Australia
in 2019/2020 fueled a continent-sized phytoplankton bloom in the iron-depleted southern
Pacific Ocean (Tang et al., 2021). The C uptaken and sequestered to ocean depths by this
bloom was reported to have offset the C emitted by the fire itself (Tang et al., 2021), although
this enhanced oceanic C uptake is likely to be temporary as changes in oceanic productivity
and C sequestration efficiency, as a function of wildfire nutrient inputs, fluctuates each fire
season (Hamilton et al., 2022; Wang et al., 2022).

**4.4   Roadmap for Advancing the State of Wildfires Report**

The fire science community is currently navigating several research frontiers to support better
societal and environmental outcomes related to wildfire. By addressing these frontiers, the
field aims to provide new observations and tools that enhance preparedness, response,
mitigation, and adaptation strategies. The field is advancing its ability to observe individual
fires, assess conditions leading to extreme fires, and predict their occurrence on timescales
ranging from hours to decades. Additionally, there is increasing focus on monitoring and
modelling the diverse impacts of extreme fires on society, the environment, and the economy.

The State of Wildfires report presents key opportunities to harness new advances and
highlight key priorities. Here, we outline the roadmap for the State of Wildfires report in
facilitating improvements in the observation, prediction, and modelling of extreme fires and
their impacts.

**4.4.1   Observability**

**The State of Wildfires report will advocate for and utilise new harmonised fire**
**observation products.**

Consistent, long-term records of fire extent and properties are fundamental for studying
extremes, which cannot be characterised without reference to historical ranges. On a global
scale, the MODIS instrument has provided valuable data on BA and active fires, underpinning
key advances in tracking daily fire progression and fire emissions models. Over two decades,
MODIS data has enabled the detection of trends and patterns in fire activity and properties.
However, the consistency of these records is now at risk as the Terra satellite, which houses
one of the two MODIS sensors, nears decommissioning.

While fire observations from other moderate-resolution datasets (such as VIIRS) and high-
resolution datasets (e.g. Landsat and Sentinel sensors) are increasingly available, there is an
urgent need to harmonise these records with MODIS data to ensure consistent observation
across datasets. The State of Wildfires report further underscores the critical strategic need
for a continuous and harmonised dataset of fire observations beyond the MODIS era. To



support future iterations of the report, we will advocate for the provision of harmonised products within the Earth Observations communities.

**The State of Wildfires Report will facilitate a community effort on a protocol for defining extreme fire events or fire seasons.**

This report emphasises important issues in the identification of fire extremes, consistent with those raised elsewhere in similar contexts such as the definition of the term 'megafire' (Linley et al., 2022). In future years, regional experts would benefit from a protocol or guidelines that can be used for categorising extreme fire events. To support future iterations of the State of Wildfires report, we will coordinate workshops with broader sections of the fire science community with the aim to produce guidance for future years. Central to this task is the inclusion of communities from broad geographies so that any output respects fire impacts that are considered to be regionally significant.

**The State of Wildfires Report will stimulate progress on combining multiple fire observation streams to better identify and characterise extreme fire.**

This report highlights the need to advance our capacity to observe fires that are impactful in diverse ways at large scales. In particular, there is a growing need to move 'beyond burned area' and towards a wider set of intensity, severity and behaviour metrics that often relate more strongly to impacts on society and the environment than BA. The integration of individual fire data from the Global Fire Atlas in this iteration of the State of Wildfires Report is one example of including wider fire parameters such as size and rate of growth, which are expected to correlate more strongly. Applications of the dataset could be explored further; for example, with the Global Fire Atlas it would be feasible to assess the occurrence of days with a large number of synchronous large fires which are known to present among the greatest challenges to fire management and suppression attempts (e.g. Abatzoglou et al., 2021). Impactful fires might also be better identified if their intersection with exposure layers, such as population centres (e.g. Modaresi Rad et al., 2023), is considered in such assessments. Similarly, combining datasets that describe individual fire behaviour with observations of fire radiative power and estimates of biomass combustion from data-driven models might further inform the identification of intense or severe fires that are often tied to the greatest consequences for ecosystems and society (e.g. Nolan et al., 2021a). The State of Wildfires report must stimulate progress on moving 'beyond burned area' and combining diverse observational capacities to better identify and characterise extreme fire events, and we intend to expand our use of such insights in future iterations.

### 4.4.2 Predictability

**The State of Wildfire will advocate for the use of extended range forecast to identify early onset of fire weather conditions**

Currently, global systems for monitoring fire danger primarily rely on short to medium-range weather forecasts, typically up to 10 days. However, evidence suggests that anomalous conditions conducive to fire weather can be confidently predicted up to one month in advance using state-of-the-art seasonal forecasting systems. In certain regions, this prediction window can extend to two months. Generally, beyond this period, forecasts do not outperform climatology.

However, an extended predictability window of up to 6-7 months is possible when anomalous fire weather results from large-scale phenomena such as the El Niño-Southern Oscillation (ENSO) and the Indian Ocean Dipole (IOD), which have teleconnections affecting regions like Indonesia and Australia.



The State of Wildfires report further emphasises the benefits of extending the forecast range to the subseasonal to seasonal timescale. This extension provides an outlook for the establishment of anomalous fire weather conditions, offering valuable advance warning and preparation time.

**The State of Wildfire will stimulate progress on the use of AI and informatics methods to aid the forecast of fire activity globally.**

Despite the increasing importance of fire management, prediction systems still rely on methodologies developed (and ongoing) since the 1960s, utilising empirical models of landscape flammability tailored to specific biomes. The Fire Weather Index (FWI), widely employed by fire management agencies due to its ease of implementation, has limited skills in fuel-limited ecosystems like savannahs. Tropical savannahs globally account for over two-thirds of mean annual BA (Jones et al., 2024). Additionally, while FWI metrics express the potential for various aspects of fire behaviour if ignited, they do not consider human or natural sources of ignition, making their correlation with fire activity limited in many parts of the planet.

Recently, there has been a notable surge in the potential of data-driven applications. Employing machine learning techniques to accurately predict fire occurrence globally at high resolution, up to ten days in advance, could become a reality. Data-driven methods could attribute a probability of fire occurrence with increasing precision, learning from available data where and when human behaviour will trigger an ignition. A notable advantage of these techniques is the facility to ingest novel input data, such as road density, without needing to directly explore the complex physical relationships that govern the occurrence of fires.

While not technically AI, ConFire uses Bayesian statistics within an informatics framework to combine a data-driven approach with process-based modelling, advancing our understanding of uncertainty around fire drivers and future projections. Current ongoing developments in ConFire, aimed at a more flexible modelling structure, could be utilised in future reports and would further reduce uncertainty, providing higher accuracy in predicting fire event drivers. For example, the controls in ConFire are currently assumed to be linearly related to burned area, and this is in the process of being updated to more accurate relationship curves that are informed and optimised by the model, based on observed data (Barbosa 2024). Another development in progress is updating the framework to enable one probability distribution to be applied to both the mean or the tails of the distribution of burned area, instead of having to select between them as is currently the case.

A priority for these developments is obtaining more data, including near real time counterfactual data, improved fuel information, and more detailed data on human/fire interactions, especially over deforestation areas such as Amazonia (See Evaluation section in the Supplement).

More generally, we also need to find better ways to represent extremes in the data, which includes an improved understanding of what classes as an extreme. A data-driven approach could be used to inform this; for example, machine-learning could be applied to large observation datasets to find any clustering of 'normal' fires, and pick out unusual fires as extremes. This would still need to be used in combination with expert solicitation though, to understand where fires have had extreme impacts on the ground that may not show up in the data.

The State of Wildfires report underscores the benefits of exploring the emergence of these new technologies to understand their potential and assess prediction skills.



### 4.4.3 Attribution

Fire attribution techniques are still relatively new compared with more established approaches for extremes such as heatwaves. Part of the challenge in attributing fire is that it is a complex hazard comprising multiple compound risks across both meteorological and human drivers, as shown in this report. A wide variety of techniques therefore exist across the attribution literature to represent fire, from fire weather indices, drought indices, and fire weather proxies such as vapour pressure deficit or individual driving variables (temperature, precipitation, relative humidity), to BA from fire models or data-driven techniques. We have applied 3 different approaches here to give a robust answer to the questions of how anthropogenic forcing and climate change are driving changes in the fires we experience today. We address changes in both fire weather and burned area. Further work could use other fire weather indices, multiple CMIP Global Climate Models, or ensembles from other climate models as another approach.

Using the ISIMIP impacts attribution framework enables us to tackle the questions of how BA has changed in response to total climate forcing, socioeconomic factors, and all forcing across multiple fire models for the first time. There is the opportunity to extend this to more fire models, other reanalysis driving datasets, and ideally to extend this up to the period covered in this report. Currently the driving data and simulations go up to 2019 for example, and we can only estimate the additional changes in BA due to climate change covering 2019 to 2023 through extrapolation. However our analysis suggests that this could make an important difference in the results, for example ConFire results for Canada show a 0.7-6.22% increase in BA over 2000-2019, whereas if the analysis is extended to include the 2023 anomaly this increases to 0.7-38.0%. Achieving near-real-time data and analysis has been one of the driving ambitions of this report, and can help us better address policy-relevant impact questions more quickly after an event has happened. This is becoming increasingly possible, with operational attribution tools coming online, and initiatives such as the World Weather Attribution tackling extreme events quickly after their occurrence. However only one of these studies has included fire so far, and a challenge for the community is therefore to be able to address these extreme fires from an attribution perspective quickly and more frequently.

### 4.4.4 Future Projections

In the coming few years, more data will become available through projects such as FireMIP and ISIMIP3b, which will provide multi-fire model projections of future BA for the first time. Using fire-enabled DGVMs enables us to more accurately model the interactions of fire and vegetation, as well as some representations of human processes of ignition and suppression. On a slightly longer time horizon, CMIP7 will also provide more Earth System Models with integrated fire feedbacks, and the community should take advantage of these modelling advances to reassess our current understanding of how fire will change in response to climate change over the next century. Currently only one model is used for future projections in this report, and in future years a more robust approach would be to include more models, as we have done in other sections of this report.

However, addressing the current limitations in wildfire modelling is crucial for this development. Models like FireMIP and ensemble approaches provide valuable insights into BA and feedbacks within the carbon cycle. To improve the utility and relevance of these models for policy and impact assessment, immediate goals include enhancing the predictions of fire beyond burned area, to consider intensity, spread and combustion rates for example. The models also provide information on emissions from fires, which could be explored in future reports using a similar weighted-multi-model ensemble to inform air quality impacts. However their performance in predicting extreme fire events is currently limited, which is where other tools such as ConFire and POF are useful to fill this gap. Addressing the critical feedback mechanisms and uncertainties in models like ConFire and POF is essential for a more



accurate representation of fire impacts, and immediate goals for development of these
methods includes integrating more comprehensive fuel data to improve predictions of burned
area, expanding the uncertainty schemes to include intensity, fire spread and emissions, and
better constraining uncertainties between SSP scenarios through improved representation of
BA drivers and their trends.
Over the next couple of years, the focus should be on the rapid integration of near-real-time
(NRT) data and leveraging advancements in remote sensing and machine learning. These
innovations will enhance the accuracy and timeliness of fire models for future projections. By
focusing on these targeted actions over the next few years, we can significantly advance the
state of wildfire modelling, making it more accurate, reliable, and relevant for policy and impact
assessment.

## 4.5 Competing Interests Statement

SV is a member of the editorial board of Earth System Science Data. The authors declare no
further conflict of interest.

## 4.6 Acknowledgements

The authors thank the following for their contributions to the identification and description of
key events in the 2023-24 fire season: Robert Ang'ila (Karatina University, Kenya); Miltiadis
Athanasiou (Institute of Mediterranean Forest Ecosystems, Greece); Davide Ascoli (University
of Turin); Chris Collins (Tasmania Fire Service, Australia); Abigail Croker (Imperial College,
London); Helen De Klerk (Stellenbosch University); Kebonyethata Dintwe (University of
Botswana); David Field (NSW Rural Fire Service, Australia); Ronald Heath (Forestry South
Africa); Konstantinos Kaoukis (Institute of Mediterranean Forest Ecosystems, Greece); Agnes
Kristina (Department of Fire and Emergency Services, Australia); Niall MacLennan (Scottish
Fire and Rescue Service); John Mendelsohn (Okavango Research Institute); Grant Pearce
(Fire and Emergency NZ, New Zealand); Galia Selaya (Ecosconsult, Bolivia); Russell
Stephens Peacock (QLD Fire and Emergency Services, Australia); Simeon Telfer (SA Country
Fire Service, Australia); Emmanuela Zevgoli (Agricultural University of Athens, Greece);
Hellenic Agricultural Organization "DIMITRA"; The Chico Mendes Institute for Biodiversity
Conservation (ICMBio, Brazil, Santarém Office). The authors thank Andrew Ciavarella (Met
Office) for guidance on using the HadGEM3-A data for the fire weather index. The authors
thank Anna Bradley (UK Met Office) for JULES-ES-ISIMIP data processing and submission to
ISIMIP repository. The authors thank the working groups "FLARE: Fire science Learning
AcRoss the Earth System" and "TerraFIRMA: Dummies Guide to using Fire Models" for
contributing to defining the report scope and establishing contributor links.

## 4.7 Author Contributions

**Conceptualization:** Jones, M. W., Kelley, D. I., Burton, C. A., Di Giuseppe, F.
**Project Administration:** Jones, M. W., Kelley, D. I., Burton, C. A., Di Giuseppe, F.
**Data Curation:** Jones, M. W., Kelley, D. I., Burton, C. A., Di Giuseppe, F., Barbosa, M. L. F.,
Brambleby, E., Lampe, S., Mataveli, G., McNorton, J., Spuler, F., Wessel, J., Parrington, M.
**Formal Analysis:** Jones, M. W., Kelley, D. I., Burton, C. A., Di Giuseppe, F., Brambleby, E.,
Lampe, S., Mataveli, G., McNorton, J., Spuler, F., Wessel, J., Parrington, M.
Jones, M. W., Kelley, D. I., Burton, C. A., Di Giuseppe, F., Barbosa, M. L. F., Brambleby, E.,
Hartley, A., Lampe, S., McNorton, J., Spuler, F., Wessel, J., Parrington, M., Hamilton, D. S.
**Resources/Software:** Andela, N., Giglio, L., Parrington, M., van der Werf, G. R., Barbosa, M.
L. F., Brambleby, E., Hartley, A., Lampe, S., McNorton, J., Spuler, F., Wessel, J., Burke, E.,
San-Miguel-Ayanz, J., Qu, Y.



**Visualisation:** Jones, M. W., Kelley, D. I., Burton, C. A., Di Giuseppe, F., Mataveli, G., McNorton, J., Qu, Y., Lombardi, A.

**Writing – Original Draft Preparation:** Jones, M. W., Kelley, D. I., Burton, C. A., Di Giuseppe, F., McNorton, J., Anderson, L., Archibald, S., Armenteras, D., Clarke, H., Doerr, S., Fernandes, P., Harris, S., Jain, P., Kolden, C., Ribeiro, N., Saharjo, B., Shuman, J., Tanpipat, V., Xanthopoulos, G., Carmenta, R., Hamilton, D. S., Hantson, S., Meier, S., Perron, M. M. G., Parrington, M., Hartley, A., J., Spuler, F., Wessel

**Writing – Review & Editing:** All authors.

## 4.8   Data Availability

BA data from NASA's MODIS BA product (MCD64A1) are extended from Giglio et al. (2018) and are available at Giglio et al. (2021, https://lpdaac.usgs.gov/products/mcd64a1v061/, last access: 2 June 2024). GFED4.1s fire C emissions data are extended from van der Werf and are available at https://globalfiredata.org/ (last access: 2 June 2024). GFAS fire C emissions data are extended from Kaiser et al. (2012) and are available at https://confluence.ecmwf.int/display/CKB/CAMS+global+biomass+burning+emissions+based+on+fire+radiative+power+%28GFAS%29%3A+data+documentation (last access: 2 June 2024). Global Fire Atlas are extended from Andela et al. (2019) and are available at Andela and Jones (2024, https://doi.org/10.5281/zenodo.11400062, last access: 2 June 2024). Regional summaries of the MODIS BA, GFED4.1s, GFAS, and the Global Fire Atlas are presented here are available at Jones et al. (2024, https://doi.org/10.5281/zenodo.11400540, last access: 2 June 2024). Studies utilising our regional summaries should cite both the current article and the primary reference for the variable(s) of interest: Giglio et al. (2018) for BA; van der Werf et al. (2017) for GFED4.1s fire C emissions; Kaiser et al. (2012) for GFAS fire C emissions; Andela et al. (2019) for the Global Fire Atlas.

Driving data, re-gridded BA target data for ConFire and ConFire outputs are available at Kelley et al. (2024, https://doi.org/10.5281/zenodo.11420743, last access: 2 June 2024). Historical (1960-2013) HadGEM3-A are available through the Centre for Environmental Data Analysis (CEDA) archive of the NERC's Environmental Data Service (EDS) at http://catalogue.ceda.ac.uk/uuid/99b29b4bfeae470599fb96243e90cde3 (last access, last access: 2 June 2024). FireMIP / ISIMIP driving and output data is available from the Inter-Sectoral Impact Model Intercomparison Project (ISIMIP) repository at https://data.ISIMIP.org/ (last access: 2 June 2024).

## 4.9   Code Availability

ConFire attribution framework code (Kelley et al., 2021; Barbosa, 2024), was incorporated into the FLAME repository (https://github.com/douglask3/Bayesian_fire_models/tree/ConFire) and will be archived with a doi upon publication. Configuration settings for **Section 3.3** are in namelists/nrt.ini, while **Section 3.4** and **Section 3.5** are in namelists/ISIMIP.ini. Scripts for reproducing plots can be found in State of Wildfire GitHub repo (currently available at https://github.com/douglask3/State_of_Wildfires_report, with an archived doi available upon final publication).

The code used to produce the FWI attribution results is available in State of Wildfire GitHub repo (currently available at https://github.com/douglask3/State_of_Wildfires_report, with an archived doi available upon final publication). FWI code can be accessed via the ECMWF GitHub (https://github.com/ecmwf-projects/geff). Met Office implementation of the FWI is based on this code, and can be accessed at https://github.com/MetOffice/impactstoolbox/tree/main with registration. All details of the data and code used for the FireMIP attribution results is documented in Burton, Lampe et al. (2023).



The current version of ibicus, used for JULES-ES bias correction, is available from PyPI (https://pypi.org/project/ibicus/, last access: 2 June 2024) under the Apache License version 2.0, and is described in detail in https://ibicus.readthedocs.io/en/latest/ (last access: 2 June 2024). The source code is available via GitHub (https://github.com/ecmwf-projects/ibicus, last access: 2 June 2024). Ibicus is also archived on Zenodo by Spuler and Wessel (2023; https://doi.org/10.5281/zenodo.8101898, last access: 2 June). Model code and evaluation for bias-correction of JULES-ES model output can be found at the State of Wildfire GitHub repo (currently available at https://github.com/jakobwes/State-of-Wildfires---Bias-Adjustment, with an archived doi available upon final publication).

## 4.10 Financial support

MWJ was funded by the UK Natural Environment Research Council (NERC) (NE/V01417X/1). DIK was supported by NERC as part of the LTSM2 TerraFIRMA project and NC-International programme (NE/X006247/1) delivering National Capability. CAB was funded by the Met Office Climate Science for Service Partnership (CSSP) Brazil project which is supported by the Department for Science, Innovation & Technology (DSIT), and by the Met Office Hadley Centre Climate Programme funded by DSIT. PMF acknowledges support by National Funds by FCT - Portuguese Foundation for Science and Technology (project UIDB/04033/2020, https://doi.org/10.54499/UIDB/04033/2020). FDG and JMC are both funded by a service contract (n 942604) issued by the Joint Research Center on behalf of the European Commission. LOA acknowledges support by the São Paulo Research Foundation (FAPESP)(projects: 2021/07660-2 and 2020/16457-3) and by the National Council for Scientific and Technological Development (CNPq), productivity scholarship (process: 314473/2020-3). GM acknowledges support by FAPESP (grants 2019/25701-8, 2020/15230-5 and 2023/03206-0). SL was supported by a PhD Fundamental Research Grant by Fonds Wetenschappelijk Onderzoek - Vlaanderen (11M7723N). SM gratefully acknowledges the support of the Dragon Capital Chair on Biodiversity Economics. ECh is being supported by the European Space Agency's Climate Change Initiative (ESA CCI) programme (FireCCI: Contract No. 4000126706/19/I-NB). CAK acknowledges support from USDA NIFA (award 2022-67019-36435). YQ was supported by the China Scholarship Council (CSC) under grant number 201906040220. MMGP was supported by a Marie Sklodowska-Curie Actions 2021 Postdoctoral Fellow funding number 101064063. HC was funded by the Westpac Scholars Trust via a Westpac Research Fellowship. SHD acknowledges support from NERC (grant NE/X005143/1) and the FirEUrisk project, which has received funding from the European Union's Horizon 2020 research and innovation programme under grant agreement No 101003890. EB was supported by the NERC ARIES Doctoral Training Partnership (grant number NE/S007334/1). JKS acknowledges support from the National Aeronautics and Space Administration (NASA) FireSense Project. NA was supported by the Sense4Fire project as part of ESA's C Cycle Cluster (ESA contract number: 4000134840/21/I-NB). MLFB was supported by the Coordination for the Improvement of Higher Education Personnel (CAPES), Finance Code 001. The contribution of SV was funded by the European Research Council through a Consolidator grant under the European Union's Horizon 2020 research and innovation program (grant agreement No. 101000987). RC is grateful to the Tyndall Centre for Climate Change Research for financial support.

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
