# Peer review of "State of Wildfires 2023-24"

_Earth System Science Data, 2024_

## Referee Comment (RC2)

**Wildfires review**

Authors have produced a remarkable assembly and collation effort! Compliments to authors. Good fit to ESSD. I will definitely recommend publication. I hope authors take some useful lessons from this review?

I understand and support this product as first-of-its-kind, a worthy initial effort intended to guide and stimulate progress on fire research. Again, good on the authors for great start! With second version promised, authors can claim 'good idea, address it in next version'. Much of what follows they will want to fix in first version! Authors and journal want best-foot-forward with this product! But some changes could delay?

I read this multiple times! Unfortunately, I feel that I understand perhaps less than half of what authors present. Why should I understand; I do not and have not engaged in wildfire research? But, as former Director of World Climate Research Programme and as co-founder and - for a too-long time - sole chief editor of this journal, I should represent at minimum a partially-competent somewhat-knowledgeable reader? Authors will of course, following scientific instincts, attempt to correct my errors and mis-interpretations. But, please take a larger view: of me as interested supportive (somewhat) knowledgeable reader. If authors have not reached  and convinced me, what have they done wrong? How do they need to change their messages? If I can't read this, how can authors expect fire managers (I deal with many, on local as well as forest-wide issues) to extract anything useful?

The manuscript remains way too long! (I deal with length issues below.)

For research audiences or (as authors hope) for "policymakers, disaster management services, firefighting agencies, and land managers" (line 131), this manuscript contains far too many deficiencies. Language ('extremes', 'serious', 'emergency', intense') remains vague and, therefore, difficult. Too many acronyms remain undefined. Maps and figures remain unreadable; figure legends in some case prove wrong. Dates of coverage and extents of observations remain inconsistent. This reader needs to scroll upward (backwards) 10s of pages and 1000s of words to find methodological details. A reader learns that satellite measurements of BA, focused necessarily on larger fires, might miss as much as 90% of burned areas (line 376; roughly 4 million km2 might actually total more like 7 million?) while, in later sections, same reader confronts author discussion of -0.2% BA anomalies? In haste to meet deadlines, authors have neglected chances and needs to check their own work and to, necessarily, impose collective oversight? To repeat: they make a great start! But they need to better guide readers and data users on their choices, their assumptions, their outcomes, and their conclusions.

**Language**: Authors take a strange but perhaps necessary approach: apply confusing ill-defined adjectives first, then later (Section 4.2.1, line 3085) highlight uncertainties and needs for improved definition. Unless authors want to dictate existing or develop new terms, I see no good alternatives to present approach. For this reader, knowledgeable disclaimers by these authors (shortened by 50%?) should have appeared earlier rather than later.

**Uncertainties**: This reader understands that wildfires represent one of our most-complex challenges. How do they start? How fast and how much do they burn? How much do fuel load, soil moisture and weather (large-scale or self-generated) influence specific fires. How will observations, predictive skills, and attributions change in unknown future? Our basic task remains identifying and then working to reduce uncertainty. To then read about 99% confidence of -0.2% changes in BA seems, frankly, absurd. I don't doubt that one can make such statements with statistical certainty. But, should we? We know that preferred fire observations carry uncertainties near $\pm$100%. We know that even our best GCMs carry very large uncertainties, particularly for next decade or one after that. We know that reanalyses, an outcome of same systems we use to specify error matrices, likewise carry substantial uncertainties. Fire models, based on these uncertain sources, very likely amplify uncertainty; at minimum they combine and must depend on all underlying uncertainty.

ESSD guidelines specify "Explicit uncertainty accounting and analysis". Here authors push those boundaries. Readers will not doubt techniques nor skills! They will, however, as I do, want to see authors' rational and best estimates to work within substantial uncertainties. Where authors need to demonstrate valid tools for handling small differences in BA, we need to know that those authors at the same time appreciate the difficulties and larger uncertainties of this science. I recommend the authors introduce an uncertainty budget early in manuscript? Show and admit where uncertainties exist? Outline your approaches: ignore, compare, test, apply two different models, whatever. Show us, however, that you understand the larger questions of uncertainties even when you must proceed.

**Atmospheric Climate Modes** At least a few of these authors adopt mode (ENSO, IOD, POD, etc.) labels and literature. (Please remember that a long time ago, before birth dates of some of you, I led a big year-long field program focused on western tropical Pacific and ENSO.) I recommend that this manuscript and this effort avoid such references and discussions. For two reasons. A) This manuscript really does not need such discussions to make basic points about S2S predictability? Your results prove, in this case, no better nor no worse - but entirely dependent on - S2S forecast skills. Difficult to dismal, at least at certain time scales, so far? B) Whatever the mode, it depends on an original identification and description now at least 20 (perhaps 40) years old. Those modes, which

initially had at-best marginal utility, have proved increasingly unstable and unreliable. These authors really do not want to step into vigorous ongoing scientific debate, nor have work judged by changing, perhaps eventually no longer valid, mode descriptions?

**Length**: too long. For many reasons.
1. Challenging for journal staff to process (typeset, proofread, etc.). We (ESSD) depend on their expertise! We should not abuse.
2. Difficult for readers. Results 10s of pages after Methods. No hot links, not easy to move around. Many apparent redundancies.
3. Impassible for fire managers. They will not use this! They will dismiss as too long, too obscure, too full of technical jargon.

Even if authors decide this first version targets other fire researchers rather than 'stakeholders'(fire & resource managers, general public), they still need to understand and respect reasons 1) and 2). I recommend a target of 80+5 pages! Even accepting necessary additions (Table of Contents, for example), authors can easily meet my 80-page target by moving large sections to Supplement, clarifying and sharpening remaining text, and - starting from a lean short product - accommodate eventual inevitable additions. You want this product to grow into respected forum? You will need, in that case, to accept substantial additional ideas and contributions!

Assumptions for 80 pages:

– Move detailed continent-by-continent descriptions to  Supplement. I know authors put a lot of work into this, but it offers very limited useful information to overall manuscript goals. Saves: 12 to 13 pages (~600 lines) plus associated figures and reference?
– Delete most discussions of ENSO, IOD, etc. Saves: 2 to 3 (estimated) pages, plus a few references?
– Keep 3.1.1 Highlights OR Section 4.1 Summary but not both. This reader would prefer Section 4.1 but moved near top of manuscript. Saves 1 (estimated) page plus a figure or two?
– Eliminate discussion and outputs from PoF because, in the end, that tool contributes very little to important outcomes? Saves perhaps 1 page?
– Check and then use acronyms correctly and consistently. Saves 100 (or more) words?
– For several sections, authors present highlight section followed by one paragraph details section. Easy to combine / reduce these in some cases?
– Reduce redundancy. Perhaps by adopting, for each major effort (descriptions, predictions, projections, etc.) an organization of methods

first followed directly by outcomes, in sequence.I predict we might gain at least one page per section, mostly by deleting redundancies?

  – Rewrite entire manuscript focusing on brevity and clarity.

I rewrote sections to test this last suggestion (my supplement). I achieved length reductions (by word counts) of 25%, 20%, 28% and 12% in these four small samples; I believe expecting 20% reduction seems reasonable. 20% * nearly 90 changeable pages saves another 18 pages? I might, for a few paragraphs or news article, assign differences between my wording and authors' wording to personal language preferences? In this case however, with length a primary problem, authors need to make substantial text reductions!

Necessary additions: Table of Contents, with hot links! One page.
Acronym list (could go in Appendix). Two pages.

By rough accumulation of assumed saving plus 3 additional pages, one easily gets to 80 net (working text) pages?

Understanding this as initial foray, I sort recommendations into three sections: now (mostly proof-type comments that authors will want to correct to make this version 'presentable'); now or future (as mostly related to length or clarity) entirely at authors' discretion; and future (for authors' discussion as they move forward).

Note: I use terms 'reader' and 'user' interchangeably.

**Now**

Line 108: "This" what? "Driven"? "Dampened"? This BA vs CO2 emissions pattern? This emission pattern? You do not want to allow uncertainty on the part of readers. 'Clearer than the truth', particularly in abstract.

Line 118: "fuel load and direct human suppression often modulated areas with anomalous burned area"? 'All' areas? 'Other' areas? Areas other than Canada and Greece?

Line 121: "extreme events". Again, use of ill-defined term 'extremes'. 'Extreme events' could include landslides, floods, etc. You want to keep reader's focus on wildfires?

Line 128: "extreme anomalies" which differ from "moderate positive anomalies" by how? Explanations, including statistical certainties / uncertainties offered later but here (again) readers will need to find exact language. So many (too many) readers will glance at abstract to check whether they want to peruse further.

Line 135, Short Summary: Excellent. ESSD editors might consider requiring similar short summaries of all papers?

Line 151: again "extreme fires", use of a fraught descriptor that authors will discuss (and, disparage?) later. I will stop recording these instances. Basic question: if authors know they confront an ill-defined (at best) term, should they use it (so often) themselves?

Lines 160 to 164: Authors should read carefully. To this reader, closing sentence lacks a conjunction ("and") or something. Confusing as written?

Line 180: "is significant" Statistically significant? Generically significant? Worrisome? (Authors used term 'concerning' in previous sentence.)

Line 193: very valid point but many ESSD readers will not know 'infamous' Wall Street (NYC) image? Slight alteration or clean up of closing sentence?

Lines 196 and following: Authors recite valid points about C emission but, as they well know, atmospheric $CO_2$ represents residual between emissions and sink. Point here: wildfires count as land emissions, no longer a land sink? Clean up and perhaps shorten particularly introductory sentence for this paragraph?

Line 205, 206: "undermine the regenerative capacity of forests (Nolan et al., 2021a) and the habitats of many endemic species being degraded in biodiversity hotspots (Ward et al., 2020). 'Undermine' regenerative capacity and habitats? With resulting degradation? Awkward sentence, needs clarification.

Line 207: Authors report correctly changed assumptions about post-fire regrowth but sink or source implications of "fire-C cycle" have always proven difficult? Perhaps a clarification or certification of one term while net equation remains unclear? Important point but needs attention to language.

Line 209: "degraded and transformed" Wildfire may degrade and transform any land, regardless of ownership or occupation. Impacts may prove more challenging or detrimental for native or Indigenous people but that represents a function of ownership, not of fire intensity? Authors allow confusion here between "fire types" (Line 211) and community impact? Valid points but deserve better itemization?

Line 215: "policymakers and involve coordination with many other stakeholders" 'that involve'? Or, 'that require'? We suffer too often when policymakers act in the abstract, without consultation with stakeholders?

Line 232: "seen as a key tool for achieving Net Zero" This reader does not see C

offsets as key nor useful tool; nor will other ESSD readers. Perhaps 'used' or 'applied'?

Line 235 & paragraph following: Very good questions. Can we expect 'stakeholders' to wade through 124 pages to find tools or answers?

Line 242 & paragraph following: Admirable sentiments but authors could reduce this paragraph by 1/2? Again, in response to final sentence about relevance: relevant perhaps but not easily available?

Line 260: Sentence here (starting from "We incorporate …") needs change in punctuation or in use of conjunctions to clarify exact intent? Later, why do authors, here and throughout, capitalize 'Earth Observations'. Do they mean to indicate satellite observations? Something different or special?

Line 292: Lose first sentence; not needed? Start with 'We identify' … in second sentence?

Line 302 "Burton, Lampe, et al., 2023": Why does this particular citation show up, throughout the document, with two names? From citation itself (Burton et al. 2023) it looks like standard reference? Something in bibliometric software? Something the authors wish to highlight?

Line 348 and following: Please confirm all dates / durations. How, for example, can this manuscript report data to Feb 2024 if BA updates only go to Andela and Jones (2023). Andela & Jones in reference list actually shows as 2024?

Please also check and confirm all acronyms in this section. This reader knows (but many will not) NASA, MODIS, ECMWF, CAMS, GFAS, GFED, etc. Consistency first then shorten where possible. Careful consistent use of these acronyms (e.g. GFAS) could eliminate many subsequent words?

Line 373 & paragraph following: important points about fire size limitations which emerge multiple times later but, here, authors could reduce this section by 20%?

Line 415, Table 1: Continent (row 2) shapefiles exist in all GIS software, open ( e.g QGIS) or proprietary (e.g. ArcGIS). Some readers will operate under institutional ArcGIS licenses but many will rely instead on open GIS software. Make this source generic rather than specific?

In my version, wrapping (at page 8) causes disconnect problem in RECCAP2 line, e.g. Ciais et al? Authors will know what they intend …

Line 425: Authors use different list numbering here (capitalized Roman numerals)

than in earlier text lists (e.g at lines 274 and following where they used Arabic numerals). Copernicus (publisher) no doubt recommends and adheres to standardized approach?

Line 430: ", and."? Superfluous or something missing?

Line 446 & paragraph following: Good important stuff, readers will need and want this info, but - again - authors could help readers (and publisher) but reduction entire section by $>$ 10%?

Line 456, "Global Fire Atlas": Why use extended title when you have already defined acronym? Shorten in all ways possible!

Line 471 & paragraph following: Useful, necessary information but authors could convey identical information in 50% fewer words?

Line 524, Table 2: Again, potentially a wrap problem around page 10 to 11, but Table 2 in my version contains duplicate entries for the Africa team? E.g. both on page 10 and again on page 11, duplicate lines?

Unfortunately, I need a map to understand section 2.3 and following.

FWI, a real-time fire forecasting tool, used to assess how 'early' (how much in advance of actual fire) users could predict that particular fire. FWI requires as inputs weather, dead fuel moisture (only dead, not living above ground?) Predicts ignition possibilities and spread rates. One advantage: moisture over three levels (depths) of soil?

Then, to address longer time scales for prediction, these authors look at FWI-compatible data (conditions) but from longer-time prediction systems (e.g. by ECMWF)?

To better understand detailed predictors (beyond weather and moisture as used by FWI), authors evaluate two additional models (applied only to their focal events?): PoF and ConFire. Here they need to introduce concept of Active Fire (AF) to supplement and complement BA? Because PoF predicts only AF? As aside, note relative incompatibility of AF with BA?

Drivers group into categories (= controls?) but drivers certainly interact (weather & moisture) and (?) predictors = drivers = single variables? Reflect interactive drivers by including them in more than one category/control?

PoF and ConFire require weather, fuel abundance, and fuel moisture, plus other? For 'other' PoF relies on vegetation type (forest vs grassland?), urban fraction and

orography, ConFire for 'other' uses land use type, urban vs rural populations, and lightning (ignition) from Table 3.

Then reader confronts / needs details of PoF and ConFire. Paragraphs, but easily shortened with no loss of info!

Finally, to develop attributions (particularly but not exclusively to climate change factors), we need to hold fire model constant (use FWI) while imposing climate scenarios (via SSP) according to definitions and recipes developed by IPCC and ISIMIP? So long as outputs of those climate scenarios meet needs of fire model (FWI)? Users need to understand that attribution works on different temporal and spatial scales?

Attribution work undoubtedly needs almost this level of detail: define terms, describe and validate climate models, set up forcing scenarios (following ISIMIP, with another Table?), describe experiments (All, Natural, climate, socio-economic, factual, counterfactual) and replicates, etc. But, do authors need to expend so much text on these processes??

Seasonal outlooks, which remain elusive for many features in many regions but good on these authors for pushing fire! Long but (in this readers view) useless discussion of ENSO, IDO, AMO, PDO, anyone's additional 'natural' mode or oscillation. Useless because best these authors can derive remains 'associations' (lines around 2680). ENSO prediction remains a fraught and failing endeavor itself, with teleconnections (on precipitation, drought, moisture) sometimes positive, sometimes negative, and occasionally, indecipherable. Those imagined non-stationary oscillations emerge only over large regions based on monthly (at best) and ocean-wide averages. These authors should not accept prior ENSO nonsense! Have the courage to show, from their data, no reliable valid correlations, positive or negative? I would start section 2.5 at line 1008: "To look more closely at impacts that historical climate oscillations (eg. ENSO, IOD, others) might have on landscape flammability …

End of diatribe. Do I have this correct? If I need a map, or graphical equivalent of a map, users will likewise?

Line 527, Bespoke Air Quality: not a main focus, move to Supplement? Or, better fit later in AQ sections/discussion?

Line 545, 2.2 Shortlisting: Good section, authors could write same content with 50% fewer words but no loss of info?

Line 589, "June 2024), and the Canadian Wildland Fire": need one fewer 'ands'

and possible punctuation change to fix this sentence. Question: wouldn't we expect Canadian Wildland Fire Information System to use FWI?

Line 596, "found to correlate": an equal number of papers show no correlations or no consistent correlations? For other regions or reasons or at other seasons, but still? At line 599, I doubt that on rigorous statistical basis, one can document any consistent ENSO etc. teleconnection to fire occurrence or intensity predictability. If authors know differently they need to cite sources? Aragao & Turbo papers do not prove what these authors hope? Authors could reduce this paragraph by ~15% with no loss of info?

Line 612, "seasonal skill is limited to 2-3 months" at best! For specific regions, specific seasons, and specific conditions!

Line 654, "Fire Weather Index": you already introduced acronym FWI. Shorten, shorten, shorten!

Line 663: I know "ESA CCI" as ESA Climate Change Initiative (I used to chair their advisory board) but users may not? Definition not until line 680?

Line 690, "that area": refers only to Greece or to all three focal events?

Line 694, Table 3: In header role, PoF control vs ConFire controls (plural)?
Here, authors describe drivers as "individual explanatory variables???
I know but many readers will not: SMOS?

Line 719, "monthly daily means": Confusing, not sure what 'monthly daily' means? Same line "FLAME"?

Line 769, "apply different modelling techniques"; use a word other than 'different' here. In this context, different can imply different from previous, not what authors intend? Paragraph that follows could reduce by 20%?

Line 857: good point about resolution of Greek data but redundant with much of prior paragraphs?

Line 861, "MODIS MCD64A1"; by now reader has seen MODIS product designations so many times, he/she no longer knows which label associates with which product? Develop (or copy) and apply standard abbreviations or acronyms?

Lines 869,870, more jargon: "MODIS Vegetation Continuous Fields collection 6.1 remote sensed data for <60°N DiMiceli et al. (2022) and collection 6 for  <60°N DiMiceli et al.": both methods applied at '<60' N or different method applied in 2015 vs 2022? Confusing or something missing?

Line 872: ibicus???

Line 882: Important paragraph. I rewrote it at ~150 words compared to authors' 185 words. See example. ~20% reduction with no loss of info?

Line 889, "maximum VPD as drivers;": I will know VPD as acronym for vapor pressure deficit but many readers will not?

Line 932, Table 5: Header row - Controls rather than Control S? 4th row (header plus 3): "temperature appriximated os ISIMIP3a/" - spelling and punctuation errors? 8th row (header plus 7): VCF needs definition? JULES acronym not defined?; 9th row (header plus 8): "tree cover plus none-tree vegetated cover simulated by JULES and bias-corrected as above": capitalize first word (as for other table entries; bias-corrected as above in this Table or in text?

Line 951, "weighted ensemble": 'weighted not explained here? Perhaps later in this paragraph? Authors could rewrite many paragraphs of this section with 15% fewer words.

Line 965, Season Forecast: S2S addresses a very complex, regionally-specific, non-stabile challenge. In any case, at much coarser spatial (and temporal?)resolution than authors need for fire prediction? Authors' figure S1 (in supplement) shows no (zero) predictability for any relevant region of North or South America and only, perhaps, weak correlations for Turkey near Greece. (What do cross-hatches designate in this figure?) So, why do authors waste time and text space on ephemeral low-resolution 'modes' (ENSO, IOD, etc.)? Emulating S2S so they have 'something' to talk about? Better (and, scientifically, more rigorous) to run their well-prepared attribution experiments to see if anything emerges. If something emerged, not valid to then label it ENSO, IOD, etc., at least not without much more work not relevant here? Getting rid of most discussion of these spurious modes would reduce manuscript by several pages?

Line 984, "ongoing debate regarding the direct influence of the IOD on Australian fires': Indeed! If we shouldn't rely on IOD with eastern Australia, why should we accept any other hypothetical correlation?

Line 995, 996, "there are few regions in the word where it is possible to establish statistically significant teleconnection between burned areas and atmospheric modes." Absolutely!! So why have authors wasted so much text on speculation?

Line 1070, "largest contributor to global mean annual totals": statistically speaking, one shouldn't identify large nor small contributions to a 'mean'?

Line 1074, North America: Perhaps a sentence about air quality impacted by Canadian fires here, consistent with South America below?

Line 1075, "contributing": 'contributed' instead?

Line 1096, "Europe: Low wildfire extent in general," Not a valid introductory phrase?

Line 1119, 1120, "Africa: Low wildfire extent in general with BA 13% below average in the African grassland, savannah, and shrubland biome.": Again, not a valid sentence? Again, grassland savannah scrubland distinctions, this time with African inflection? IPCC made definitions not widely accepted by terrestrial ecologists?

Line 1132: reader encountered identical information a few lines earlier. Details here, highlights there? We don't need both? Same comments as above apply here. "MODIS BA product": which, what version, etc. Authors need to settle on clear convenient naming system for MODIS burn products?

Lines 1143, 1144, "African grassland, savannah and shrubland biome, which is the largest contributor to global mean annual BA totals": I appreciate that authors know this but nothing in Figs 1, 2 or 3 proves this point?

Line 1146, "lower BA in savannah-like systems in 2023-24 was not observed in Australia": not clear to this reader whether this constitutes a real decline or a labeling discrepancy?

Line 1175, "58% towards total global BA and 40% towards total global fire C emissions.": this statement needs a certifying citation?

Line 1180, "BA extent was in the top three years on record": because Fig 1 shows only a single fire year, hard to credit this observation from Fig 1?

Line 1205, "prominent regional feature": refers to northern South America but not evident in biome data and not particularly notable compared to Africa or much of Asia. Authors have, no doubt, valid reason for this claim but not evident in their maps? Much of the information presented by the remainder of this paragraph repeats what reader encounters later, e.g. in Section 3.1.3.6 (starting at line 1845). We don't need two repeated versions of same data?

Line 1240, "three good rainfall years have resulted in grass fuel accumulation": here multiple years of abundant rainfall increases fire risk and occurrence? Other places in this manuscript, multiple years of higher-than-average rainfall decreases fire risk? Do authors want to or need to address this discrepancy? If, for attribution

studies, they accept months of wetness (dryness) as indicative of low likelihood (high likelihood) of fire,  does this statement expose a weakness in their assumptions? Repeated at lines 1340 & following?

Line 1281 & following paragraph: hasn't reader already received this info? Better here, where more relevant? Not in both places, please.

Line 1282, "moderate resolution satellite data": another term for MODIS burn data?

Line 1297, "higher resolution and higher overpass frequency": but reader learned a few sentences earlier that high-res satellite information confirmed what GFA implied? High-res in this case refers to Sentinel-2 at 10m, compared to VIIRS at ~400m and MODIS at 500m? Words such as 'moderate' and 'high' not particularly helpful in this case. Fig S3 very helpful!

Line 1310, "of the fire also showed high": 'also'? But, reader learned in previous sentence about ~10% difference among various products?

Line 1329: What do authors want readers to learn from this section? That GFA, when it works and when validated by other products, proves reliable? In other cases, however, particularly in urban areas, GFA often fails? Not clear to this reader? When lucky, GFA works great? When unlucky, GFA fails? Plus it detects only larger fires? So?

Line 1335, Review by Continent: Unfortunately, this section reads like WMO annual weather report: warm and wet here, dry there, only station reports, no correlations. Both here and there authors invoke ENSO when they have nothing else to pin to? This reader begins to question value of this entire section. Next year will prove different in detail but no better in overall content or conclusions? Fires always occur. Damage always ensues. Evacuations, suppressions, death: likewise. Nothing new here? "a fairly typical fire season" (line 1406), some places hot, others cold, seems the only useful summary? ESSD should not publish 'fairly typical' reports! But, without this (and WMO reports on weather), where would one find such records? Authors will have saved readers lots of work? Nothing about livestock, please! For this manuscript, readers also need zero info about fire fighting experiences or equipment. For me, excessive detail about Evros fire; how will this info prove relevant to a) other fires, and to b) larger questions about changing fire regimes? Evros fire notable only by BA? So? A fire killed two civilians? Condolences, but what use to a reader to know that info?

Line 1729, Figure 7: Potentially informative but provided here at very poor resolution?

Line 1738, "across the Country that were well in excess" why 'County' not Canada

or country?

Line 1758, "if the USA had also experienced a high fire year, the air quality would have worsened in many states": is this a guess or a statistically-valid prediction as one outcome of this work?

Line 1941: Again, what do authors want readers to have learned from this section? Admiration that authors have gathered lots of info in one place? Granted. Can a single reader assimilate all this info? No. Does this accounting, by continent or cumulatively, make any difference? For this reader: no.

Line 1983, "declines in deforestation rates and deforestation-related fires have fallen": 'declines' have 'fallen'? Silva Junior et al. paper very confusing but title probably best conveys their message "Brazilian Amazon deforestation rate in 2020 is the greatest of the decade". Not clear what the authors intend here?

Lines 2125, 2126, "Some anomalous BA starting in August and extending to November": not a complete sentence, readers need to guess at authors' intent.

Line 2136, Figure 10: Despite good intents and good efforts by authors, this figure not really readable enough to prove useful? +10% for Canada, +2% for Greece and western Amazon. This reader does not believe, based on information gained earlier from this paper, that authors can expect readers to distinguish +1% from - 1%. Reader needs some basis to minimize uncertainties here! Resolution of figures in my version = very poor?

Line 2153, 2154, "inadequacies in predicting certain ignition sources or accurately representing fire propagation across vast landscapes in current forecasting systems" A very important conclusion, noted here in case it does not re-appear later.

Line 2161, 2162: "inadequacies in predicting certain ignition sources or accurately representing fire propagation across vast landscapes in current forecasting systems": Another strong important conclusion, conveyed in highlights but repeated in detailed section? Same text!!!

Line 2165, "linked to the strong El Niño.": for reasons already highlighted, including by authors, this reader doubts that authors can assert this link?

Line 2173, "no single factor can explain the most severe fires": another strong positive statement, in highlights, waiting for details to follow?

Line 2199, Figure 11: Very important helpful figure! This reader wonders how a different fire prediction model, something other than Canadian FWI, might have

worked? At longer time scales (longer than 2 weeks or so), no fire model would show significant skill? Authors have skillfully defended choice of FWI, and it seems to make sense given importance of Canadian fires, but they must know about other fire prediction systems? (Thinking here of earlier [perhaps now dated] work with GEE by Gray et al. in ESSD: https://doi.org/10.5194/essd-10-1715-2018). Some discussion or assessment? Who else but these authors would know?

Line 2260, Figure 12: Again, informative and helpful. Here, however, reader confronts (for the first time?) 'dead' fuel vs 'live' fuel, both in reference to fuel moisture. If this dead vs live difference proves important, should authors highlight these factors for readers? 'Deap' should instead read as 'deep' in all cases? Lightning basically constant in later months in Amazonia?

Lines 2273, 2274, "lightning as a key source of ignitions in the region": referring to Canada but flash rates much higher in Greece and higher yet in Amazonia?

Line 2281, "vast, densely vegetated": referring to Canada, have authors given readers any bases for accepting this statement?

Line 2293, "intrinsic difficulties in forecasting isolated extreme events": Indeed! Major primary conclusion, but not emphasized by authors?

Line 2329, "anomalous weather conditions subsided later in the fire season": not true, at least as shown in Canadian section of Fig 14? There, weather controls look nearly identical Apr vs Sep?

Line 2356, 2357, "direct human-induced landscape changes exerted minimal influence on the extent of burned areas in Greece": to my eye, human influence (row 5 in Fig 14) looks vastly different Canada to Greece, and very important for Greece?

Line 2361, legend to Fig 14, "increases/decreased": should read as 'increases/decreases'? BA range data, which varies for each site, not clearly evident? Hidden by overlapping graphics? Time spans (X axis) also vary in this figure. Gives this reader even less confidence that I understand the figure? I would have said: a) minor fuel influence at all three sites but different signal Canada to other two; b) strong drought signal in Amazonia; c) solid weather signal at all three sites but perhaps dropping off late in season at Amazonia; d) minor human influence Canada and Amazonia but strong in Greece case? Personally, I don't understand how authors got probabilities to tenths of percent and I find no basis to trust $\pm$ 5%? A highly uncertain model (ConFire) run multiple times on multiple cases does not improve reliability? If Fig 14 presents site-specific BA anomalies relative to site-specific means, but all with same magnitude (same Y-axis units) then I get even more confused? It seems key to me to understand Fig 14 but I don't?

Line 2366, legend to Fig 14, "as a fraction of land area": I appreciate effort by authors to quote and illustrate uncertainties but this reader still does not understand the bottom graphs? Fraction of land area?

Line 2420, Figure 15. Not at all understandable by this reader. The figure legend seems wrong? E.g. this reader sees, for July and June (why this order?), top to bottom: BA anomaly (top), increase from controls (mid), decrease from controls (bot). The legend instead implies left right distributions: "left of the other two maps looks at anomalies in controls that would cause higher BA" while "right map shows drivers that would have led to lower than normal levels of burning". Not correct? June had greater -7 to +7 BA anomalies than July (-3 to+3) so, strictly speaking, user should not attempt monthly intercomparisons? Greyed-out areas should inversely correspond to top BA anomalies, for mid plots, but directly correspond for lower plots?  This seems very hard to detect. Then, four size and color dependent dots in each 0.5 degree grid box? I think I understand goals for this figure, but fails for me.

Same problems extend to Figure S13, but for two additional months (Sep, May). Also to Fig S14 for Amazonia. Again: unreadable. Sorry, these old eyes can make no sense of this arrangement. I give up entirely; skip this section.

Line 2492, Attribution. Similar to above: highlights followed by details. At first I liked this arrangement. Now I see it mostly as redundant; largely a waste of space?

Line 2501, 2502, "All forcings combined have led to an overall reduction in today's average BA across Canada." Interesting. But this contradicts all previous assessments for Canada?

Line 2509, 2510, "Climate change has increased today's average BA in the Mediterranean region, but this has been mainly offset by socio-economic factors.": reference Greece but same conclusion as for Canada: socio-economic offsets climate leading to no net change?

Line 2517, 2518, "Climate change has increased today's average BA in the Northwest region, and all forcings have led to an overall increase in burning". In this conclusion, western Amazon differs from Mediterranean and from North America?

Line 2529 to 2531, "probability of experiencing the high fire weather observed during June 2023 is more likely in a climate forced with anthropogenic emissions.": Referencing high BA across Canada, this conclusion (plus Fig 17) contradicts what reader found in highlights?

Line 2578, 2579, "BA in Canada in June 2023 was 0.8-38.0% greater due to total climate forcing in the 2003-2019 period" Perhaps labels have confused this reader, but if 'total climate forcing' = 'All', then this conclusion contradicts what we read in highlights?

Line 2599 to 2601, "uncertainties around whether total climate forcing and socioeconomic factors caused an increase or decrease in BA are higher, and the smaller region size makes detecting a strong signal of change more challenging". About Greece, safest to draw no firm conclusions?

Line 2610, 2611, "additional burning could have been up to 0.8-36.2%" but, from Line 2615 "socioeconomic conditions (95.98% likelihood) increasing BA by 0.18-5.32%. Authors lost this reader long ago on how to trust 1 to 36 vs 0 to 5. In any case, for western Amazon region we can only say with confidence: perhaps?

Line 2623, legend for Fig 18. No explanation/assignment of orange color? Oops, now I get it: orange = factual, blue = counterfactual, combination (overlap) = purple. Not immediately clear? Taking western Amazon, this reader finds 1.53 to 7.66 (2 to 8) modest increases in high BA (how authors derive 99% confidence for curves that basically overlap remains beyond me) vs 0.18 to 5.32 (0 to 5) increase in socio-economic factors. These numbers DO NOT accord with those at lines 2610 and 2615 in text. This reader now concludes, using corrected numbers, no net change for western Amazon in high BA?

Line 2667, legend to Figure 19: In this Fig, authors did not use transparent colors so colors do not blend as they did in Fig 18? If authors did not present Y-axis (probability) in log scale, users would see nothing? I figured out what bothers me about these ultra-precise uncertainty estimates: authors compare final products of long train of manipulations, each with its own (sometimes acknowledged, sometimes not) uncertainties, source observations (satellite or reanalysis), aggregations or disagregrations, final manipulation for effective presentation, etc. but only report (valid?) uncertainties deriving from the last step.

For ESSD, we need full end-to-end, uncertainties! A full uncertainty budget! For satellite data: orbit changes, sensor degradation, processing levels, cloud or aerosol or competing species absorption, etc. For reanalysis: data ingestions, processing, discarding, spatial inhomogeneity, etc.; reanalyses derive from need to produce error matrix! Processing: use of anomalies, use of relative anomalies, changes in axis ranges, etc. Uncertainties at every step, propagating, canceling, amplifying, whatever. To report only the statistical inter-comparison of final steps (e.g factual vs counterfactual) ignores a host of uncertainties inherent in how authors got that far? I would rather not check all numbers (although prior comments relative to Figure 18 suggest that someone needs to validate), but I

react negatively, as will most readers, to tenths of a percent in final precision when readers know and authors well know that basic uncertainty proves higher in every case! How can authors report -0.2% differences between model outcomes when we and they know that we can't even measure BA from satellite to better than $\pm$ 10%? I would feel very pleased to get proven wrong on this, but, with no disrespect intended for authors, they make much in these sections of very tiny differences only by ignoring much larger underlying uncertainty! ESSD readers need to see underlying systematic end-to-end uncertainty budget. Very difficult to produce, especially in this case? Perhaps, but tell us. Might mean that one can not, in fact, distinguish factors that influence max BA or median BA? Tough news, but tell us; we need to know. And we need to hear recommended solutions from these experts!

Summary: great respect for efforts and reports, but manuscript lacks an encompassing uncertainty budget.

Line 2674, Seasonal and decadal: Again, highlight followed by details, leads unfortunately  to substantial redundancy.

Line 2678 & following paragraph, IOD and ENSO: But, authors stated earlier that no statistically valid links exist between BA and atmosphere modes (line 995, 996). Why then do readers confront more discussion of IOD, ENSO ("ENSO-neutral"), etc?

Line 2688 & following paragraph: But readers learned in previous paragraph that (for Canada) "no clear signal for extreme anomalies is present". Why then does this paragraph focus on scenario differences (SSP585 vs SSP126) and mitigation actions. To remedy signal that does not exist?

Line 2701, "above PI": readers will not understand PI without definition?

Line 2704, 2705, 2707, 2708: as if readers did not already suspect ,"projections show a high level of uncertainty in all regions" and "we cannot determine if current mitigation efforts are effective". Valid, honest conclusions, but don't those render this entire section moot?

Line 2745, legend to Fig 20. If authors already concluded lack of statistical connections, and one of our best modeling groups predicts this wide (useless?, >3C in every case?) range of near-future temperature ranges (one for ocean SST in central equatorial Pacific region, the other we would have to search for) why should reader give any credence to any of this? Prove challenges by showing current weak model skill but don't waste our time with speculations?

Line 2679, legend to Fig 21: Based on careful reading of all prior guidance, this

reader has learned to disregard this information entirely. July & August for 95th % basically empty. Reader could easily have guessed at areas for 75th in July and August? Nothing against ECMWF, just that we should have learned by this point in this manuscript to trust no intense FWI probability < 100%, which basically don't occur anywhere on the planet after June?

Line 2780, future changes section: Direct mapping of BA pixels to model output pixels? No intervening fire model, predictive or attributional? Methods at Lines 1030? JULES never defined? But figure time extents do not accord with method-specified boundaries. Fig 20 definitely not, Fig 21 perhaps but only for Canada?

Line 2786, 2787, "ISIMIP3a and ISIMIP3b": this reader understands a few differences between 3a and 3b but authors provide this and most readers no help and no guidance?

Lines 2796, 2797, "likelihood of a 2023 fire event recurrence increases to 0.31% - 0.9% - two to almost four times as likely as today (Table 7; Figure 22). But, remains highly unlikely? This outcome (these numbers) very hard to extract from Table 7 - one needs to confuse min and max or cross scenarios - so data come from Fig 22? This reader unfortunately gets nothing from Fig 22, particular at level of detail implied here? In next 10 years? Nothing.

Line 2798, "different SSP scenarios diverge significantly by 2070": Really? By what criteria? Lack of range overlap, as authors specified at line 2790? Very hard to confirm authors' conclusions from Fig 22!

Lines 2802, 2803, "at least one event similar to or worse than the 2023 event occurring again is estimated to be between 16% and 25%": If reader hopes to confirm these author-specified changes (at least these seem more moderate), where should they turn. For Canada in Fig 22, likelihood range only goes 1 to 5%? Does one need to convert and guess from BA event numbers? Not acceptable.

Line 2804, legend to Table 7: No explanation of colors? Obviously related to magnitude, but how? Many cells report insignificant results (48 of 160, 30%)? Predominantly not valid for W Amazon, regardless of return rate (1 in 6 vs 1 in 100)? Marginally valid for Greece? Please fix wrapping and display for 2nd column. Entire table needs improvement! Minimum likelihood outcomes for Canada 'increase' from 0.08 to 0.31? Readers should learn what from this? Finally, here, a definition of 'extreme', in a UNEP report? But, in text later (lines 2845, 2846), UNEP 2024 seems to specify something different? This reader does not want to search for UNEP 2024; we need authors' best explanations!

Line 2825, "changes in both [djc: fuel load and fuel moisture] controls are needed to explain divergence between SSPs": very hard, perhaps impossible, for reader to

credit this conclusion based on these data!

Line 2828, legend to Fig 22: No indication of when or if signals emerge from uncertainty bounds, either in time or in event return times or % likelihood? No explanation of means, medians, whatever, indicated by lines within each bar? This reader guesses: perhaps a scenario difference by 2100 in Canada, driven perhaps by fuel abundance? Before that, nothing significant? For Greece, perhaps significant differences by 2090 and 2100, driven by moisture? For W Amazon, nothing (zero significance) for any decade or any scenario? But these personal conclusions diverge substantially from authors' narrative? Here they compare ConFire simulations to re-analyses? No indication of large uncertainties in each?

Line 2823, legend to Fig 23: Not the US-Canada border that I know? Even if outlines stay north of Great Lakes (border actually intersects lakes via mostly straight midpoint lines), remainder to west looks step-wise distorted whereas actual border remains dead (latitudinally) straight, Minnesota to Vancouver Is? Relevant because only signal seems to occur along southern border, near N Rockies and/or Glacier NP. Surprised to see, in col 1, an apparent minimum along southern border? (Do authors think readers will have any concept or trust of 0.001% difference in BA? Or any concept of 0.05 vs 0.5 % change?)  In middle and right columns, at least for SSP585, fire extents and likelihood of return seem to max in that area? This reader develops no confidence in any part of this figure? Dot and change scales not linear? Arbitrary?

Line 2844, "some areas" ... see increases ... "almost everywhere": No sense in that statement. Authors expect users to accept that areas of low BA today will become areas of preferred BA in the future? Authors or ConFire or both haven't the faintest idea?

Lines 2862, "from 1.3% to 0.67-1.78%": relevant to Greece, authors expect users to accept or understand significance of such a negligible change? Later (line 2872) authors report "no significant difference between the scenarios by 2100. Why, then, have they wasted user time and their own page budget?

Line 2889, legend for Fig 24: Same questions as for Fig 23, but why? Speculation? Waste of valuable space?

Line 2941, Summary: Useful organization here: paragraph responses to each original objective. But these paragraphs read mostly as redundant restatements of previously-discussed results? Reader does not need both? This section seems better organized and more concise?

Line 2934, "suggests": authors can do no better than 'suggest'? Reader conclusion: nothing significant, why waste my time?

Line 2993: here, reader confronts mention of "Alexandroupolis" fire, A few lines earlier, authors mentioned "Evros" fire (line 2972). Two names for same event?

Lone 3012, "release break": New term for readers? Not used previously?

Line 3044, "socioeconomic factors outweigh the increase from climate change", many readers will agree with this statement, even sans hard evidence, but preceding paragraph seems to express contradictory opinions: climate change increases probability of large BA while socioeconomic inhibit? Decide what you want reader to take from this paragraph?

Lines 3055 to 3057, "At extended lead times (greater than 2 months), … no discernible signal about moderate or anomalous conditions is identified" Very worthy and valid conclusion, but negates much or what authors worked to present? Earlier in this paragraph, more vacillating speculation about ENSO or IOD, etc?

Lines 3060 & following paragraph: Cautious but confusing? From final sentences, this reader concludes: even with best current mitigation options, warmth coupled with drought (interspersed with floods) will increase fire vulnerability? I truly worry that better weather, soil, vegetation data in Canada influence all these lncomparisons!

Line 3076, Frontiers: I and most readers will welcome and agree with these concerns. Definitions, observations, predictions, attributions, future projections. Too long in every case but see recommendations above. Very valid helpful cautious summary, ideal coming from this group? Mention SI?Also a place to encourage other contributions to proposed or other future special issues?

Line 3393: Frontiers. Good section! Here, particularly, you need to focus on research audience or fire management audience? Way too long! Could reduce by 30% with no loss of information (see above),

Line 3430, "extensive tree mortality": First (and sole?) hint at infestations. For much of northern Rockies, as for many forests in Germany, climate-enhanced (activity, reproduction) insect infestations kill 1000's of trees. At very least, this large-scale die-off moves vegetation from 'live' to 'dead' categories, a distinction invoked once or twice but not much discussed here? Impact seems to involve climate, weather, vegetation and human controls? Where would this fit in existing modeling? Or in future needs? Insect-mediated factors could grow?

Line 3460, data about fires and fire impacts: in an earlier ESSD paper, Karen Short of Montana Forest Service reported difficulties in rectifying versions of fire data

into one accurate valid record. As I remember, nearly 40% of fire reports in her US-based database proved invalid or redundant. Emphasizes dependence on remote sensing data? Strength or weakness? Also emphasizes that we currently cannot, at least not in near-real time, furnish local experts with accurate local data? Causes us to rely on media, which have even worse unstable biases? Also raises issues of industrial (including transport) fires. Not currently monitored or logged? Often, in media accounts, prove sensational. Notable impacts on emissions, air quality, evacuations, human safety, etc? Another aspect of fire currently out-of-bounds for fire researchers?

Line 3490, air quality impacts: ground-based measurements, dispersion and transport studies, plume dynamics, animal-based human exposure/response models? Add intermittent ill-defined fire emissions? Almost an impossible problem?

Line 3526, authors here tend to focus on fire management, by federal or state agencies vs IP&LC communities. Good points. But, Canadian projections (e.g. from Fig 23) pass through - on both sides of Canadian-US border) - large reservations. Smoke air quality exposures might prove adverse in those communities  and locations?

Line 3564 & following paragraphs, economic impacts: omission of or ignorance of industrial fires might prove key here? Economic impact estimates miss many long-term health costs already mentioned?

Line 3740 & following paragraphs: Back to observations, predictions, attributions, future projections. Good stuff but serious redundancies with Frontiers Section. Too long but authors need to think here about readers: what new recommendations do you want readers to remember from this section that we did not already learn from Frontiers section?

Lines 4982, 4983: URLs (e.g. https//doi.org/10.xxxx/longer number that wrap across lines don't work as hot links.

**Perhaps now**

Need a table of contents? Or, something different or better to help readers find specific information. Perhaps accompanied by short paragraph explaining what you keep in main manuscript and what additional info users can find in supplement. Supplement would also need a table of contents or equivalent?

Document would benefit from list of acronyms? Developing such a list might help authors check & track such a long list of acronyms?

Line 102: "extreme" fires? As authors repeatedly point out, community presently confronts inability to reliably quantify nor communicate 'extreme? Should these authors apply a term they later, on valid grounds, question?

Lines 131, 132: "insights relevant to policymakers, disaster management services, firefighting agencies, and land managers," But, this important target group of readers will find 124 pages daunting or forbidding?

Line 133: Recommendations - well-grounded - that follow cover basic understanding as well as "preparedness, mitigation, and adaptation"?

Line 258 & paragraph following: initial caution & questions about what constitutes extreme events arises here. Good! But, authors have already used terms 'extreme', 'extreme fire', 'extreme wildfire', leaving this reader with two questions. A) Should one use such a complex fraught term? B) What will these authors, best in their field, suggest as alternate or as quantitative?

Line 268 & following "Objectives": Good section, appreciated. Shorten? Opening paragraph suggests tools and information (good), prospective, but user just read that authors will focus in BA while first objective definitely leans toward recent observations. Somehow, perhaps in introductory paragraph of this section, remind users about intent to apply and report data and observations?

Line 326, paragraph starting here: Good paragraph, first discussion of uncertainty in application of term 'extreme', very useful caution, but authors could clean this up by 10% to 15%?

Line 396 & paragraph following: Important paragraph outlining timing assumptions and choices. This paragraph should perhaps move to top of Methods section?

Line 701 & following, Section 2.3.2.3 on Drivers: authors could rewrite this entire section with 15% fewer words but no loss of info?

**For future discussions and planning**

Move everything not global nor related to specific focal events to supplement?

Authors have, in this version, 'teased' readers with three specially-chosen focus events. Because each of those focus events includes long discussions of present predictability and future probability, won't readers want, in second version, some update on authors' conclusions for these regions? Meanwhile, authors will select a different three regions for 2024-25? Eventually, doesn't one end up with oldest, older, prior and present events?

Background, global, regional, predictions: different tools and different conclusions for each. Should these evolve into separate products? Mention here ideas about a special issue? Perhaps parse under several journals, according to?

---

## Referee Comment (RC3)

Dear Authors,

Firstly, I would like to extend my warm congratulations on your comprehensive assessment of the state of extreme wildfires for 2023-24. This manuscript represents a significant and valuable contribution to the understanding of wildfires, and I believe it holds great potential for impactful publication after addressing a few minor revisions and additions.

**Comments:**

- I understand that this is the inaugural assessment, necessitating the inclusion of many details. However, the paper is quite lengthy, spanning over 100 pages (and more than 50 pages of supplementary material), which makes it challenging to read. Some sections could benefit from greater conciseness as some repetitions and details could be referenced rather than fully detailed within the paper. To improve readability and impact, I suggest the following:
    - Streamline sections to remove repetitive content.
    - Condense detailed explanations where possible by referencing relevant literature.
    - Focus on making the key findings and methodologies more concise and directly accessible.
    - The conclusion section is currently too long and reads more like an incomplete review rather than a focused conclusion of this study. I suggest strongly reducing it and concentrating on summarizing the key results and findings of this report. This will provide a clear and concise conclusion that highlights the main contributions and implications of your study.

- For future updates, it could be interesting to include regional data where available that can cover longer periods compared to satellite data. This would provide a more comprehensive historical context and enhance the analysis.

- Existing approaches are not considered in this inaugural report. For instance, hybrid models for seasonal fire risk, fire propagation models, or new fire susceptibility/risk mapping tools are not included. Adding a discussion on these approaches could provide valuable insights and indicate potential directions for future reports.

- In future versions, it might be beneficial to include a summary of the main scientific initiatives currently in action (e.g., projects, consortia, collaborating groups). Additionally, consider adding comments on ongoing work in the main scientific agenda, including how these efforts align with the Sustainable Development Goals (SDGs).

- In future versions, more attention should be given to the Wildland-Urban Interface (WUI), human exposure, and the broader impacts of fires, such as on air quality. These aspects are crucial for a comprehensive understanding of wildfire risks and their societal impacts.

**Minor Comments:**

L100-101: The authors state in the abstract that global wildfires are increasing in frequency and intensity due to climate change. This statement may oversimplify the complexity of wildfire trends and drivers. It would be beneficial to provide a more nuanced discussion or to soften the statement to reflect the multifaceted influences on wildfire activity. This will ensure the abstract accurately represents the detailed analysis provided in the paper. In addition, a very recently published paper assesses trends in extreme fires (Cunningham et al., 2024). I think this is an important reference to add to this paper, and their methodology may be applied to future versions of this report.

L124: The values like "2.22" should be rounded to an integer or at least to one decimal place. The associated uncertainty is likely significant, and using two decimal places may artificially increase the perceived confidence in these future estimates.

L150: You can reference IPCC AR6.

L175: Reference Zheng et al., 2021 Science Adv.?

L180: Any reference for this statement?

L196-213, L215-233: While the information is accurate, it would be beneficial to break these paragraphs into smaller, more focused sections that group related information together. This will enhance readability and ensure a more logical flow of ideas.

L255-256: I agree, and for this reason, I stress the importance of being as concise as possible to enhance readability and ensure the key messages are communicated.

L258-266: Why is this paragraph here? This can be discussed later. In any case, I agree that defining extreme fires is challenging. Your title indicates "wildfires," but you consider all fires in the BA statistics, including agricultural fires. A possibility is to use "fires" instead of "wildfires" in the title and reduce the discussion on the definition of "extreme fires" here and later in the paper.

L270: I suggest moving before "This inaugural edition […]".

L275: "March 2023 to February 2024"? I suggest indicating briefly in the introduction why this period.

L284: Which fire season? Please be specific.

L290: I suggest including a broader range of sources, not just publications.

L292-315: This can be merged within the Method section.

L297: The term "fire risk" is used here. Please ensure consistency and be mindful of the distinction between "risk" and "danger" in natural hazards studies. It is important to use these terms accurately to avoid confusion.

L298: All acronyms should be expanded at their first appearance or included in an auxiliary table for reference. This will help ensure clarity for all readers.

L324: You consider BA values since 2001 while the combined Terra and Aqua datasets started in 2003. Could the trend be affected by this inhomogeneity? Please address this potential issue.

L382-283: Not clear, please reformulate.

L396-405: I suggest explaining the target season shortly in the introduction.

Table 1: For the IPCC regions, you can reference: Iturbide M. et al. (2020). An update of IPCC climate reference regions for subcontinental analysis of climate model data: definition and aggregated datasets. Earth System Science Data, 12(4), 2959-2970.

L429: "(2003 for C emissions)" applies also for points "I" and "II"?

L430: "and;"?

L442-444: Not clear.

L451: Always using the March-February years?

L486: For future updates, I suggest correcting the p-values of the Mann-Kendall test for multiple testing using a false discovery rate test, as described in Ventura et al. (2004). "Controlling the proportion of falsely rejected hypotheses when conducting multiple tests with climatological data." Journal of Climate, 17, 4343–4356.

Table 2: There are some typos (e.g., "Ang'ila"), inconsistencies (such as country names being abbreviated or not), and repetition (the Africa region). Please correct the typos, ensure consistency in the formatting of country names, and eliminate the repetition in the Africa region.

L607: I think the comparison should be made fairer by using observed Fire Weather Index (FWI) data. This will ensure a more accurate assessment.

L612: I believe the seasonal skill is limited to less than 2-3 months in most regions. Please verify this information and adjust accordingly.

L614: "Predictability" or skill?

L618-621: Not clear. As I previously indicated, observed FWI should be the target for the verification.

L632: "Shapley values"?

L644: "and;"?

L658-659: Not clear.

L755: Please update the references considering the latest one: Forster et al. (2024). Indicators of Global Climate Change 2023: annual update of key indicators of the state of the climate system and human influence. Earth System Science Data, 16(6), 2625-2658. Also, please clearly indicate the quantity and reference for the temperature increase. For instance, Forster et al. (2024) wrote "The indicators show that, for the 2014–2023 decade average, observed warming was 1.19 [1.06 to 1.30] °C, of which 1.19 [1.0 to 1.4] °C was human-induced. For the single-year average, human-induced warming reached 1.31 [1.1 to 1.7] °C in 2023 relative to 1850–1900."

L805: Which models?

L827: There appears to be an issue with the numbering. I suggest referencing the first supplementary figures first to ensure a logical sequence.

L828: I suggest replacing "correct" with "adjust".

L854: I suggest explaining here why two thresholds.

L870: Please ensure proper formatting for citations, correcting any issues with parentheses.

L943: Burton et al. (2023) is a preprint, so I'm not sure if the methodology could be applied here without a proper examination and discussion. Is this approach mature enough to provide reliable information for this inaugural report and potentially for the next ones? I suggest a thorough review and discussion to ensure the methodology's robustness and suitability.

L951: I suggest comparing the results also against a standard no-weighted ensemble.

L963: The term "decadal outlook" might be misleading as you are considering decadal projections and not decadal predictions. Please adjust the terminology accordingly to avoid confusion.

L967: I suggest removing the ENSO description. An appropriate reference should suffice.

L967-996: The description is not clear. Is this the application of an already verified method or a new method? Additionally, please clarify how you establish significant teleconnections.

L1063: It might be useful to include a brief introduction to prepare the reader for the following section. Additionally, I notice that for some continents you provide specific quantities, while for others you only indicate "above" or "below." I suggest maintaining consistency in this section by providing quantities or clear indications for all continents.

L1076: Is it possible to provide an estimation of this contribution?

L1082, L1087, L1340, L1679, etc.: It might be useful to include a reference to demonstrate that the main driver here is drought, as this will substantiate your assertion. If such evidence is not robust or widely agreed upon, consider softening the statement to reflect the complexity of the factors involved.

L1085: I suggest using "BA" or "burned area" consistently throughout the text instead of terms like "fire extent," "burned areas" (at L1111), "Burnt Area" (at Figure 11), or similar, to avoid confusion.

L1096: How do you define "wildfire extent" here?

L1132-1137: This paragraph is partly repetitive. Please revise it to eliminate redundancy and ensure the information is presented concisely.

L1162: Where can I see these results?

Figure 1: To enhance readability, consider using a plot with two y-axes. This could help present the data and make it easier for readers to interpret the information.

Figure 5 (BA): Why since 2003 and not from 2001?

Figure 7: The low quality of this figure makes it difficult to read the text inside it. Please improve the resolution to enhance readability.

L1788: A more specific reference is needed.

L1843: I suggest expanding the names of the months, maintaining consistency throughout the paper.

L1915: Which variables are being referred to here? Did you perform any tests to validate these links? Please provide more details or specify the tests conducted.

L2188: Did you calculate this correlation?

L2193: I suggest avoiding references to "not-shown" results. If the results are important, consider including them in the main text or supplementary materials for transparency and completeness.

L2210-2221: If I understood correctly, the FWI aligns with peak fire counts. This observation may merit additional comments to highlight its significance and implications.

L2225: All South America?

L2236: Is soil moisture derived from ERA5?

L2239-2256: This paragraph may not be entirely necessary. Consider condensing it to make the text more concise and to the point.

Figure 12: Do you think there is any added value in including relative humidity, which is a key ingredient of the FWI?

L2340: "arae".

L2358: The sentence mentioning "fuel load" appears incomplete. Please review and complete the sentence to ensure clarity and coherence.

L2381: Can you clarify how you estimate the statistical significance here?

Figure 15: "anomoly".

L2498: It is unclear whether this refers to the direct human influence on fires or the influence of human-caused climate change. Please clarify to ensure the distinction is clear.

L2499: The term "total climate forcing" is unclear. Please provide a definition or rephrase for better understanding.

L2501-2502: Not clear.

L2505: "total".

L2586: Why are two decimal places used here? It might be more appropriate to round to one decimal place or an integer to reflect the associated uncertainty more accurately.

L2607: Please expand the name of the month.

L2612: "(Figure 14)".

L2701: "PI"?

Table 7: Do two decimal places accurately reflect the uncertainties here? Consider rounding to one decimal place or an integer to better represent the associated uncertainty. The color bar is missing.

L3022: I agree, and for this reason, the comparison is not fair. A discussion of this limitation would be helpful to provide context and ensure clarity.

L3048-3057: Why not conduct a reforecast evaluation for 2023-24? Including this analysis would provide valuable insights and enhance the robustness of your findings.

L3111: I suggest clearly stating that a major limitation in assessing trends is that we only have global data for a few years. This limitation should be explicitly mentioned to provide context for the trend analysis and the definition of extreme events.

L3252-3258: I suggest adding that there are data-driven methods that work regionally and globally, but for this inaugural report, these methods are not included. This acknowledgment will provide a more comprehensive understanding of the current limitations and future directions for the report.

L3308: 2024 or 2023?

L3357: CMIP5?

L3393: "Impacts" is underlined.

L3547: IFN?

L3650: Agree, but you do not use many regional datasets that can provide very useful information.

L3669: Not sure if all of this paragraph is necessary here.

L3774: Agree, but you do not use many regional datasets that can provide very useful information.

L3798: Why do you use only the Global Fire Atlas and no other similar datasets?

L3809: Maybe a good possibility here is to mention datacube datasets (e.g., Alonso et al., 2022).

L3829: What about decadal prediction?

L3831-3838: I believe that throughout the paper, the important role of antecedent weather on fire activity is not adequately addressed. It would be beneficial to emphasize how prior weather conditions influence fire behavior and risk.

L3870: The science behind the rigorous detection and attribution methods used to assess the anthropogenic contribution to climate change is well established. However, when it comes to fires, as presented in this report, the approach is useful but still somewhat approximate. It would be helpful to acknowledge these limitations in this section to provide a more balanced and accurate perspective.

L3896: Only one? Which one?

L3944: "Ang'ila"

L4567: "System†"

L4620: Are there any scientific papers that can be used to replace this reference?

Supplementary Material

L176: ")))"

FigS24: Bias adjustment with cross-validation?

L255: Is this the correct reference for this section?

L259-261: Not clear.

L348: "per (Barbosa 2024)": Please check that other references are formatted consistently and do not have extra parentheses.

**References:**

Alonso, L., Gans, F., Karasante, I., Ahuja, A., Prapas, I., Kondylatos, S., ... & Cremer, F. (2022). SeasFire Cube: A Global Dataset for Seasonal Fire Modeling in the Earth System. Zenodo: Geneva, Switzerland.

Bowman, D. M., Williamson, G. J., Abatzoglou, J. T., Kolden, C. A., Cochrane, M. A., & Smith, A. M. (2017). Human exposure and sensitivity to globally extreme wildfire events. Nature Ecology & Evolution, 1(3), 0058.

Cunningham, C.X., Williamson, G.J., & Bowman, D.M.J.S. (2024). Increasing frequency and intensity of the most extreme wildfires on Earth. Nature Ecology & Evolution. https://doi.org/10.1038/s41559-024-02452-2

---

## Author Comment (AC1)

**RC1 Response**

This is a very ambitious and exciting project. The authors are to be applauded for the comprehensiveness of what will hopefully become an annual effort. The strengths are the consistent approaches, the regional expert panel assessment, the efforts at attribution for key wildfires based on the COnFire and PoF models and the future outlook. The authors are also open about some of the issues and limitations of the first assessment.

The online assets on Zenodo are really comprehensive and clear. The supplement nicely details extra figures and extended methods which i found useful

Thank you for taking time to review our work, we appreciate that this is not a small task. We are pleased to receive your positive and constructive comments. Indeed, we hope to make great progress on this report over the next decade and beyond.

Please note:

- Due to the volume of changes made, particularly through re-structuring of the manuscript and to significant word-cutting, it became impractical to use track changes. In addition, the re-structuring is so substantial that our attempts to 'compare' versions (to highlight changes) has failed.
- Due to final editing/re-wording before submission, some of the quotes provided in the responses below may be out of sync with the submitted document.
- We sincerely apologise for any inconvenience this causes in the processes of reviewing the refinements to the report.

I outline some considerations below.

1. The authors state in the abstract that global wildfires are increasing in frequency and intensity. I thought this was debated and depends on the definition used? I thought that biomass burning has been decreasing in topical regions and some longer term datasets showed more fires in past decades going back further than the last 20 years, the focus of these datasets. I think in this first paper it would be really worthwhile to address this debate head on as I expect the paper to be widely read.

While the decline in burned area in tropical savannahs does indeed heavily skew global trends in observed fire extent, fires are nonetheless expanding into areas where they burn more severely or intensely and carry greatest potential to impact society, particularly into forests. The expansion of fire into forests (see **Figure 9** [now figure 5]]) generates increasing carbon emissions from forest fires and hence total fire emissions (forest + non-forest) are approximately flat over the past 20 years. Zheng et al. (2021, *Science*) and Cunningham et al. (2024a) highlight that the decline in burned area in tropical savannahs has disguised major shifts in the frequency and intensity of fires that are most impactful to people (e.g. examples given in our introduction) and the environment (e.g. generating a deficit in C storage). ]

We feel that the slightly softened statement, "*Climate change contributes to the increased frequency and intensity of wildfires globally, with significant impacts on society and the environment.*", is appropriate.

2. Related to 1, I wondered whether any global indexes could be perhaps be presented in a key figure and discussed. Although, I understand the regional focus.

Figure 1 portrays some of the headline results from section "Extreme Fire Seasons and Events of 2023-24"; specifically, regionally significant anomalies with greatest influence on global BA and C emissions totals.

However, we agree that moving forwards we must develop better ways to efficiently synthesise and communicate the results from several of our other analyses (e.g. thematic maps/figures that visualise the key drivers of the focal events and the results of the attribution analysis). We hope to make time to develop these over the coming year so that we can improve the report for future iterations.

3. The expert panel is a really nice idea, it might be nice to have some more information on how it made its decisions and what data it used.

Thanks! We considered this to be one of the most valuable contributions to the fire science community and had great feedback from our international partners. We have moved the results from this section to Appendix A to improve the flow of the manuscript, following suggestions from another reviewer, however we hope that readers will still consult the section for details about the wide range of impacts generated by extreme fires in 2023-24 and we have references Appendix A clearly in the main text.

In the methods section, we revised the text to more clearly explain the types of regionally significant wildfires and wildfire impacts that were captured by the expert panel (over and above what is be observable in Earth Observations).

4. Section 3, it would be useful to specify the averaging period used for "higher than average" statements etc.

Thank you, we have repeated the periods covered by the datasets more regularly throughout the section.

5. Table 2 repeats Africa

Thanks, now fixed.

6. Section 4 might additionally briefly discuss the role of PM2.5 and ozone from fires on ecosystem health?

e.g. Tian, C. G., Yue, X., Zhu, J., Liao, H., Yang, Y., Lei, Y. D., Zhou, X. Y., Zhou, H., Ma, Y., and Cao, Y.: Fire-climate interactions through the aerosol radiative effect in a global chemistry-climate-vegetation model, Atmospheric Chemistry and Physics, 22, 12353-12366, 10.5194/acp-22-12353-2022, 2022.

Thanks for making this important point. We have now added a sentence on the detrimental impacts of PM2.5 and ozone on plants and ecosystem health at lines 3658-3661 (originally section 4, and now moved to Appendix B section 8.3.5 "Loss of Biodiversity, Ecosystem Function and Carbon Storage"):

"Furthermore, release of ozone and particulate matter from fires including PM2.5 negatively impacts plant and ecosystem health (Saxena et al., 2017), and reduces leaf area index through drought induced by aerosol radiative effects, leading to reduced carbon uptake (Tian et al., 2022)."

**RC2 Response**

Authors have produced a remarkable assembly and collation effort! Compliments to authors. Good fit to ESSD. I will definitely recommend publication. I hope authors take some useful lessons from this review?

I understand and support this product as first-of-its-kind, a worthy initial effort intended to guide and stimulate progress on fire research. Again, good on the authors for great start! With second version promised, authors can claim 'good idea, address it in next version'. Much of what follows they will want to fix in first version! Authors and journal want best-foot-forward with this product! But some changes could delay?

I read this multiple times! Unfortunately, I feel that I understand perhaps less than half of what authors present. Why should I understand; I do not and have not engaged in wildfire research? But, as former Director of World Climate Research Programme and as co-founder and - for a too-long time - sole chief editor of this journal, I should represent at minimum a partially-competent somewhat knowledgeable reader? Authors will of course, following scientific instincts, attempt to correct my errors and mis-interpretations. But, please take a larger view: of me as interested supportive (somewhat) knowledgeable reader. If authors have not reached and convinced me, what have they done wrong? How do they need to change their messages? If I can't read this, how can authors expect fire managers (I deal with many, on local as well as forest-wide issues) to extract anything useful?

The manuscript remains way too long! (I deal with length issues below.)

Thank you very much for taking the time to review our work. We appreciate the significant effort involved. We read these comments as being those of an committed and thoughtful reviewer and have sought to add clarity to our manuscript wherever it is needed to improve the reader experience. We feel the manuscript has been improved, and we are grateful for your guidance.

Regarding your overarching point about audience: We specifically state that the *findings of* this report are relevant to various stakeholders beyond the primary scientific audience. This article is part of a package of materials that will enable us to reach a variety of stakeholders. In our experience of publishing policy-facing pieces such as the Global Carbon Budget and UNEP report on wildfires, concerned stakeholders read lengthy documents with interest. In our experience of publishing policy-facing pieces such as the Global Carbon Budget and UNEP report on wildfires, engaged stakeholders are willing to read lengthy documents with interest.

Regarding your overarching point about length: We have dramatically re-structured the report to improve readability. Our actions included:

> (1) Moving the "Year in Review by Continent" section to "Appendix A: Year in Review by Continent". In the appendix, this content interferes less with the flow of the manuscript, an issue raised by two reviewers.

> (2) Moving the elements of review ("Frontiers…" and "Roadmap") to "Appendix B: Defining A Roadmap for the State of Wildfires Report". In the appendix, this content interferes less with the flow of the manuscript, an issue raised by two reviewers.

> (3) Cutting the word count of the "Frontiers … " and "Roadmap …" sections substantially (these now appear in "Appendix B: Defining A Roadmap for the State of Wildfires Report").

(4) Presenting the methods and results for each theme of the report in sequence (i.e.: Observations-Methods; Observations-Results; Drivers-Methods; Drivers-Results; Attribution-Methods; Attribution-Results; Outlook-Methods; Outlook-Results. This was suggested by at least one reviewer. This streamlines the text by improving the proximity of methods and results material through, thus circumventing the need to scatter reminders about the methodology to the reader throughout the text.

(5) Removing the original "highlights" from each individual sub-section of the report and instead placing this material in the streamlined Conclusions section.

Other less substantial actions, such as removing figures and shortening sections of text throughout the manuscript, also led to a reduction in length.

Please note:

- Due to the volume of changes made, particularly through re-structuring of the manuscript and to significant word-cutting, it became impractical to use track changes. In addition, the re-structuring is so substantial that our attempts to 'compare' versions (to highlight changes) has failed.
- Due to final editing/re-wording before submission, some of the quotes provided in the responses below may be out of sync with the submitted document.
- We sincerely apologise for any inconvenience this causes in the processes of reviewing the refinements to the report.

For research audiences or (as authors hope) for "policymakers, disaster management services, firefighting agencies, and land managers"(line 131), this manuscript contains far too many deficiencies. Language ('extremes', 'serious', 'emergency', intense') remains vague and, therefore, difficult. Too many acronyms remain undefined. Maps and figures remain unreadable; figure legends in some case prove wrong. Dates of coverage and extents of observations remain inconsistent. This reader needs to scroll upward (backwards) 10s of pages and 1000s of words to find methodological details. A reader learns that satellite measurements of BA, focused necessarily on larger fires, might miss as much as 90% of burned areas (line 376; roughly 4 million km2 might actually total more like 7 million?) while, in later sections, same reader confronts author discussion of -0.2% BA anomalies? In haste to meet deadlines, authors have neglected chances and needs to check their own work and to, necessarily, impose collective oversight? To repeat: they make a great start! But they need to better guide readers and data users on their choices, their assumptions, their outcomes, and their conclusions.

We segment our response into specific critiques:

- *For research audiences or (as authors hope) for "policymakers, disaster management services, firefighting agencies, and land managers"(line 131), this manuscript contains far too many deficiencies.*
  As mentioned above, this manuscript primarily serves a scientific audience but is part of a package of materials designed to allow relevant stakeholders to receive targeted information. We highlight in the text that we are "consolidating state-of-the-art wildfire science and delivering key insights _relevant_ to policymakers" rather than being targeted at this audience directly.

- Language ('extremes', 'serious', 'emergency', intense') remains vague and, therefore, difficult. Too many acronyms remain undefined.

Taking on board this comment and others expressed later in your reviewer comments, we make substantial efforts to avoid loose use of these words. We performed a keyword search of each term, evaluated if its use is appropriate, and adjusted accordingly. We now also point readers towards the full discussion of the problematic definition of "extremes" when this issue first arises in the methods section. We have checked acronyms throughout.

- Maps and figures remain unreadable; figure legends in some case prove wrong.
  We address specific comments below, as this comment seems to apply to only specific maps/figures.

- Dates of coverage and extents of observations remain inconsistent.

  We used the best coverage available based on the input datasets described for each analysis. We address specific comments below.

- This reader needs to scroll upward (backwards) 10s of pages and 1000s of words to find methodological details.

  This has been addressed through the restructuring described above.

- "A reader learns that satellite measurements of BA, focused necessarily on larger fires, might miss as much as 90% of burned areas (line 376; roughly 4 million km2 might actually total more like 7 million?) while, in later sections, same reader confronts author discussion of -0.2% BA anomalies?"

  - The stated -0.2% anomaly value is from analysis of background change in BA in the median month of the periods 2003-2019, based on simulations using DGVMs. The analysis targets background changes in modelled BA during factual and counterfactual scenarios. We added detail to the methods section to explain the limitations of the DGVM representation of fire as well as the types/nature of the uncertainties quantified by that analysis.

  [- Please note, we state "Recent work has shown that including detections of small active fires increases global BA estimates by 93% (Chen et al., 2023)." Hence, satellite measurements of BA, focused necessarily on larger fires, might miss as much as 48% of burned areas (not 93%). Still significant of course, but not a ~factor 2 difference.]

**Language:** Authors take a strange but perhaps necessary approach: apply confusing ill-defined adjectives first, then later (Section 4.2.1, line 3085) highlight uncertainties and needs for improved definition. Unless authors want to dictate existing or develop new terms, I see no good alternatives to present approach. For this reader, knowledgeable disclaimers by these authors (shortened by 50%?) should have appeared earlier rather than later.

We view this approach as pragmatic and necessary. While few would argue against the need to study extremes in the fire "spectrum", they are not straightforward to define at present. The paper is very open about that.

The report must serve as a platform for discussion in the community and eventually support progress towards a consensus. Our revised methods section flags the challenges of defining "extreme" (near the top of **section 2.1**).

**Uncertainties:** This reader understands that wildfires represent one of our most complex challenges. How do they start? How fast and how much do they burn? How much do fuel load, soil moisture and weather (large-scale or self-generated) influence specific fires. How will observations, predictive skills, and attributions change in unknown future? Our basic task remains identifying and then working to reduce uncertainty. To then read about 99% confidence of -0.2% changes in BA seems, frankly, absurd. I don't doubt that one can make such statements with statistical certainty. But, should we? We know that preferred fire observations carry uncertainties near +100%. We know that even our best GCMs carry very large uncertainties, particularly for next decade or one after that. We know that reanalyses, an outcome of same systems we use to specify error matrices, likewise carry substantial uncertainties. Fire models, based on these uncertain sources, very likely amplify uncertainty; at minimum they combine and must depend on all underlying uncertainty.

We respond to the various points raised above in isolation:

(1) This reader understands that wildfires represent one of our most complex challenges. How do they start? How fast and how much do they burn? How much do fuel load, soil moisture and weather (large-scale or self-generated) influence specific fires. How will observations, predictive skills, and attributions change in unknown future? Our basic task remains identifying and then working to reduce uncertainty. To then read about 99% confidence of -0.2% changes in BA seems, frankly, absurd.

Please see the response above regarding the 0.2% number.

In the re-submitted report, we added clarity about the limitations and uncertainties of the approaches applied. Please see further details in response to specific questions throughout this document for more detail.

In terms of the high confidence levels given by ConFire, the model is designed to quantify the uncertainties by constraining the model by observations, which is why we are able to give very high likelihood of the results.

(2) We know that preferred fire observations carry uncertainties near +100%.

The "uncertainties" in fire observations are less uncertainties, more biases related to the boundaries of which fire types are observable by different sensors used to observe the Earth's surface. Sensor-specific biases are discussed in both the methods section (i.e., details about small fires missing from MODIS-based observations) and Appendix B (i.e., description of MODIS limitations and discussion of how other sensors such as VIIRS are becoming available with different observational boundaries).

The use of MODIS data is justified in this study because it is important to contextualise extremes with respect to the past decades of consistent observation, and its demonstrated capacity to detect landscape fires with potential to be impactful. When studying extreme fires, small fires such as agricultural fires, captured by some other sensors, are *generally* of lesser priority).

(3) I don't doubt that one can make such statements with statistical certainty. But, should we?

We are proponents of making statements with statistical certainty, though we appreciate the importance of clearly communicating which types of uncertainty are/are not included. This communication of uncertainties is addressed further below.

(4) We know that even our best GCMs carry very large uncertainties, particularly for next decade or one after that.

In our future outlook section, we utilize GCM future projections that have been bias-corrected through ISIMIP. This bias correction removes biases in the present-day mean climate state across the variables used to drive the ConFire framework. It effectively eliminates uncertainties that GCMs show in today's observable climate while not correcting trends and interannual variability of the GCM. As trends and variability are not corrected, we preserve the uncertainty in future climate change.

Our future projections are based on sampling uncertainty in forcings from different socioeconomic pathways (via multiple SSPs) and the climate response to these forcings (via multiple climate models). ConFire itself was specifically designed to account for and propagate uncertainties in how drivers influence burned area, as outlined under the next point. In the results, we present the full range of these uncertainties, which we agree is of critical importance of maintaining integrity in our projections.

We present all these uncertainties in our future outlook section. We outline where uncertainties are constrained enough to make confident projections and where they are not, highlighting the need to constrain uncertainties in the development of future projections.

(5) We know that reanalyses, an outcome of same systems we use to specify error matrices, likewise carry substantial uncertainties. Fire models, based on these uncertain sources, very likely amplify uncertainty; at minimum they combine and must depend on all underlying uncertainty.

The tools we used to perform the analyses, particularly ConFire, have uncertainty quantification built-in to them. ConFire is used standalone in section X and in combination with DGVM output in sections X and X. As noted above, the critical need is to clearly communicate which types of uncertainty are included.

ConFire is designed specifically as a uncertainty quantification system, and all results for ConFire in this report are based on that uncertainty quantification - either as 5-95% ranges or likelihood. It uses a Bayesian inference system to quantify the uncertainty of the impact of various factors on fire occurrence. It starts with basic assumptions on drivers' influence on burned area: fuel drivers increase burned area; moisture decrease; ignitions increase and suppression decrease. It then quantifies the likelihood of different levels of influence through training towards historic burned area. When doing this, it also accounts uncertainties through stochasticity of fire and factors not directly considered.

The model's probability distribution is a logit zero-inflated function, designed to assess changes in extreme fires even when observed burned areas are small. The model has successfully handled uncertainties from unconsidered factors and the stochastic nature of fire occurrence, and it can also accommodate larger uncertainties like unpredictable weather changes and varying vegetation responses.

To test ConFire's quantification of uncertainty, we use evaluation techniques specifically designed to test uncertainty quantification from Bayesian inference. Detailed results from testing show that the model effectively captures out-of-sample observations within its uncertainty range and aligns well with real-world data.

For clarity, throughout this work, uncertainty estimates quantified by ConFire include:

- Uncertainty contribution from different drivers, obtained by constraining by observations
- Model structural uncertainty: we represent only one relationship of control to BA here, but the uncertainty across the potential range of different relationships is accounted for in the probability distribution
- Noise: stochasticity of burned area is explicitly accounted for, both in terms of areas of no fire, and in terms of the probability of fire elsewhere in the model

Uncertainties not represented in the ConFire's uncertainty quantification include:

- Any potential changes in feedbacks between fire and vegetation when moving to a new state of fire regimes which is out of sample of the training period.
- DGVM model uncertainty: we only use one bias-corrected model for fuel load, JULES, so we assume here that the vegetation responses to future climate change in JULES are correct
- We use anomalies of BA so observational bias is not directly relevant, but when there is disagreement in the bias across space or time then this is not included in the training for ConFire.

We added the following details to the main text:

"For both models we include an estimate of uncertainty. ConFire is designed as an uncertainty quantification model, providing BA probabilities and their likelihoods for each region. Results in this report are based on 5-95% ranges or likelihoods. ConFire uses Bayesian inference to quantify how various factors impact fire occurrence, assuming that fuel increases burned area, moisture decreases it, ignitions increase it, and suppression decreases it. The model trains on historical data to determine influence levels and accounts for uncertainties from fire stochasticity and unconsidered factors. Its probability distribution is a logit zero-inflated function, assessing changes in extreme fires even with small observed areas. ConFire manages uncertainties from unpredictable weather and vegetation responses. ConFire quantifies uncertainty estimates from different drivers by constraining them with observations and addresses structural uncertainties, such as missing explanatory variables and errors in mechanistic relationships. While it represents one relationship of control to burned area, the probability distribution accounts for uncertainty across various potential relationships. Noise is considered, with the stochasticity of burned area accounted for both in areas with no fire and in the probability of different potential levels of burning where fire does occur. We test ConFire's uncertainty quantification using Bayesian inference evaluation techniques. However, ConFire does not account for some uncertainties, such as potential changes in feedbacks between fire and vegetation when transitioning to a new fire regime out of sample. The model assumes the accuracy of one bias-corrected model for fuel load (JULES), neglecting DGVM model uncertainty. While BA anomalies are used to reduce observational bias, any disagreements in bias across space or time are not included in ConFire's training.

Meanwhile, PoF outputs are provided in terms of probabilities calculated using ensemble predictions from weather forecasts each to generate a set of binary classifiers. The probabilities are therefore based on a wide parameter space, taking into account uncertainties in both the input parameters and the stochasticity of the classification algorithm itself."

ESSD guidelines specify "Explicit uncertainty accounting and analysis". Here authors push those boundaries. Readers will not doubt techniques nor skills! They will, however, as I do, want to see authors' rational and best estimates to work within substantial uncertainties. Where authors need to demonstrate valid tools for handling small differences in BA, we need to know that those authors at the same time appreciate the difficulties and larger uncertainties of this science. I recommend the authors introduce an uncertainty budget early in manuscript? Show and admit where uncertainties exist? Outline your approaches: ignore, compare, test, apply two different models, whatever. Show us, however, that you understand the larger questions of uncertainties even when you must proceed.

We respond separately for different methods employed in this work.

OBSERVATIONAL DATASETS:

We now expand on the uncertainties in BA observation and C emissions in the method section.

FIRE WEATHER FORECAST/HINDCASTS:

Fire weather index predictions (e.g. used in figure 11 [now figure 7]) are performed using ensemble forecasts. The full set of ensemble simulations are available and can be used to estimate uncertainties. Variability across the ensemble was previously estimated to be on the order of 10% -15% (https://www.nature.com/articles/s41597-020-0554-z), which we now state in the main text. We have added in the supplementary material (Figure S24-S25 and S26) the spread on the FWI prediction for reference.

For the driver attribution methods (POF and ConFire), we provide a comprehensive estimation of uncertainties by using a probabilistic framework as the default. All results are presented in terms of probabilities rather than deterministic values. We have included the following explanation in the text: "For both models we include an estimate of uncertainty. ConFire is designed as an uncertainty quantification model, providing BA probabilities and their likelihoods for each region. ConFire is an uncertainty quantification system. Results in this report are based on 5-95% ranges or likelihood. It uses Bayesian inference to quantify how various factors impact fire occurrence, assuming that fuel increases burned area, moisture decreases it, ignitions increase it, and suppression decreases it. The model trains on historical data to determine the influence levels and accounts for uncertainties from fire stochasticity and unconsidered factors. Its probability distribution is a logit zero-inflated function, assessing changes in extreme fires even with small observed areas. ConFire manages uncertainties from unpredictable weather and vegetation responses. We test ConFire's uncertainty quantification using Bayesian inference evaluation techniques. Meanwhile, PoF outputs are provided in terms of probabilities calculated using ensemble predictions from weather forecasts each to generate a set of binary classifiers. The probabilities are therefore based on a large parameter space, taking into account uncertainties in both the input parameters and the stochasticity of the classification algorithm itself."

ATTRIBUTION:

We include uncertainty in each of the attribution methods presented in the paper. We have now added an extra paragraph into section 2.4.2 "Overview of Attribution Approaches" which describes precisely how uncertainties are represented in the 3 approaches to make this clear:

"In each approach we include an explicit estimate of uncertainty. ConFire is designed as an uncertainty quantification model, giving likelihood of all possible burned areas for each region based on a probabilistic analysis of past burn patterns and environmental conditions. We combine the information from the FireMIP models in a weighted multi-model ensemble to give uncertainty

ranges across the models, and we use bootstrapping to give uncertainty estimates for the FWI risk ratios. Each result therefore presents a 5-95[th] percentile probability estimate."

SEASONAL AND DECADAL OUTLOOK SECTION:

Seasonal outlooks, both for meteorological variables and fire weather index, are provided as ensemble predictions. In Figure 20 [now removed], we presented a multimodel envelope of simulations to clearly show the dispersion and uncertainties in the predictions. As this figure is indeed not from our analysis but available through the C3S service and in the interest of shortening the manuscript we have decided to remove it. Figure 21 [now figure 16] is presented in terms of probabilities above thresholds as they provide probabilistic information which include uncertainties as requested by the reviewer.

**Atmospheric Climate Modes** At least a few of these authors adopt mode (ENSO, IOD, POD, etc.) labels and literature. (Please remember that a long time ago, before birth dates of some of you, I led a big year-long field program focused on western tropical Pacific and ENSO.) I recommend that this manuscript and this effort avoid such references and discussions. For two reasons. A) This manuscript really does not need such discussions to make basic points about S2S predictability? Your results prove, in this case, no better nor no worse - but entirely dependent on - S2S forecast skills. Difficult to dismal, at least at certain time scales, so far? B) Whatever the mode, it depends on an original identification and description now at least 20 (perhaps 40) years old. Those modes, which initially had at-best marginal utility, have proved increasingly unstable and unreliable. These authors really do not want to step into vigorous ongoing scientific debate, nor have work judged by changing, perhaps eventually no longer valid, mode descriptions?

Modes of variability are highly relevant to fires, and we would be remiss not to mention their role in driving regional increases year-to-year. We adopt established, widely-used descriptions for the modes which are used in many facets of science, including fire science.

The ENSO forecasts and fire weather forecasts are produced by distinct analyses; ENSO forecasts do not feed into fire weather forecasts.

We respect your experience in this area. However, we were not aware of any evidence to suggest that indices such as ENSO have proven increasingly unstable and unreliable. Indeed this is a highly predictable and observable phenomenon, on which the well-established field of seasonal forecasting is based. We supplied the search terms "ENSO" AND ("unstable" OR "unreliable" OR "invalid") to Web of Science and Google Scholar. Amongst the top 50 hits in both cases, we did not observe a consensus that indices such as ENSO have proven increasingly unstable and unreliable or that they are invalid.

ENSO is widely used to anticipate and manage impacts on various sectors, such as agriculture and disaster preparedness. For example, ENSO projections are distributed to farmers in the tropics so that they are able to adjust planting and irrigation schedules based on expected changes in rainfall, which can significantly affect crop yields. Additionally, governments utilise these forecasts to prepare for natural disasters, such as floods and droughts, by implementing early warning systems and resource allocation strategies to mitigate potential damages. In the Global Carbon Budget, the year 2023 projection of atmospheric $CO_2$ is made based on the forecasts of ENSO specifically because the strongest covariate of interannual changes in the land sink flux is the ENSO signal, including due to drought and increased fires during the El Niño

phase (Betts et al., 2016; https://doi.org/10.1038/nclimate3063; see also section 2.4.2 of Friedlingstein et al., 2023).

If we look at applications connected with fire, there are also many examples across the literature establishing clear links of fires with ENSO. An incomplete list of the papers reporting on the role of ENSO in modifying the global patterns of wildfire is provided below. Collectively, these papers have shown that there is a predictable cascade of fire across tropical continents during ENSO events, highlighting staggered global responses of wildfire to ENSO.

- Chen et al 2017 https://www.nature.com/articles/s41558-017-0014-8
- Field et al 2009 https://doi.org/10.1038/ngeo443
- Field et al 2016 https://www.pnas.org/doi/full/10.1073/pnas.1524888113
- Page et al 2002 https://www.nature.com/articles/nature01131
- Malhi et al 2018 https://royalsocietypublishing.org/doi/full/10.1098/rstb.2017.0298
- Farfan et al 2021 https://link.springer.com/article/10.1007/S10661-021-09494-0
- Kitzberger 2002 https://www.publish.csiro.au/wf/wf01041
- Zheng et al 2023 https://agupubs.onlinelibrary.wiley.com/doi/full/10.1029/2023JD039688
- Fang et al 2021https://www.nature.com/articles/s41467-021-21988-6
- Nurdiati et al 2022 https://www.mdpi.com/2073-4433/13/4/537
- Shikwambana et al 2022 https://www.tandfonline.com/doi/full/10.1080/10106049.2022.2113449
- Wooster et al 2012 https://bg.copernicus.org/articles/9/317/2012/

The utility of using ENSO as a predictor of fire is emphasised by its practical application in Indonesia. The strong El Nino of 2015 led to severe fires in Indonesia, causing hazardous haze levels in Singapore and sparking diplomatic tensions, prompting calls for better regional cooperation and enforcement of laws against those responsible for the fires. During the 2019 El Nino, another severe haze episode linked to dry conditions reignited calls for stronger regional action and improved land management practices in Indonesia. In response, Indonesian legislators now implement preemptive bans on agricultural burning based on ENSO predictions. This proactive measure has proven successful, as in 2023, despite a strong positive ENSO event, there have been no significant anomalies in fire activity recorded. This is a clear example where ENSO forecasts are used in fire management without even transforming the information into fire weather predictions.

To clarify the utility of the ENSO index to readers, and our strategy we have rewritten this section.

*Among the modes of variability in the climate system most relevant to wildfire activity globally is the El Niño-Southern Oscillation (ENSO) (Chen et al., 2017; Mariani et al., 2016; Fuller and Murphy, 2006; Cardil et al., 2023). Numerous studies have demonstrated that there is a predictable cascade of fire across tropical continents during ENSO events, highlighting staggered responses of wildfire to ENSO. The utility of using ENSO as a predictor of fire is highlighted by its application in Indonesia, where severe fires during the 2015 El Nino led to hazardous haze in Singapore and diplomatic tensions in the region , prompting better regional cooperation and enforcement of anti-burning laws(CITATIONS). Consequently, Indonesia now implements preemptive bans on agricultural burning based solely on ENSO predictions, a measure that proved successful in 2023 when no significant fire anomalies were recorded despite a strong positive ENSO event.*

*Another phenomenon demonstrably linked to global fire activity is the Indian Ocean Dipole (IOD), which occurs in the Indian Ocean. However, there is still ongoing debate regarding the direct influence of the IOD on Australian fires, for example as the signal is often obscured by changes in land management practices (Harris and Lucas, 2019). Other atmospheric modes of variability in the Southern, Northern hemisphere and in the arctic regions can also have strong influence on the*

*seasonal trend of regional burned area and, **Figure S1** shows the climate modes with strongest influence on burned areas globally.*

*Outputs available from the Copernicus Climate Change Service (C3S) multi-model seasonal prediction system are used to evaluate large-scale climate modes with the most proven links to variation in fire activity: ENSO and IOD (CDS, 2018). As not all regions display similar seasonal direct correlations between fire activity and ENSO, we also use seasonal outlooks of the Fire Weather Index (FWI) from one of the models from the aforementioned multi-model system, ECMWF-System 5, to identify probabilities for the establishment of anomalous landscape flammability in the next season. This is done using a 51-member forecast ensemble and a 24-year model climatological distribution (derived from a 25-member ensemble re-forecast) covering the period 1993-2016. The probability of exceedance is determined based on the proportion of forecast members meeting an anomaly criteria at any given geographical point. We consider the 75th percentile indicative of moderate anomalous conditions and the 95th percentile indicative of extreme anomalous conditions over a month for the next season.*

Our dates of birth are not relevant to the peer review process. However, as this point has been raised, we highlight that this authorship group has around 500 years of combined experience in science and includes authors of the paper titled "A pan-tropical cascade of fire driven by El Niño/Southern Oscillation" by Chen et al. (2017).

**Length:** too long. For many reasons.

- Challenging for journal staff to process (typeset, proofread, etc.). We (ESSD) depend on their expertise! We should not abuse.
- Difficult for readers. Results 10s of pages after Methods. No hot links, not easy to move around. Many apparent redundancies.
- Impassible for fire managers. They will not use this! They will dismiss as too long, too obscure, too full of technical jargon.

Even if authors decide this first version targets other fire researchers rather than 'stakeholders'(fire & resource managers, general public), they still need to understand and respect reasons 1) and 2). I recommend a target of 80+5 pages! Even accepting necessary additions (Table of Contents, for example), authors can easily meet my 80-page target by moving large sections to Supplement, clarifying and sharpening remaining text, and - starting from a lean short product - accommodate eventual inevitable additions. You want this product to grow into respected forum? You will need, in that case, to accept substantial additional ideas and contributions!

Please see our response above to general comments about length and audience.

Assumptions for 80 pages:

- Move detailed continent-by-continent descriptions to Supplement. I know authors put a lot of work into this, but it offers very limited useful information to overall manuscript goals. Saves: 12 to 13 pages (~600 lines) plus associated figures and reference?
  We moved this section to an appendix to enhance the readability of the manuscript without hiding an element of the work that the authorship felt was a novel and exciting.
- Delete most discussions of ENSO, IOD, etc. Saves: 2 to 3 (estimated) pages, plus a few references?
  For the reasons detailed above, we didn't remove discussion of ENSO though we did remove one figure to reduce the page count.

- Keep 3.1.1 Highlights OR Section 4.1 Summary but not both. This reader would prefer Section 4.1 but moved near top of manuscript. Saves 1 (estimated) page plus a figure or two?
  Addressed through the restructuring described above.
- Eliminate discussion and outputs from PoF because, in the end, that tool contributes very little to important outcomes? Saves perhaps 1 page?
  Addressed through the restructuring described above.
- Check and then use acronyms correctly and consistently. Saves 100 (or more) words?
  We improved our definition and use of acronyms.
- For several sections, authors present highlight section followed by one paragraph details section. Easy to combine / reduce these in some cases?
  Addressed through the restructuring described above.
- Reduce redundancy. Perhaps by adopting, for each major effort (descriptions, predictions, projections, etc.) an organization of methods – – first followed directly by outcomes, in sequence.I predict we might gain at least one page per section, mostly by deleting redundancies?
  Addressed through the restructuring described above.
- Rewrite entire manuscript focusing on brevity and clarity.

  We sought to implement efficiencies throughout.

I rewrote sections to test this last suggestion (my supplement). I achieved length reductions (by word counts) of 25%, 20%, 28% and 12% in these four small samples; I believe expecting 20% reduction seems reasonable. 20% * nearly 90 changeable pages saves another 18 pages? I might, for a few paragraphs or news article, assign differences between my wording and authors' wording to personal language preferences? In this case however, with length a primary problem, authors need to make substantial text reductions!

Necessary additions: Table of Contents, with hot links! One page. Acronym list (could go in Appendix). Two pages.

We will enquire with the publisher about adding a table of contents. We did not add one in our re-submitted version because page numbers will differ between this version and the published typeset article.

By rough accumulation of assumed saving plus 3 additional pages, one easily gets to 80 net (working text) pages?

Understanding this as initial foray, I sort recommendations into three sections: now (mostly proof-type comments that authors will want to correct to make this version 'presentable); now or future (as mostly related to length or clarity) entirely at authors' discretion; and future (for authors' discussion as they move forward).

Note: I use terms 'reader' and 'user' interchangeably.

Regarding length: Please see our earlier response to similar/repeated comments above.

Regarding audience: Please see our earlier response to similar/repeated comments above.

**Now**

Line 108: "This" what? "Driven"? "Dampened"? This BA vs CO2 emissions pattern? This emission pattern? You do not want to allow uncertainty on the part of readers. 'Clearer than the truth', particularly in abstract.

Correction: "Global fire C emissions were increased by record emissions in Canadian boreal forests (over 9 times the average) and reduced by low emissions from African savannahs."

Line 118: "fuel load and direct human suppression often modulated areas with anomalous burned area"? 'All' areas? 'Other' areas? Areas other than Canada and Greece?

This is in reference to the boundaries of higher than average burned areas that often correspond to regions that transition to lower burned areas in Figure 15 [now 11], 16 [now 12]. We have rephrased to read:

"whereas burned area anomalies were weaker in regions with lower fuel loads and higher direct suppression, particularly in Canada."

Line 121: "extreme events". Again, use of ill-defined term 'extremes'. 'Extreme events' could include landslides, floods, etc. You want to keep reader's focus on wildfires?

Correction: "extreme fire seasons"

Line 128: "extreme anomalies" which differ from "moderate positive anomalies" by how? Explanations, including statistical certainties / uncertainties offered later but here (again) readers will need to find exact language. So many (too many) readers will glance at abstract to check whether they want to peruse further.

Correction: Clarified that these refer to 75th vs. 90th percentile thresholds.

Line 135, Short Summary: Excellent. ESSD editors might consider requiring similar short summaries of all papers?

Thank you.

Line 151: again "extreme fires", use of a fraught descriptor that authors will discuss (and, disparage?) later. I will stop recording these instances. Basic question: if authors know they confront an ill-defined (at best) term, should they use it (so often) themselves?

Correction: "extremely large, fast-moving or intense fires"

Lines 160 to 164: Authors should read carefully. To this reader, closing sentence lacks a conjunction ("and") or something. Confusing as written?

Correction: added conjunction

Line 180: "is significant" Statistically significant? Generically significant? Worrisome? (Authors used term 'concerning' in previous sentence.)

Correction: "particularly important"

Line 193: very valid point but many ESSD readers will not know 'infamous' Wall Street (NYC) image? Slight alteration or clean up of closing sentence?

Correction: "images of North American cities blanketed…"

Lines 196 and following: Authors recite valid points about C emission but, as they well know, atmospheric CO2 represents residual between emissions and sink. Point here: wildfires count as

land emissions, no longer a land sink? Clean up and perhaps shorten particularly introductory sentence for this paragraph?

This comment is not clear to us. For clarity:

Line 196 refers to the importance of intact forests and the detrimental effect of forest loss to fire.

Wildfires are accounted for in the natural sink flux, represented by DGVMs in the global carbon budget. Increasing wildfire extent or severity leads to a reduced natural sink.

Land use fires are implicitly included in the land use emissions flux, represented by bookkeeping models in the global carbon budget. These trigger a net loss of carbon to the atmosphere. The representation is implicit rather than explicit because bookkeeping models "see" the land use changes associated with land use fires, not the fires themselves.

We have cross-checked that the content of the paragraph is accurate and clear. We broke the first sentence into two for added clarity.

Line 205, 206: "undermine the regenerative capacity of forests (Nolan et al., 2021a) and the habitats of many endemic species being degraded in biodiversity hotspots (Ward et al., 2020). 'Undermine' regenerative capacity and habitats? With resulting degradation? Awkward sentence, needs clarification.

Correction: "Ecosystem function is also impacted by extreme wildfires through widespread mortality of forest stands, which reduces the regenerative capacity of forests and constrains the habitat of many endemic species."

Line 207: Authors report correctly changed assumptions about post-fire regrowth but sink or source implications of "fire-C cycle" have always proven difficult? Perhaps a clarification or certification of one term while net equation remains unclear? Important point but needs attention to language.

Correction: "Persistent increases in fire C emissions can result in a change in local to regional terrestrial C budgets from sink to source in cases where the gross emissions from fires outweigh gross sinks to vegetation during post-fire recovery."

Line 209: "degraded and transformed" Wildfire may degrade and transform any land, regardless of ownership or occupation. Impacts may prove more challenging or detrimental for native or Indigenous people but that represents a function of ownership, not of fire intensity? Authors allow confusion here between "fire types" (Line 211) and community impact? Valid points but deserve better itemization?

Correction: "The lands and territories of various landowners can be degraded and transformed through wildfires, and particular climate justice issues have been raised in cases where wildfires impact traditional communities and Indigenous Peoples."

Line 215: "policymakers and involve coordination with many other stakeholders" 'that involve'? Or, 'that require'? We suffer too often when policymakers act in the abstract, without consultation with stakeholders?

Correction: "require"

Line 232: "seen as a key tool for achieving Net Zero" This reader does not see C offsets as key nor useful tool; nor will other ESSD readers. Perhaps 'used' or 'applied'?

Correction: "which often feature as a key tool in national policies and international initiatives for achieving Net Zero emissions".

Line 235 & paragraph following: Very good questions. Can we expect 'stakeholders' to wade through 124 pages to find tools or answers?

This article is part of a package of materials that will enable us to reach a variety of stakeholders. In our experience of publishing policy-facing pieces such as the Global Carbon Budget and UNEP report on wildfires, concerned stakeholders read lengthy documents with interest.

Line 242 & paragraph following: Admirable sentiments but authors could reduce this paragraph by 1/2? Again, in response to final sentence about relevance: relevant perhaps but not easily available?

Line 242 and the paragraph following are not sentimental. They are an introduction to the contents and points of novelty of this report.

We made some in-text changes to shorten the paragraph.

We specifically state that the *findings of* this report are relevant to various stakeholders beyond the primary scientific audience. It is our responsibility to disseminate those findings to stakeholders. As mentioned above, this article is part of a package of materials that will enable us to reach a variety of stakeholders. In our experience of publishing policy-facing pieces such as the Global Carbon Budget and UNEP report on wildfires, concerned stakeholders read lengthy documents with interest.

Line 260: Sentence here (starting from "We incorporate …") needs change in punctuation or in use of conjunctions to clarify exact intent?

Correction: The sentence has been re-written for clarity.

Later, why do authors, here and throughout, capitalize 'Earth Observations'. Do they mean to indicate satellite observations? Something different or special?

Correction: Now "Earth observations" throughout the manuscript.

Line 292: Lose first sentence; not needed? Start with 'We identify' … in second sentence?

Correction: Dropped first sentence, start with "We identify".

Line 302 "Burton, Lampe, et al., 2023": Why does this particular citation show up, throughout the document, with two names? From citation itself (Burton et al. 2023) it looks like standard reference? Something in bibliometric software? Something the authors wish to highlight?

Burton and Lampe co-led this paper and are credited equally. We prefer to credit the authors accurately but we will adhere to any editorial requirements at the copyediting stage.

Line 348 and following: Please confirm all dates / durations. How, for example, can this manuscript report data to Feb 2024 if BA updates only go to Andela and Jones (2023). Andela & Jones in reference list actually shows as 2024?

Correction: Andela and Jones (2023) becomes Andela and Jones (2024).

Please also check and confirm all acronyms in this section. This reader knows (but many will not) NASA, MODIS, ECMWF, CAMS, GFAS, GFED, etc. Consistency first then shorten where possible. Careful consistent use of these acronyms (e.g. GFAS) could eliminate many subsequent words?

Correction: Verified that each of these acronyms are defined on first use and used consistently thereafter, except in citations or tables.

Line 373 & paragraph following: important points about fire size limitations which emerge multiple times later but, here, authors could reduce this section by 20%?

We reduced the length of this paragraph by around 20%.

Line 415, Table 1: Continent (row 2) shapefiles exist in all GIS software, open ( e.g QGIS) or proprietary (e.g. ArcGIS). Some readers will operate under institutional ArcGIS licenses but many will rely instead on open GIS software. Make this source generic rather than specific?

The specific layer that we used is that defined by ArcGIS. It is available open source and can be opened in other software. For example, we have accessed this layer using QGIS, Python and R.

In my version, wrapping (at page 8) causes disconnect problem in RECCAP2 line, e.g. Ciais et al? Authors will know what they intend …

Correction: In the re-submission, we ensure the table occupies its own page with no wrapping issues.

Line 425: Authors use different list numbering here (capitalized Roman numerals) than in earlier text lists (e.g at lines 274 and following where they used Arabic numerals). Copernicus (publisher) no doubt recommends and adheres to standardized approach?

Later, we refer to this list as "(see i-iii above)" and it is important that the specific list referred to is clear. This saves some words because we need not repeat how anomalies were calculated.

Line 430: ", and."? Superfluous or something missing?

Correction: Remove superfluous string.

Line 446 & paragraph following: Good important stuff, readers will need and want this info, but - again - authors could help readers (and publisher) but reduction entire section by > 10%?

We removed/edited one or two clauses, but did not think that these methods could be described in a plainer / more efficient way without losing key details.

Line 456, "Global Fire Atlas": Why use extended title when you have already defined acronym? Shorten in all ways possible!

Correction: "Global Fire Atlas" is preferred as its acronym (GFA) could be easily confused with the more widely-known acronym for the Global Fire Assimilation System (GFAS).

Line 471 & paragraph following: Useful, necessary information but authors could convey identical information in 50% fewer words?

We significantly reduced the length of this section.

Line 524, Table 2: Again, potentially a wrap problem around page 10 to 11, but Table 2 in my version contains duplicate entries for the Africa team? E.g. both on page 10 and again on page 11, duplicate lines?

[Now table A1]

Correction: this is fixed in the latest submission.

Line 102: "extreme" fires? As authors repeatedly point out, community presently confronts inability to reliably quantify nor communicate 'extreme? Should these authors apply a term they later, on valid grounds, question?

The actionable element of this comment is not entirely clear to us. To be clear on what we did do:

In this work, we raise the need for the community to work towards more unified terminology. As communicated throughout the text, we adopted a conservative approach to be inclusive of what constitutes "extreme" to a broad section of the fire science community. For quantitative elements (e.g. metrics such as BA or C emissions), metrics of anomaly magnitude and ranking amongst fire seasons of the past two decades are used (i.e. statistical extremes of the distribution). On the other hand, the expert panel is given freedom to identify additional extremes according to their experience of working in specific regions. This is explained in the methods section and reiterated in the discussion and conclusions.

Lines 131, 132: "insights relevant to policymakers, disaster management services, firefighting agencies, and land managers," But, this important target group of readers will find 124 pages daunting or forbidding?

We specifically state that the *findings (insights) of* this report are relevant to various stakeholders beyond the primary scientific audience. It is our responsibility to disseminate those findings to stakeholders. As mentioned above, this article is part of a package of materials that will enable us to reach a variety of stakeholders. In our experience of publishing policy-facing pieces such as the Global Carbon Budget and UNEP report on wildfires, concerned stakeholders read lengthy documents with interest.

Line 133: Recommendations - well-grounded - that follow cover basic understanding as well as "preparedness, mitigation, and adaptation"?

This comment is not entirely clear to us. The sentence is structured as follows:

First clause - we deliver the insights to stakeholders (referring to results presented in the foregoing sentences).

Second clause - those insights should promote improved preparedness, mitigation, and adaptation (by raising knowledge and highlighting urgency).

Line 258 & paragraph following: initial caution & questions about what constitutes extreme events arises here. Good! But, authors have already used terms 'extreme', 'extreme fire', 'extreme wildfire', leaving this reader with two questions. A) Should one use such a complex fraught term? B) What will these authors, best in their field, suggest as alternate or as quantitative?

Consensus on the definition of specific terms is a pervasive issue in fire science (as in many other fields of science). We are very pragmatic in this paper.

Studying extremes of the fire distribution is clearly important; the introduction explains the case for this.

In the methods section, we highlight the difficulties of doing so and explain our practical (and conservative) solution.

In the discussion section, we discuss how progress towards unified terminology can be fostered and achieved.

Introducing a different term would not solve the challenge; it would leave the current one behind and likely introduce a new one to ponder over.

Line 268 & following "Objectives": Good section, appreciated. Shorten? Opening paragraph suggests tools and information (good), prospective, but user just read that authors will focus in BA while first objective definitely leans toward recent observations. Somehow, perhaps in introductory paragraph of this section, remind users about intent to apply and report data and observations?

Correction: We added the following text to the section to clarify the new dataset that we built in order to deliver objective 1.

"To support objective 1, we build a novel dataset of fire metrics including BA, fire counts, fire C emissions, and individual fire properties (size and rate of growth) for consistent world regions."

Line 326, paragraph starting here: Good paragraph, first discussion of uncertainty in application of term 'extreme', very useful caution, but authors could clean this up by 10% to 15%?

We re-formulated the paragraph to cut its length by around 20% prior to adding detail about the March-February fire season definition (see below).

Line 396 & paragraph following: Important paragraph outlining timing assumptions and choices. This paragraph should perhaps move to top of Methods section?

The methods described in this paragraph relate specifically to how regional statistics were aggregated spatially and temporally. Moving this detail would result in a disjuncture in the logical flow of the methods. However, to flag this earlier to the reader, we add the following to section sentence to section 2.1:

"In this work, the global fire season is defined as occurring in March-February windows oriented around the annual minima of global fire activity in boreal spring (see section 2.1.1.2)."

Line 701 & following, Section 2.3.2.3 on Drivers: authors could rewrite this entire section with 15% fewer words but no loss of info?

We have reduced this section from 500 to around 320 words

Unfortunately, I need a map to understand section 2.3 and following.

We will enquire with the publisher about adding a table of contents. We did not add one in our re-submitted version because page numbers will differ between this version and the published typeset article.

FWI, a real-time fire forecasting tool, used to assess how 'early' (how much in advance of actual fire) users could predict that particular fire. FWI requires as inputs weather, dead fuel moisture (only dead, not living above ground?) Predicts ignition possibilities and spread rates. One advantage: moisture over three levels (depths) of soil?

The Fire Weather Index (FWI) is a measure of landscape flammability and not a measure of fire activity. Many regions of the world may experience extreme landscape flammability year-round but never witness a fire (e.g., the Sahel). As the term suggests, FWI only considers weather variables. The equations were derived for a specific fuel types, considering how weather conditions would affect that fuel type by simulating the drying of the fuel bed at different depths.

Nevertheless, since FWI is the most commonly used metric by forest agencies nowadays, it is crucial to understand where it works and where it does not. This understanding is the rationale behind the section dedicated to predictability.

Then, to address longer time scales for prediction, these authors look at FWI compatible data (conditions) but from longer-time prediction systems (e.g. by ECMWF)?

Yes, that is precisely what we did. We calculated the fire weather index based on forecasts of temperature, humidity, precipitation and wind speed from the ECMWF seasonal forecast.

To better understand detailed predictors (beyond weather and moisture as used by FWI), authors evaluate two additional models (applied only to their focal events?): PoF and ConFire. Here they need to introduce concept of Active Fire (AF) to supplement and complement BA? Because PoF predicts only AF? As aside, note relative incompatibility of AF with BA?

These two models are very different from FWI. First, they take into account all aspects of fire drivers, not just weather. Most importantly, they are trained on either fire activity or burned area, so they attempt to predict not just landscape flammability but actually the probability for a fire being observed.

PoF *could* be trained to predict burned areas, and likewise ConFire could be adapted to predict active fires. However, this work would require important re-structuring and the resulting methods would be unpublished.

The fact that we train on active fire is not a limitation but a choice and an advantage. Our use of two observational fire products and two distinct model approaches provides a way to account for inherent uncertainties in the definition of the fire events and the uncertainties in the methodologies. The advantage of applying two different models to two different metrics and getting broadly consistent and complementary results, which adds weight to our conclusions (agreement between models) and also informs uncertainty (any differences between models would alert us to the dependence of results on methodological choices).

section 2.3.2.1 *Our use of two observational fire products and two distinct model approaches provides a way to account for inherent uncertainties in the definition of the fire events and the uncertainties in the methodologies.*

Drivers group into categories (= controls?) but drivers certainly interact (weather & moisture) and (?) predictors = drivers = single variables? Reflect interactive drivers by including them in more than one category/control?

We grouped the fire drivers into the four controls based on established theory surrounding the connections between mechanistic processes affecting fire risk (controls) and observable metrics that are available as proxies for those controls (drivers). The grouping of drivers into controls is somewhat novel and so our choices are explained and justified thoroughly and available for critique and refinement from other scientists in this field.

PoF and ConFire require weather, fuel abundance, and fuel moisture, plus other? For 'other' PoF relies on vegetation type (forest vs grassland?), urban fraction and orography, ConFire for 'other' uses land use type, urban vs rural populations, and lightning (ignition) from Table 3.

The actionable element of this comment is not clear to us. Indeed, Table 3 is provided to indicate which drivers are grouped into which controls per the available modelling approaches. We also added the following sentence to clarify that 'other' includes noise related to uncertainties in input data and imperfect modelling:

*"Others not only include factors related to ignitions but also the fraction of predictions missed by the models. This is important because this category weights the importance of unaccounted-for factors and provides an estimation of the uncertainty in the prediction"*

Then reader confronts / needs details of PoF and ConFire. Paragraphs, but easily shortened with no loss of info!

We significantly condensed this section from around 500 to around 320 words. Both statistical infrastructures/methods are published in the literature so we just reference it as suggested by the reviewer.

Finally, to develop attributions (particularly but not exclusively to climate change factors), we need to hold fire model constant (use FWI) while imposing climate scenarios (via SSP) according to definitions and recipes developed by IPCC and ISIMIP? So long as outputs of those climate scenarios meet needs of fire model (FWI)? Users need to understand that attribution works on different temporal and spatial scales?

The reviewer makes a good suggestion for future iterations of the report: assessing future projections of FWI across different SSP scenarios. We note however that we do already use consistent tools across the drivers, attribution and future projections analyses, for example we consistently use ConFire to both attribute and make projections of burned area.

We have added the suggestion to look at future scenarios with FWI into the Roadmap section (8.4.4), which could be addressed by future studies:

"In the coming years, new outputs from the Fire Model Intercomparison Project and the  Inter-Sectoral Impact Model Intercomparison Project are anticipated and will provide multi-fire model projections of future BA for the first time. The State of Wildfires report will make use of these simulations as soon as they are available, thus improving upon the single model used here and improving characterisation of uncertainty in the projections. Other options could include an assessment of fire weather across future SSP scenarios."

Attribution work undoubtedly needs almost this level of detail: define terms, describe and validate climate models, set up forcing scenarios (following ISIMIP, with another Table?), describe experiments (All, Natural, climate, socio-economic, factual, counterfactual) and replicates, etc. But, do authors need to expend so much text on these processes??

It is important to make the distinction between the experiments very clear for the reader, and we have tried to do this in an efficient way. We have reviewed this section again and have removed ~200 words in an effort to make the description even more concise, including the reviewer's suggested revision at the end of this document.

Seasonal outlooks, which remain elusive for many features in many regions but good on these authors for pushing fire! Long but (in this readers view) useless discussion of ENSO, IDO, AMO, PDO, anyone's additional 'natural' mode of oscillation. Useless because best these authors can derive remains 'associations' (lines around 2680). ENSO prediction remains a fraught and failing endeavor itself, with teleconnections (on precipitation, drought, moisture) sometimes positive, sometimes negative, and occasionally, indecipherable. Those imagined nonstationary oscillations emerge only over large regions based on monthly (at best) and ocean-wide averages. These authors should not accept prior ENSO nonsense! Have the courage to show, from their data, no reliable valid correlations, positive or negative? I would start section 2.5 at line 1008: "To look more closely at impacts that historical climate oscillations (eg. ENSO, IOD, others) might have on landscape flammability … End of diatribe. Do I have this correct? If I need a map, or graphical equivalent of a map, users will likewise?

Thank you for your feedback. We do not fully follow the reviewer's perspective on the predictability of ENSO in numerical weather predictions. ENSO is a highly predictable mode of

variability, and although not perfect it has enormous utility and is the very reason that so many seasonal forecasting systems have been created (ECMWF,MET-OFFICE,NOAA,BoM,JMA). While ENSO prediction does have its complexities and inherent challenges, a substantial volume of work has demonstrated the connection between ENSO phases and fire weather and fire activity (please see our response to your general comment "**Atmospheric Climate Modes**").

In the current manuscript, there is no attribution between fire activity and ENSO. Instead, there is a discussion about the relationship between ENSO and meteorological drivers in areas known to be affected by teleconnection patterns. We simply summarise current literature on this topic and are not aware of any consensus that ENSO indices are unstable, unreliable or invalid (quite the opposite; please see our response to your general comment "**Atmospheric Climate Modes**").

We have revised section 2.5 to reflect our intentions more clearly, reference literature and shorten the text as suggested.

Please see also our response to your general comment "**Atmospheric Climate Modes**".

Line 527, Bespoke Air Quality: not a main focus, move to Supplement? Or, better fit later in AQ sections/discussion?

Removed here - this information was instead efficiently communicated in the figure captions.

Line 545, 2.2 Shortlisting: Good section, authors could write same content with 50% fewer words but no loss of info?

We reduced the text length in section 2.2.

Line 589, "June 2024), and the Canadian Wildland Fire": need one fewer 'ands' and possible punctuation change to fix this sentence. Question: wouldn't we expect Canadian Wildland Fire Information System to use FWI?

Yes, we expect the Canadian FWI to be employed in Canada. We have revised the sentence and shortened the text accordingly.

Line 596, "found to correlate": an equal number of papers show no correlations or no consistent correlations? For other regions or reasons or at other seasons, but still?

There are clear correlations between ENSO and landscape flammability in some regions of the world. Here we are discussing fire weather and not fire activity. Teleconnections during ENSO are a proven fact with a vast supporting literature. Nevertheless, the correlation between fire activity and ENSO can be disrupted by human intervention, as demonstrated by the case of Indonesia and discussed by the experts as well.

Please see also our response to your general comment "**Atmospheric Climate Modes**".

At line 599, I doubt that on rigorous statistical basis, one can document any consistent ENSO etc. teleconnection to fire occurrence or intensity predictability. If authors know differently they need to cite sources?

The links between ENSO and fire have been demonstrated by a volume of literature. Please see also our response to your general comment "**Atmospheric Climate Modes**" for further details.

Moreover, this text discusses the correlation between fire weather and ENSO, not between fire activity and ENSO, since fire activity is influenced by human activities to some extent and its more difficult to prove.

Aragao & Turbo papers do not prove what these authors hope? Authors could reduce this paragraph by ~15% with no loss of info? Line 612, "seasonal skill is limited to 2-3 months" at best! For specific regions, specific seasons, and specific conditions!

We have analysed the expected skill in FWI seasonal forecasts, with a map of skills available in referenced publications (Di Giuseppe et al 2024). As the reviewer rightly pointed out, we should shorten the paper and reference previous works. A particularly interested reader can dig into the publication provided and assess which region of the world have shorter/longer windows of predictability at the seasonal time scale.

Line 654, "Fire Weather Index": you already introduced acronym FWI. Shorten, shorten, shorten! Line 663: I know "ESA CCI" as ESA Climate Change Initiative (I used to chair their advisory board) but users may not? Definition not until line 680?

Thanks for pointing this out. This was a missed acronym, now corrected.

Line 690, "that area": refers only to Greece or to all three focal events?

It was in Greece but that sentence has now been removed

Line 694, Table 3: In header role, PoF control vs ConFire controls (plural)?

As noted on lines 717-718 in the original m/s, ConFire's design allows drivers to be incorporated into more than one of the controls (hence the plural). We have added "Note that for ConFire, explanatory variables can be associated with multiple controls (Kelley et al. 2019)." to the table caption to reinforce this point.

Here, authors describe drivers as "individual explanatory variables??? I know but many readers will not: SMOS?

Confirmed, drivers are individual explanatory variables which become grouped into controls as per table 3 caption (revised for clarity).

SMOS has now been defined on first use (Soil Moisture and Ocean Salinity satellite).

Line 719, "monthly daily means": Confusing, not sure what 'monthly daily' means?

Months can have different lengths ranging from 28 to 31 days, so instead of using monthly total which can be affected by the monthly length, we use a daily means over the month. We have altered this phrase to "monthly averages of daily values" to avoid confusion.

Same line "FLAME"?

FLAME is an acronym for "Fogo Local Analisado pela Máxima Entropia", which we have now added to the text.

Line 769, "apply different modelling techniques"; use a word other than 'different' here. In this context, different can imply different from previous, not what authors intend? Paragraph that follows could reduce by 20%?

Correction: We now use the word "various"

We reduced this text, however a 20% reduction was ambitious without loss of clarity in our view.

Line 857: good point about resolution of Greek data but redundant with much of prior paragraphs?

We re-phrased this for clarity.

Line 861, "MODIS MCD64A1"; by now reader has seen MODIS product designations so many times, he/she no longer knows which label associates with which product? Develop (or copy) and apply standard abbreviations or acronyms?

We removed unnecessary uses of "MCD64A1"

Lines 869,870, more jargon: "MODIS Vegetation Continuous Fields collection 6.1 remote sensed data for <60°N DiMiceli et al. (2022) and collection 6 for <60°N DiMiceli et al.": both methods applied at '<60' N or different method applied in 2015 vs 2022? Confusing or something missing?

This is a typo, and now reads "MODIS Vegetation Continuous Fields collection 6.1 remote sensed data for <60°N DiMiceli et al. (2022) and collection 6 for >60°N"

Line 872: ibicus???

ibicus is the name of the software package we used and is not an acroymn.

Line 882: Important paragraph. I rewrote it at ~150 words compared to authors' 185 words. See example. ~20% reduction with no loss of info?

We have revised this paragraph to reduce the length as the reviewer suggests, and word count is now 153.

Line 889, "maximum VPD as drivers;": I will know VPD as acronym for vapor pressure deficit but many readers will not?

We now use "Vapour Pressure Deficit (VPD)" in the text

Line 932, Table 5: Header row - Controls rather than Control S? 4th row (header plus 3): "temperature appriximated os ISIMIP3a/" - spelling and punctuation errors? 8th row (header plus 7): VCF needs definition? JULES acronym not defined?; 9th row (header plus 8): "tree cover plus none-tree vegetated cover simulated by JULES and bias-corrected as above": capitalize first word (as for other table entries; bias-corrected as above in this Table or in text?

We have corrected all these items

Line 951, "weighted ensemble": 'weighted not explained here? Perhaps later in this paragraph? Authors could rewrite many paragraphs of this section with 15% fewer words.

Correct, this is explained in the following sentence of the same paragraph:

"A weighted ensemble of the monthly outputs of BA, based on regional performance against observational data from GFED5 and FireCCI is used for the analysis. Due to large differences in absolute values of BA between the GFED5 and FireCCI observational datasets and across the models, the weightings in the ensemble are based on model capability to capture relative anomalies present in the observational datasets on a regional basis, and all changes are reported as relative anomalies."

We have revised this section to reduce the length as requested by the reviewer.

Line 965, Season Forecast: S2S addresses a very complex, regionally-specific, non-stabile challenge. In any case, at much coarser spatial (and temporal?)resolution than authors need for fire prediction? Authors' figure S1 (in supplement) shows no (zero) predictability for any relevant region of North or South America and only, perhaps, weak correlations for Turkey near Greece. (What do cross-hatches designate in this figure?) So, why do authors waste time and text space on ephemeral low-resolution 'modes' (ENSO, IOD, etc.)? Emulating S2S so they have

'something' to talk about? Better (and, scientifically, more rigorous) to run their well-prepared attribution experiments to see if anything emerges. If something emerged, not valid to then label it ENSO, IOD, etc., at least not without much more work not relevant here? Getting rid of most discussion of these spurious modes would reduce manuscript by several pages?

The reviewer addresses several aspects, including the spatial scale at which atmospheric variability can be predicted, the predictability and skill of long-range forecasts, S2S, and beyond. We understand that our initial explanation may not have been clear, so we have revised this section to state the following:

1. Modes of atmospheric variability are relevant for predicting burned areas.
2. ENSO is the most relevant mode.
3. We provide an example of how ENSO forecasts are used directly without performing fire danger assessments in some countries, such as Indonesia.
4. For many other regions where large modes of variability are relevant through teleconnections, and the link with fire activity is not direct, we assess anomalies in landscape flammability, which may or may not result in extended fire activity.

This is now stated more clearly. Additionally, we have included a section summarizing the current state of understanding of the connection between large-scale patterns, S2S prediction, and fire activity. The report is not intended to advance fire science in any specific area but rather to review the state of the art of knowledge and applied tools.

S2S prediction of landscape flammability (FWI) is a published dataset available in the CDS as part of the Copernicus Emergency Management Service offering. There is a supporting paper published in Nature Scientific Reports, so we would say that we are not 'emulating' S2S; we are just using available resources for this time scale.

Line 984, "ongoing debate regarding the direct influence of the IOD on Australian fires': Indeed! If we shouldn't rely on IOD with eastern Australia, why should we accept any other hypothetical correlation?

As explained above there are regions where large modes of variability are relevant through teleconnections, and the link with fire activity is not direct, we assess anomalies in landscape flammability, which may or may not result in extended fire activity.

Line 995, 996, "there are few regions in the word where it is possible to establish statistically significant teleconnection between burned areas and atmospheric modes." Absolutely!! So why have authors wasted so much text on speculation?

As said above this has now been streamlined and made it more clear.

Line 1070, "largest contributor to global mean annual totals": statistically speaking, one shouldn't identify large nor small contributions to a 'mean'?

We re-phrased for clarity.

Line 1074, North America: Perhaps a sentence about air quality impacted by Canadian fires here, consistent with South America below?

Yes, good detail now included.

Line 1075, "contributing": 'contributed' instead?

Rephrased for clarity.

Line 1096, "Europe: Low wildfire extent in general," Not a valid introductory phrase?

Rephrased for clarity.

Line 1119, 1120, "Africa: Low wildfire extent in general with BA 13% below average in the African grassland, savannah, and shrubland biome.": Again, not a valid sentence? Again, grassland savannah scrubland distinctions, this time with African inflection? IPCC made definitions not widely accepted by terrestrial ecologists?

Rephrased for clarity.

"African grassland, savannah, and shrubland biome" is the specific (albeit long-winded) name of a biome as defined by Olson et al. https://academic.oup.com/bioscience/article/51/11/933/227116

Line 1132: reader encountered identical information a few lines earlier. Details here, highlights there? We don't need both? Same comments as above apply here. "MODIS BA product": which, what version, etc. Authors need to settle on clear convenient naming system for MODIS burn products?

This information is repeated because the point raised was deemed to be worth emphasising to the more casual reader in the highlights section.

"MODIS BA product" is now preferred throughout the text except where the definition of the precise product (MCD64A1) is deemed particularly necessary.

Lines 1143, 1144, "African grassland, savannah and shrubland biome, which is the largest contributor to global mean annual BA totals": I appreciate that authors know this but nothing in Figs 1, 2 or 3 proves this point?

We added citations to sources of this information.

Figures 1, 2, and 3 are not referenced in this sentence.

Line 1146, "lower BA in savannah-like systems in 2023-24 was not observed in Australia": not clear to this reader whether this constitutes a real decline or a labeling discrepancy?

Rephrased for clarity and references figures 1 and 2..

Line 1175, "58% towards total global BA and 40% towards total global fire C emissions.": this statement needs a certifying citation? Line 1180, "BA extent was in the top three years on record": because Fig 1 shows only a single fire year, hard to credit this observation from Fig 1?

Added citation to the dataset compiled here.

Line 1205, "prominent regional feature": refers to northern South America but not evident in biome data and not particularly notable compared to Africa or much of Asia. Authors have, no doubt, valid reason for this claim but not evident in their maps? Much of the information presented by the remainder of this paragraph repeats what reader encounters later, e.g. in Section 3.1.3.6 (starting at line 1845). We don't need two repeated versions of same data?

The feature is prominent in the panels showing BA anomalies at the level of administrative regions (bottom panels of both Figure 2 and Figure 3).

This information is repeated because the point raised was deemed to be worth emphasising to the more casual reader in the highlights section.

Line 1240, "three good rainfall years have resulted in grass fuel accumulation": here multiple years of abundant rainfall increases fire risk and occurrence? Other places in this manuscript, multiple years of higher-than-average rainfall decreases fire risk? Do authors want to or need to address this discrepancy? If, for attribution studies, they accept months of wetness (dryness) as indicative of low likelihood (high likelihood) of fire, does this statement expose a weakness in their assumptions? Repeated at lines 1340 & following?

The models do not make assumptions about the direction of each factor's influence on BA. Hence, wetness can correlate with increased fire likelihood in some locations and reduced fire likelihood in other regions.

This is consistent with theory in our field: In fuel-limited regions where grass/herbaceous fuels dominate (when available), high rainfall tends to promote fuel accumulation and increase fire extent. In fuel-rich regions with high tree cover, high rainfall tends to increase fuel moisture and thereby reduce fire extent.

We have added these in the following sentence

*Another important aspect is that models do not assume a specific direction for each factor's influence on fire activity. Consequently, wetness can correlate with increased fire likelihood in some locations and reduced fire likelihood in others. This aligns with established theory in our field: in fuel-limited regions where grass and herbaceous fuels dominate, high rainfall promotes fuel accumulation and increases fire extent. Conversely, in fuel-rich regions with high tree cover, high rainfall increases fuel moisture and reduces fire extent.*

Line 1281 & following paragraph: hasn't reader already received this info? Better here, where more relevant? Not in both places, please.

This information is repeated because the point raised was deemed to be worth emphasising to the more casual reader in the highlights section.

Line 1282, "moderate resolution satellite data": another term for MODIS burn data?

Each of the products cited uses a selected dataset as input, which can be either from MODIS MCD64A1, a different MODIS-derived BA product, or another moderate resolution satellite sensor. The phraseology intentionally covers all bases.

Line 1297, "higher resolution and higher overpass frequency": but reader learned a few sentences earlier that high-res satellite information confirmed what GFA implied? High-res in this case refers to Sentinel-2 at 10m, compared to VIIRS at ~400m and MODIS at 500m? Words such as 'moderate' and 'high' not particularly helpful in this case. Fig S3 very helpful!

This sentence was redundant given the prior statement and has been removed.

Line 1310, "of the fire also showed high": 'also'? But, reader learned in previous sentence about ~10% difference among various products?

Rephrased for clarity.

Line 1329: What do authors want readers to learn from this section? That GFA, when it works and when validated by other products, proves reliable? In other cases, however, particularly in urban areas, GFA often fails? Not clear to this reader? When lucky, GFA works great? When unlucky, GFA fails? Plus it detects only larger fires? So?

Line 1335, Review by Continent: Unfortunately, this section reads like WMO annual weather report: warm and wet here, dry there, only station reports, no correlations. Both here and there authors invoke ENSO when they have nothing else to pin to? This reader begins to question value of this entire section. Next year will prove different in detail but no better in overall content or conclusions? Fires always occur. Damage always ensues. Evacuations, suppressions, death: likewise. Nothing new here? "a fairly typical fire season" (line 1406), some places hot, others cold, seems the only useful summary? ESSD should not publish 'fairly typical' reports! But, without this (and WMO reports on weather), where would one find such records? Authors will have saved readers lots of work? Nothing about livestock, please! For this manuscript, readers also need zero info about fire fighting experiences or equipment. For me, excessive detail about Evros fire; how will this info prove relevant to a) other fires, and to b) larger questions about changing fire regimes? Evros fire notable only by BA? So? A fire killed two civilians? Condolences, but what use to a reader to know that info?

A collective response to the two comments above.

The authorship sees these elements of the report as an important contribution to progress in the fire science community, and we are not in favour of removing them from the report completely. This section has been moved to appendix A.

A key objective of this report is to catalogue extremes in the past fire season. Synchronously, there is a push within the fire science community to consider not only bulk regional fire statistics but also other metrics of fire properties, behaviour and impact. Statistics derived from Earth Observations are of fundamental importance in fire science, however they may not directly correlate with the impacts of fires on people and the environment. While the impacts of fires are not always quantifiable (i.e. data-oriented, a focus of ESSD), the descriptions of fire impacts from our regional experts do add context and complement our data-oriented findings.

Hence, utilising the current tools available for studying individual fires (section 3.1.2.2) and gathering information about individual fires that were impactful in diverse ways (section 3.1.2.3) are seen by this authorship as important and progressive contributions to the field of fire science. By reading across a decade of reports, a reader in 2034 will be able to construct a timeline of wildfire impacts on people and the environment.

We deleted sentences about fire impacts on livestock predation in Botswana and Mongolia.

Line 1729, Figure 7: Potentially informative but provided here at very poor resolution?

Apologies, we improved the resolution of the figure [now figure A3].

Line 1738, "across the Country that were well in excess" why 'County' not Canada or country?

Correction: country

Line 1758, "if the USA had also experienced a high fire year, the air quality would have worsened in many states": is this a guess or a statistically-valid prediction as one outcome of this work?

Perhaps this is best described as an application of logic. We removed it as it was not central to the point made in the paragraph.

Line 1941: Again, what do authors want readers to have learned from this section? Admiration that authors have gathered lots of info in one place? Granted. Can a single reader assimilate all this info? No. Does this accounting, by continent or cumulatively, make any difference? For this reader: no.

We expect section 3.1.3 to prove useful to returning readers who wish to derive information from specific sections of the report, rather than to read the report in its entirety and attempt to assimilate every piece of information presented.

Line 1983, "declines in deforestation rates and deforestation-related fires have fallen": 'declines' have 'fallen'? Silva Junior et al. paper very confusing but title probably best conveys their message "Brazilian Amazon deforestation rate in 2020 is the greatest of the decade". Not clear what the authors intend here?

Correction: removed "declines in"

Lines 2125, 2126, "Some anomalous BA starting in August and extending to November": not a complete sentence, readers need to guess at authors' intent.

Correction: "The most widespread BA anomalies emerged in August 2023 and extended through to November 2023 (Figure S11)."

Line 2136, Figure 10: Despite good intents and good efforts by authors, this figure not really readable enough to prove useful? +10% for Canada, +2% for Greece and western Amazon. This reader does not believe, based on information gained earlier from this paper, that authors can expect readers to distinguish +1% from - 1%. Reader needs some basis to minimize uncertainties here! Resolution of figures in my version = very poor?

[Now figure 6]

We exported the figure at higher resolution.

Please see responses to more general comments elsewhere regarding how uncertainties are conveyed. In this revision we are clearer throughout on what uncertainties are/are not included across the analyses.

Line 2153, 2154, "inadequacies in predicting certain ignition sources or accurately representing fire propagation across vast landscapes in current forecasting systems" A very important conclusion, noted here in case it does not re-appear later.

We have ensured that this finding is represented in the conclusions section: "In addition, unaccounted factors such as ignition, suppression, and landscape fragmentation, likely played important roles in modulating the Western Amazonian and Greece events."

Line 2161, 2162: "inadequacies in predicting certain ignition sources or accurately representing fire propagation across vast landscapes in current forecasting systems": Another strong important conclusion, conveyed in highlights but repeated in detailed section? Same text!!!

Removed the repetition.

Line 2165, "linked to the strong El Niño.": for reasons already highlighted, including by authors, this reader doubts that authors can assert this link?

Western Amazonia is significantly affected by drought during positive ENSO events. These two papers have now been added to support this statement.

https://agupubs.onlinelibrary.wiley.com/doi/full/10.1029/2006GL028946

https://www.nature.com/articles/srep33130

Line 2173, "no single factor can explain the most severe fires": another strong positive statement, in highlights, waiting for details to follow?

This important conclusion has now been made into the highlight of this section

*The synchronous occurrence of anomalies in both weather and fuel characteristics created the conditions leading to the extremely large burned areas recorded in all three events.*

Line 2199, Figure 11: Very important helpful figure! This reader wonders how a different fire prediction model, something other than Canadian FWI, might have worked? At longer time scales (longer than 2 weeks or so), no fire model would show significant skill? Authors have skillfully defended choice of FWI, and it seems to make sense given importance of Canadian fires, but they must know about other fire prediction systems? (Thinking here of earlier [perhaps now dated] work with GEE by Gray et al. in ESSD: https://doi.org/10.5194/essd-10-1715-2018). Some discussion or assessment? Who else but these authors would know?

Thanks, this is a very good point and something that was touched upon in the conclusion section, which has now been moved to Appendix B. We acknowledge that the FWI is not the only index for fire danger, and sub-indices of this system or other fire danger systems developed for other regions (such as NFDRS and Mark5) can be used to improve the information provided to forestry agencies in biomes different from the boreal forest of Canada, for which the FWI was derived.

It is true that the performance of fire weather indices could, in absolute terms, be better for some cases than what is shown here. However, the scope of this section is to highlight the significant role of fuel rather than to provide an overall assessment of all the indices available. As the reviewer suggested, it needs some clarification, and we have added the following sentence:

*The FWI is not the only index for fire danger, and other fire danger systems or sub-indices of this system may correlate more strongly with BA or fire behaviour metrics in some environments. Nonetheless, FWI is widely applied due to its good performance across a range of environments (Di Giuseppe, 2016; Jones et al., 2022) and so we adopt it in the current work.*

Line 2260, Figure 12: Again, informative and helpful. Here, however, reader confronts (for the first time?) 'dead' fuel vs 'live' fuel, both in reference to fuel moisture. If this dead vs live difference proves important, should authors highlight these factors for readers? 'Deap' should instead read as 'deep' in all cases? Lightning basically constant in later months in Amazonia?

Thanks for pointing this out. An explanatory sentence has been added

*Dead fuel, with its lower moisture content and higher combustibility, often plays a significant role in determining fire ignition. During extreme events, it is the dry live fuel that burns, contributing to the overall severity and intensity of the fire.*

Lines 2273, 2274, "lightning as a key source of ignitions in the region": referring to Canada but flash rates much higher in Greece and higher yet in Amazonia?

True, an explanation of this sentence is needed here. The lightning density reported here is composed of two contributions: it includes cloud-to-cloud (intracloud) flashes, which are not useful for us, and cloud-to-ground flashes, which are useful for fires. The percentage of cloud-to-cloud (intracloud) lightning compared to cloud-to-ground lightning varies, but generally, cloud-to-cloud lightning is more common. Approximately 70-90% of lightning occurs within clouds (intracloud), while about 10-30% reaches the ground (cloud-to-ground). It is the cloud-to-ground lightning that interests us.

Due to the convective nature of tropical weather, the Amazon experiences, in absolute terms, many more lightning flashes than Canada. However, lightning in Canada can have worse consequences because there is a higher percentage of cloud-to-ground lightning. In mid-latitude regions, particularly during the summer months, the proportion of cloud-to-ground lightning can be higher compared to tropical regions. Additionally, in these regions, it is not infrequent to have dry lightning in summer. Dry lightning refers to lightning that occurs without significant accompanying rainfall.

We have added a sentence

*Most of the explainability of the Canada event comes from anomalous weather conditions. Increased lightning activity often coincides with or precedes significant fire periods, indicating lightning as a key source of ignitions in the region given the contribution of the cloud-to-ground flashes to the total predicted lightning activity. This is in agreement with the attribution of 59% of wildfires and 93% of total BA to lightning ignition sources in Canada during 2023 (Jain et al., 2024).*

Line 2281, "vast, densely vegetated": referring to Canada, have authors given readers any bases for accepting this statement?

Removed as not strictly necessary.

Line 2293, "intrinsic difficulties in forecasting isolated extreme events": Indeed! Major primary conclusion, but not emphasized by authors?

This point is now reiterated in the conclusions section:

*This highlights the intrinsic difficulties in forecasting isolated extreme events and underscores the need to enhance early warning systems beyond fire weather to consider fuel availability and ignition variability.*

Line 2329, "anomalous weather conditions subsided later in the fire season": not true, at least as shown in Canadian section of Fig 14? There, weather controls look nearly identical Apr vs Sep?

We corrected to: "*However, the anomalous weather conditions subsided in May through the early summer, though by September showed an increased likelihood of contributing to the increase in burned area anomaly seen in the late fire season.*"

Line 2356, 2357, "direct human-induced landscape changes exerted minimal influence on the extent of burned areas in Greece": to my eye, human influence (row 5 in Fig 14) looks vastly different Canada to Greece, and very important for Greece?

Compared to the influence of dryness in Western Amazonia, which is the dominant control on burned area for most of the period assessed, human influence in all three regions is minimal. However the reviewer correctly points out that there is a slightly larger influence in Greece compared to Canada, which we have now added to the text:

"Although direct human-induced landscape changes exerted greater influence on BA extent in Greece than in the other focal regions, this influence remained small compared with weather factors."

Line 2361, legend to Fig 14, "increases/decreased": should read as 'increases/ decreases'?

Corrected [now figure 10]

BA range data, which varies for each site, not clearly evident? Hidden by overlapping graphics?

BA range is set as the same for each focal region, hence the need for only one colourbar.

Time spans (X axis) also vary in this figure. Gives this reader even less confidence that I understand the figure? I would have said: a) minor fuel influence at all three sites but different signal Canada to other two; b) strong drought signal in Amazonia; c) solid weather signal at all three sites but perhaps dropping off late in season at Amazonia; d) minor human influence Canada and Amazonia but strong in Greece case? Personally, I don't understand how authors got probabilities to tenths of percent and I find no basis to trust + 5%? A highly uncertain model (ConFire) run multiple times on multiple cases does not improve reliability? If Fig 14 presents site-specific BA anomalies relative to site-specific means, but all with same magnitude (same Y-axis units) then I get even more confused? It seems key to me to understand Fig 14 but I don't?

The reviewer's interpretation of the figure is correct, although for (d) we would interpret human influence as secondary or tertiary in all regions. The X-axis was chosen so that it is consistent with Figs. 11-13 in the rest of this section. In terms of the reliability of ConFire, we argue that the model in fact clarifies the uncertainties further than most other fire models, by constraining the model by observations. The uncertainty range given by the model may therefore be large, but the confidence in this range can be very high because it incorporates all possible relationships between burned area and its drivers. By comparison, most other fire models that are integrated within land surface models give one answer that is highly parameterised, and uncertainty is subsequently derived by alternative means. Therefore we disagree with the reviewer that ConFire is a highly uncertain model, rather that it is a model designed to assess uncertainty and is therefore highly reliable.

Line 2366, legend to Fig 14, "as a fraction of land area": I appreciate effort by authors to quote and illustrate uncertainties but this reader still does not understand the bottom graphs? Fraction of land area?

The uncertainties in the bottom panels refer to the standard deviation of the probability distribution in the plots above. To make this clearer we have changed the units from fraction of land area to uncertainty in BA extent to match the rest of the plot.

Line 2420, Figure 15. Not at all understandable by this reader. The figure legend seems wrong? E.g. this reader sees, for July and June (why this order?),

We have now swapped the ordering so that June comes first. [now figure 11]

top to bottom: BA anomaly (top), increase from controls (mid), decrease from controls (bot). The legend instead implies left right distributions: "left of the other two maps looks at anomalies in

controls that would cause higher BA" while "right map shows drivers that would have led to lower than normal levels of burning". Not correct?

This was an old legend from a previous figure layout and has now been corrected to "top" "middle" and "bottom".

June had greater -7 to +7 BA anomalies than July (-3 to+3) so, strictly speaking, user should not attempt monthly intercomparisons?

Correct, the comparison should be between burned area in the top map and control influence in the lower maps, as it is used within the text

Greyed-out areas should inversely correspond to top BA anomalies, for mid plots, but directly correspond for lower plots? This seems very hard to detect. Then, four size and color dependent dots in each 0.5 degree grid box? I think I understand goals for this figure, but fails for me. Same problems extend to Figure S13, but for two additional months (Sep, May).

Correct, in the middle plots the areas that are not greyed out correspond to greater than monthly average BA (red squares in the top plots), and in the bottom plots the areas that are not greyed out correspond to lower than monthly average BA (blue squares in the top plot). We do acknowledge that there is a lot of information presented in this plot, but we feel it is important to present how controls can co-vary; as we state in the legend, some controls act in unison in some areas to cause or prevent extreme levels of burning, whereas in other areas there is one dominant control. The complexity of the figure highlights the complexity of fire as a process - there is often more than one driver that is responsible for an extreme fire, because by definition they are highly unusual and many factors have contributed to making it extreme.

Also to Fig S14 for Amazonia. Again: unreadable. Sorry, these old eyes can make no sense of this arrangement. I give up entirely; skip this section.

As we state above, we believe it is important to show all the controls that contribute to making these fires extreme, and sometimes that means presenting complex information where multiple controls combine together. It takes a little more explanation than a simpler plot, but we hope we have given enough detail in the text for a reader to navigate the key messages, especially through giving 'highlight' summaries (now in the Conclusion) which allow a reader who doesn't want to go into depth in the details to still understand the results.

Line 2492, Attribution. Similar to above: highlights followed by details. At first I liked this arrangement. Now I see it mostly as redundant; largely a waste of space?

Highlights were removed.

Line 2501, 2502, "All forcings combined have led to an overall reduction in today's average BA across Canada." Interesting. But this contradicts all previous assessments for Canada?

The previous assessments for Canada are identifying the impact of anomalous controls over the last 20 years (in the previous section), and the impact of anthropogenic forcing and total climate forcing on fire weather and burned area. When we refer to 'All forcings combined' we are also assessing the impact of changes in land use on the net median burned area, which we find has mitigated some of the influence of climate change on fires. Both statements are thus not contradictory. However, as this is not an important highlight, we have removed it to avoid confusion.

Line 2509, 2510, "Climate change has increased today's average BA in the Mediterranean region, but this has been mainly offset by socio-economic factors.": reference Greece but same conclusion as for Canada: socio-economic offsets climate leading to no net change?

Correct, socio-economic factors offset the influence of climate change, so there is a smaller change in median burned area than we would see from climate change alone (slightly positive in MED, slightly negative in NWN and NEN).

Line 2517, 2518, "Climate change has increased today's average BA in the Northwest region, and all forcings have led to an overall increase in burning". In this conclusion, western Amazon differs from Mediterranean and from North America?

Correct.

Line 2529 to 2531, "probability of experiencing the high fire weather observed during June 2023 is more likely in a climate forced with anthropogenic emissions.": Referencing high BA across Canada, this conclusion (plus Fig 17) contradicts what reader found in highlights?

We don't believe that this conclusion contradicts the highlight section, which states that: "In Canada, anthropogenic forcing increased the chance of high fire weather in 2023" and "Human influence at least doubled the probability of experiencing high fire weather in June 2023". We also find that "It is virtually certain that total climate forcing increased recent high BA in the region by up to 38%", which is assessing high burnt area as a separate metric from fire weather, but again these results agree that climate change and anthropogenic emissions have increased both high fire weather and high burnt area in Canada.

Line 2578, 2579, "BA in Canada in June 2023 was 0.8-38.0% greater due to total climate forcing in the 2003-2019 period" Perhaps labels have confused this reader, but if 'total climate forcing' = 'All', then this conclusion contradicts what we read in highlights?

Please see Table 4 for descriptions: "Total climate forcing" refers to changes in burnt area due to climate change, whereas "All forcing" refers to changes in burnt area due to climate change and socio-economic factors. Socio-economic factors are mitigating the influence of climate change on burnt area, so we assess All forcing to understand the net impact.

Line 2599 to 2601, "uncertainties around whether total climate forcing and socioeconomic factors caused an increase or decrease in BA are higher, and the smaller region size makes detecting a strong signal of change more challenging". About Greece, safest to draw no firm conclusions?

We acknowledge that the uncertainties are higher for Greece than the other regions, but there is enough information for us to be able to draw some important conclusions nonetheless. We give the confidence level in likelihood for each result as well as the range of potential increase / decrease, and thus our conclusions are completely clear about what we can and can't say with confidence about the fires in each region.

Line 2610, 2611, "additional burning could have been up to 0.8-36.2%" but, from

Line 2615 "socioeconomic conditions (95.98% likelihood) increasing BA by 0.18-5.32%. Authors lost this reader long ago on how to trust 1 to 36 vs 0 to 5. In any case, for western Amazon region we can only say with confidence: perhaps?

Western Amazonia is actually one of the regions where we are very confident in our attribution statements, as we show with likelihoods of 99% for increased burned area from total climate forcing, and 95.98% likelihood of socio-economic factors exacerbating the fires. Perhaps the reviewer is confusing the amount of additional burning with the confidence level (likelihood)?

Line 2623, legend for Fig 18. No explanation/assignment of orange color? Oops, now I get it: orange = factual, blue = counterfactual, combination (overlap) = purple. Not immediately clear? Taking western Amazon, this reader finds 1.53 to 7.66 (2 to 8) modest increases in high BA (how authors derive 99% confidence for curves that basically overlap remains beyond me) vs 0.18 to 5.32 (0 to 5) increase in socio-economic factors. These numbers DO NOT accord with those at lines 2610 and 2615 in text. This reader now concludes, using corrected numbers, no net change for western Amazon in high BA?

We think there must be some confusion here, as there are no numbers quoted in line 2610, and the numbers in line 2615 are the second of those quoted by the reviewer (0.18-5.32%). Over 2003-2019 the change in high BA for Western Amazonia is 1.53-7.66% due to total climate forcing, and extending the analysis up to 2023 we estimate an increase of 2.3-46.7%.

Line 2667, legend to Figure 19: In this Fig, authors did not use transparent colors so colors do not blend as they did in Fig 18? If authors did not present Y-axis (probability) in log scale, users would see nothing? I figured out what bothers me about these ultra-precise uncertainty estimates: authors compare final products of long train of manipulations, each with its own (sometimes acknowledged, sometimes not) uncertainties, source observations (satellite or reanalysis), aggregations or disagregrations, final manipulation for effective presentation, etc. but only report (valid?) uncertainties deriving from the last step. For ESSD, we need full end-to-end, uncertainties! A full uncertainty budget!

For satellite data: orbit changes, sensor degradation, processing levels, cloud or aerosol or competing species absorption, etc.

We agree with the reviewer that including uncertainty estimates is important within our analysis, which is why all of our results in this section are presented with uncertainty ranges. In particular, we acknowledge the large uncertainty range across observations of burned area, and within Fig 19 [now figure 15] to which the reviewer refers here, have established a methodology which enables us to still report useful results. For example, we report our results in terms of relative anomalies rather than absolute values so that we focus on the relative *change* in burned area due to climate forcing, rather than highly uncertain absolute totals. When we analyze relative anomalies as opposed to absolute values, any consistent biases in the satellite data will be eliminated when converted to relative anomalies. One example to demonstrate this is the fact that the GFED5 product registers ~2 times more burned area than FireCCI5.1. However, when transformed to relative anomalies, both products are almost identical. It is possible that sensor degradation and the quality of cloud and aerosol formation could potentially have an impact, but these factors are already adjusted for in the MODIS product that we are using. If any of these factors still exist, they may contribute to small-scale noise or consistent biases.

For reanalysis: data ingestions, processing, discarding, spatial inhomogeneity, etc.; reanalyses derive from need to produce error matrix! Processing: use of anomalies, use of relative anomalies, changes in axis ranges, etc. Uncertainties at every step, propagating, canceling, amplifying, whatever. To report only the statistical inter-comparison of final steps (e.g factual vs counterfactual) ignores a host of uncertainties inherent in how authors got that far? I would rather not check all numbers (although prior comments relative to Figure 18 suggest that someone needs to validate), but I react negatively, as will most readers, to tenths of a percent in final precision when readers know and authors well know that basic uncertainty proves higher in every case! How can authors report -0.2% differences between model outcomes when we and they know that we can't even measure BA from satellite to better than + 10%? I would feel very pleased to get proven wrong on this, but, with no disrespect intended for authors, they make much in these sections of very tiny differences only by ignoring much larger underlying

uncertainty! ESSD readers need to see underlying systematic end-to-end uncertainty budget. Very difficult to produce, especially in this case? Perhaps, but tell us. Might mean that one can not, in fact, distinguish factors that influence max BA or median BA? Tough news, but tell us; we need to know. And we need to hear recommended solutions from these experts! Summary: great respect for efforts and reports, but manuscript lacks an encompassing uncertainty budget.

The main objective in this section is to establish multiple lines of evidence of attribution, which involves using the FireMIP attribution analysis as one of three tools available, as stated in the methods. All three of the attribution tools we use are based around comprehensive uncertainty quantification techniques. With FireMIP attribution and ConFire specifically, uncertainty quantification is more rigorous than in any previous attempts, as it encompasses more sources of uncertainty, or we find ways to reduce the impact of inherent model and observational noise. Unlike WWA attribution, which is often rightly considered rigorous and robust but often reports meteorological variables, our frameworks target fire variables (fwi, extreme and median burned areas). Combined with our uncertainty quantification, that means that if our frameworks show an attributable result accounting for uncertainties, we can have much more confidence in that result than previous studies - especially when all three frameworks provide qualitatively similar attributions, as they do for Canada and Western Amazonia.

While there are uncertainties associated with satellite data, using two products with large differences in absolute values but agreeing on relative anomalies cancels out consistent discrepancies, as explained above. Additionally, relative anomalies cancel out certain factors in models as well. When dealing with large regional variations over extended periods, the "noise" becomes less relevant.

It is indeed important to acknowledge that there are uncertainties associated with inputs as well. However, our focus here is on attribution to climate change. We are not assessing the uncertainty across different driving datasets and how that translates into burned area, but rather how different forcings translate into differences between the two simulations i.e. how much burned area is changed due to total climate forcing. If the reanalysis dataset would in fact be 10% too dry in a certain region, then both factual and counterfactual simulations are likely predicting more burned area because of this. But again, we are not taking these at face value, but rather transform them to relative anomalies. Thus, uncertainties in the inputs do not directly translate into uncertainties in the end results. One way of estimating the effect of uncertainties associated with the climate reanalysis dataset on these results would be to repeat the simulations with the three alternative reanalysis datasets provided by ISIMIP. The runs used here involved 7 modelling groups and took several years to collate, incorporating additional runs would be well beyond the scope of this study.This does not mean that additional uncertainties could not be assessed in the future. But the attribution in this report represents the current state of the field, and serves as a starting point to incorporate and/or constrain all these additional sources of uncertainty in a separate study.

Line 2674, Seasonal and decadal: Again, highlight followed by details, leads unfortunately to substantial redundancy.

"Highlights" removed.

Line 2678 & following paragraph, IOD and ENSO: But, authors stated earlier that no statistically valid links exist between BA and atmosphere modes (line 995, 996). Why then do readers confront more discussion of IOD, ENSO ("ENSOneutral"), etc?

We have stated that there is a clear connection between ENSO and burned areas in many places, following the current state of the art research. This section is dedicated to an outlook for the next fire season, and from the available forecasts, we learn that there is a transition toward ENSO-neutral conditions. Forecasts from all the available seasonal systems confirm this transition, and it is reported here in one sentence for information.

Line 2688 & following paragraph: But readers learned in previous paragraph that (for Canada) "no clear signal for extreme anomalies is present". Why then does this paragraph focus on scenario differences (SSP585 vs SSP126) and mitigation actions. To remedy signal that does not exist?

This paragraph is for multi-decadal outlook, whereas the previous is for seasonal outlook. We have added "In the Multi-Decadal Outlook, effective strong mitigation…" to make this clear.

Line 2701, "above PI": readers will not understand PI without definition?

Pre-industrial, which we now define on first use.

Line 2704, 2705, 2707, 2708: as if readers did not already suspect ,"projections show a high level of uncertainty in all regions" and "we cannot determine if current mitigation efforts are effective". Valid, honest conclusions, but don't those render this entire section moot?

In addition to giving firm answers, part of the aim of this report is also to clearly outline what we are not able to robustly comment on, so that we inform the current state of science and to give a roadmap for where the science needs to focus on next.

Line 2745, legend to Fig 20. If authors already concluded lack of statistical connections, and one of our best modeling groups predicts this wide (useless?, >3C in every case?) range of near-future temperature ranges (one for ocean SST in central equatorial Pacific region, the other we would have to search for) why should reader give any credence to any of this? Prove challenges by showing current weak model skill but don't waste our time with speculations?

The ENSO index is not a temperature range but the difference in temperature between two areas in the Pacific. The legend describes it as an anomaly plume. A 3-degree SST anomaly is very high, and in fact, ENSO 2023 was a strong ENSO and has been highly predictable. The models all show a positive ENSO status converging into an ENSO-neutral status. The variability of the prediction measures the uncertainties in the prediction, but they all indicate the same trend. Therefore, we are not sure why the reviewer finds this not useful. In any case, this picture, which is available on the NOAA, C3S, and ECMWF websites, has now been removed from the report to keep its length short.

Line 2679, legend to Fig 21: Based on careful reading of all prior guidance, this reader has learned to disregard this information entirely. July & August for 95th % basically empty. Reader could easily have guessed at areas for 75th in July and August? Nothing against ECMWF, just that we should have learned by this point in this manuscript to trust no intense FWI probability < 100%, which basically don't occur anywhere on the planet after June?

Figure 21 [now figure 16] shows the probability of an anomaly, not the probability of a value. For example, if in a certain region the FWI is consistently around 70 (extreme) every year, and this year it is forecasted to be 70 in all ensemble members, the probability of being above the 75th percentile will be zero. However, if 25 ensemble members show values around 100, the probability of being above the 75th percentile could be around 30%. This metric allows for calibration across different regions.

Seeing a probability greater than 0 for extremely anomalous conditions at month 3 is very unlikely, as predictability is mostly limited to 1-2 months. However, it can occur when large modes of variability are active as shown in Di Giuseppe et al 2024 https://www.nature.com/articles/s41597-024-02948-3

Line 2780, future changes section: Direct mapping of BA pixels to model output pixels? No intervening fire model, predictive or attributional? Methods at Lines 1030? JULES never defined? But figure time extents do not accord with method specified boundaries. Fig 20 definitely not, Fig 21 perhaps but only for Canada?

Figure 20 has been removed. Figure 21 [now figure 16] is global

We now define "Joint UK Land Environment Simulator Earth System (JULES-ES)" on first use.

and "The Inter-Sectoral Impact Model Intercomparison Project (ISIMIP)" on first use.

Line 2786, 2787, "ISIMIP3a and ISIMIP3b": this reader understands a few differences between 3a and 3b but authors provide this and most readers no help and no guidance?

We have now provided additional explanation of ISIMIP3a when first used in section 2.4.2:

"*Our attribution to total climate forcing considers changes driven by climate change since the pre-industrial period, including both anthropogenic forcing and natural variability in line with the IPCC WGII and the Intersectoral Impacts Model Intercomparison Project 3a (ISIMIP3a) definition of climate change impact attribution (IPCC, 2023b; IPCC 2023c; Mengel et al., 2021)*"

and define the difference between ISIMIP3a and ISIMIP3b in section 2.5.2:

"*While ISIMIP3a provides reanalysis datasets to drive models for impact assessments, ISIMIP3b provides driving data from 5 bias-corrected GCMs,*

Lines 2796, 2797, "likelihood of a 2023 fire event recurrence increases to 0.31% - 0.9% - two to almost four times as likely as today (Table 7; Figure 22). But, remains highly unlikely? This outcome (these numbers) very hard to extract from Table 7 - one needs to confuse min and max or cross scenarios - so data come from Fig 22?

[now figure 17] [now table 6]

This projection is across scenarios, which we have made clear in the text: " By the middle of the century, the likelihood of a 2023 fire event recurrence in any given year increases to 0.31% - 0.9% across scenarios - two to almost four times as likely as today (Table 7; Figure 17). While projections are slightly higher for SSP585 (0.46-0.9) compared to the other scenarios (0.31-0.6 for SSP126 and 0.36-0.69 for SSP370), there is still overlap between these projections."

This reader unfortunately gets nothing from Fig 22, particular at level of detail implied here? In next 10 years? Nothing.

We are surprised to find that the reviewer gets nothing from Fig. 22, given we show that burned area of the level seen in Canada in 2023 will be up to 3.4 times more likely by the end of this century. Even though changes in burned area may not be large in the next 10 years, the important point of this figure is showing the changes in the context of the rest of this century where there are potentially significant increases in burned area projected by 2100.

However, to make the plot clearer we have updated the following:

- Caption: "Future projections from ConFire of the likelihood of BA extremes OF THE MAGNITUDE SEEN IN 2023-24, and their corresponding fuel and moisture controls";
- The axis label to read "Annual Likelihood of an Event like 2023-24" instead of likelihood (%)
- "Control Strength" now changed to "Influence of Change in Control on Annual Likelihood"
- "Burnt Area times more likely" now changed to "Risk Ratio of an Event like 2023-24"

Line 2798, "different SSP scenarios diverge significantly by 2070": Really? By what criteria? Lack of range overlap, as authors specified at line 2790? Very hard to confirm authors' conclusions from Fig 22!

It's SSP126 and SSP585 that diverge from 2070, as seen in Fig. 22, and we are now more specific in the text "The  SSP126 scenario diverges significantly from SSP370 and SSP585 after  2070."

Lines 2802, 2803, "at least one event similar to or worse than the 2023 event occurring again is estimated to be between 16% and 25%": If reader hopes to confirm these author-specified changes (at least these seem more moderate), where should they turn. For Canada in Fig 22, likelihood range only goes 1 to 5%? Does one need to convert and guess from BA event numbers? Not acceptable.

In the text we use an interpretation of the likelihood that's shown in Fig. 22, to make it more relatable to the reader and to explain the results in a different way. We now outline how this calculation is done just before this piece of text so that it is fresh in the reader's mind.

Line 2804, legend to Table 7: No explanation of colors? Obviously related to magnitude, but how? Many cells report insignificant results (48 of 160, 30%)? Predominantly not valid for W Amazon, regardless of return rate (1 in 6 vs 1 in 100)? Marginally valid for Greece? Please fix wrapping and display for 2nd column. Entire table needs improvement! Minimum likelihood outcomes for Canada 'increase' from 0.08 to 0.31? Readers should learn what from this? Finally, here, a definition of 'extreme', in a UNEP report? But, in text later (lines 2845, 2846), UNEP 2024 seems to specify something different? This reader does not want to search for UNEP 2024; we need authors' best explanations!

We break this down into the individual points we believe the reviewer is making:

- *Explain the colours in the legend to Table 7*

We have now added an explanation of the colours to the legend [now table 6]: "Colours show linear increase of likelihood (red) and frequency (orange), where darker shade indicates higher values."

- *Fix wrapping for 2nd column*

This has now been fixed

- *The min. likelihood for Canada SSP126 is 0.08 (2010-2020) up to 0.31 (2040-2050). What should the readers learn from this?*

As we state in the text, the fact that there is a small change in likelihood under SSP126 means that climate mitigation can have a big impact on reducing the chance of an extreme fire happening again. The fact that there is a big divergence between SSP126 and the higher SSP scenarios is an important result that can help inform decisions around mitigation policy.

- *What is the definition of extreme according to UNEP 2024?*

In Table 7 we consider the change in likelihood of burned area of a similar magnitude to the 2023-2024 event in each region. In Western Amazonia this is a 1-in-6 year event, which may not be considered "extreme" by the definition of the other 2 events which were 1-in-700 year and 1-in-80 year events. We therefore also consider what the likelihood of a 1-in-100 year event would be, taking the UNEP definition of extreme. This is explained in the legend and in the text.

Line 2825, "changes in both [djc: fuel load and fuel moisture] controls are needed to explain divergence between SSPs": very hard, perhaps impossible, for reader to credit this conclusion based on these data!

If you look at the difference between the scenarios in the 2070s for Canada in figure 17, fuel controls the larger differences between SSP126 and SSP585, whereas by the end of the century the SSPs follow the changes in moisture more closely. Therefore both controls are needed to explain the divergence between the SSPs. However we have edited the text slightly to make it more accurate:

"The divergence of likelihoods between the two scenarios is associated with increases in both fuel load and fuel dryness."

Line 2828, legend to Fig 22: No indication of when or if signals emerge from uncertainty bounds, either in time or in event return times or % likelihood? No explanation of means, medians, whatever, indicated by lines within each bar? This reader guesses: perhaps a scenario difference by 2100 in Canada, driven perhaps by fuel abundance? Before that, nothing significant? For Greece, perhaps significant differences by 2090 and 2100, driven by moisture? For W Amazon, nothing (zero significance) for any decade or any scenario? But these personal conclusions diverge substantially from authors' narrative? Here they compare ConFire simulations to re-analyses? No indication of large uncertainties in each?

We split this paragraph into individual questions as follows:

- No indication of when or if signals emerge from uncertainty bounds, either in time or in event return times or % likelihood?

The signal emerges from the uncertainty bounds when the SSPs don't overlap, which we state in the text.

- No explanation of means, medians, whatever, indicated by lines within each bar?

We have now added to legend that each small line is a GCM: "with individual bars representing different GCMs".

- This reader guesses: perhaps a scenario difference by 2100 in Canada, driven perhaps by fuel abundance? Before that, nothing significant? For Greece, perhaps significant differences by 2090 and 2100, driven by moisture? For W Amazon, nothing (zero significance) for any decade or any scenario? But these personal conclusions diverge substantially from authors' narrative?

The reviewer's conclusions are mostly correct here, although there are changes in western Amazonia in SSP585. This is described in the text:

For Canada: [line 2025] "The SSP126 scenario diverges significantly from SSP370 and SSP585 after 2070. Under SSP126, the likelihood of an event like 2023 stabilises at 0.3-0.8% in the 2070s and remains largely unchanged until the 2090s" and [line 2043] "The divergence of likelihoods between the two scenarios is associated with increases in both fuel load and fuel dryness"

For Greece: "[line 2118] SSP350 and SSP585 show significant increases in the likelihood of an event like 2023 by the 2070s (relative to the 2010s), and also diverge significantly beyond SSP126 in the 2080s" and [line 2123] "The divergence of likelihoods between SSP126 and the two other scenarios (SSP350 and SSP585) is associated with increases in both fuel load and fuel dryness, with particularly striking differences in the latter across the scenarios (Figure 17)."

Western Amazonia: [line 2163]: "Under the SSP585 scenario, the likelihood of an event like 2023 increases significantly in the 2090s to 18.2-21.0%, representing a factor 1.2-1.3 increase versus the 2010s. This increase is primarily attributed to lower fuel moisture, in line with declines in fire weather in this region (Section 4.2.1). On the other hand, in the SSP126 scenario representing strong climate change mitigation, the likelihood of an event like 2023 does not change significantly at any point this century (Table 6)."

- Here they compare ConFire simulations to re-analyses? No indication of large uncertainties in each?

Correct, ConFire is compared to reanalysis in Fig 22 (reanalysis being the dashed line in each sub plot). In both ConFire and the reanalysis, we have already used the uncertainty distribution to calculate the probabilities, therefore it is not appropriate to include additional uncertainty on the uncertainty. This is similar to what has been done in other papers including Kelley et al., 2021 and UNEP 2024.

Line 2823, legend to Fig 23: Not the US-Canada border that I know? Even if outlines stay north of Great Lakes (border actually intersects lakes via mostly straight midpoint lines), remainder to west looks step-wise distorted whereas actual border remains dead (latitudinally) straight, Minnesota to Vancouver Is? Relevant because only signal seems to occur along southern border, near N Rockies and/or Glacier NP. Surprised to see, in col 1, an apparent minimum along southern border?

[now figure 18]

This minimum along the southern border is also clearly seen in the observations in Fig. S29, which shows that the simulation generally picks up high and low BA areas.

 (Do authors think readers will have any concept or trust of 0.001% difference in BA? Or any concept of 0.05 vs 0.5 % change?)

Please see our response regarding non-linear scales below

In middle and right columns, at least for SSP585, fire extents and likelihood of return seem to max in that area? This reader develops no confidence in any part of this figure?

As stated in the caption, the middle column shows the change in "BA extent projected for 2090-2100 expressed as a multiplier of 2010-2020 values." The area on the southern border, therefore, shows a likely doubling in BA. However, BA was low in this region between 2010 and 2020, so this doubling still isn't a large burned area. The region also shows a small likelihood of no change in BA (outer blue dot). The right, which shows the increase in the likelihood of a local (i.e. within grid cell) 1-in-100 BA event, doesn't show anything unusual for that region compared to the rest of Canada.

We have adapted the legend to clarify that the range on the colour bar extends between a halving and a doubling in BA extent.

Dot and change scales not linear? Arbitrary?

Using non-linear colour bars with human-readable levels is a typical way of plotting non-normally distributed fire metrics, and is routinely used for mapping burned area, fire counts and other fire metrics. Examples of this approach are numerous but include examples:

Hantson, S., Arneth, A., Harrison, S. P., Kelley, D. I., Prentice, I. C., Rabin, S. S., Archibald, S., Mouillot, F., Arnold, S. R., Artaxo, P., Bachelet, D., Ciais, P., Forrest, M., Friedlingstein, P., Hickler, T., Kaplan, J. O., Kloster, S., Knorr, W., Lasslop, G., Li, F., Mangeon, S., Melton, J. R., Meyn, A., Sitch, S., Spessa, A., van der Werf, G. R., Voulgarakis, A., and Yue, C.: The status and challenge of global fire modelling, Biogeosciences, 13, 3359–3375, https://doi.org/10.5194/bg-13-3359-2016, 2016.

Rabin, S. S., Melton, J. R., Lasslop, G., Bachelet, D., Forrest, M., Hantson, S., Kaplan, J. O., Li, F., Mangeon, S., Ward, D. S., Yue, C., Arora, V. K., Hickler, T., Kloster, S., Knorr, W., Nieradzik, L., Spessa, A., Folberth, G. A., Sheehan, T., Voulgarakis, A., Kelley, D. I., Prentice, I. C., Sitch, S., Harrison, S., and Arneth, A.: The Fire Modeling Intercomparison Project (FireMIP), phase 1: experimental and analytical protocols with detailed model descriptions, Geosci. Model Dev., 10, 1175–1197, https://doi.org/10.5194/gmd-10-1175-2017, 2017.

Hantson S, Kelley DI, Arneth A, Harrison SP, Archibald S, Bachelet D, Forrest M, Hickler T, Lasslop G, Li F, Mangeon S. Quantitative assessment of fire and vegetation properties in simulations with fire-enabled vegetation models from the Fire Model Intercomparison Project. Geoscientific Model Development. 2020 Jul 17;13(7):3299-318.

Kelley, D. I., Harrison, S. P., and Prentice, I. C.: Improved simulation of fire–vegetation interactions in the Land surface Processes and eXchanges dynamic global vegetation model (LPX-Mv1), Geosci. Model Dev., 7, 24goo11–2433, https://doi.org/10.5194/gmd-7-2411-2014, 2014.

Adzhar, R., Kelley, D. I., Dong, N., George, C., Torello Raventos, M., Veenendaal, E., Feldpausch, T. R., Phillips, O. L., Lewis, S. L., Sonké, B., Taedoumg, H., Schwantes Marimon, B., Domingues, T., Arroyo, L., Djagbletey, G., Saiz, G., and Gerard, F.: MODIS Vegetation Continuous Fields tree cover needs calibrating in tropical savannas, Biogeosciences, 19, 1377–1394, https://doi.org/10.5194/bg-19-1377-2022, 2022.

Mathison, C., Burke, E., Hartley, A. J., Kelley, D. I., Burton, C., Robertson, E., Gedney, N., Williams, K., Wiltshire, A., Ellis, R. J., Sellar, A. A., and Jones, C. D.: Description and evaluation of the JULES-ES set-up for ISIMIP2b, Geosci. Model Dev., 16, 4249–4264, https://doi.org/10.5194/gmd-16-4249-2023, 2023.

Line 2844, "some areas" … see increases … "almost everywhere": No sense in that statement.

We have rephrashed this sentence so that it reads "While some areas see increases in BA and fire extremes in all scenarios, the greatest rates of change are projected in Southern Alberta and Saskatchewan and under SSP585."

Authors expect users to accept that areas of low BA today will become areas of preferred BA in the future? Authors or ConFire or both haven't the faintest idea?

The 1-in-100 definition used by UNEP pertains to local events, representing the burned area that would occur once every 100 years in a specific location given today's climate and land surface conditions. UNEP utilizes this metric recognizing that increased fire activity is relative to the local

infrastructure, management, and ecosystems' adaptation. This metric has been extensively used since UNEP released it.

A substantial increase in these events does not equate to a significant increase in the percentage of burned area, although it is often (but not always) correlated with a large increase in the relative burned area extent. The regions along the southern border exhibit a high likelihood of a doubling in burned area (indicated by colours associated with '2+'). However, since this region currently has a low burned area, the percentage of burned area change is actually quite small. We have made this more obvious in the figure by changing the units in the legend to read 'half' rather than '½' and 'double' rather than '2'.

The increasing trend in burned area in these regions aligns with trend analyses conducted earlier in the report, as well as UNEP et al.'s (2022) future projections and recent independently derived projections by Haas et al. (2024).

We have made this clearer in the text, adding "We also use the 1-in-100 definition at a grid cell level to determine spatial variations in the change in extreme fire for each region" to the methods for this section.

Olivia Haas, Iain Prentice, Sandy Harrison et al. Global wildfires on a changing planet, 21 May 2024, PREPRINT (Version 1) available at Research Square [https://doi.org/10.21203/rs.3.rs-4359943/v1]

Lines 2862, "from 1.3% to 0.67-1.78%": relevant to Greece, authors expect users to accept or understand significance of such a negligible change?

We have added the following sentence to make it clearer to the reader that the difference we are seeing here is when we switch from using reanalysis to model data. This is important context for the future projections, to show how reliable the models are for the current period compared to reanalysis before using them for future projections.

"Here, using GCMs instead of reanalysis changed the likelihood of Greece's August 2023 event from 1.3% to 0.67-1.78% for 2010-2020 (Table 7; Figure 17), indicating that our models reproduce the likelihood given by reanalysis for the current period, although with a larger uncertainty range."

Later (line 2872) authors report "no significant difference between the scenarios by 2100. Why, then, have they wasted user time and their own page budget?

We have clarified now in the text that there is no significant difference between SSP370 and SSP585 - it is still the case that there is a large difference between these two scenarios and SSP126 which is an important point showing that mitigation would be effective for reducing extreme BA. The second important point that we make within this sentence is that while there is no significant difference between the scenarios, both show an increase in extreme BA by 2100, so without strong climate mitigation action we can expect extreme fires to be much more likely in the future. We disagree that giving future projections of extreme BA under different SSP scenarios is a waste of user time and page budget; we in fact suggest that this is a really important piece of analysis that is extremely relevant to policy making. Hopefully the clarification in the text helps elucidate this for the reviewer:

"There is no significant difference between SSP370 and SSP585 by 2100, which both indicate a likelihood of seeing an annual recurrence of 2023's event of between 2.5-3.3% by 2100 – a 1.83-to-4.28-fold increase today and an average return time of every 39-31 years."

Line 2889, legend for Fig 24: Same questions as for Fig 23, but why? Speculation? Waste of valuable space?

See our response above for Figure 23.

Line 2934, "suggests": authors can do no better than 'suggest'? Reader conclusion: nothing significant, why waste my time?

The science results we present provide solid evidence required for policy makers to make decisions around mitigation options, however our role is not to prescribe policy solutions. We feel that the meaning of the sentence here has been misconstrued; we are not saying the results are suggestions, nor are we saying they are insignificant. What we are saying is that the results "suggests that strong mitigation measures may offset the risk of extreme burned areas in the region in the future".

Line 2941, Summary: Useful organization here: paragraph responses to each original objective. But these paragraphs read mostly as redundant restatements of previously-discussed results? Reader does not need both? This section seems better organized and more concise?

Having removed "highlights" sections, this section is now less repetitive.

Line 2993: here, reader confronts mention of "Alexandroupolis" fire, A few lines earlier, authors mentioned "Evros" fire (line 2972). Two names for same event?

We checked all uses of "Evros" and "Alexandroupolis" and corrected usage to accurately reflect the following: The Evros fire started NE of Alexandroupolis in the prefecture of Evros, Macedonia and Thrace Greece.

Lone 3012, "release break": New term for readers? Not used previously?

Rephrased for clarity.

Line 3044, "socioeconomic factors outweigh the increase from climate change", many readers will agree with this statement, even sans hard evidence, but preceding paragraph seems to express contradictory opinions: climate change increases probability of large BA while socioeconomic inhibit? Decide what you want reader to take from this paragraph?

The statements are in agreement: socioeconomic factors inhibit and outweigh the increase from climate change. We have written "in all three areas, socioeconomic factors have reduced BA, dampening the effects of climate change that would otherwise have been experienced. When the impact of all forcings is assessed, we find there is …an overall reduction in mean BA in North West and East North America, signalling that socioeconomic factors outweigh the increase from climate change in these areas."

Lines 3055 to 3057, "At extended lead times (greater than 2 months), … no discernible signal about moderate or anomalous conditions is identified" Very worthy and valid conclusion, but negates much or what authors worked to present? Earlier in this paragraph, more vacillating speculation about ENSO or IOD, etc?

The plot in Figure 21 [now figure 16] shows a typical example of the predictability span expected in seasonal forecasts, with relaxation to climatology occurring within the first few months. The consistent message across the report is that for FWI forecasts, anything beyond 10 days is subject to the general predictability of the atmosphere. While in some places and under certain

circumstances, forecasts can extend to a seasonal timescale, they are generally limited to 1-2 months, especially for extreme anomalies.

We do not speculate about the ENSO prediction. The predictions are available through the C3S multi-model system, and we only report an analysis of their simulations to provide an outlook for the coming season. For example, Figure 20 [now removed] was from the C3S portfolio of products and credited to them.

Lines 3060 & following paragraph: Cautious but confusing? From final sentences, this reader concludes: even with best current mitigation options, warmth coupled with drought (interspersed with floods) will increase fire vulnerability? I truly worry that better weather, soil, vegetation data in Canada influence all these Incomparisons!

Apologies, we are unsure what the reviewer is asking for here.

Line 3076, Frontiers: I and most readers will welcome and agree with these concerns. Definitions, observations, predictions, attributions, future projections. Too long in every case but see recommendations above. Very valid helpful cautious summary, ideal coming from this group? Mention SI?Also a place to encourage other contributions to proposed or other future special issues?

Addressed through restructuring described above.

Line 3393: Frontiers. Good section! Here, particularly, you need to focus on research audience or fire management audience? Way too long! Could reduce by 30% with no loss of information (see above),

We have shortened this section and moved it to Appendix B, where it is now less intrusive.

Line 3430, "extensive tree mortality": First (and sole?) hint at infestations. For much of northern Rockies, as for many forests in Germany, climate-enhanced (activity, reproduction) insect infestations kill 1000's of trees. At very least, this large-scale die-off moves vegetation from 'live' to 'dead' categories, a distinction invoked once or twice but not much discussed here? Impact seems to involve climate, weather, vegetation and human controls? Where would this fit in existing modeling? Or in future needs? Insect-mediated factors could grow?

Indeed, this is an important challenge. In fact, this comes under a broader challenge of quantifying and characterising fuel loads at scale.

We added various sentences to section 4.2.3 to better describe this issue and prepare the reader better for the mention of tree mortality on line 3430:

*"Likewise, the response of fuel moisture to meteorological factors can be influenced by external factors that are challenging to observe and model at scale, such as mortality triggered by insect infestation or disease (Canelles et al., 2021)."*

*"There are also likely to be some stubborn issues with the detail provided large-scale observational data available to predictive systems, particularly in the case of fuel loads and fuel state (e.g. living versus dead). New global biomass observations, such as those from airborne and spaceborne Light Detection and Ranging (Lidar) and Synthetic Aperture Radar (SAR), provide insights into fuel loading but they are not currently providing information regarding fuel state that would be useful for modelling fuel moisture response to meteorological conditions (Santoro et al., 2022; Hunke et al., 2023)."*

Line 3460, data about fires and fire impacts: in an earlier ESSD paper, Karen Short of Montana Forest Service reported difficulties in rectifying versions of fire data into one accurate valid record. As I remember, nearly 40% of fire reports in her US-based database proved invalid or redundant. Emphasizes dependence on remote sensing data? Strength or weakness? Also emphasizes that we currently cannot, at least not in near-real time, furnish local experts with accurate local data? Causes us to rely on media, which have even worse unstable biases? Also raises issues of industrial (including transport) fires. Not currently monitored or logged? Often, in media accounts, prove sensational. Notable impacts on emissions, air quality, evacuations, human safety, etc? Another aspect of fire currently out-of-bounds for fire researchers?

The reviewer is absolutely correct in pointing out that Short (2011) found numerous redundancies in the federal and state-level ignition databases that she synthesised; these errors have long been known and published about in the US, as these data were never meant for scientific use (the reporting occurs to track finances and personnel committed to fires), and which stem primarily from duplicate reporting across different state and federal agencies. The level of accuracy was particularly poor prior to the digital age, but has improved substantially since the implementation of digital national reporting databases in the last two decades. This is why so many fire researchers have developed and relied upon remotely-sensed fire databases derived from MODIS, Landsat, and other remote sensing resources, as these were developed by scientists for science and are more consistent globally. Given the digital age advances we are seeing, we don't suggest this aspect of fire science is out-of-bounds for researchers at all, rather, researchers are critical to its advancement.

This is exactly why this paragraph and the one prior (lines 3435 - 3469) highlight the need for systematic data collection of fire impacts beyond area burned. Air quality and emissions impacts are perhaps the easiest data to verify and collect globally thanks to the development of low cost sensor networks and improvement of remote sensing detection, but for many other types of impacts, methods for measuring and systematically cataloguing such impacts are still nascent. The US state of California Fire agency (CALFIRE) now systematically quantifies structure losses on every wildfire, but the California system has not been adopted across other states. Canada has developed an evacuation database that was just published (https://www.publish.csiro.au/wf/WF23097), but such an approach has not been adopted elsewhere. We do not suggest a reliance on the media, which is variable in accuracy and can indeed sensationalise, but instead suggest that governments have a critical role in recording impacts systematically in a way that databases like EM-DAT are unable to do. Additionally, we might note that one current barrier to creating scientifically vetted systematic databases is that funding agencies in several countries do not generally support database development, so part of the reason for our inclusion of this discussion is to highlight the need for funding that would allow scientists to develop these data, potentially in collaboration with governments, to drive more accurate understanding and modelling of wildfire impacts. We thank the reviewer for highlighting why this paragraph on data development is so important to include in our report.

Line 3490, air quality impacts: ground-based measurements, dispersion and transport studies, plume dynamics, animal-based human exposure/response models? Add intermittent ill-defined fire emissions? Almost an impossible problem?

We added the following detail to highlight the issues raised by the comment:

"Issues contributing to the challenge of quantifying the impact of fire pollution on human health are the same as those for other pollution sources, including, but not limited to, a lack of ground-based measurements in many regions of the world, a need for more pollution dispersion and transport studies, a deeper understanding of plume dynamics and chemistry, and a partial

reliance on animal-based human exposure/response models (e.g., Fioreet al. 2012, Fuzzi et al. 2015)."

Line 3526, authors here tend to focus on fire management, by federal or state agencies vs IP&LC communities. Good points. But, Canadian projections (e.g. from Fig 23) pass through - on both sides of Canadian-US border) - large reservations. Smoke air quality exposures might prove adverse in those communities and locations?

With apologies, we were not able to grasp what is being suggested in this comment.

Line 3564 & following paragraphs, economic impacts: omission of or ignorance of industrial fires might prove key here? Economic impact estimates miss many longterm health costs already mentioned?

With apologies, we were not able to grasp what is being suggested in this comment. This report and the referenced work have focussed on fires on the open landscape, not industrial fires.

The sentence "While tangible costs (e.g., insured losses such as property or infrastructure) are relatively straightforward to measure, intangible costs (e.g., lives lost, shortened lifespans and impaired performance/productivity due to smoke exposure, damage to species and habitats) are more challenging to quantify owing to issues of data availability and their occurrence over varying temporal and spatial (geopolitical) scales." already states that intangible costs stemming from long term health effects are difficult to quantify.

Line 3740 & following paragraphs: Back to observations, predictions, attributions, future projections. Good stuff but serious redundancies with Frontiers Section. Too long but authors need to think here about readers: what new recommendations do you want readers to remember from this section that we did not already learn from Frontiers section?

We have we re-written large parts of this section to better emphasise its objective: to describe how *this report* will play a role in promoting advances in the science and overcoming challenges that were identified in the Frontiers sections.

Lines 4982, 4983: URLs (e.g. https//doi.org/10.xxxx/longer number that wrap across lines don't work as hot links. Perhaps now Need a table of contents? Or, something different or better to help readers find specific information. Perhaps accompanied by short paragraph explaining what you keep in main manuscript and what additional info users can find in supplement. Supplement would also need a table of contents or equivalent? Document would benefit from list of acronyms? Developing such a list might help authors check & track such a long list of acronyms?

URLs: this was not an issue in our local PDF. We trust this will be resolved in the final published MS file.

We will enquire with the publisher about adding a table of contents. We did not add one in our re-submitted version because page numbers will differ between this version and the published typeset article.

To the best of our knowledge, we have now defined all acronyms on first use.

**For future discussions and planning**

Thank you, we have noted the following comments and will consider them next year.

Move everything not global nor related to specific focal events to supplement?

Authors have, in this version, 'teased' readers with three specially-chosen focus events. Because each of those focus events includes long discussions of present predictability and future probability, won't readers want, in second version, some update on authors' conclusions for these regions? Meanwhile, authors will select a different three regions for 2024-25? Eventually, doesn't one end up with oldest, older, prior and present events? Background, global, regional, predictions: different tools and different conclusions for each. Should these evolve into separate products? Mention here ideas about a special issue? Perhaps parse under several journals, according to?

**RC3 Response**

Dear Authors, Firstly, I would like to extend my warm congratulations on your comprehensive assessment of the state of extreme wildfires for 2023-24. This manuscript represents a significant and valuable contribution to the understanding of wildfires, and I believe it holds great potential for impactful publication after addressing a few minor revisions and additions.

Thank you for this helpful review and for taking the time to study our work. We are grateful for the many constructive points raised and we have worked hard to address them.

Comments:

- I understand that this is the inaugural assessment, necessitating the inclusion of many details. However, the paper is quite lengthy, spanning over 100 pages (and more than 50 pages of supplementary material), which makes it challenging to read. Some sections could benefit from greater conciseness as some repetitions and details could be referenced rather than fully detailed within the paper. To improve readability and impact, I suggest the following:
    - Streamline sections to remove repetitive content.
    - Condense detailed explanations where possible by referencing relevant literature.
    - Focus on making the key findings and methodologies more concise and directly accessible.
    - The conclusion section is currently too long and reads more like an incomplete review rather than a focused conclusion of this study. I suggest strongly reducing it and concentrating on summarizing the key results and findings of this report. This will provide a clear and concise conclusion that highlights the main contributions and implications of your study.

    We have dramatically re-structured the report to achieve all of the important points. Our actions included:

    (1) Moving the "Year in Review by Continent" section to "Appendix A: Year in Review by Continent". In the appendix, this content interferes less with the flow of the manuscript, an issue raised by two reviewers.

    (2) Moving the elements of review ("Frontiers…" and "Roadmap") to "Appendix B: Defining A Roadmap for the State of Wildfires Report". In the appendix, this content interferes less with the flow of the manuscript, an issue raised by two reviewers.

    (3) Cutting the word count of the "Frontiers … " and "Roadmap …" sections substantially (these now appear in "Appendix B: Defining A Roadmap for the State of Wildfires Report").

(4) Presenting the methods and results for each theme of the report in sequence (i.e.: Observations-Methods; Observations-Results; Drivers-Methods; Drivers-Results; Attribution-Methods; Attribution-Results; Outlook-Methods; Outlook-Results. This was suggested by at least one reviewer. This streamlines the text by improving the proximity of methods and results material through, thus circumventing the need to scatter reminders about the methodology to the reader throughout the text.

(5) Removing the original "highlights" from each individual sub-section of the report and instead placing this material in the streamlined Conclusions section.

Other less substantial actions, such as removing figures and shortening sections of text throughout the manuscript, also led to a reduction in length.

Please note:

- Due to the volume of changes made, particularly through re-structuring of the manuscript and to significant word-cutting, it became impractical to use track changes. In addition, the re-structuring is so substantial that our attempts to 'compare' versions (to highlight changes) has failed.
- Due to final editing/re-wording before submission, some of the quotes provided in the responses below may be out of sync with the submitted document.
- We sincerely apologise for any inconvenience this causes in the processes of reviewing the refinements to the report.

● For future updates, it could be interesting to include regional data where available that can cover longer periods compared to satellite data. This would provide a more comprehensive historical context and enhance the analysis.

Thank you. We added the following to our review of "Frontiers in Observing and Modelling Extreme Fire Occurrence > Observing Extreme Fire Events" and noted the utility of new datasets that are being compiled globally and offer insights into extremes on multi-decadal timescales.

"A natural starting point for this global assessment of the 2023-24 fire season was global data provided by the MODIS BA dataset, though we note that various national, state-level or regional systems exist and can add longer-term context to the extremity of fire seasons (e.g. Canadell et al., 2021; Short, 2014; Gincheva et al., 2024). Regional datasets generally depend on manual logging of fires via field approaches or desk-based identification with high-resolution imagery, or alternatively harness different blends of satellite observation with fire detection algorithms that can be regionally optimised. These approaches carry their own uncertainties and are limited by design to targeted regions, however their major advantage is multi-decadal coverage. Advances in compiling

regional datasets into gridded records with global coverage are bringing these advantages to formats compatible with global scale analysis (Gincheva et al. 2024) and will thus be explored in future efforts to characterise regional extremes at scale."

And later:

"In addition, regional products often provide scope to characterise the extremity of events over multi-decadal timescales and are now being provided in globally harmonised formats compatible with global analyses such as ours. These regional dataset should be utilised in future iterations of the State of Wildfires report."

- Existing approaches are not considered in this inaugural report. For instance, hybrid models for seasonal fire risk, fire propagation models, or new fire susceptibility/risk mapping tools are not included. Adding a discussion on these approaches could provide valuable insights and indicate potential directions for future reports.

  This is a very fair point, and we are fully aware that while the State of Wildfire report is a fantastic idea, it is far from exhaustive. Aspects such as impacts on air quality, general global trends of fire danger, and shifts in fire regimes are also important considerations that are not addressed here but will need to be included in the following edition as we broaden the scope of the report. In the appendix where we discuss the roadmap of the report, we have expanded on some of the suggested recommendations into different areas.

  *The current report does not include hybrid models for seasonal fire risk, fire propagation models, or fire susceptibility/risk mapping tools. Incorporating these approaches could offer valuable insights and will b e considered in future reports. These advanced models and tools, which account for both past and present weather conditions as well as other critical factors such as soil moisture and vegetation dryness and occurred fires, can enhance our understanding of fire dynamics and improve predictive capabilities. By exploring these methods, future editions of the report could provide a more comprehensive overview of fire risk assessment and management strategies.*

- In future versions, it might be beneficial to include a summary of the main scientific initiatives currently in action (e.g., projects, consortia, collaborating groups). Additionally, consider adding comments on ongoing work in the main scientific agenda, including how these efforts align with the Sustainable Development Goals (SDGs).

  Thank you, this is an interesting idea. Indeed, this would enable readers from "outside science" to more quickly identify points of contact which would be beneficial for impact. Thinking towards future years, we will begin to compile a list and provide it as an additional supplement to the paper.

- In future versions, more attention should be given to the Wildland-Urban Interface (WUI), human exposure, and the broader impacts of fires, such as on air quality. These aspects are crucial for a comprehensive understanding of wildfire risks and their societal impacts. Minor

We completely agree. Most (all?) of these points are included in the "Frontiers" section, though perhaps not as enthusiastically as they should be.

Comments:

L100-101: The authors state in the abstract that global wildfires are increasing in frequency and intensity due to climate change. This statement may oversimplify the complexity of wildfire trends and drivers. It would be beneficial to provide a more nuanced discussion or to soften the statement to reflect the multifaceted influences on wildfire activity. This will ensure the abstract accurately represents the detailed analysis provided in the paper. In addition, a very recently published paper assesses trends in extreme fires (Cunningham et al., 2024). I think this is an important reference to add to this paper, and their methodology may be applied to future versions of this report.

We softened the language to "Climate change contributes to… ".

We now added a citation to Cunningham in several places, including in the very first sentence of the introduction.

L124: The values like "2.22" should be rounded to an integer or at least to one decimal place. The associated uncertainty is likely significant, and using two decimal places may artificially increase the perceived confidence in these future estimates.

Now rounded appropriately.

L150: You can reference IPCC AR6.

Thanks, good point, we added the reference to Seneviratne et al. (2021)

L175: Reference Zheng et al., 2021 Science Adv.?

Added citation.

L180: Any reference for this statement?

Not necessary here in our opinion - we have built the argument for this throughout the paragraph.

L196-213, L215-233: While the information is accurate, it would be beneficial to break these paragraphs into smaller, more focused sections that group related information together. This will enhance readability and ensure a more logical flow of ideas.

L196-213: Thanks for highlighting the issues with readability. We re-organised to improve flow/clarity.

L215-233: We felt that the original text was already sufficiently focussed on the point of the paragraph, i.e. that "Mitigating and adapting to increases in wildfire potential are growing priorities of policymakers and require coordination with many other stakeholders". Each sentence of the paragraph addresses this point from the perspective of an identified stakeholder.

L255-256: I agree, and for this reason, I stress the importance of being as concise as possible to enhance readability and ensure the key messages are communicated.

Thanks, we bear this helpful feedback in mind.

L258-266: Why is this paragraph here? This can be discussed later. In any case, I agree that defining extreme fires is challenging. Your title indicates "wildfires," but you consider all fires in the BA statistics, including agricultural fires. A possibility is to use "fires" instead of "wildfires" in the title and reduce the discussion on the definition of "extreme fires" here and

later in the paper.

We moved the paragraph to the methods section 1.2.

While the MODIS BA may include some agricultural fires, this is certainly not its specialty (hence the reason why some datasets such as the GFED BA dataset have to retrospectively add active fire to capture signals of agriculture/small fire burning - van der Werf et al., 2017, Chen et al., 2024). Nonetheless, your point stands that some fires in the dataset are not wildfires. On balance, we feel that it is obvious enough from the materials presented that we are interested in the extreme end of the fire "spectrum", which are typically considered wildfires. It would be rare for fires with large size to be anything other than a wildfire. Likewise, it would be rare for periods with regionally exceptional fire activity to be driven by agricultural/land management fires alone.

Indeed, we note that "variability and trends in regional BA totals using datasets that include small fires do not differ significantly from the variability and trends present in the MODIS MCD64A1 BA data (Chen et al., 2023)" which indicates that regional-scale variability is predominantly sourced from the BA records associated with larger fires, rather than smaller fires. Even in Amazonia, a region where the fire regime is heavily driven on people and land use, the largest fires tend to be related to wildfires rather than deforestation fires or other fires associated with land use (Andela et al., 2022). Combining these reasons with the relative ease of communicating the term "wildfire" to wider public audiences, we were not motivated to alter the title.

Although the discussion on the definition of "extreme fires" might seem somewhat awkward and interruptive, we think it is nonetheless an important aspect to raise as a point of discussion to fellow researchers in fire science. We know that consensus on terminology is an issue in many aspects of this field, and we think this paper takes a pragmatic approach: we open up the discussion of what should be considered "extreme" and avoid introducing new definitions that might be subject to debate (for now). To summarise: we believe that most researchers would agree that studying the extremes of the fire spectrum is important, but there is fuzziness in where the line between "normal" and "extreme" should be drawn. (Especially when one considers the need to avoid an over-reliance on what is observable by satellites and specific catch-all metrics such as BA that may not scale as strongly to impact as other metrics of fire behaviour).

L270: I suggest moving before "This inaugural edition […]".

Apologies, but we did not understand this suggestion as it is not clear what should be moved.

L275: "March 2023 to February 2024"? I suggest indicating briefly in the introduction why this period.

This information is now given very close to the top of methods section 2.1 as there wasn't a natural place to introduce it in the introduction.

L284: Which fire season? Please be specific.

Added detail "(commencing March 2024)"

L290: I suggest including a broader range of sources, not just publications.

We clarified that this as we intended it to read: "between the annual iterations of the report"

L292-315: This can be merged within the Method section.

This section was re-written to act as intended: a very brief summary of the methods to help prepare the reader for what comes later in the methods section.

L297: The term "fire risk" is used here. Please ensure consistency and be mindful of the distinction between "risk" and "danger" in natural hazards studies. It is important to use these terms accurately to avoid confusion.

Thanks for pointing this out; this was a misuse of the word "risk" in this context. When we talk about metrics of fire danger, we never use the term "fire risk." However, the term "fire risk" might come up in the discussion when we really mean risk in terms of combining hazard with exposure.

L298: All acronyms should be expanded at their first appearance or included in an auxiliary table for reference. This will help ensure clarity for all readers.

We have taken care to do define all acronyms on first use.

L324: You consider BA values since 2001 while the combined Terra and Aqua datasets started in 2003. Could the trend be affected by this inhomogeneity? Please address this potential issue.

We checked for structural bias towards the years 2001 and 2002 appearing as the minima across the BA record of regions included in this work.

Indeed, we found a slight (+20%) bias versus other years for the year 2001 (but not 2002) to be the minima in the regional records.

In response, we have corrected our results to disregard the year 2001 from the analysis. Figures and in-text values have been updated accordingly. Reassuringly, this did not have a major impact on observed anomalies in BA and did not alter the conclusions of this work. But we are glad to have checked.

L382-283: Not clear, please reformulate.

This sentence was dropped as it effectively replicates the sentence prior.

L396-405: I suggest explaining the target season shortly in the introduction.

Now added this at the top of the methods section.

Table 1: For the IPCC regions, you can reference: Iturbide M. et al. (2020). An update of IPCC climate reference regions for subcontinental analysis of climate model data: definition and aggregated datasets. Earth System Science Data, 12(4), 2959-2970.

Excellent, thanks. Corrected.

L429: "(2003 for C emissions)" applies also for points "I" and "II"?

Yes, corrected.

L430: "and;"?

Corrected.

L442-444: Not clear.

Removed for brevity as this is not central to the current sub-section and is explained elsewhere.

L451: Always using the March-February years?

Clarified: "but limited to the March 2023-February 2024 period"

L486: For future updates, I suggest correcting the p-values of the Mann-Kendall test for multiple testing using a false discovery rate test, as described in Ventura et al. (2004). "Controlling the proportion of falsely rejected hypotheses when conducting multiple tests with climatological data." Journal of Climate, 17, 4343– 4356.

Thanks, that's interesting and has been noted.

Table 2: There are some typos (e.g., "Ang'ila"), inconsistencies (such as country names being abbreviated or not), and repetition (the Africa region). Please correct the typos, ensure consistency in the formatting of country names, and eliminate the repetition in the Africa region.

Thanks -

Corrected the country names.

Corrected the repetition.

Ang'ila is not a typo (it's a surname).

L607: I think the comparison should be made fairer by using observed Fire Weather Index (FWI) data. This will ensure a more accurate assessment.

Very good point. We have both, reanalysis data from ERA5 Land, which serves as our proxy for observations, and fire activity data. The comparison with ERA5 Land assesses the predictability of the FWI itself in terms of how well weather forecasts can reproduce the 'observed' FWI. However, since these metrics are used to predict fire activity, we also need to provide an indication of how they correlate with observed fires. We understand that this might not be clear to a reader and an explanation has been added:

*Specifically, the best estimates of the Fire Weather Index (FWI) are derived from reanalysis products, particularly ERA5-Land (Muñoz-Sabater et al., 2021), which serves as a proxy for observed FWI. Forecasts at different lead times are taken from the operational high-resolution ECMWF weather system, and seasonal predictions are sourced from ECMWF's long-range forecasting system, SEAS5 (Johnson et al., 2019; Di Giuseppe et al., 2020; Di Giuseppe et al., 2024). A comparison between reanalysis and forecast provides an indication of how weather forecast errors translate into FWI uncertainties (predictability). Additionally, the predictions are compared to recorded peaks in fire activity, both in terms of burned areas and active fires as observed by the MODIS satellites, to provide a qualitative assessment (skill) of the correlations between landscape flammability and actual fire events.*

L612: I believe the seasonal skill is limited to less than 2-3 months in most regions. Please verify this information and adjust accordingly.

Yes indeed from Di Giuseppe et al 2024 prediction is mostly 1 month and only in some regions

Di Giuseppe, F., Vitolo, C., Barnard, C. et al. Global seasonal prediction of fire danger. Sci Data 11, 128 (2024). https://doi.org/10.1038/s41597-024-02948-3

This has now been amended to 1-2 months.

L614: "Predictability" or skill?

Corrected to "skill"

L618-621: Not clear. As I previously indicated, observed FWI should be the target for the verification.

As explained in the previous point, we have provided an assessment against observed FWI using ERA5 Land.

However, the goal is not just to be accurate in our forecast of FWI  but also to provide  useful information. If we can predict FWI excellently but it does not correlate with fire activity, then it becomes a futile exercise.

L632: "Shapley values"?

Yes, from the Nobel Prize winner Lloyd Shapley. They translate the contribution of a driver to the final predictions.

We have a detailed explanation in the supplementary material with references which reads:

*The relative contribution of each input control to the model prediction is evaluated using Shapley values, computed using the Shapley Additive exPlanations python library [Lundberg & Lee, 2017]. The SHAP value indicates the importance of each feature in a model, where a positive SHAP value reflects a positive impact on the model prediction and a negative SHAP value reflects a negative impact. Specifically for this study we use the TreeExplainer, which computes the SHAP values by interrogating the structure of the decision trees within the model based on the input feature values. The probability controls are then normalised and grouped into the four categories given in **Table 3** of the main text. By combining these with the total amount of fires predicted for a given area we can attribute those fires into one of the four controls. The 'Other' control also includes fire occurrences not predicted by the model. This is computed given by:*

$$Other = SHAP[Other] + max\left(0, Area\_Total\_Fires\_Observed - Area\_Total\_Fires\_Predicted\right)$$

*(2)*

*Where, SHAP[Other], is the contribution of the 'Other' control to the total predicted fires for a given region and,* `Area_Total_Fires_Observed` *and* `Area_Total_Fires_Predicted` *are the total number of observed and predicted fires for the same region.*

L644: "and;"?

Corrected

L658-659: Not clear.

We could examine every single variable; for example, among the control fuel moisture, we could analyse the individual contributions of live vegetation versus dead vegetation. However, we decided to analyse the impact of controls rather than single drivers. We have now expanded on this approach.

*Both ConFire and PoF are capable of disentangling the contributions of individual drivers within the same control category (for example, the separate contributions of dead or live vegetation) and quantifying these contributions (Kelley et al., 2019; McNorton et al., 2024). However, we will focus our analysis on the impact of the controls.*

L755: Please update the references considering the latest one: Forster et al. (2024). Indicators of Global Climate Change 2023: annual update of key indicators of the state of the climate system and human influence. Earth System Science Data, 16(6), 2625-2658. Also, please clearly indicate the quantity and reference for the temperature increase. For instance, Forster et al. (2024) wrote "The indicators show that, for the 2014–2023 decade average, observed warming was 1.19 [1.06 to 1.30] °C, of which 1.19 [1.0 to 1.4] °C was human-induced. For the single-year average, human-induced warming reached 1.31 [1.1 to 1.7] °C in 2023 relative to 1850–1900."

Thanks, corrected the referenced and stated "between 1.1 and 1.3°C" for a catch-all summary.

L805: Which models?

We have clarified this now in the text "which most other fire models embedded within land surface models are unable to capture well"

L827: There appears to be an issue with the numbering. I suggest referencing the first supplementary figures first to ensure a logical sequence.

We have removed this reference to the Supplementary figures here, so that they are first referenced in the following paragraph.

L828: I suggest replacing "correct" with "adjust".

Thank you, we have updated this in the text for this and following instances.

L854: I suggest explaining here why two thresholds.

If the reviewer will allow, we feel that adding the explanation of the two thresholds into the brackets at line 854 would make the sentence very long and difficult to follow, whereas keeping it in the next sentence makes it clearer.

L870: Please ensure proper formatting for citations, correcting any issues with parentheses.

Thank you, we have corrected this in the text.

L943: Burton et al. (2023) is a preprint, so I'm not sure if the methodology could be applied here without a proper examination and discussion. Is this approach mature enough to provide reliable information for this inaugural report and potentially for the next ones? I suggest a thorough review and discussion to ensure the methodology's robustness and suitability.

We agree that the Burton and Lampe et al (2023) work is novel, as it is the first time an ensemble of land surface models (via fireMIP) have been used for fire attribution, which is why we believe it is an important line of evidence to include in this State of Wildfires report. Although that work itself was still in pre-print at the time of submission, it has subsequently been accepted for publication in Nature Climate Change. We would also reassure the reviewer that the experimental design and uncertainty quantification presented in Burton and Lampe et al (2023) is very similar to that used in ConFire, documented and reviewed in this paper and in Kelley et al (2019; 2022 cited in the m/s). The attribution framework used in both cases is published in Mengel et al (2021), the model weighting is based on Knutti et al (2017), and the NME term used to benchmark the models is from Kelley et al (2013). So we believe the individual elements of the methodology have been reviewed and are established enough in the literature to include here.

We have made this clear in the supplement description of the fireMIP ensemble: "Based on model performance by AR6 region, a region-specific weighting is also applied following  Knutti et al (2017). The weighting is based on the model's distance to the observed BA temporal RA using both FireCCI5.1 and GFED5, measured using NME as per Kelley et al. (2013). To measure the uncertainty, random noise is generated and scaled by the climatological RMSE of each model. This noise is then added to the modelled relative anomaly, this process is repeated 1000 times. This performs the same function and the uncertainty quantification from model error as Equation 4 does for ConFire"

Knutti, R. et al. A climate model projection weighting scheme accounting for performance and interdependence. Geophys Res Lett 44, 1909–1918 (2017).

Kelley, D. I. et al. A comprehensive benchmarking system for evaluating global vegetation models. Biogeosciences 10, (2013).

L951: I suggest comparing the results also against a standard no-weighted ensemble.

Apologies that this was unclear in our explanation, the unweighted models are indeed compared against the observations in the first instance in order to create the weighted ensemble. We believe this is what the reviewer means, rather than repeating the analysis with the unweighted model ensemble and comparing the results to the weighted model ensemble, which would serve little purpose and present a biased result. We have made this clearer in the text now: "A weighted ensemble of the monthly outputs of BA, based on the regional performance of the unweighted models against observational data from GFED5 and FireCCI, is used for the analysis."

L963: The term "decadal outlook" might be misleading as you are considering decadal projections and not decadal predictions. Please adjust the terminology accordingly to avoid confusion.

Yes, agreed. We replaced with "multi-decadal outlook"

L967: I suggest removing the ENSO description. An appropriate reference should suffice. L967-996: The description is not clear. Is this the application of an already verified method or a new method? Additionally, please clarify how you establish significant teleconnections.

Yes we agree and the section could have been written more clearly, and now it has been completely revised. We have removed anything that could be referenced and rather make an example of how ENSO forecast is used in some countries to implement fire management practices. The section reads now as follows:

*'Among the modes of variability in the climate system most relevant to wildfire activity globally is the El Niño-Southern Oscillation (ENSO) (Chen et al., 2017; Mariani et al., 2016; Fuller and Murphy, 2006; Cardil et al., 2023). Numerous studies have demonstrated that there is a predictable cascade of fire across tropical continents during ENSO events, highlighting staggered responses of wildfire to ENSO. The utility of using ENSO as a predictor of fire is highlighted by its application in Indonesia, where severe fires during the 2015 El Nino led to hazardous haze in Singapore and diplomatic tensions in the region , prompting better regional cooperation and enforcement of anti-burning laws(CITATIONS). Consequently, Indonesia now implements preemptive bans on agricultural burning based solely on ENSO predictions, a measure that*

*proved successful in 2023 when no significant fire anomalies were recorded despite a strong positive ENSO event.*

*Another phenomenon demonstrably linked to global fire activity is the Indian Ocean Dipole (IOD), which occurs in the Indian Ocean. However, there is still ongoing debate regarding the direct influence of the IOD on Australian fires, for example as the signal is often obscured by changes in land management practices (Harris and Lucas, 2019). Other atmospheric modes of variability in the Southern, Northern hemisphere and in the arctic regions can also have strong influence on the seasonal trend of regional burned area and, Figure S1 shows the climate modes with strongest influence on burned areas globally.'*

L1063: It might be useful to include a brief introduction to prepare the reader for the following section. Additionally, I notice that for some continents you provide specific quantities, while for others you only indicate "above" or "below." I suggest maintaining consistency in this section by providing quantities or clear indications for all continents.

This "highlights" section no longer exists following re-organisation of the report.

L1076: Is it possible to provide an estimation of this contribution?

Great idea. Added "*(24%, up from a mean value of 3% from prior fire seasons)*"

L1082, L1087, L1340, L1679, etc.: It might be useful to include a reference to demonstrate that the main driver here is drought, as this will substantiate your assertion. If such evidence is not robust or widely agreed upon, consider softening the statement to reflect the complexity of the factors involved.

Statements involving the role of drought were removed from the section on data-driven identification of extreme events.

In the expert assessment section (now in Appendix A), we softened language used in reference to droughts or heatwaves where necessary, for example by mentioning when these events were based on reports or expert knowledge, rather than analysis.

L1085: I suggest using "BA" or "burned area" consistently throughout the text instead of terms like "fire extent," "burned areas" (at L1111), "Burnt Area" (at Figure 11), or similar, to avoid confusion.

[now figure 7]

We added a clarification "burned area (BA), synonymous with fire extent" in the methods section.

All remaining instances of "burnt" have now been removed.

L1096: How do you define "wildfire extent" here?

All instances of "wildfire extent" have been corrected to "fire extent"

L1132-1137: This paragraph is partly repetitive. Please revise it to eliminate redundancy and ensure the information is presented concisely.

Indeed! Thanks, this was removed and integrated into the methods.

L1162: Where can I see these results?

Added citation to the accompanying data set at: Jones et al. (2024)

Figure 1: To enhance readability, consider using a plot with two y-axes. This could help present the data and make it easier for readers to interpret the information.

Added a second axis.

Figure 5 (BA): Why since 2003 and not from 2001?

[now figure A1]

Perhaps this was mis-reading of "2023"? The figure caption says: "Summary of the 2023-2024 fire season in Greece."

Figure 7: The low quality of this figure makes it difficult to read the text inside it. Please improve the resolution to enhance readability.

[now figure A3]

Fixed in this version.

L1788: A more specific reference is needed.

Thanks, the citation's date was missing (2024).

L1843: I suggest expanding the names of the months, maintaining consistency throughout the paper.

Agreed, corrected, thanks.

L1915: Which variables are being referred to here? Did you perform any tests to validate these links? Please provide more details or specify the tests conducted.

Language was softened and some of the awkward phraseology of this paragraph was addressed.

L2188: Did you calculate this correlation?

The correlation between FWI and fire activity in the boreal forest is notably high. This correlation is provided in Di Giuseppe et al. (2016), and this citation has now been added.

L2193: I suggest avoiding references to "not-shown" results. If the results are important, consider including them in the main text or supplementary materials for transparency and completeness.

We have removed the sentence and also added a discussion acknowledging that there are other metrics and models that might perform better in some cases than the FWI. However, we decided to stick with the FWI as it is the most widely used and provides comparable results to other indices. The previous sentence has been removed, and we have added the following:

*The FWI is not the only index for fire danger, and other fire danger systems or sub-indices of this system may correlate more strongly with BA or fire behaviour metrics in some environments. Nonetheless, FWI is widely applied due to its good performance across a range of environments (Di Giuseppe, 2016; Jones et al., 2022) and so we adopt it in the current work.*

L2210-2221: If I understood correctly, the FWI aligns with peak fire counts. This observation may merit additional comments to highlight its significance and implications.

Yes we have expanded a bit but this is exactly what we have said before

*It provides valuable insights into the sequence and extent of extreme fire weather days during such events. Notably, in this region, peaks of fire activity correspond to peaks of landscape flammability, and there is a good correlation between observed fire activity and predicted fire danger.*

L2225: All South America?

Western Amazonia - now corrected here and in a few other places throughout the manuscript.

L2236: Is soil moisture derived from ERA5?

Yes is derived from ERA5-Land as specified in the table. All variables used to train the PoF and ConFire model are from ERA5-Land

L2239-2256: This paragraph may not be entirely necessary. Consider condensing it to make the text more concise and to the point.

Thanks. We shortened the text as follows.

*Analysing the time series of key drivers contextualises the conditions under which events occurred (see **Figure 8**). However, leveraging the PoF and ConFire models allows for a statistical causality attribution of the four controls for observed fire occurrence (see **Figure 9**). These models will provide control attribution even if no fire event is recorded, with a low probability across all controls indicating an accurate prediction. High fire probability without recorded fire activity could indicate successful suppression or fire-prevention policies. Unaccounted human influence is categorised under "other," encompassing variables and not forecasted by the models. This analysis enhances our understanding of fire activity controls and helps identify missing information that degradates the quality of the prediction.*

Figure 12: Do you think there is any added value in including relative humidity, which is a key ingredient of the FWI?

[now figure 8]

In Figure 12, we consider four variables: relative humidity at two levels and fuel moisture for both dead and live fuels. The dead and live fuel moisture are strongly correlated with 2 m relative humidity, which is the main reason they were not included in the main analysis.

L2340: "arae".

Corrected.

L2358: The sentence mentioning "fuel load" appears incomplete. Please review and complete the sentence to ensure clarity and coherence.

Apologies, the end of the sentence got cut when formatting the figure on the subsequent page. The sentence now reads: "While the analysis indicates a slightly higher than normal fuel load with only a low likelihood of having a substantial influence on increased levels of burning."

L2381: Can you clarify how you estimate the statistical significance here?

We have changed this to describe the likelihood that fuel load has either a positive or negative effect on burned area to make confident with the statistical reports in the rest of this section. I.e. "There is little confidence in the direction of the effect on BA anomaly, with potential influences ranging from a slight suppressive effect (26.5% likelihood) to potentially explaining the majority of the increased BA in September, to virtually no impact in October."

Figure 15: "anomoly".

[now figure 11]

Fixed

L2498: It is unclear whether this refers to the direct human influence on fires or the influence of human-caused climate change. Please clarify to ensure the distinction is clear.

Thank you, we have corrected this to match our definitions in Table 4, and have similarly corrected the other highlight statements in this section.

L2499: The term "total climate forcing" is unclear. Please provide a definition or rephrase for better understanding.

The terms are as defined in Table 4 of the Methods. We have also added "total climate forcing from climate change" into the text at line 2499 to make this clear again at this point of the paper.

L2501-2502: Not clear.

We agree that without additional context this sentence is unclear, and in order to keep the highlights brief we have removed this sentence and only refer to it later on in the results together will the full explanation.

L2505: "total".

Thank you, we have corrected this to "Total"

L2586: Why are two decimal places used here? It might be more appropriate to round to one decimal place or an integer to reflect the associated uncertainty more accurately.

We have updated all the results in this section now to one decimal place

L2607: Please expand the name of the month.

Thank you, we have corrected this.

L2612: "(Figure 14)".

[now figure 10]

Thank you, we have removed the brackets around "Figure 14"

L2701: "PI"?

"PI" stands for "Pre-industrial", which we have included in the text.

Table 7: Do two decimal places accurately reflect the uncertainties here? Consider rounding to one decimal place or an integer to better represent the associated uncertainty. The color bar is missing.

We have now used one decimal place.

L3022: I agree, and for this reason, the comparison is not fair. A discussion of this limitation would be helpful to provide context and ensure clarity.

As discussed in the previous section, the focus is on utility rather than fairness. If the only metric we have is only loosely correlated with fire activity, it is evident that we need to look for additional information to complement it. This is not merely about predicting the FWI; it is about determining whether the FWI is actually a useful metric for anticipating possible fires.

L3048-3057: Why not conduct a reforecast evaluation for 2023-24? Including this analysis would provide valuable insights and enhance the robustness of your findings.

Analysis of meteorological anomalies at the seasonal time scale is available through www.charts.ecmwf.int. To keep the report concise and focused, we have decided to remove the figure showing ENSO and IOD plumes, as these are accessible on the CDS and ECMWF websites.

L3111: I suggest clearly stating that a major limitation in assessing trends is that we only have global data for a few years. This limitation should be explicitly mentioned to provide context for the trend analysis and the definition of extreme events.

This topic comes under the later section "Observing Extreme Fire Events".

L3252-3258: I suggest adding that there are data-driven methods that work regionally and globally, but for this inaugural report, these methods are not included. This acknowledgment will provide a more comprehensive understanding of the current limitations and future directions for the report.

These methods are not included in the analysis of predictability but are used to study the drivers of the focal events. In Appendix B, where we discuss the roadmap of the report, we have expanded on this.

*However, it is the emergence of the data-driven revolution that holds the promise of significantly enhancing our predictive capabilities for extreme fires (McNorton et al., 2024). This has driven the development of semi-empirical tools at regional and global scales that could improve fire predictions, and their effectiveness will be assessed in the next edition of the report.*

L3308: 2024 or 2023?

Corrected

L3357: CMIP5?

Yes, and we have added this to the sentance: "Beyond that, we still show similar uncertainties in responses of Canada and Western Amazonia under different scenarios as highlighted by UNEP (2022), which uses the previous generation of climate models from CMIP5"

L3393: "Impacts" is underlined.

Corrected.

L3547: IFN?

Thanks, good catch. Integrated Fire Management - acronym now defined on first use.

L3650: Agree, but you do not use many regional datasets that can provide very useful information.

We added: "*In all cases, efforts to harmonise local to regional-scale observations of impact are required, in a similar manner to emerging compilations of regional fire monitoring systems (e.g. Gincheva et al. 2024).*"

L3669: Not sure if all of this paragraph is necessary here.

We feel that impacts of fire on the success of NBS may become an increasingly important topic for identifying fires with high impact on society and the environment.

L3774: Agree, but you do not use many regional datasets that can provide very useful information.

We added "In addition, regional products often provide scope to characterise the extremity of events over multi-decadal timescales and are now being provided in globally harmonised formats compatible with global analyses such as ours. These regional dataset should be utilised in future iterations of the State of Wildfires report."

L3798: Why do you use only the Global Fire Atlas and no other similar datasets?

This sentence was removed during revisions. In the revised text we added a clarification "or other individual fire atlases (e.g. Laurent et al., 2018; Artés et al., 2019)".

L3809: Maybe a good possibility here is to mention datacube datasets (e.g., Alonso et al., 2022).

This sounds great! But unfortunately we were not able to find the reference on Zenodo.

L3829: What about decadal prediction?

This topic comes in the later section "Future Projections"

[remaining 3,589 characters of this post omitted]